# Single-cell atlas of the human brain vasculature across development, adulthood and disease

Thomas Wälchli[1,2,3,4,37,38 ✉], Moheb Ghobrial[1,2,3,4,5,37], Marc Schwab[1,2,3,4,6], Shigeki Takada[2,7,8], Hang Zhong[1,2,3,4,5], Samuel Suntharalingham[1,2,9], Sandra Vetiska[2,8], Daymé Rodrigues Gonzalez[10], Ruilin Wu[11,12], Hubert Rehrauer[10], Anuroopa Dinesh[13], Kai Yu[11,12], Edward L. Y. Chen[13,14], Jeroen Bisschop[1,2,3,4,9], Fiona Farnhammer[1,2,3,4,9], Ann Mansur[2,8,11], Joanna Kalucka[15], Itay Tirosh[16], Luca Regli[4], Karl Schaller[17], Karl Frei[3,4], Troy Ketela[18], Mark Bernstein[2,19], Paul Kongkham[2,19,20], Peter Carmeliet[21,22,23], Taufik Valiante[2,19,24,25,26], Peter B. Dirks[2,14,27], Mario L. Suva[28,29], Gelareh Zadeh[2,19,30], Viviane Tabar[31], Ralph Schlapbach[10], Hartland W. Jackson[13,14,32], Katrien De Bock[5], Jason E. Fish[11,12,33], Philippe P. Monnier[9,34,35], Gary D. Bader[13,14,18,36,38] & Ivan Radovanovic[2,8,11,19,38]

A broad range of brain pathologies critically relies on the vasculature, and cerebrovascular disease is a leading cause of death worldwide. However, the cellular and molecular architecture of the human brain vasculature remains incompletely understood[1]. Here we performed single-cell RNA sequencing analysis of 606,380 freshly isolated endothelial cells, perivascular cells and other tissue-derived cells from 117 samples, from 68 human fetuses and adult patients to construct a molecular atlas of the developing fetal, adult control and diseased human brain vasculature. We identify extensive molecular heterogeneity of the vasculature of healthy fetal and adult human brains and across five vascular-dependent central nervous system (CNS) pathologies, including brain tumours and brain vascular malformations. We identify alteration of arteriovenous differentiation and reactivated fetal as well as conserved dysregulated genes and pathways in the diseased vasculature. Pathological endothelial cells display a loss of CNS-specific properties and reveal an upregulation of MHC class II molecules, indicating atypical features of CNS endothelial cells. Cell–cell interaction analyses predict substantial endothelial-to-perivascular cell ligand–receptor cross-talk, including immune-related and angiogenic pathways, thereby revealing a central role for the endothelium within brain neurovascular unit signalling networks. Our single-cell brain atlas provides insights into the molecular architecture and heterogeneity of the developing, adult/control and diseased human brain vasculature and serves as a powerful reference for future studies.

The brain vasculature is important for both the proper functioning of the normal brain as well as for a variety of vascular-dependent CNS pathologies such as brain tumours, brain vascular malformations, stroke and neurodegenerative diseases[1–9]. A better understanding of the underlying cellular and molecular mechanisms and architecture of the vasculature during brain development, in the healthy adult brain, as well as in vascular-dependent brain diseases, has broad implications for both the biological understanding as well as the therapeutic targeting of the pathological brain vasculature[10–15]. Vascular growth and network formation, involving endothelial cells (ECs) and other cells of the neurovascular unit (NVU), are highly dynamic during brain development, almost quiescent in the healthy adult brain and reactivated in a variety of angiogenesis-dependent brain pathologies, including brain tumours and brain vascular malformations[3,7,16–21], thereby activating ECs and perivascular cells (PVCs) of the NVU and other tissue-derived cells (hereafter collectively referred to as PVCs). However, it is unclear which molecular signalling cascades are reactivated and how they regulate brain tumour and brain vascular malformation vascularization and growth.

The CNS vasculature has unique features such as the blood–brain barrier (BBB) and the NVU[22–24]. During development, various CNS-specific and general signalling pathways drive CNS angiogenesis[3,7,23,25–27]. The brain vasculature also displays an arteriovenous (AV) endothelial hierarchy similar to peripheral vascular beds[28–30]. Developmentally regulated signalling axes in ECs are thought to contribute to the establishment of CNS-specific properties as well as AV specification of the endothelium in the healthy adult brain and to their alteration in disease[13,14]. Over the past years, single-cell transcriptome atlases of human peripheral organs[31–33], the human brain vasculature[34–37], and the mouse brain and peripheral vasculature[28,38] were established. Nevertheless, a landscape of the human brain vasculature at the single-cell level across fetal development, adulthood and various vascular-dependent diseases with a focus on the brain vascular endothelium is lacking. Here we created a comprehensive

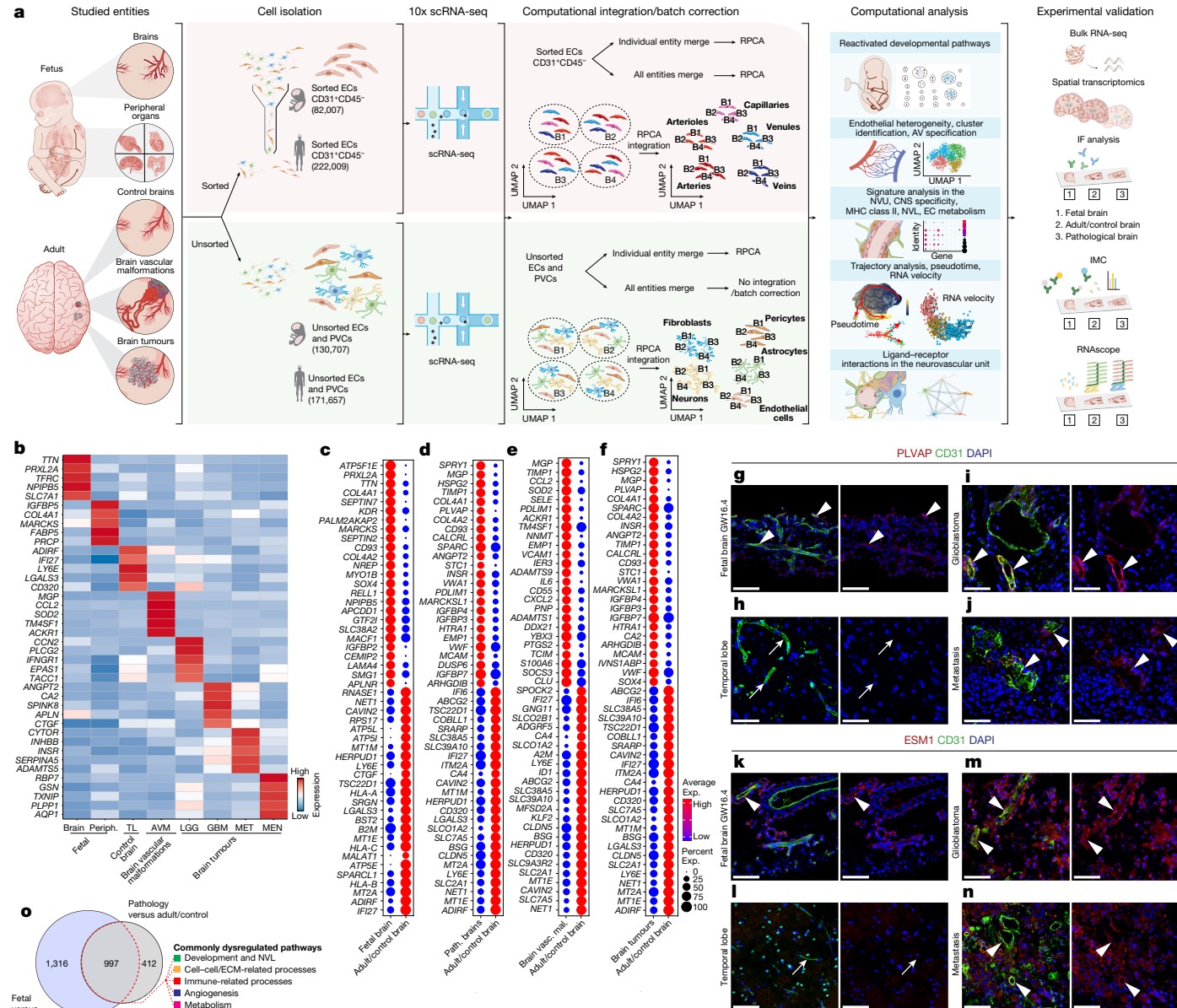

**Fig. 1 | Construction of a molecular sc-atlas of the human brain vasculature and reactivation of fetal programmes in pathological brain vascular ECs.** **a**, Schematic of the experimental workflow including scRNA-seq, computational analysis summary and validation experiments. **b**, Expression heat map of the top ranking marker genes in the indicated tissues. For the colour scale, red shows high expression, white shows intermediate expression and blue shows low expression. **c**–**f**, Dotplot heatmap of the fetal versus adult/control brain endothelium (**c**); pathological (path.) versus adult/control brain endothelium (**d**); brain vascular malformations (brain vasc. mal.) versus adult/control brain endothelium (**e**); and brain tumours versus adult/control brain endothelium (**f**) signatures based on differential gene expression analyses.

**g**–**n**, IF imaging of tissue sections from the indicated entities, stained for *PLVAP* (red; **g**–**j**), *ESM1* (red; **k**–**n**) and *CD31* (green). Nuclei are stained with DAPI (blue). The arrowheads indicate expression of PLVAP or ESM1 in blood-vessel ECs in the different tissues, and the arrows indicate the absence of expression in blood-vessel ECs in the different tissues. For **g**–**n**, scale bars, 50 µm. **o**, The overlap between the 2,313 significant pathways enriched in fetal brain ECs as compared to adult/control brain ECs and the 1,409 significant gene sets enriched in pathological brain ECs as compared to adult/control brain ECs. ECM, extracellular matrix; NVL, neurovascular link; periph., periphery; RPCA, reciprocal principal component analysis; UMAP, uniform manifold approximation and projection.

molecular atlas of the human brain vasculature using single-cell RNA sequencing (scRNA-seq) analysis in the developing, adult/control and diseased human brain (Fig. 1a, Extended Data Fig. 1 and Supplementary Methods). We identified extensive heterogeneity among ECs as well as common hallmarks across a spectrum of multiple brain pathologies, including commonly regulated angiogenic signalling pathways that significantly overlap with the fetal signalling axes, altered AV specification and CNS specificity, upregulation of MHC class II signalling and strong EC–EC/EC–PVC communication networks.

## Constructing a sc-atlas of the human brain vasculature

We constructed a human brain vasculature single-cell atlas (sc-atlas) using samples from fetal as well as adult control (undiseased atlas) and diseased brains, including adult brain vascular malformations and brain tumours (diseased atlas). We acquired freshly isolated cells (both fluorescence-activated cell sorting (FACS)-sorted ECs and unsorted ECs and PVCs; Supplementary Tables 1–4) from 8 individual fetuses[39–41] and from 61 adult brain samples (from 61 individual patients), covering adult temporal lobe (TL) controls and adult vascular-dependent

pathologies, including brain vascular malformations, namely, brain AV malformation (AVM)[36,42], and brain tumours, notably, lower-grade glioma (LGG)[22,43], glioblastoma (GBM)[22,44,45], lung cancer brain metastasis (MET)[22,46] and meningioma (MEN)[47] (Fig. 1a, Extended Data Fig. 1a,b and Supplementary Tables 1–4). Brain tissue samples were dissociated into single-cell suspensions, which were either FACS-sorted for ECs (CD31⁺CD45⁻) or processed as unsorted samples to examine all cells of the NVU (Fig. 1a). Single-cell transcriptomes were collected using the 10x Genomics Chromium system[48] and analysed. CD31⁺CD45⁻ ECs showed consistent expression of classical endothelial markers, such as *CD31*, *VWF* and *CLDN5*, while not expressing PVC markers (Supplementary Fig. 1a–o), thereby confirming the purity of EC isolations. In summary, 606,380 single cells, including 304,016 sorted ECs and 302,364 unsorted ECs and PVCs, passed the quality-control criteria (Fig. 1a and Supplementary Tables 3–5). The number of sorted ECs analysed here substantially exceeds the number of ECs analysed using scRNA-seq[36,37] or single-nucleus RNA-seq[34,35] in previous studies, and we directly compared single-cell transcriptomics of sorted ECs from the vasculature of the fetal and adult brain and of various brain pathologies. Notably, we report higher numbers of sorted ECs and of unsorted ECs and PVCs in the different brain entities compared with previous studies[34–37,49], enabling us to assess EC heterogeneity across development, adulthood and disease at a high resolution.

To address the role of the endothelium within the brain NVU across different entities, fetal, adult/control and pathological unsorted EC and PVC transcriptomes from 31 patients were analysed (Fig. 1a and Supplementary Fig. 2a–f). We identified 18 major brain cell types, including all known vascular, perivascular and other tissue-derived cell types in the human brain. The detected cell type distributions within the NVU differed between the fetal, adult control and pathological brain samples (Supplementary Figs. 2e–g and 3a–n and Supplementary Tables 10–16). Key signatures and differentially expressed genes (DEGs) were validated using bulk RNA-seq, RNAscope, spatial transcriptomics, immunofluorescence (IF) and imaging mass cytometry (IMC) (Fig. 1a).

We next compared ECs in the sorted samples across entities and found that ECs from different entities exhibited prominent transcriptomic heterogeneity (Extended Data Fig. 1c,f) as well as distinct gene expression signatures (Fig. 1b–f, Extended Data Fig. 1d,e,g,h and Supplementary Fig. 5). We defined major EC signatures, including a human fetal CNS (and peripheral) signature characterizing CNS and periphery-specific markers of the fetal vasculature (Extended Data Fig. 1g and Supplementary Table 6), a human fetal/developmental CNS/brain signature revealing properties of the developing and mature human brain vasculature, and a pathological signature of the diseased brain vasculature including a brain vascular malformation and a brain tumour signature (Fig. 1d–f, Supplementary Fig. 5a–c and Supplementary Table 25). The fetal and pathological brain EC signatures revealed differential expression of the well-known angiogenic markers *PLVAP* and *ESM1*[36,38,50–56], which we confirmed in the fetal and diseased brain entities using IF analysis[36,38,52–56] (Fig. 1g–n and Extended Data Fig. 2).

Although all of the entities revealed distinct EC markers, some EC markers were conserved across two or more entities (such as *ADIRF*, *EGR1*, *PLPP1* and *ANGPT2*) (Fig. 1b, Extended Data Fig. 1e,h and Supplementary Tables 8 and 9).

## Reactivation of fetal programmes in pathological brain ECs

We further assessed the differences in ECs across developmental stages and in pathological conditions (Supplementary Tables 7 and 25). DEGs between the fetal and adult/control stage and between the adult/control and pathological brain showed developmental and pathology-specific gene and pathway enrichments (Supplementary Fig. 5d–f), providing insights into functional specialization of the human brain vasculature

across development, homeostasis and disease (Supplementary Fig. 5). Using various approaches, including statistical regression, we found no evidence that age and sex[57–59] are confounders of our findings (Supplementary Tables 13 and 14). The top differentially regulated pathways in both fetal versus adult/control as well as in pathological versus adult/control brain EC signatures belonged to five main groups, including development and neurovascular link[3], cell–cell/extracellular-matrix-related processes, immune-related processes, angiogenesis and metabolism (Supplementary Fig. 5d–f). Notably, of the 1,409 differentially regulated pathways in pathology versus adult/control brain ECs, more than half (997) also showed differential regulation in fetal versus adult/control brain ECs (Fig. 1o and Supplementary Fig. 5f), highlighting the importance of developmental pathways in vascular-dependent brain pathologies. Bulk RNA-seq analysis confirmed the scRNA-seq findings, including the dysregulated pathways across pathologies (Supplementary Fig. 6 and Supplementary Table 26). Together, these data indicate that signalling axes driving vascular growth during fetal brain development are silenced in the adult control brain and reactivated in the vasculature of brain tumours and brain vascular malformations and that common dysregulated pathways are observed in the pathological brain vascular endothelium across diseases.

## Inter-tissue heterogeneity and AV zonation of brain ECs

To further address EC heterogeneity across different brain entities at the single-cell level, we pooled, integrated and batch-corrected (using RPCA)[60–63], clustered and visualized all fetal (21,512), adult/control (76,125) and pathological (145,884) sorted brain EC transcriptomes from 43 patients (Fig. 2a and Supplementary Figs. 7a–e and 9). Brain vascular ECs are organized along the human brain AV axis, referred to as AV zonation[28,64–69]. Endothelial clusters were biologically annotated using DEGs across entities, and we identified 44 EC subclusters (Fig. 2e, Supplementary Fig. 7 and Supplementary Table 18) that were arranged according to AV zonation, which we grouped into 14 major EC clusters for further downstream analysis (Fig. 2a and Supplementary Fig. 7a,h).

We characterized AV zonation markers[28,70] with arterial (subclustered into large artery, artery, arteriole) and venous (subclustered into large vein, vein, venule) clusters located at the opposite ends of the uniform manifold approximation and projection (UMAP), separated by major capillary clusters (subdivided into capillary and angiogenic capillary) in the fetal, adult and pathological brains (Fig. 2a,e, Supplementary Figs. 7 and 10 and Supplementary Tables 18 and 19), providing unprecedented transcriptional resolution by AV zonation[34–36].

While we confirmed differential expression of known marker genes of AV specification[28,38], we also identified AV-zonation markers that have not to our knowledge been identified previously in the human brain: *LTBP4* (large arteries); *ADAMTS1* (arteries); *VSIR*, *AIF1L*, *CD320* and others (arterioles); *SLC38A5*, *BSG*, *SLC16A1* and *SLCO1A2* (capillaries); *JAM2*, *PRCP*, *PRSS23* and *RAMP3* (venules); *PTGDS*, *POSTN* and *DNASE1L3* (veins); *CCL2* (large veins); and *PLVAP*, *ESM1* and *CA2* (angiogenic capillaries) (Fig. 2e, Supplementary Fig. 7i and Supplementary Tables 18 and 19). *PLVAP* and *ESM1* were among the top markers of the angiogenic capillary cluster and we confirmed *PLVAP* and *ESM1* expression in diseased brain entities and in the fetal brain, indicating its role in developmental and pathological vascular growth[50,51]. Indeed, *PLVAP* and *ESM1* exhibited RNA and protein expression in human fetal brain and human brain vascular malformation/tumour ECs on the basis of RNAscope and IF analysis (Extended Data Fig. 2).

We also assigned EC clusters outside AV zonation, notably (proliferating) stem-to-endothelial cell transdifferentiating (stem-to-EC) clusters and (proliferating) endothelial-to-mesenchymal transition (EndoMT) clusters (Fig. 2e and Supplementary Fig. 7a,i), for which we identified specific molecular markers. We found proliferating ECs (such as *TOP2A*

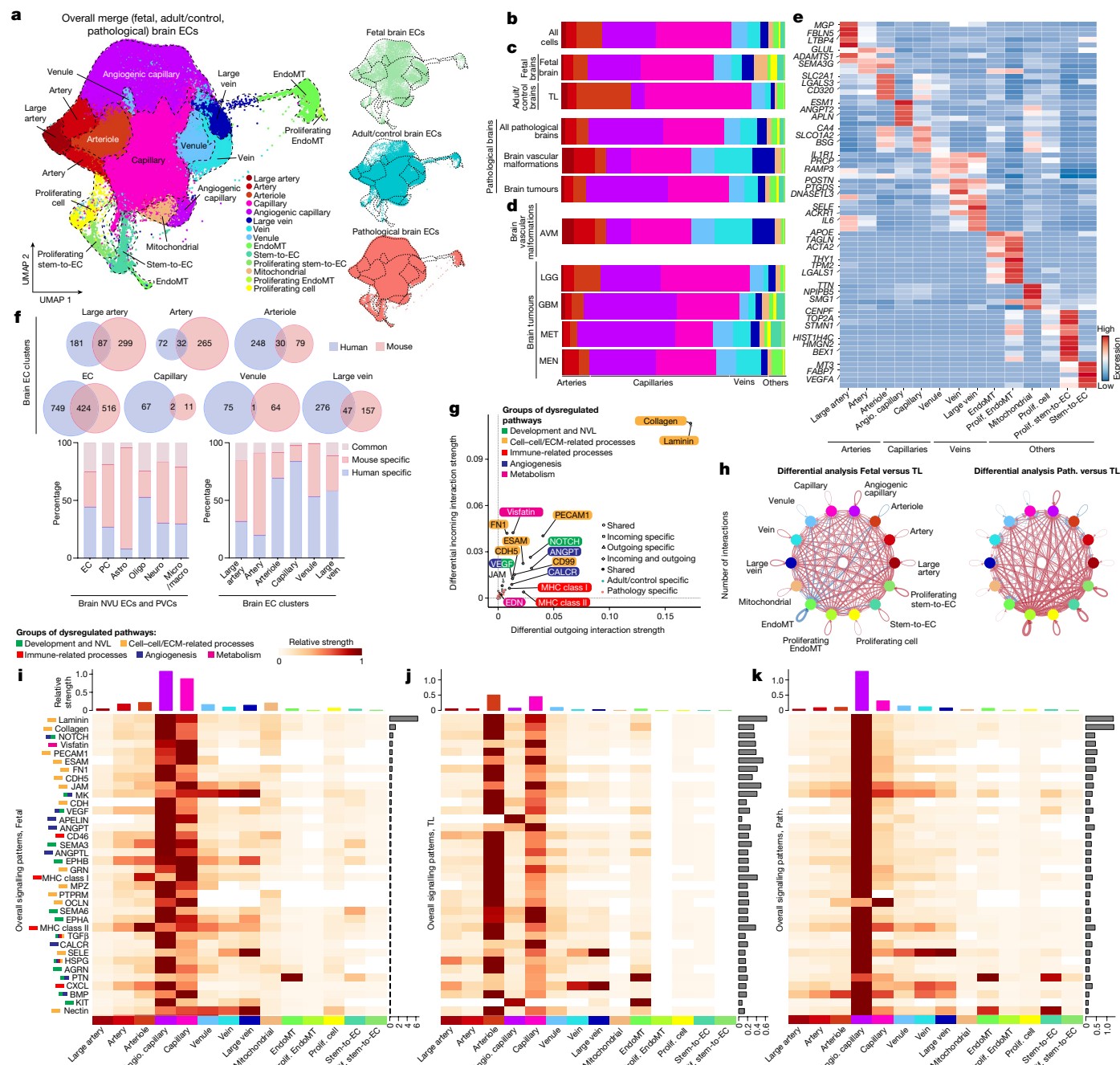

**Fig. 2 | Inter-tissue heterogeneity and AV zonation of brain vascular ECs.**
**a**, UMAP plot of the 243,521 integrated/batch corrected fetal, adult/control and pathological brain ECs across 5 (fetal), 9 (adult/control) and 29 (pathological) individuals (Supplementary Table 3), colour coded by EC AV specification, and UMAP plots split by tissue of origin: fetal brain (5 individuals), adult/control brain (9 individuals) and brain pathologies (29 individuals). **b–d**, The relative abundance of EC subtypes (AV specification cluster) from the indicated tissue of origin. **b–d** are coloured according to the colour code in **a** (Supplementary Table 10). The number of individuals analysed was as follows: $n = 43$ (all entities), $n = 5$ (fetal brain), $n = 9$ (adult/control brain (TL)), $n = 29$ (all pathological brains), $n = 5$ (brain vascular malformations), $n = 24$ (brain tumours), $n = 5$ (AVM), $n = 6$ (LGG), $n = 8$ (GBM), $n = 5$ (MET) and $n = 5$ (MEN). **e**, The top ranking marker gene expression levels in different EC subtypes. For the colour scale, red shows high expression, white shows intermediate expression and blue shows low expression. Angio., angiogenic; prolif., proliferating. **f**, The overlap between

human and mouse AV specification markers (of large artery, artery, arteriole, capillary, venule and large vein) and endothelial (EC) markers (top). Bottom, the percentage of common, human-specific and mouse-specific cell/AV specification markers. Astro, astrocytes; micro/macro, microglia/macrophages; neuro, neurons; PC, pericytes. **g**, Scatter plot showing the differential incoming and outgoing interaction strength of pathways in angiogenic capillaries, identifying signalling changes in those cells in pathological as compared to the control conditions. **h**, The number of statistically significant ligand–receptor interactions between EC subtypes in fetal versus adult/control brains (left) and pathological (path.) versus adult/control brains (right). The circle plots show a differential analysis of the intercellular signalling interactions; red indicates upregulation and blue indicates downregulation. **i,j,k**, The overall signalling patterns of different EC subtypes in fetal (**i**), adult/control (**j**) and pathological (**k**) brains. Grey bars indicate signalling strength.

and *MKI67*) in the fetal (3.54%), adult (0.4%) and pathological brains (1.37%) (Fig. 2e, Supplementary Fig. 7i and Supplementary Table 19). We identified EndoMT clusters expressing both mesenchymal (such as *APOE*, *ACTA2* and *TAGLN*) and endothelial markers[71] (Fig. 2e, Supplementary Fig. 7i and Extended Data Fig. 2q$_i$–b$_{ii}$,g$_{ii}$–j$_{ii}$′). Notably, we observed two subsets of EndoMT ECs (proliferating EndoMT and EndoMT) that were increased in certain pathologies (MEN > LGG > GBM). Proliferating EndoMT ECs expressed both EndoMT and proliferation markers (for example, *ACTA2* and *MKI67*) (Fig. 2e, Supplementary Fig. 7i and Supplementary Tables 8 and 9). Using RNAscope, IF and IMC, we found ACTA2⁺CD31⁺CLDN5⁺ co-expressing but PDGFRβ⁻ (suggesting no pericyte identity) ECs across pathologies (Extended Data Fig. 2q$_i$–b$_{ii}$,g$_{ii}$–j$_{ii}$′,k$_{ii}$–q$_{ii}$⁵), indicating the presence of EndoMT ECs in the diseased vasculature. In GBMs and METs, we observed stem-to-EC clusters that expressed classical EC markers (such as *CD31*, *CLDN5*, *CDH5* and *VWF*) to a lower level compared with other EC clusters, as well as some markers of (tumour) stem cells (Fig. 2e, Extended Data Fig. 3 and Supplementary Fig. 7i), suggesting ECs undergoing stem-to-EC transdifferentiation. In GBMs, we identified a stem-to-EC cluster expressing the GBM stemness markers *SOX2*, *PTPRZ1*, *POUR3F2* and *OLIG1*[72,73], and EC markers[74,75] (Supplementary Fig. 7i and Extended Data Fig. 3a–i,a$_i$–n$_i$′). In METs, we noted a previously undescribed stem-to-EC population that co-expressed EC markers and stem cell markers of lung cancers (for example, *SOX2*, *EPCAM*, *CD44* and *SFTPB*)[76] (Extended Data Fig. 3n–v,g$_i$–z$_i$′ and Supplementary Fig. 7i). In GBMs and METs, we identified groups of stem-to-ECs that co-expressed stemness (for example, *SOX2*, *PTPRZ1*, *EPCAM1* and *SFTPB*) and proliferation markers (for example, *MKI67*, *BEX1*, *HMGB2* and *UBE2C*) (Fig. 2e and Extended Data Fig. 3). To validate the stem-to-EC clusters in GBM and MET, we used double immunostaining for EC and stemness markers. In GBM, we found SOX2⁺CD31⁺ and PTPRZ1⁺CD31⁺ co-expressing ECs, whereas, in MET, we observed EPCAM⁺CD31⁺ and SFTBP⁺CD31⁺ co-expressing ECs (Extended Data Fig. 3). The confirmation of tumour stemness marker enrichment in a subset of tumour ECs suggests the presence of stem-to-ECs in GBM and MET vasculature.

We next addressed the distributions of EC clusters between the fetal, adult/control and pathological brains. Capillaries accounted for around 60.5% of ECs, arterial ECs accounted for 18.2% and venous ECs accounted for 16.2%, in agreement with previous studies[3,17]. We further uncovered previously unrecognized EC heterogeneity across a wide range of human brain tissues (Fig. 2b–d, Supplementary Fig. 10 and Supplementary Tables 10–16). Angiogenic capillary proportions were significantly higher in the fetal brain and in brain tumours (GBM > MET > MEN > LGG), illustrating their angiogenic capacity[3,13,22,77], whereas brain vascular malformations (AVM) revealed significantly elevated proportions of venous clusters, indicating their venous character[78,79] (Fig. 2c,d, Supplementary Fig. 10 and Supplementary Table 12). We next evaluated whether AV-zonation markers were conserved across species[34–36,49,80] (Supplementary Fig. 4). Although the overall structure of AV zonation was conserved between human and mouse, the number of conserved AV-zonation genes in the different AV compartments was low. Accordingly, we found the highest proportion of human-specific AV-zonation markers in small > large-calibre and venous > arterial vessel ECs (Fig. 2f, Supplementary Fig. 4z$_{xxix}$,z$_{xxx}$ and Supplementary Table 17), and we validated these human-specific markers referring to the Human Protein Atlas (HPA)[35,81–83] (Supplementary Fig. 8).

In the brain NVU, mapping of our dataset to the freshly isolated mouse dataset[70] revealed high transcriptomic similarity between species for ECs and PVCs[84] (Supplementary Fig. 4z$_{xvii}$–xvix). We further observed that neurons and astrocytes showed the greatest transcriptional divergence (Fig. 2f and Supplementary Table 17), while ECs and oligodendrocytes displayed the highest percentage of human-specific markers, in agreement with previous studies[34–36]. These species-specific differences along AV-zonation suggest fundamental disparities in brain vascular properties, indicating the necessity to directly study sorted/enriched ECs and PVCs of the human brain vasculature at the single-cell level.

As EC clusters reside in close proximity to each other along the AV tree, we next inferred cell–cell communication pathways[85,86]. Differential analysis revealed increased cellular cross-talk among EC clusters in pathological and fetal ECs, highlighting a key role for angiogenic capillaries. Angiogenic capillaries displayed upregulation of several signalling pathways, including the five above-mentioned groups in both the diseased and fetal (versus adult/control) brain (Fig. 2g–k and Supplementary Fig. 11), highlighting this cluster as a major signalling mediator within brain EC networks.

## Alteration of AV specification in pathological brain ECs

Failure of proper AV specification in brain vascular malformations and formation of tortuous arteries and veins in brain tumours has been reported[77,87], but AV specification in brain pathologies and fetal (brain) development remains poorly understood. We ordered ECs along a one-dimensional transcriptional gradient using Monocle[88] and TSCAN[89] to examine the AV axis in the different entities. Whereas arterial and venous markers peaked at opposite ends, capillary markers showed peaks in the mid-section throughout all entities (Fig. 3b,f,j, Extended Data Fig. 4b,f,j,n,r,v and Supplementary Fig. 13III), indicating that in silico pseudospace and pseudotime recapitulate in vivo anatomical topography of EC clusters in the human brain vasculature[28,38]. We observed AV zonation throughout the fetal, control and pathological brains, but observed a partial alteration of EC ordering along the AV axis in disease (Fig. 3b,c,f,g,j,k and Extended Data Fig. 4).

We defined an AV signature comprising genes revealing significant expression gradients of ECs along the AV axis (Fig. 3d,h,l and Extended Data Fig. 4d,h,l,p,t,x). The seamless zonation continuum was recapitulated in all entities but again showed alteration in pathologies. While AV markers revealed a clear distinction between AV compartments in the fetal and adult/control brain, showing specific markers of large arteries (for example, *VEGFC*, *FBLN5*), arterioles (such as *LGALS3*, *AIF1L*), capillaries (for example, *SLC35A5*, *MFSD2A*), angiogenic capillaries (such as *ESM1*, *ANGPT2*), venules (for example, *JAM2* and *PRCP*) and large veins (such as *SELE* and *SELP*), some zonation markers showed a less-specific presence in pathologies (Figs. 2 and 3d,h,l and Extended Data Fig. 4d,h,l,p,t,x).

Whereas almost all fetal and pathological ECs were quite similar to temporal-lobe EC clusters[60], small-calibre vessel ECs were more different compared with their healthy temporal lobe counterparts (Extended Data Fig. 4z$_v$–z$_{viii}$), probably pertaining to a higher vulnerability of small-calibre vessel ECs to alterations in the local tissue microenvironment[38].

To further address lineage relationships in AV specification, we referred to RNA velocity[90,91] and diffusion map[92], revealing vectors from multiple EC clusters towards the angiogenic capillary cluster in angiogenic entities (tumours > fetal brain > vascular malformations), whereas, in vascular malformations, we observed vectors from various EC clusters towards venous clusters (Fig. 3a,e,i, Extended Data Fig. 5 and Supplementary Fig. 13). These results indicate that RNA velocity can suggest timeline relationships among human brain vascular ECs[23,29].

We next addressed whether EC markers of AV clusters were conserved between vascular beds or expressed in a more tissue-specific manner[38]. While we identified multiple conserved markers for large arteries and large veins, capillaries were more tissue/entity specific, indicating a more pronounced transcriptional heterogeneity of the capillary bed across the different brain tissues[38]. Accordingly, capillaries showed more tissue-specific markers than large-calibre vessels (Fig. 3m–p), indicating a higher susceptibility of capillary ECs to the local tissue microenvironment.

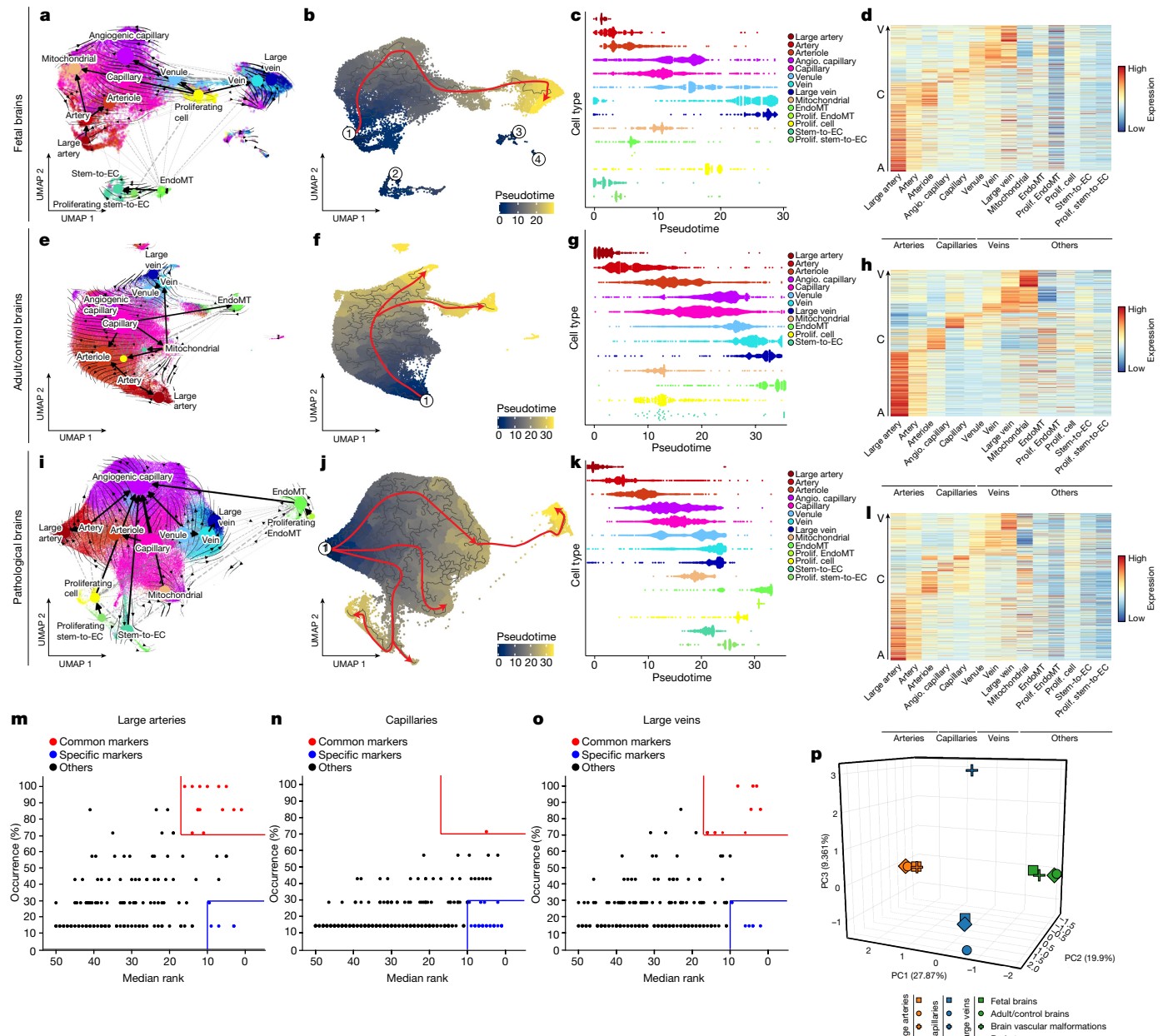

**Fig. 3 | Alteration of AV specification in pathological brain vascular ECs.** **a**,**e**,**i**, UMAP plots of human brain ECs isolated from fetal brains (21,512 ECs from 5 individuals; **a**), adult/control brains (76,125 ECs from 9 individuals; **e**) and pathological brains (145,884 ECs from 29 individuals; **i**), coloured by AV specification. RNA velocity streamlines and partition-based graph abstraction (PAGA) vectors extended by velocity-inferred directionality are superimposed onto the UMAPs. **b**,**f**,**j**, UMAP plots of human brain ECs isolated from fetal brains (**b**), adult/control brains (**f**) and pathological brains (**j**), coloured by pseudotime. The red line, which was drawn manually, indicates the major trajectory flow. **c**,**g**,**k**, Pseudotime order of ECs, colour coded according to AV specification from fetal brains (**c**), control adult/control brains (**g**) and pathological brains (**k**). **d**,**h**,**l**, Heat map of adult/control brain EC AV specification signature gene expression in human brain ECs isolated from fetal brains (**d**), adult/control brains (**h**) and pathological brains (**l**). A, arterial; C, capillary; V, venous. **m**,**n**,**o**, Common and tissue-specific markers in ECs from large arteries (**m**), capillaries (**n**) and large veins (**o**) in different tissue types (fetal brain, adult/control brain, brain vascular malformations and brain tumours). The red boxes highlight conserved markers between ECs from different tissues; the blue boxes highlight tissue-specific markers. Dots are coloured as defined in the legend. **p**, Three-dimensional principal component analysis visualization of pairwise Jaccard similarity coefficients between the indicated ECs from the different tissues.

## Alteration of CNS specificity in pathological brain ECs

We next examined CNS-specific properties distinguishing brain ECs from ECs outside the CNS[3,5]. Bulk RNA-seq analysis in mice revealed a BBB-enriched transcriptome[93], but how the human brain EC CNS properties differ at the single-cell level and whether they are heterogeneous across developmental stages and in disease remains largely unclear. The development of the human fetal BBB (occurring between gestational week 8 and 18)[39,40,94] is controversial[41,95]. We therefore studied human fetal BBB development at a high resolution. Referring to a human/mouse BBB signature (Supplementary Table 20 and Extended Data Fig. 6a–j), we observed an increased BBB signature expression with increased gestational age (Extended Data Fig. 6b–j), in agreement with a previous study[96]. Along the AV compartments, the BBB signature revealed a higher expression in small- versus large-calibre vessels across developmental stages (Extended Data Fig. 6d), in agreement

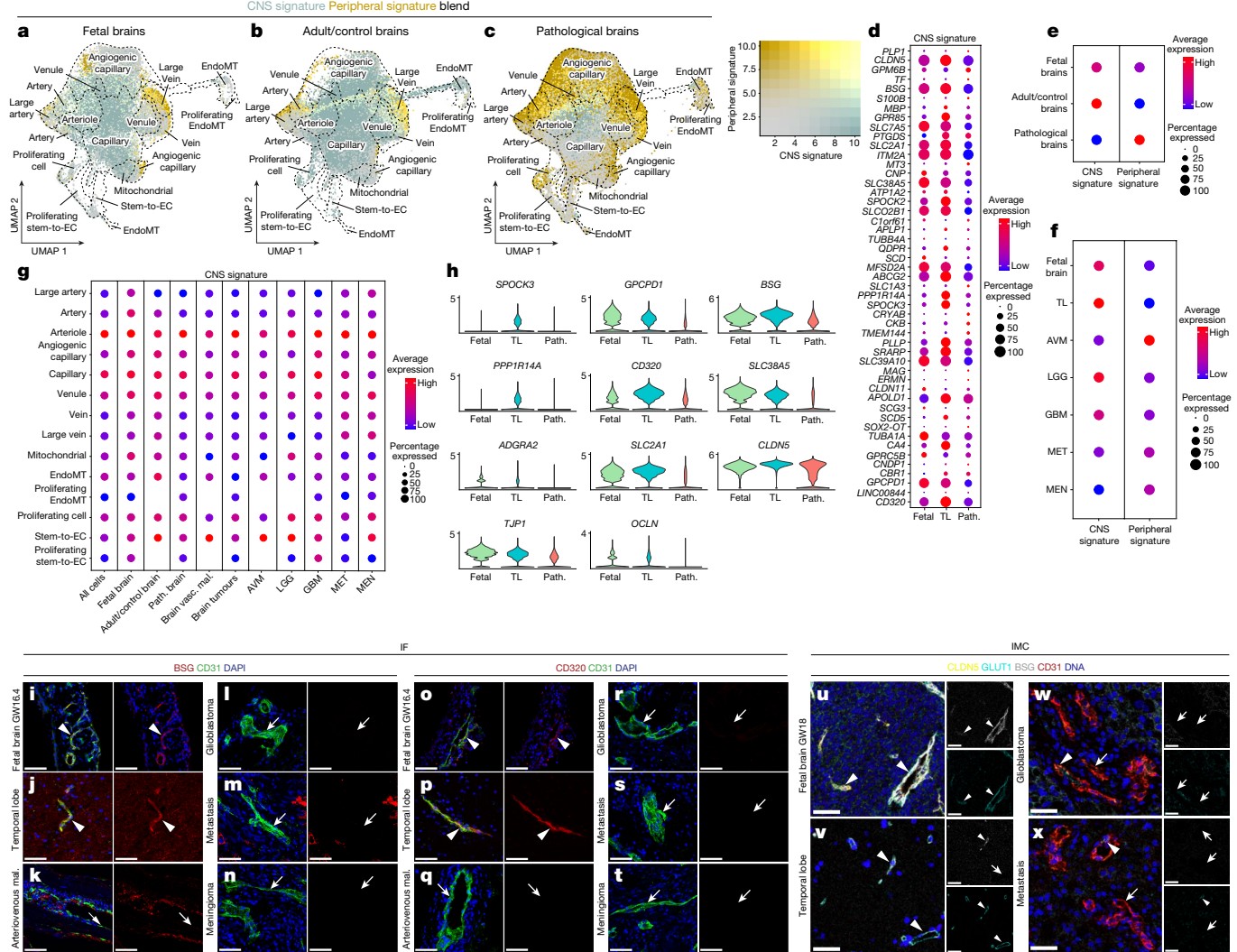

**Fig. 4 | Alteration of CNS specificity in pathological brain vascular ECs.**
**a–c**, UMAP plots of the ECs from fetal brains (21,512 ECs from 5 individuals; **a**), adult/control brains (76,125 ECs from 9 individuals; **b**) and pathological brains (145,884 ECs from 29 individuals; **c**). Plots are colour coded for CNS signature (green) and peripheral signature (yellow). **d**, CNS signature genes expression in fetal brain, adult/control brains (temporal lobes) and pathological brain ECs. **e**,**f**, CNS and peripheral signature expression in fetal brain, adult/control brain and pathological brain ECs (**e**) and in each individual entity (**f**). **g**, The CNS signature at the level of AV specification for the indicated entities. For the colour scale, red shows high expression and blue shows low expression. The dot size represents the percentage expression within the indicated entity. **h**, The expression of representative CNS-specific and BBB marker genes for fetal brain versus adult/control brain versus pathological brain ECs. **i–t**, IF images for the protein expression of *BSG* (**i–n**) and *CD320* (**o–t**) in fetal brain (**i** and **o**), adult/control brain (TL; **j** and **p**), brain AVMs (**k** and **q**), GBM/high-grade glioma (**l** and **r**), metastasis (**m** and **s**) and meningioma (**n** and **t**). For **i–t**, scale bars, 80 μm (fetal brain) and 50 μm (adult control and pathological brains). **u–x**, IMC imaging of fetal brain (**u**), adult/control brain (TL; **v**), GBM/high-grade glioma (**w**) and metastasis (**x**) tissue samples visualizing five metal-conjugated antibodies (Supplementary Table 24). Representative pseudocolour images of *CLDN5*, *GLUT1*, *BSG* and *CD31* combined, and individual *GLUT1* and *BSG* channels are shown; white, overlap; yellow, *CLDN5*; cyan, *GLUT1*; grey, *BSG*; red, *CD31*; blue, DNA intercalator. For **u–x**, scale bars, 50 μm. The arrowheads indicate blood-vessel ECs expressing the indicated markers in the different tissues. The arrows indicate blood-vessel ECs not expressing the indicated markers in the different tissue.

with previous findings[97] and probably pertaining to the susceptibility of capillaries to the local microenvironment[38].

Next, to address molecular differences of CNS and peripheral ECs at the single-cell level, we defined a human adult and fetal CNS signature (Fig. 4a–f, Supplementary Fig. 14l–q and Supplementary Table 20). These include known BBB (*MFSD2A*[98] and *CLDN5*[99]) and capillary markers (*CA4*[28,100] and *SPOCK2*[38]), as well as novel genes enriched in the CNS vasculature such as *SPOCK3*, *BSG* and *CD320* (Supplementary Table 20). The adult and fetal CNS signatures showed elevated expression with increasing gestational age and in small- versus large-calibre vessels across developmental stages, reminiscent of the BBB signature expression pattern (Fig. 4g and Extended Data Fig. 6d).

CNS properties were observed in the fetal and adult brains, whereas alteration of the CNS signature and concomitant acquisition of the peripheral signature was seen in pathologies (Fig. 4a–f and Supplementary Figs. 14 and 15). Comparing the CNS signature between pathological and adult/control brain ECs, we found downregulation of *SLC2A1*, which is dysregulated in neurodegenerative conditions[69]; the lipid transporter *MFSD2A*, which is expressed in brain ECs and restricts caveolae-mediated transcytosis at the BBB[98,101,102]; and the BBB marker *CLDN5* (Fig. 4d–f and Supplementary Table 20), therefore suggesting BBB alteration in pathologies[22]. The CNS signature was highest in the temporal lobe, followed by intra-axial primary brain tumours and fetal brain (LGG > fetal brain > GBM), brain vascular malformations, intra-axial secondary brain tumour MET and extra-axial brain tumour

MEN, whereas the peripheral signature followed an inversed pattern (Fig. 4e,f and Supplementary Fig. 14p,q).

We next addressed CNS-specific properties along the AV axis. In the fetal and adult brain, the CNS signature was mainly expressed by small-calibre vessels, while the peripheral signature was predominantly present in large arteries and large veins. We observed a similar pattern in pathological brains with, however, a notable decrease in cells expressing the CNS signature (most pronounced for angiogenic capillaries > capillaries), paralleled by an increase in cells expressing the peripheral signature predominantly for angiogenic capillaries and large-calibre vessels (Fig. 4g and Supplementary Fig. 14h).

We next examined how the CNS, BBB and peripheral signatures changed along the AV axis in each pathological entity. The CNS and BBB signatures were downregulated in every pathology with a similar pattern to that described above and reaching the highest baseline values of CNS specificity at the capillary and arteriole levels, with the capillaries being the cluster mostly affected by pathologies[38] (Fig. 4g and Supplementary Fig. 14h), probably pertaining to the influence of the local microenvironment for small-calibre vessels. The peripheral signature was upregulated in disease, peaking for AVM > MEN > MET > GBM and lower expression for LGG, predominantly affecting large-calibre vessels and angiogenic capillaries (Fig. 4h and Supplementary Figs. 14 and 15). These data indicate that CNS ECs acquire CNS-specific properties during fetal-to-adult transition and take on a peripheral EC signature in disease conditions.

The CNS and BBB signatures are tightly linked to a functional BBB in vivo[103] and BBB dysfunction affects the CNS properties of ECs[93]. We therefore investigated the human and mouse BBB dysfunction modules, with the latter being upregulated in CNS ECs after various disease triggers in the mouse brain, shifting CNS ECs into peripheral EC-like states[93]. We found that the human and mouse BBB dysfunction modules were upregulated in human brain tumours and brain vascular malformations as well as in the fetal brain (Supplementary Fig. 16a–n), probably due to pathways related to BBB dysfunction[93]. Both the human and mouse BBB dysfunction modules were highest in AVM > GBM > MET > MEN (Supplementary Fig. 16a–n). The BBB dysfunction modules expression along the AV axis revealed enrichment in large-calibre vessels and angiogenic capillaries, mimicking the peripheral signature expression, again indicating that pathological CNS ECs take on a peripheral endothelial gene expression[93] (Supplementary Fig. 16h–n). Comparison to BBB dysfunction modules in human Alzheimer's disease[35], Huntington's disease[34] and AVMs[36] revealed some overlap with the human and mouse BBB dysfunction modules[93] (Supplementary Fig. 16o–s and Supplementary Table 21), indicating common and distinct features among brain diseases.

We confirmed decreased expression of the CNS-signature genes *SPOCK3*, *BSG*, *CD320*, *PPP1R14A* and *SLC38A5* in all brain tumours and brain vascular malformations (Fig. 4h–x and Extended Data Figs. 6k–t′₄, 9 and 10) using IF and IMC, thereby highlighting the alteration of CNS properties in the diseased human cerebrovasculature.

## Upregulation of MHC class II in pathological brain ECs

We identified EC populations expressing the MHC class II genes *CD74*, *HLA-DRB5*, *HLA-DMA*, *HLA-DPA1* and *HLA-DRA* in pathological CNS tissues. This antigen-presenting signature, indicating possible immune functions of human brain ECs, prompted us to investigate the heterogeneity of MHC class II transcripts between tissues at the single-cell level.

scRNA-seq identified endothelial MHC class II expression in peripheral human and mouse tissues[31,100], but assessment of MHC class II expression in human brain vascular beds at the single-cell level is lacking. To assess MHC class II gene expression across brain development and disease, we defined a human MHC class II signature including MHC class II receptors (Fig. 5d and Supplementary Table 22). The MHC class II signature was upregulated in pathologies, and low in the

fetal brain (Fig. 5a–f and Extended Data Fig. 9), in agreement with a previous study[31]. We found that the MHC class II signature was highest in AVM > MEN > MET, followed by LGG > GBM, the temporal lobe and the fetal brain, grossly following the peripheral signature expression gradient (Fig. 5f). We examined MHC class II signature expression patterns according to AV zonation. Whereas, in the fetal brain, only very few ECs (large arteries and arterioles) expressed the MHC class II signature, mainly large-calibre vessel (large arteries and large veins) ECs expressed a signature of genes involved in MHC class II-mediated antigen presentation in the adult brain (Fig. 5a–c,g).

The MHC class II signature was upregulated in all pathologies according to the pattern described above[38] (Fig. 5a–c,g–z). We observed a partial overlap of the MHC class II and peripheral signatures and of the BBB dysfunction module with a common predominance in large-calibre vessels, but a more widespread/stronger expression of the peripheral signature and BBB dysfunction module in angiogenic capillaries consistent with previous findings[69]. These data suggest that pathological CNS ECs upregulate MHC class II receptors in brain tumours and brain vascular malformations.

We confirmed enrichment of MHC class II genes including *CD74* and others in the pathological human cerebrovasculature using IF, IMC and RNAscope (Fig. 5h–w and Extended Data Figs. 10, 11, 12r and 13).

Spatial transcriptomics confirmed spatial co-localization of MHC class II ligands and receptors on ECs in the temporal lobe and in GBM (Supplementary Fig. 18). MHC class II signalling seems to be mediated mainly by *APP*, *COPA* and *MIF* ligands and the *CD74* receptor in AV clusters (Supplementary Fig. 17), and *APP–CD74*, *COPA–CD74* and *MIF–CD74* have been described as ligand–receptor pairs[104,105].

## A key role for ECs in the human brain NVU

Single-cell transcriptomics of unsorted ECs and PVCs offers the opportunity to address cellular cross-talk and the role of ECs within the NVU. To address cell–cell interactions across entities, we constructed ligand–receptor interaction maps[85,86]. In the majority of entities, ECs were at the centre of the network displaying numerous interactions with other ECs and PVCs (Extended Data Fig. 12a–i and Supplementary Fig. 19), indicating a crucial role of ECs in NVU function and EC–PVC cross-talk. In fetal and adult/control brains, ECs showed most interactions with fibroblasts, pericytes and astrocytes (Extended Data Fig. 12a–f). In brain pathologies, ECs displayed increased interaction numbers as well as increased interactions with immune cells (Extended Data Fig. 12g–i and Supplementary Figs. 19 and 20). Intercellular signalling pathways were substantially increased in fetal and pathological ECs (and PVCs) (Extended Data Fig. 12j,k and Supplementary Fig. 20a–g). Cell–cell communication analysis predicted upregulation of signalling pathways in the developing versus control brain as well as diseased versus control brain NVUs, including similar pathways as observed among EC clusters in EC–EC networks (Fig. 2i–k, Extended Data Fig. 12l–n and Supplementary Fig. 20k–n). Intercellular cross-talk analysis predicted a key role for the ECs within the fetal, adult/control and diseased brain NVU signalling networks (Extended Data Fig. 12l–n and Supplementary Fig. 20e–g).

During fetal brain development and in brain pathologies, we observed upregulation of ligands and receptors on ECs and PVCs as well as of the corresponding pathways, which partially overlapped with the ones in EC–EC cross-talk (Extended Data Fig. 12l–n), suggesting that these ligand–receptor interactions contribute to brain EC–PVC signalling.

On the basis of our observation of MHC class II signalling in EC–EC communication, we next addressed MHC class II signalling in EC–PVC intercellular cross-talk, which predicted elevated MHC class II signalling (predominantly in microglia and macrophages, ECs and tumour cells/oligodendrocytes) in brain pathologies (Extended Data Fig. 12o,p,q and Supplementary Fig. 20h–j). Spatial transcriptomics confirmed spatial co-localization of MHC class II ligands and receptors on ECs and PVCs

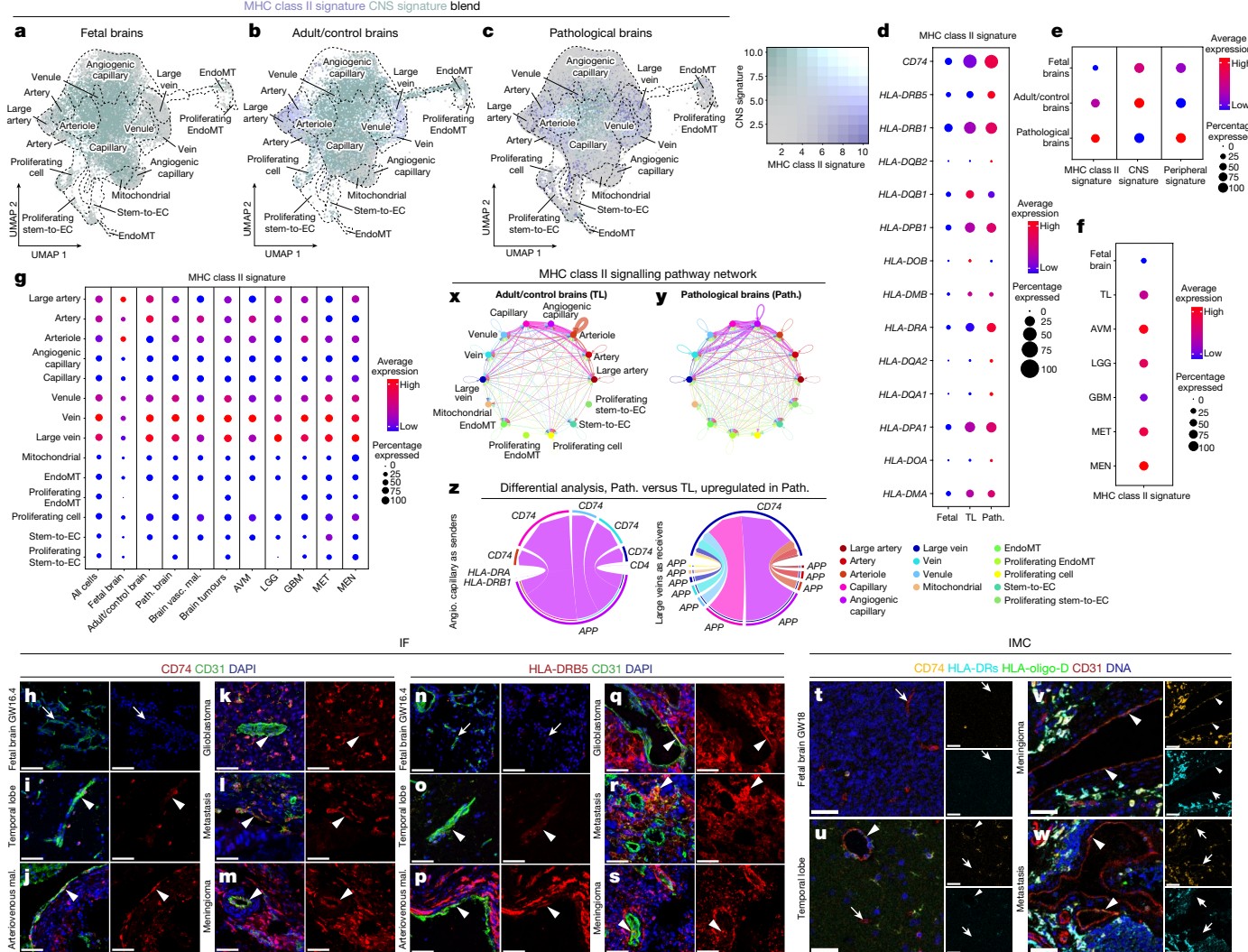

**Fig. 5 | Upregulation of MHC class II receptors in pathological brain vascular ECs. a–c**, UMAP plots of ECs from fetal (21,512 ECs from 5 individuals; **a**), adult/control (76,125 ECs from 9 individuals; **b**) and pathological brains (145,884 ECs from 29 individuals; **c**). Plots are colour coded for MHC class II (violet) and CNS (green) signatures. **d**, MHC class II signature gene expression in fetal, adult/control (temporal lobes) and pathological brain ECs. **e,f**, MHC class II, CNS and peripheral signature expression in fetal, adult/control and pathological brain ECs (**e**), and MHC class II signature expression in each individual entity (**f**). **g**, The MHC class II signature at the level of AV specification for the indicated entities. For the colour scale, red shows high expression and blue shows low expression. The dot size represents the percentage expression within the indicated entity. **h–s**, IF images for the protein expression of *CD74* and *HLA-DRB5* in the fetal brain (**h** and **n**) and the adult/control brain (TL; **i** and **o**), in brain AVMs (**j** and **p**), in GBM/high-grade glioma (**k** and **q**), in metastasis (**l** and **r**) and in meningioma (**m** and **s**). For **h–s**, scale bars, 80 μm (fetal brain)

and 50 μm (adult control and pathological brains). **t–w**, IMC imaging of fetal brain (**t**), adult/control brain (**u**), meningioma (**v**) and metastasis (**w**) tissue samples visualizing metal-conjugated *CD74*, pan-HLA-DR, oligo-HLA-D and *CD31* primary antibodies. An overlay of pseudocolour images as well as individual channels for *CD74* and pan-HLA-DR are shown; white, overlap; orange, *CD74*; cyan, pan-HLA-DR; green, oligo-HLA-D; red, *CD31*; blue, DNA intercalator. For **t–w**, scale bars, 50 μm. **x,y**, The strength of MHC class II signalling interactions between the different EC subtypes of the adult/control brain (**x**) and pathological brain (**y**) ECs at the AV specification level. **z**, Differential analysis of MHC class II ligand–receptor pairs. Chord/circos plots showing upregulated MHC class II signalling in angiogenic capillaries as the source and all other cell clusters as targets (left), and large veins as receivers (right). The edge thickness represents its weight. The edge colour indicates the sender cell type. The arrowheads indicate ECs expressing the indicated markers. The arrows indicate ECs not expressing the indicated markers.

(Supplementary Fig. 18) in the temporal lobe and in GBM, while IMC illustrated physical proximity between MHC class II-expressing ECs and microglia/macrophages across all entities (Extended Data Figs. 12r and 13). Notably, the *APP–CD74*, *COPA–CD74* and *MIF–CD74* ligand–receptor pairs that were predicted to mediate MHC class II signalling in EC–EC interactions were also predicted ligand–receptor pairs in the developing, adult and diseased NVU (Supplementary Fig. 20h–j), with ECs notably strongly expressing *CD74* (Supplementary Fig. 19e,j,o,t,y). These data suggest involvement of MHC class II in EC–immune cell interactions and indicate that the *APP–CD74*, *COPA–CD74* and *MIF–CD74* ligand–receptor pairs contribute to NVU signalling.

## Discussion

Here we generated a large-scale single-cell molecular atlas of the developing fetal, adult/control and diseased human brain vasculature at a very high resolution, using scRNA-seq, composed of 606,308 freshly isolated endothelial, perivascular and other tissue-derived cells covering a substantial diversity of human brain tissue.

We have provided molecular definitions of human brain cell types and their differences by brain developmental stage and pathology, thereby unravelling organizational principles of ECs and PVCs composing the human brain vasculature. Our experimental methodology relies on

transcriptional profiles of human cerebrovascular cells generated from fresh human neurosurgical resections and fresh fetal abortions, reducing the likelihood of transcriptional alterations associated with post-mortem tissue acquisition (Supplementary Discussion).

Our human vascular brain atlas provides a basis for understanding the organizing principles and single-cell heterogeneity of universal, specialized and activated endothelial and PVCs with broad implications for physiology and medicine, and serves as a powerful publicly available reference for the field.

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

¹Group Brain Vasculature and Perivascular Niche, Division of Experimental and Translational Neuroscience, Krembil Brain Institute, Krembil Research Institute, Toronto Western Hospital, University Health Network, University of Toronto, Toronto, Ontario, Canada. ²Division of Neurosurgery, Department of Surgery, University of Toronto, Toronto, Ontario, Canada. ³Group of CNS Angiogenesis and Neurovascular Link, Neuroscience Center Zurich, University of Zurich and University Hospital Zurich, Zurich, Switzerland. ⁴Division of Neurosurgery, University Hospital Zurich, Zurich, Switzerland. ⁵Laboratory of Exercise and Health, Institute of Exercise and Health, Department of Health Sciences and Technology; Swiss Federal Institute of Technology (ETH Zurich), Zurich, Switzerland. ⁶Institute for Regenerative Medicine, University of Zurich, Zurich, Switzerland. ⁷Department of Neurosurgery, Kyoto University Graduate School of Medicine, Kyoto, Japan. ⁸Division of Experimental and Translational Neuroscience, Krembil Brain Institute, Krembil Research Institute, Toronto Western Hospital, University Health Network, University of Toronto, Toronto, Ontario, Canada. ⁹Department of Physiology, Faculty of Medicine, University of Toronto, Toronto, Ontario, Canada. ¹⁰Functional Genomics Center Zurich, ETH Zurich/University of Zurich, Zurich, Switzerland. ¹¹Department of Laboratory Medicine and Pathobiology, University of Toronto, Toronto, Ontario, Canada. ¹²Toronto General Hospital Research Institute, University Health Network, Toronto, Ontario, Canada. ¹³The Lunenfeld-Tanenbaum Research Institute, Mount Sinai Health System, Toronto, Ontario, Canada. ¹⁴Department of Molecular Genetics, University of Toronto, Toronto, Ontario, Canada. ¹⁵Department of Biomedicine, Aarhus University, Aarhus, Denmark. ¹⁶Department of Molecular Cell Biology, Weizmann Institute of Science, Rehovot, Israel. ¹⁷Department of Neurosurgery, University of Geneva Medical Center & Faculty of Medicine, University of Geneva, Geneva, Switzerland. ¹⁸The Donnelly Centre, University of Toronto, Toronto, Ontario, Canada. ¹⁹Division of Neurosurgery, Sprott Department of Surgery, University of Toronto, Toronto, Ontario, Canada. ²⁰MacFeeters-Hamilton Centre for Neuro-Oncology Research, University Health Network, Toronto, Ontario, Canada. ²¹Laboratory of Angiogenesis and Vascular Metabolism, Center for Cancer Biology, VIB & Department of Oncology, KU Leuven, Leuven, Belgium. ²²State Key Laboratory of Ophthalmology, Zhongshan Ophthalmic Center, Sun Yat-sen University, Guangzhou, P. R. China. ²³Laboratory of Angiogenesis and Vascular Heterogeneity, Department of Biomedicine, Aarhus University, Aarhus, Denmark. ²⁴Krembil Brain Institute, Division of Clinical and Computational Neuroscience, Krembil Research Institute, Toronto Western Hospital, University Health Network, University of Toronto, Toronto, Ontario, Canada. ²⁵Institute of Biomaterials and Biomedical Engineering and Electrical and Computer Engineering, University of Toronto, Toronto, Ontario, Canada. ²⁶Institute of Medical Science Faculty of Medicine, University of Toronto, Toronto, Ontario, Canada. ²⁷Division of Neurosurgery, Arthur and Sonia Labatt Brain Tumor Research Center, Departments of Surgery and Molecular Genetics, Hospital for Sick Children, Toronto, Ontario, Canada. ²⁸Department of Pathology and Center for Cancer Research, Massachusetts General Hospital and Harvard Medical School, Boston, MA, USA. ²⁹Broad Institute of Harvard and MIT, Cambridge, MA, USA. ³⁰Princess Margaret Cancer Centre, University Health Network, Toronto, Ontario, Canada. ³¹Department of Neurosurgery, Memorial Sloan Kettering Cancer Center, New York, NY, USA. ³²Ontario Institute of Cancer Research, Toronto, Ontario, Canada. ³³Peter Munk Cardiac Centre, University Health Network, Toronto, Ontario, Canada. ³⁴Krembil Research Institute, Vision Division, Krembil Discovery Tower, Toronto, Ontario, Canada. ³⁵Department of Ophthalmology and Vision Sciences, Faculty of Medicine, University of Toronto, Toronto, Ontario, Canada. ³⁶Department of Computer Science, University of Toronto, Toronto, Ontario, Canada. ³⁷These authors contributed equally: Thomas Wälchli, Moheb Ghobrial. ³⁸These authors jointly supervised this work: Thomas Wälchli, Gary D. Bader, Ivan Radovanovic. ✉e-mail: thomas.waelchli2@uzh.ch

## Reporting summary

Further information on research design is available in the Nature Portfolio Reporting Summary linked to this article.

## Data availability

All data are now accessible under Gene Expression Omnibus accession number GSE256493. An interactive website is available at https://waelchli-lab-human-brain-vasculature-atlas.ethz.ch, https://brain-vasc.cells.ucsc.edu and https://cellxgene.cziscience.com/collections/c95ca269-68a7-47c5-82db-da227f31b598 on which the data can be visualized and downloaded.

## Code availability

Most of our analyses are standard workflows and we have now deposited the source code at GitHub (https://github.com/Waelchli-lab/Single-cell-atlas-of-the-human-brain-vasculature-accross-development-adulthood-and-disease) to improve reproducibility our results. The generated Seurat objects of FACS-sorted (CD31$^+$CD45$^-$) ECs for the individual entities can be downloaded from https://doi.org/10.5281/zenodo.10058183, and for the overall merges from https://doi.org/10.5281/zenodo.10057779. The generated Seurat objects of unsorted ECs and PVCs for individual entities can be downloaded from https://doi.org/10.5281/zenodo.10058371, and for the overall merges from https://doi.org/10.5281/zenodo.10058563. The generated Monocle 3 CDS pseudotime objects of FACS-sorted (CD31$^+$CD45$^-$) ECs can be downloaded from https://doi.org/10.5281/zenodo.10058766. The generated diffusion map objects of FACS-sorted (CD31$^+$CD45$^-$) ECs can be downloaded from https://doi.org/10.5281/zenodo.10060876. The generated RNA velocity objects of FACS-sorted (CD31$^+$CD45$^-$) ECs of individual pathological entities can be downloaded from https://doi.org/10.5281/zenodo.10065659; the fetal and adult/control brains from https://doi.org/10.5281/zenodo.10066390; the overall merge of brain tumours from https://doi.org/10.5281/zenodo.10066538; and the overall merge of pathological entities from https://doi.org/10.5281/zenodo.10066703.

Acknowledgements We thank N. Krayenbühl, M. Germans, O. Bozinov, P. Bijlenga, P.-Y. Dietrich and V. Dutoit for help with the human adult tissue asservation; the donors and the staff at the RCWIH Biobank, the Lunenfeld-Tanenbaum Research Institute, the Mount Sinai Hospital/UHN Department of Obstetrics and Gynaecology, and Maximilian Niit for help with the human fetal specimen asservation, preparation and IHC staining (https://biobank.lunenfeld.ca); E. Speck and the members of the Flow Cytometry Facility, Krembil Discovery Tower, University Health Network for help with the FACS sorting; G. Basi, J. Cirlan, C. Dumrese and M. Kisielow for help with the scRNA-seq experiments; A. M. Sababi and M. M. Saad for help with the computational analysis; J. L. Gorman for help with the IMC experiments; the members of the University Health Network, Laboratory Medicine-Pathology Research Program and Melanie Peralta for help with adult specimen preparation and IHC staining; N. C. Ji for help with the illustrations; P.P.M. Thomson for help with English proofreading; A. Keller, F. Kern, T. Nowakowski, E. Winkler, M. Kellis, N. Sun, A. Regev, G. Eraslan, F. J. Theis, J. Shin, M. Prinz, R. Sankowski, R. Adams, S. Teichmann, L. Yang, A. Maria Cujba and N. Huang for their scientific inputs and advice regarding computational integration methods, the covariates age and sex, and overall discussion of our manuscript and figures. We acknowledge the following financial support for the research, and/or publication of this Article: T.W. was supported by the OPO Foundation, the Swiss Cancer Research foundation (KFS-3880-02-2016-R, KFS-4758-02-2019-R), the Stiftung zur Krebsbekämpfung, the Kurt und Senta Herrmann Foundation, Forschungskredit of the University of Zurich, the Zurich Cancer League, the Theodor und Ida Herzog Egli Foundation, the Novartis Foundation for Medical-Biological Research and the HOPE Foundation; I.R. by the Canadian Institutes of Health Research (funding reference number 155922) and the Irwin & Mariel Michael and Family through the University Health Network Foundation; I.R. and T.W. by the Gill Family Charitable Trust through the University Health Network Foundation; P.P.M. by the Canadian Institutes of Health Research; G.D.B. by NRNB (US National Institute of Health, National Center for Research Resources grant number P41GM103504). IMC work was supported by an NSERC Discovery grant (RGPIN-2021-03404) and an Early Career Investigator Award from Ontario Institute for Cancer Research (IA-1-020) and the Canada Research Chairs program to H.W.J. This publication is part of the Human Cell Atlas, www.humancellatlas.org/publications.

Author contributions T.W. had the idea for the study. T.W. and I.R. conceived the study. T.W. designed the experiments, wrote the manuscript, analysed the data, designed the figures and made the figures with M.G. with the help of M.S.; I.R., T.W., L.R., K.S., M.B., P.K., G.Z. and T.V. acquired the tissue. M.G., M.S., S.S. and S.V. performed the cell isolation experiments. T.W. and K.F. developed the initial isolation experiments. M.G. (majority of the analysis), M.S., H.Z., D.R.G. and H.R. performed the single-cell data processing and analysed the data with T.W.; T.W. and G.D.B. supervised the data analysis and interpreted data. R.S. and T.K. helped with single-cell data processing. M.G., M.S. and G.D.B. developed the computational methods with the help of T.W.; S.T., S.S., R.W. and K.Y. performed the IF stainings and RNA scope experiments, and S.T., R.W., K.Y., M.G., M.S., J.E.F. and T.W. analysed and interpreted the data. M.S., R.W., J.E.F. and T.W. selected the final IF images. A.D. E.L.Y.C. and H.W.J. performed the IMC staining, and A.D., M.S., H.W.J. and T.W. analysed and interpreted the data. M.S., H.W.J. and T.W. selected the final IMC images. T.W., I.R., G.D.B. and P.P.M. acquired funding. T.W., P.P.M., I.R. and G.D.B. edited the final version of the manuscript. M.G., M.S. and J.B. helped edit the manuscript. P.P.M., K.D.B., J.E.F., M.L.S., P.B.D., P.C., V.T., G.Z., T.V., I.R. and G.D.B. provided critical inputs to the manuscript. T.W. supervised all of the research. All of the authors read and approved the final manuscript.

Competing interests The authors declare no competing interests.

Additional information

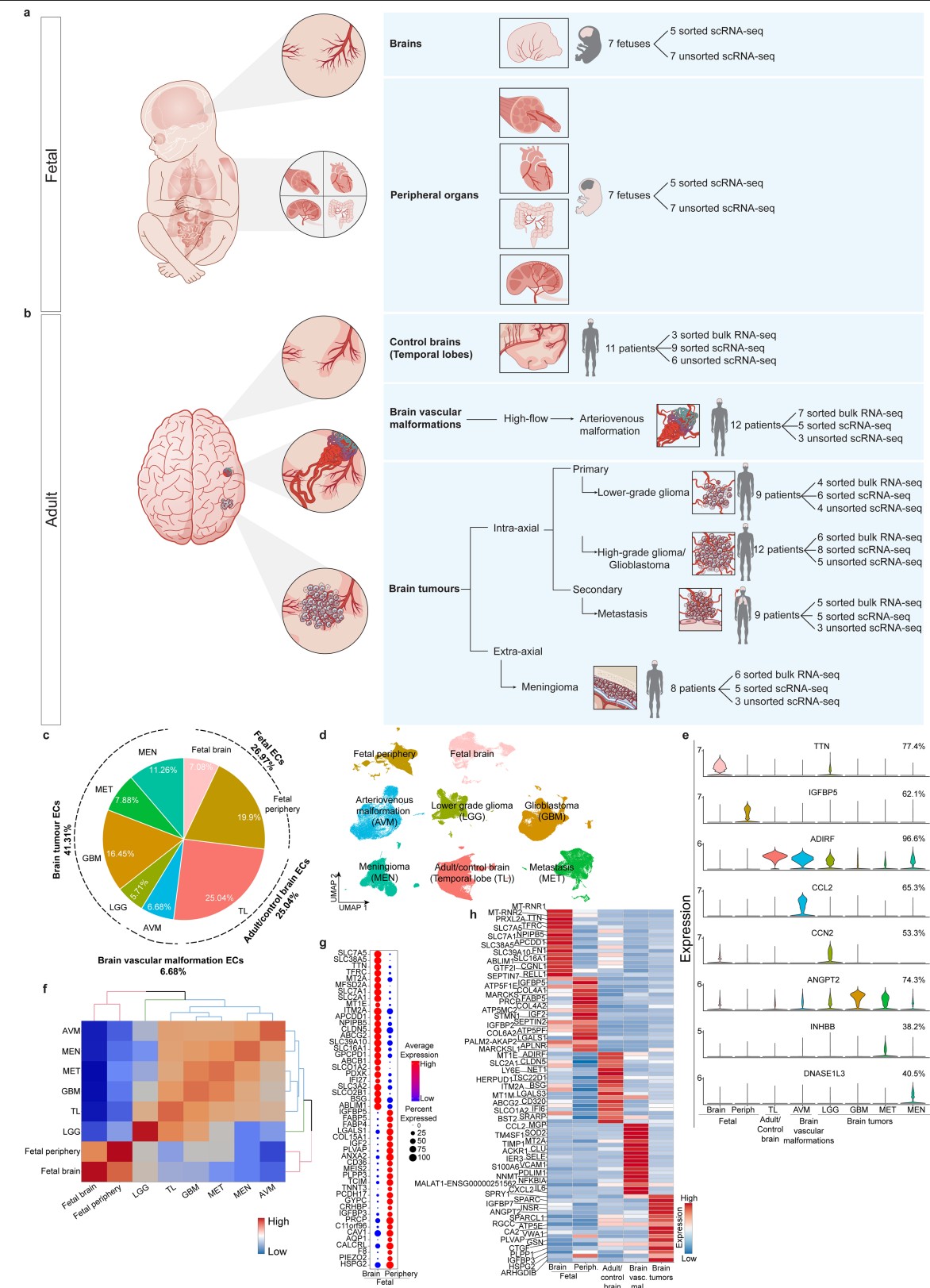

**Extended Data Fig. 1 | Construction of a molecular single-cell atlas of the human brain vasculature across development, adulthood and disease and of the fetal peripheral vasculature - studied entities and inter-tissue heterogeneity.** Figure shows studied entities and inter-tissue heterogeneity of sorted vascular endothelial cells. **a,b,** Scheme of the different tissue types present in the study with respective sample/patient numbers of fetal (**a**) and adult (**b**) origins. **c,** Piechart showing relative abundance and percentage of ECs from each tissue collected. **d,** Composite UMAP plots of sorted and in silico-quality checked (ECs, coloured by tissue of origin. **e,** Violin plots of the expression of the top marker genes of each tissue type, percentage of cells expressing the marker gene is indicated on the right. **f,** Endothelial cells transcriptome correlation heatmap and hierarchical clustering of all tissues analysed. **g,** Dotplot heatmap of the fetal brain vs fetal periphery endothelial signature. **h,** Expression heatmap of top ranking marker genes in the indicated tissues. Colour scale: red, high expression; white, intermediate expression; blue, low expression.

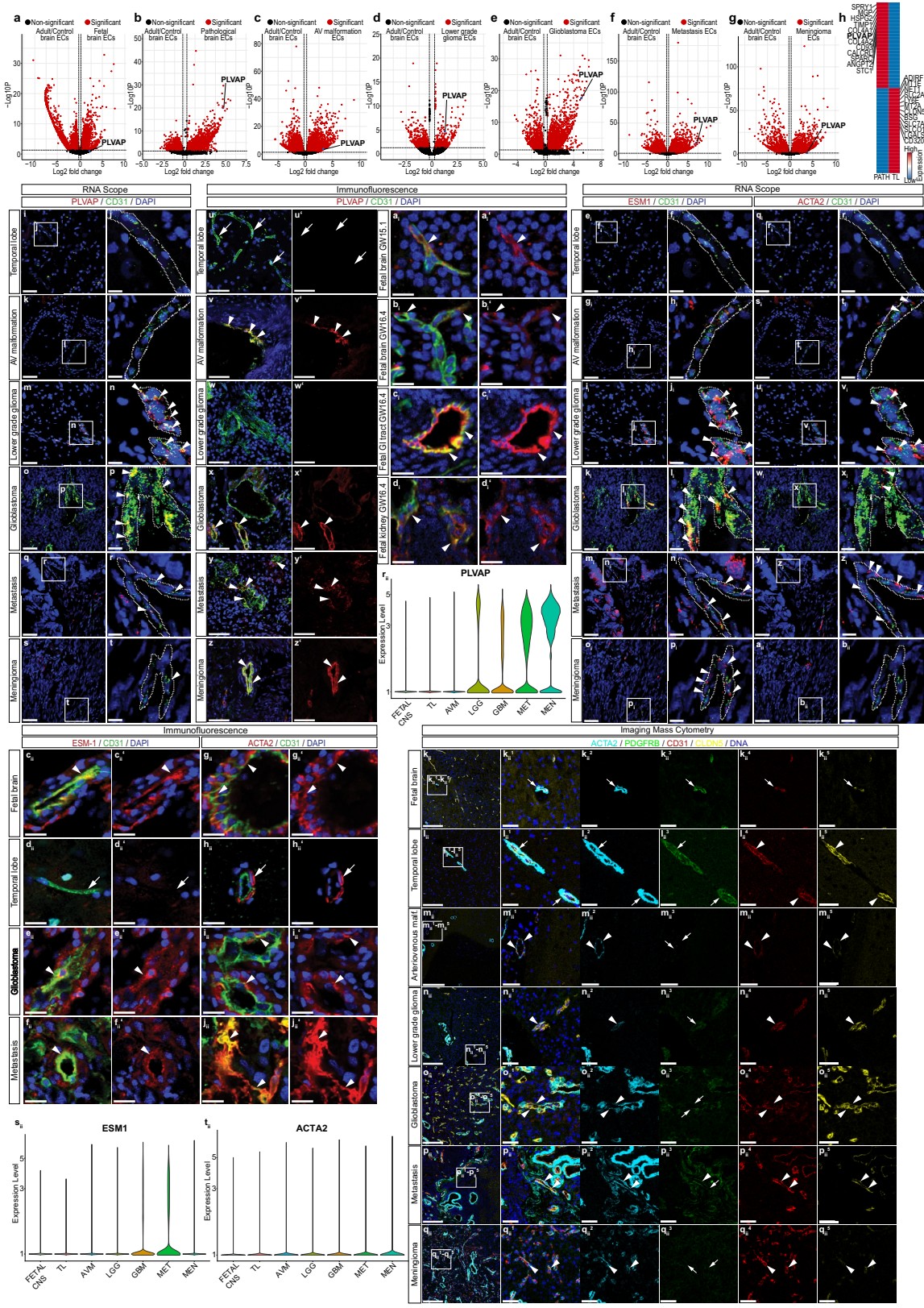

**Extended Data Fig. 2** | See next page for caption.

**Extended Data Fig. 2 | Validation of angiogenic and EndoMT markers in vascular ECs of the fetal, adult and pathological brain vasculature using RNAscope, IF and IMC. a-g**, Volcano plots showing the differential expression analysis comparing endothelial cells from adult/control brains (left) and the indicated entity (right) (Wald test, Benjamini Hochberg correction; *P*-value < 0.05 and log2FC ≥ 0.25 coloured significant in red, PLVAP dot is coloured blue). Number of individuals analysed is as follows: Fetal brain=5, Adult/control brain (TL) = 9, AV malformations=5, AVM = 5, LGG = 6, GBM = 8, MET = 5, MEN = 5. **h**, Expression heatmap of the top 25 differentially expressed genes in adult/control ECs (TL) vs. pathological brain ECs (PATH). **i-d$_i$'**, Immunofluorescence (IF) and RNAscope imaging of tissue sections from the indicated entities, stained for PLVAP (red), (RNAscope: **i-t**; IF: **u-d$_i$'**) and CD31 (green). Nuclei are stained with DAPI (blue). Arrowheads indicate ECs expressing PLVAP and dotted lines (RNAscope) delineate vascular structures in the different tissues. Scale bars: 50 µm in overviews (RNAscope); 50 µm in (IF) and 12.5 µm in zooms (RNAscope). **a$_i$-d$_i$'**, IF of fetal brain tissue sections from the indicated gestational ages, stained for PLVAP (red) and CD31 (green). Nuclei are stained with DAPI (blue). Arrowheads indicate ECs expressing PLVAP. Arrows indicate absence of expression in ECs. Scale bars: 20 µm. **e$_i$-b$_{ii}$**, RNAscope imaging of tissue sections from the indicated entities, stained for ESM1 (red; **e$_i$-p$_i$**), ACTA2

(red; **q$_i$-b$_{ii}$**) and CD31 (green). Nuclei are stained with DAPI (blue). Boxed area is magnified on the right; arrowheads indicate ECs expressing ESM1 or ACTA2 and dotted lines delineate vascular structures in the different tissues. Scale bars: 50 µm in overviews and 12.5 µm in zooms. **c$_{ii}$-j$_{ii}$'**, IF of tissue sections from the indicated entities, stained for ESM1 (red; **c$_{ii}$-f$_{ii}$'**), ACTA2 (red; **g$_{ii}$-j$_{ii}$'**) and CD31 (green). Nuclei are stained with DAPI (blue). Boxed area is magnified on the right; arrowheads indicate ECs expressing ESM1 or ACTA2. Arrows indicate absence of expression in ECs. Scale bars: 80 µm in overviews and 20 µm in zooms. **k$_{ii}$-q'⁴**, Mass cytometry (IMC) imaging of the indicated fetal, adult/control and pathological brain tissue samples visualizing metal-conjugated ACTA2, PDGFRB, CD31 and CLDN5 primary antibodies out of all 39 stained panel (Supplementary Table 24). Overlay of pseudocolor images as well as individual channels of the different markers (white, overlap; cyan, ACTA2; green, PDGFRB; red, CD31; yellow, CLDN5; blue, DNA intercalator), Scale bars: 250 µm in overviews, 50 µm in zooms. Arrowheads identify blood vessels ECs expressing the indicated markers in the different tissues. Arrows identify blood vessel ECs not expressing the indicated markers in the different tissues. **r$_{ii}$-t$_{ii}$**, Violin plots showing the expression of PLVAP (**r$_{ii}$**), ESM1 (**s$_{ii}$**) and ACTA2 (**t$_{ii}$**) in the indicated fetal brain (CNS), adult/control and pathological brain tissues.

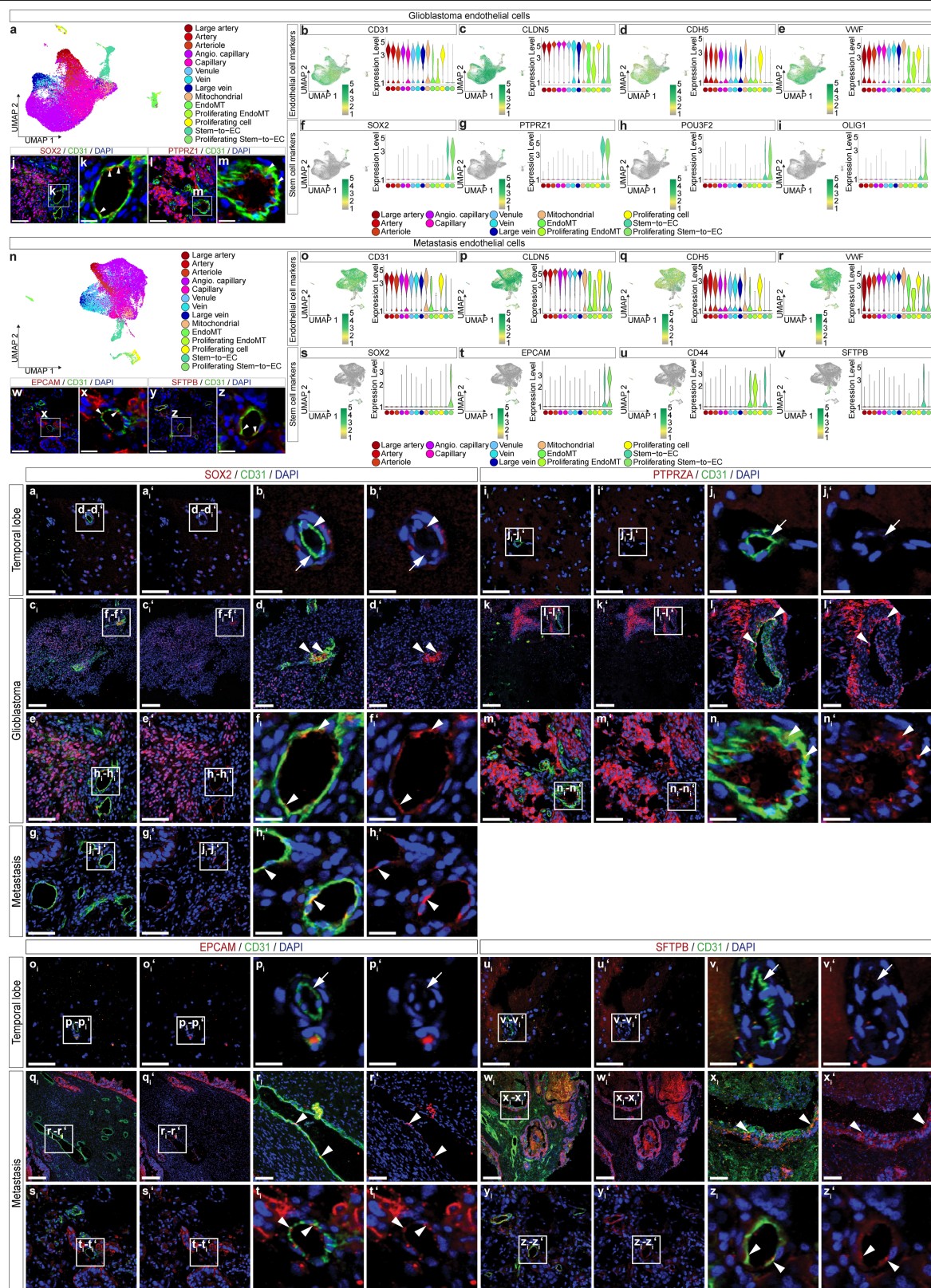

**Extended Data Fig. 3 | Validation of stem-to-endothelial transdifferentiation in vascular ECs of glioblastoma and of lung cancer brain metastasis using IF.** **a,n**, UMAP plot of glioblastoma (49,999 ECs from 8 individuals) (GBM) and lung cancer brain metastasis (23,962 ECs from 5 individuals) (MET) endothelial cells, coloured by AV specification. **b-i**, **o-v**, Violin plots showing the expression of the indicated endothelial- and stem cell specific markers in the different EC subtypes (AV clusters). **j-m**, **$a_i$-$z_i$′**, Immunofluorescence staining of *SOX2* (**j,k**, **$a_i$-$h_i$′**), *PTPRZ1* (**l,m**, **$i_i$-$n_i$′**), *EPCAM* (**w,x**, **$o_i$-$t_i$′**), *SFTPB* (**y,z**, **$u_i$-$z_i$′**) (red) and *CD31* (green) in the indicated tissue samples. Arrowheads identify blood vessels ECs expressing the indicated markers in the different tissues. Arrows identify blood vessel ECs not expressing the indicated markers in the different tissues. Boxed area is magnified on the right. Scale bars: 80 µm in overviews except for **$c_i$-$c_i$′**,**$k_i$-$k_i$′**, **$q_i$-$q_i$′**, **$w_i$-$w_i$′** (200 µm), 20 µm in zooms except for **$d_i$-$d_i$′**, **$l_i$-$l_i$′**, **$r_i$-$r_i$′**, **$x_i$-$x_i$′** (50 µm).

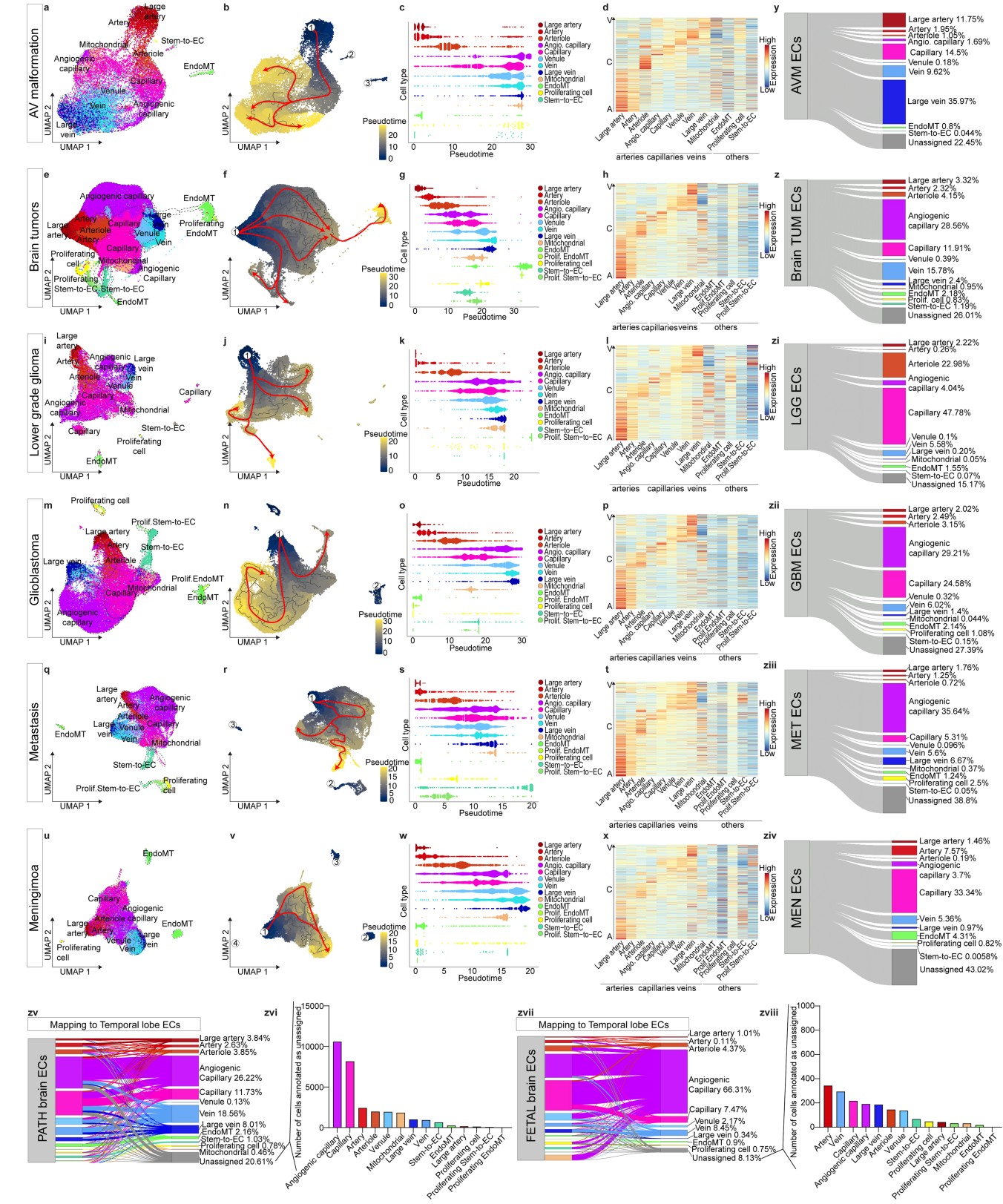

**Extended Data Fig. 4** | See next page for caption.

**Extended Data Fig. 4 | Alteration of AV-specification in pathological brain vascular ECs.** UMAP plots of human brain ECs isolated from the brain arteriovenous malformations (20,305 ECs from 5 individuals) (**a**,**b**), all brain tumours (125,579 ECs from 24 individuals) (**e**,**f**), lower-grade glioma (17,373 ECs from 6 individuals) (**i**,**j**) glioblastoma (49,999 ECs from 8 individuals) (**m**,**n**), brain metastasis (23,962 ECs from 5 individuals) (**q**,**r**), and meningioma (34,245 ECs from 5 individuals) (**u**,**v**) coloured by AV specification (**a**,**e**,**i**,**m**,**q**,**u**) and by pseudotime (**b**,**f**,**j**,**n**,**r**,**v**), the red line drawn manually indicates the major trajectory flow. **c**,**g**,**k**,**o**,**s**,**w**, Pseudotime order of ECs colour-coded according to AV specification from all the indicated entities. **d**,**h**,**l**,**p**,**t**,**x**, Heatmap of adult/control brain ECs AV specification signature gene expression in human brain ECs isolated from the indicated entities. **y-ziv**, Sankey plot showing the predicted annotation of the ECs of the indicated entities as mapped to adult/control brain (TL) ECs. Unassigned cells are indicated in grey. **zv**, Sankey plot of pathological brain ECs, showing the predicted annotation (right nodes) – as mapped to adult/control brain (TL) ECs – of the AV clusters assigned based on top cluster marker analysis (left nodes). Unassigned cells are indicated in grey. **zvii**, Sankey plot of fetal brain ECs, showing the predicted annotation (right nodes) – as mapped to adult/control brain (TL) ECs – of the AV clusters assigned based on top cluster marker analysis (left nodes). Unassigned cells are indicated in grey. **zvi**,**zviii**, Barplots showing the composition of unassigned cells – based on mapping to adult/control brain (TL) ECs – in pathological brain ECs (**zv**) and fetal brain ECs (**zvii**).

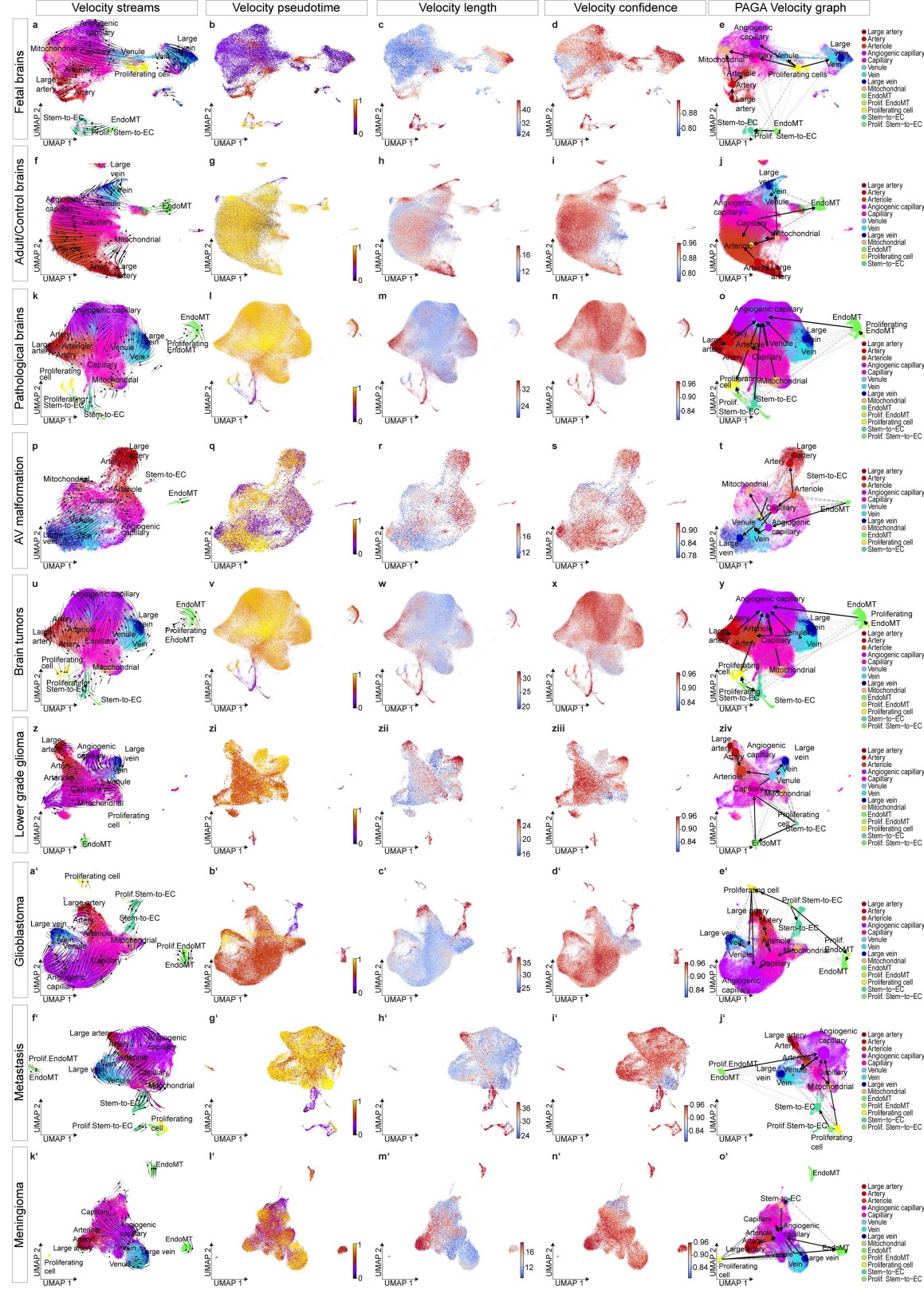

**Extended Data Fig. 5** | See next page for caption.

**Extended Data Fig. 5 | Trajectory analysis of fetal, adult and pathological human brain vascular ECs via RNA velocity along the AV-zonation/ specification axis. a**,**f**,**k**,**p**,**u**,**z**,**a'**,**f'**,**k'**, UMAP plots of human brain ECs isolated from the indicated entities (fetal brains (21,512 ECs from 5 individuals) (**a**), adult/ control brains (76,125 ECs from 9 individuals) (**f**), pathological brains (145,884 ECs from 29 individuals) (**k**), brain arteriovenous malformations (20,305 ECs from 5 individuals) (**p**), all brain tumours (125,579 ECs from 24 individuals) (**u**), lower-grade glioma (17,373 ECs from 6 individuals) (**z**) glioblastoma (49,999 ECs from 8 individuals) (**a'**), brain metastasis (23,962 ECs from 5 individuals) (**f'**), and meningioma (34,245 ECs from 5 individuals) (**k'**)) coloured by AV specification, onto the UMAPs are superimposed RNA velocity streamlines extended by velocity-inferred directionality. **b**,**g**,**l**,**q**,**v**,**zi**,**b'**,**g'**,**l'**, UMAP plots of the indicated entities, coloured by RNA velocity pseudotime. **c**,**h**,**m**,**r**,**w**,**zii**,**c'**, **h'**,**m'**, UMAP plots of the indicated entities, coloured by RNA velocity length. **d**,**i**,**n**,**s**,**x**,**ziii**,**d'**,**i'**,**n'**, UMAP plots of the indicated entities, coloured by RNA velocity confidence. **e**,**j**,**o**,**t**,**y**,**ziv**,**e'**,**j'**,**o'**, UMAP plots of the indicated entities, coloured by AV specification, onto the UMAPs are superimposed RNA velocity PAGA vectors extended by velocity-inferred directionality.

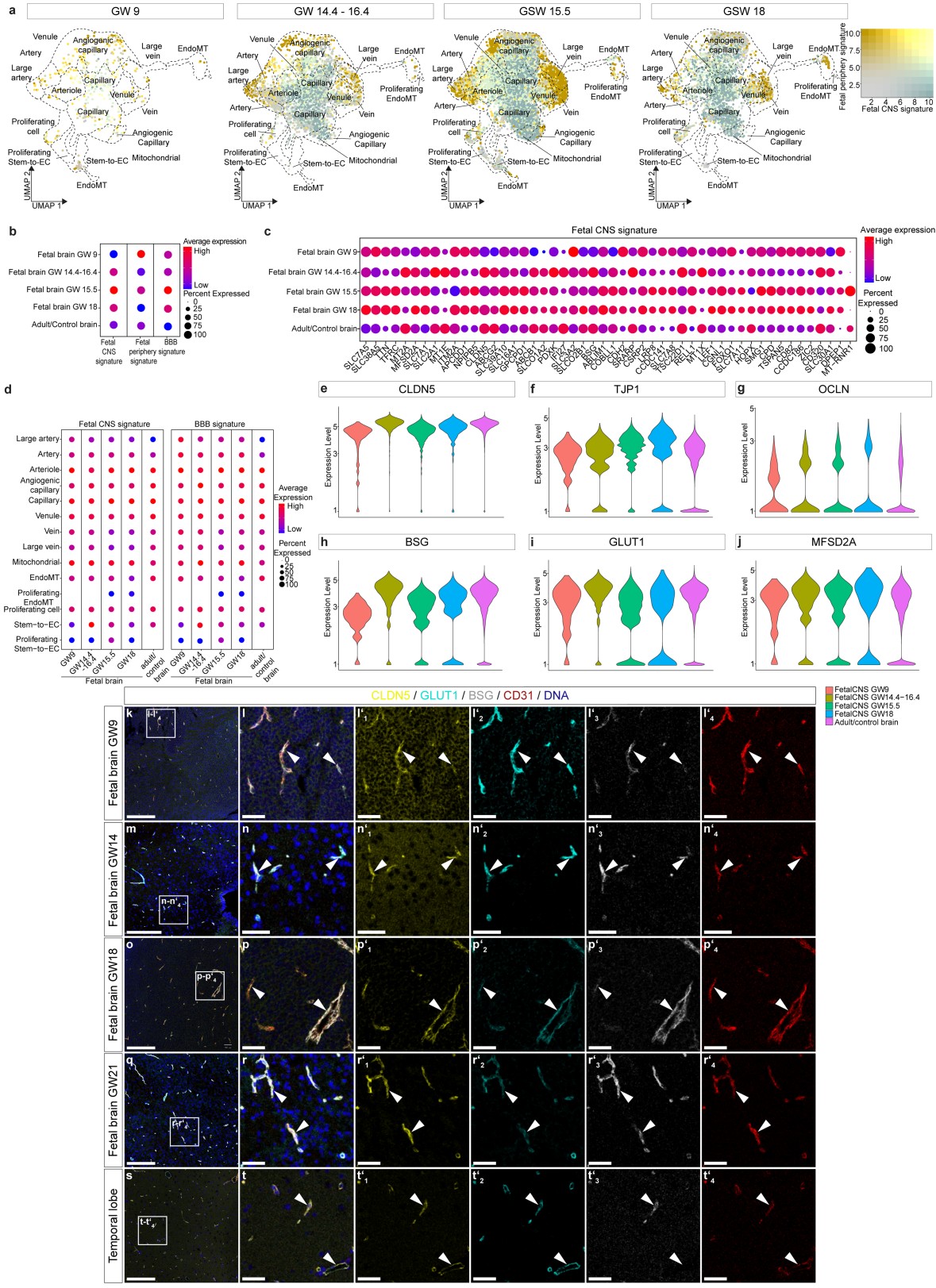

**Extended Data Fig. 6** | See next page for caption.

**Extended Data Fig. 6 | Establishment of CNS, peripheral and BBB signatures in vascular ECs from the developing fetal to the adult brain vasculature. a**, UMAP plots of the ECs from fetal brains in ascending fetal age ranging from gestational week 9 (444 ECs), gestational week 14.4 and 16.4 (3762 ECs), gestational week 15.5 (18,314 ECs) to gestational week 18 (3601 ECs); plots are colour-coded for fetal CNS signature (green) and fetal peripheral signature (yellow). **b**, Dotplot heatmap of fetal CNS, fetal peripheral, and BBB signatures' expression in fetal brains of indicated fetal age and adult/control brains (temporal lobes). **c**, Dotplot heatmap of fetal CNS signature genes' expression in fetal brains of indicated fetal age and adult/control brains (temporal lobes). **d**, Dotplot heatmaps of fetal CNS and BBB signatures expression at the level of AV specification for the indicated entities. Colour scale: red, high expression; blue, low expression, whereas the dot size represents the percentage expression within the indicated entity. **e-j**, Violin plots showing the expression of selected CNS specific and BBB marker genes in fetal brains of indicated fetal age and adult/control brains (TL). Markers shown were further validated by IF and/or IMC. **k-t'$_4$** Mass cytometry (IMC) imaging of fetal brains of fetal age GW9 (**k-l'$_4$**), GW14 (**m-n'$_4$**), GW18 (**o-p'$_4$**), GW21 (**q-r'$_4$**) and adult/control brains (temporal lobe) (**s-t'$_4$**) tissue samples using metal-conjugated antibodies. Representative pseudocolor images of combined and individual channels of the different markers (white, overlap; yellow, CLDN5; cyan, GLUT1; grey, BSG; red, CD31; blue, DNA intercalator), Scale bars: 250 μm in overviews, 50 μm in zooms. Shown are 5 antibodies stained for (Supplementary Table 24). Arrowheads identify blood vessels ECs expressing the indicated markers in the different tissues. Arrows identify blood vessel ECs not expressing the indicated markers in the different tissues.

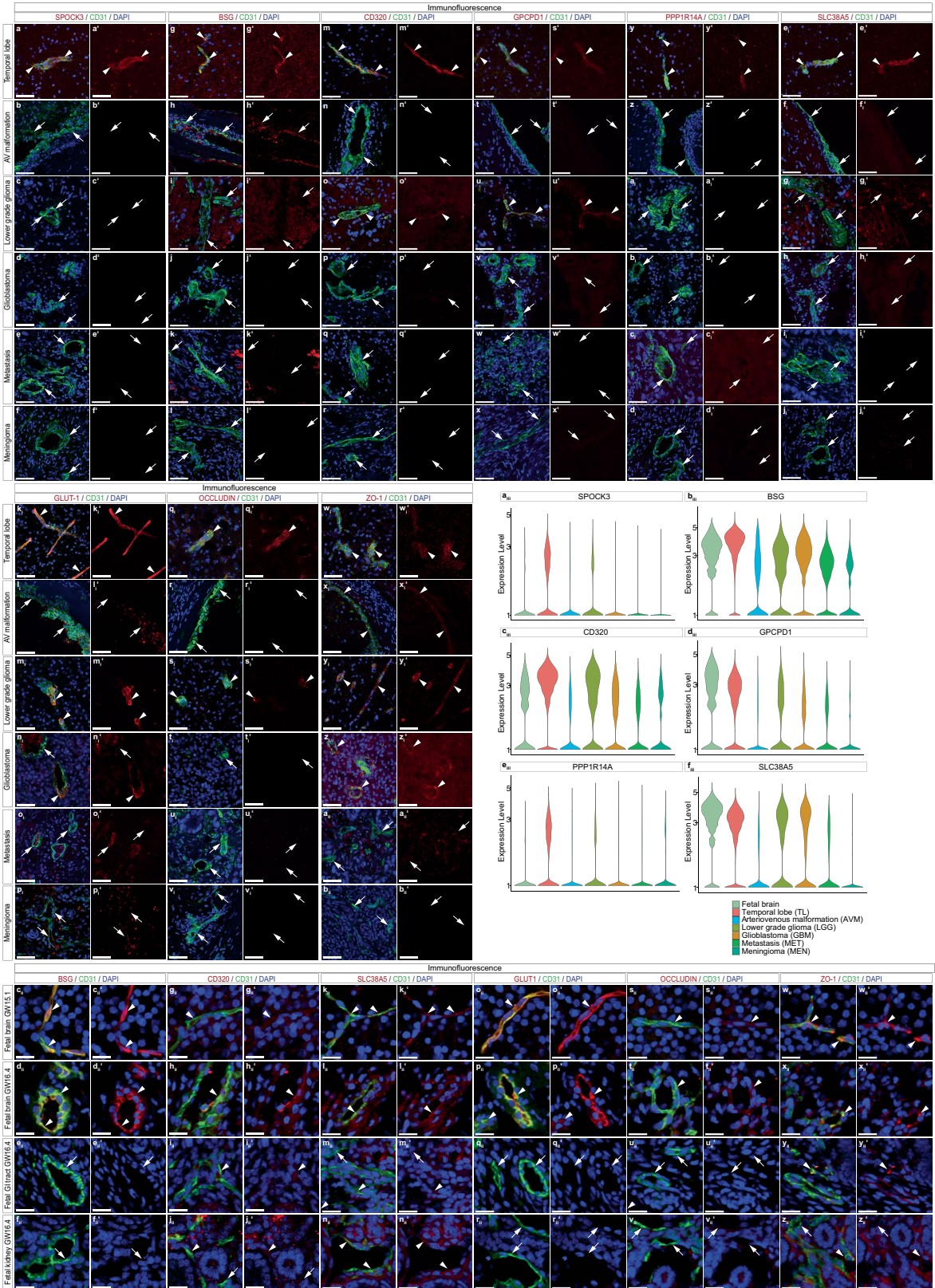

**Extended Data Fig. 7 | Validation of CNS signature markers in vascular ECs of the developing fetal, adult and pathological brain vasculature using IF. a-z$_{ii}$'**, Immunofluorescence (IF) imaging of tissue sections from the indicated entities, stained for *SPOCK3* (red; **a-f'**), *BSG* (red; **g-l'**, **c$_{ii}$-f$_{ii}$'**), *CD320* (red; **m-r'**, **g$_{ii}$-j$_{ii}$'**), *GPCPD1* (red; **s-x'**), *PPP1R141* (red; **y-d$_i$'**), *SLC38A5* (red; **e$_i$-j$_i$'**, **k$_{ii}$-n$_{ii}$'**), *GLUT1* (red; **k$_i$'-p$_i$'**, **o$_{ii}$-r$_{ii}$'**), *OCLN* (red; **q$_i$'-v$_i$'**, **s$_{ii}$-v$_{ii}$'**), *ZO-1* (red; **w$_i$-b$_{ii}$'**,

**w$_{ii}$-z$_{ii}$'**) and *CD31* (green). Nuclei are stained with DAPI (blue). Arrowheads identify blood vessel ECs expressing the indicated markers in the different tissues. Arrows identify blood vessel ECs not expressing the indicated markers in the different tissues. Scale bars: 200 µm in overviews, 50 µm in zooms. **a$_{iii}$-f$_{iii}$**, Violin plots showing the expression of CNS-specific genes in the indicated adult control and pathological brain tissues.

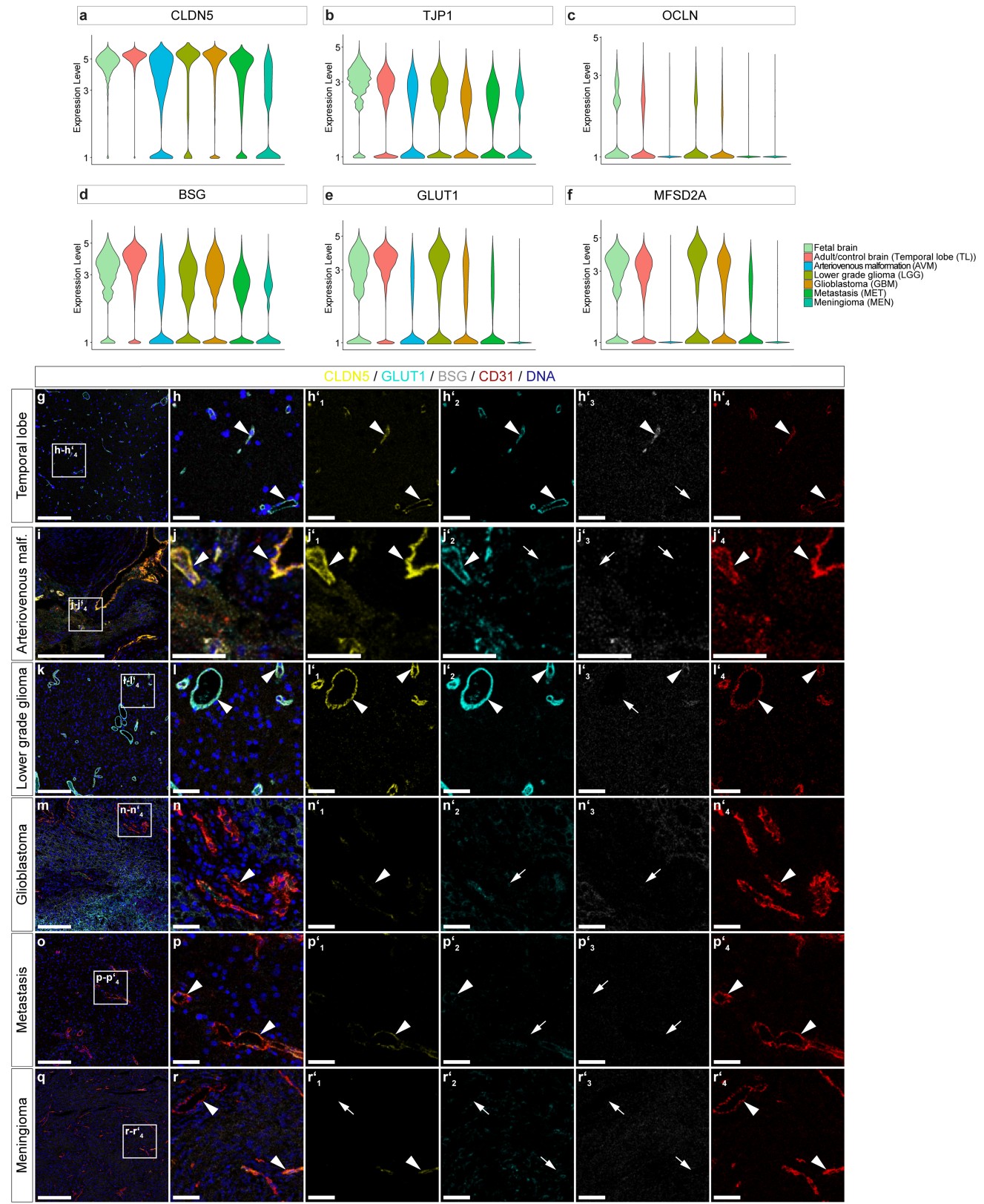

**Extended Data Fig. 8 | Validation of CNS signature markers in vascular ECs of the developing fetal, adult and pathological brain vasculature using IMC. a-f**, Violin plots showing the expression of CNS-specific and BBB marker genes in the indicated adult control and pathological brain tissues. Markers were further validated by IF and/or IMC. **g-r′₄** Mass cytometry (IMC) imaging of the indicated adult control and pathological brain tissue samples using metal-conjugated antibodies. Representative pseudocolour images of combined and individual channels of the different markers (white, overlap; yellow, CLDN5; cyan, GLUT1; grey, BSG; red, CD31; blue, DNA intercalator), Scale bars: 250 μm in overviews, 50 μm in zooms. Shown are 5 antibodies stained for (Supplementary Table 24). Arrowheads identify blood vessels ECs expressing the indicated markers in the different tissues. Arrows identify blood vessel ECs not expressing the indicated markers in the different tissues.

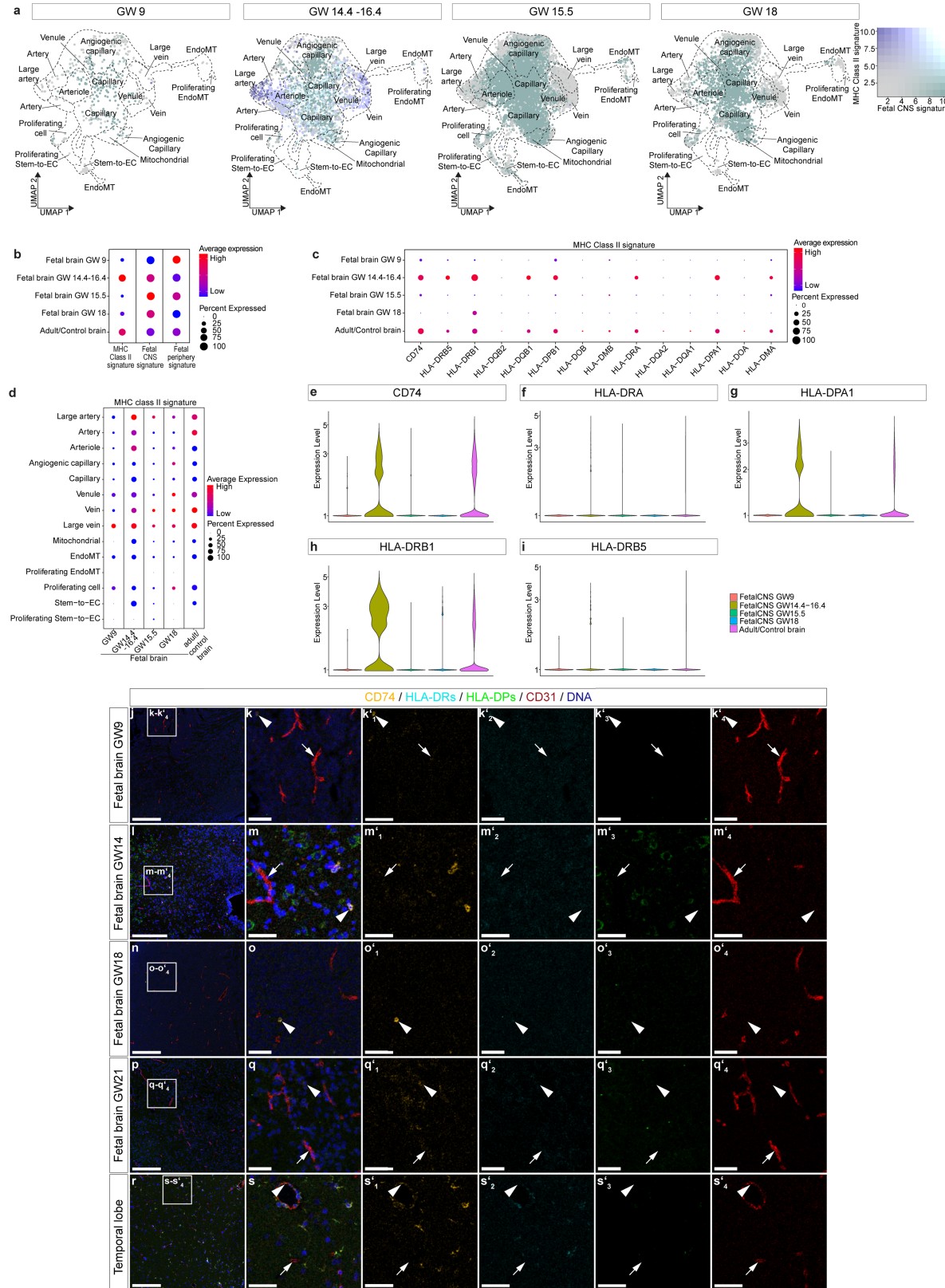

**Extended Data Fig. 9** | See next page for caption.

**Extended Data Fig. 9 | Establishment of MHC class II and CNS signatures in vascular ECs from the developing fetal to the adult brain vasculature.**
**a**, UMAP plots of the ECs from fetal brains in ascending fetal age ranging from gestational week 9 (444 ECs), gestational week 14.4 and 16.4 (3762 ECs), gestational week 15.5 (18,314 ECs) to gestational week 18 (3601 ECs); plots are colour-coded for fetal CNS signature (green) and MHC class II signature (blue).
**b**, Dotplot heatmap of MHC class II, fetal CNS and peripheral signatures expression in fetal brains of indicated fetal age and adult/control brains (temporal lobes). **c**, Dotplot heatmap of MHC class II signature genes' expression in fetal brains of indicated fetal age and adult/control brains (temporal lobes).
**d**, Dotplot heatmaps of MHC class II signature expression at the level of AV specification for the indicated entities. Colour scale: red, high expression; blue, low expression, whereas the dot size represents the percentage expression within the indicated entity. **e-i**, Violin plots showing the expression of MHC class II marker genes in fetal brains of indicated fetal age and adult/control brains. Markers shown were further validated by IF and/or IMC. **j-s'$_4$** Mass cytometry (IMC) imaging of fetal brains of fetal age GW9 (**j-k'$_4$**), GW14 (**l-m'$_4$**), GW18 (**n-o'$_4$**), GW21 (**p-q'$_4$**) and adult/control brain (**r-s'$_4$**) tissue samples using metal-conjugated primary and secondary antibodies. Representative pseudocolour images of combined and individual channels of the different markers (white, overlap; orange, CD74; cyan, pan-HLA-DR; green, oligo-HLA-D; red, CD31; blue, DNA intercalator), Scale bars: 250 µm in overviews, 50 µm in zooms. Shown are 5 antibodies stained for (Supplementary Table 24). Arrowheads identify blood vessels ECs expressing the indicated markers in the different tissues. Arrows identify blood vessel ECs not expressing the indicated markers in the different tissues.

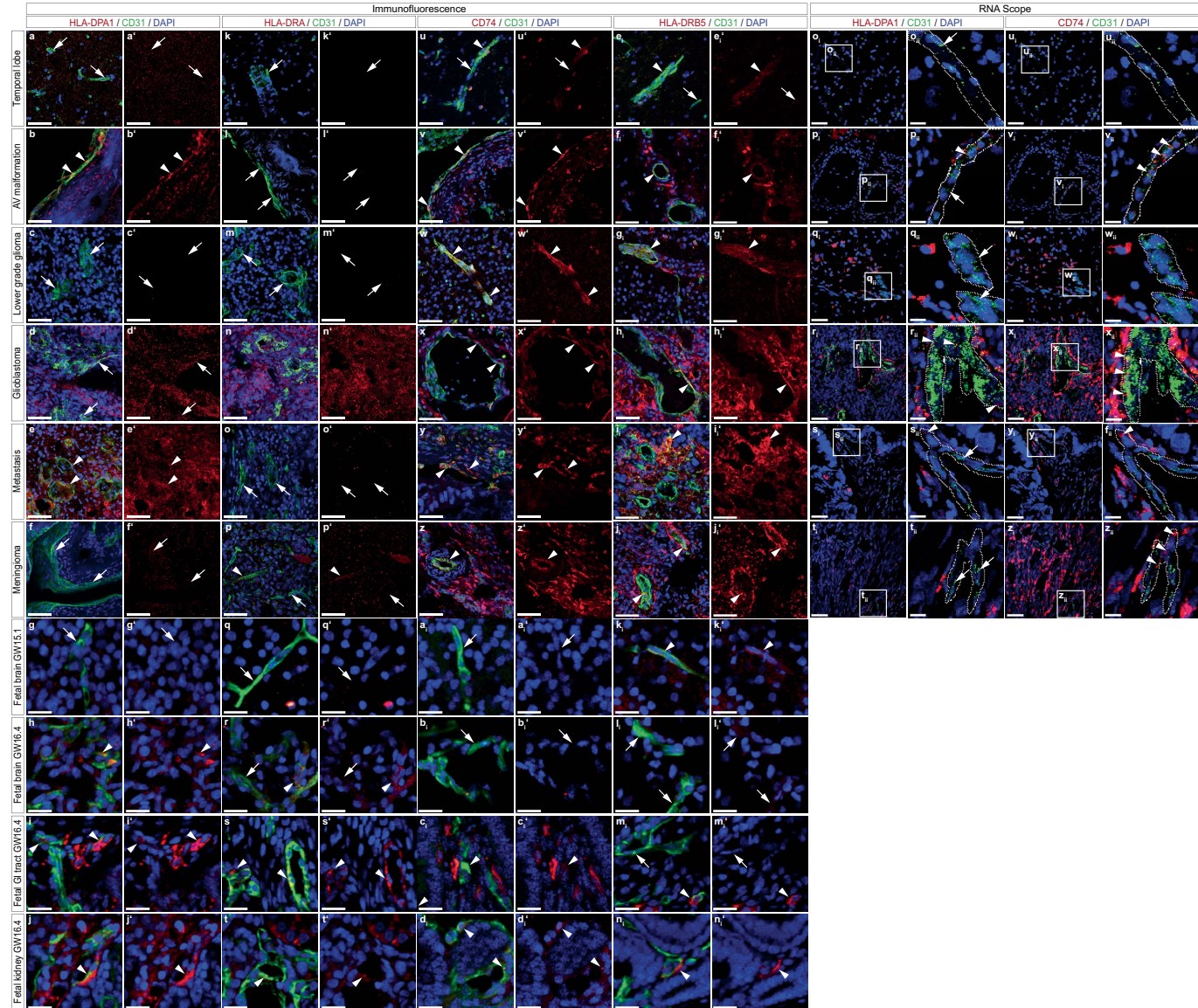

**Extended Data Fig. 10 | Validation of MHC class II signature markers in vascular ECs of the developing fetal brain and periphery vasculature, and of the adult and pathological brain vasculature using RNAscope and IF.** Immunofluorescence (IF) and RNAscope imaging of tissue sections from the indicated entities, stained for HLA-DPA1 (red; **a-j'**, IF; **o$_i$-t$_{ii}$**, RNAscope), HLA-DRA (red; **k-t'**, IF), CD74 (red; **u-d$_i$'**, IF; **u$_i$-z$_{ii}$**, RNAscope), HLA-DRB5 (red; **e$_i$-n$_i$'**, IF) and CD31 (green). Nuclei are stained with DAPI (blue). Arrowheads identify blood vessel ECs expressing the indicated markers, arrows identify blood vessel ECs not expressing the indicated markers in the different tissues and dotted lines (RNAscope) indicate vascular structures in the different tissues. Scale bars: 50 μm in (IF), 50 μm in overviews (RNAscope) and 12.5 μm in zooms (RNAscope).

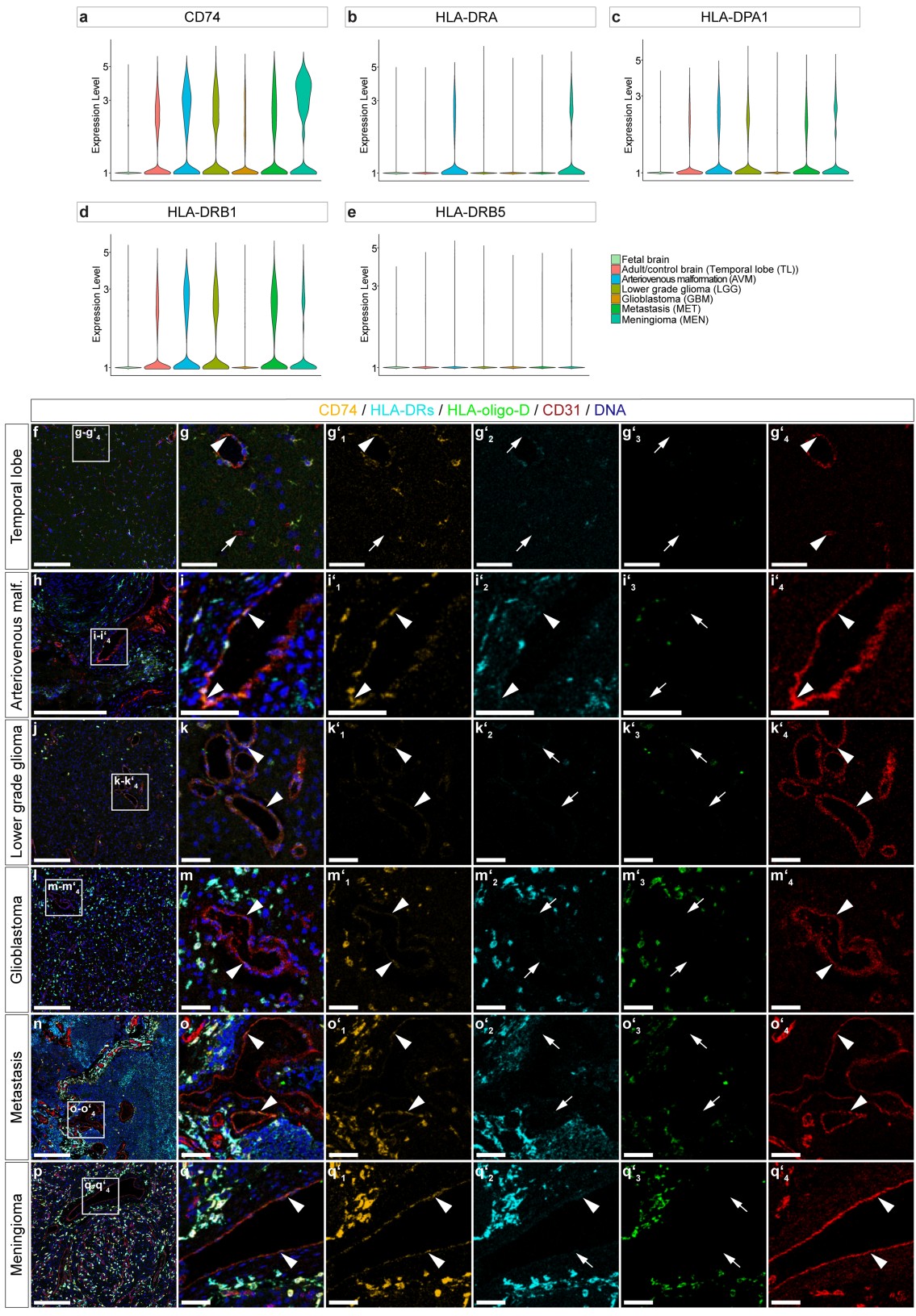

**Extended Data Fig. 11 | Validation of MHC class II signature markers in vascular ECs of the adult and pathological brain vasculature using IMC.** **a-e**, Violin plots showing the expression of MHC class II marker genes in the indicated fetal brain (CNS), adult/control and pathological brain tissues. Markers shown were further validated by IF and/or IMC. **f-q'₄** Mass cytometry (IMC) imaging of the indicated adult control and pathological brain tissue samples using metal-conjugated primary and secondary antibodies. Representative pseudocolor images of combined and individual channels of the different markers (white, overlap; orange, CD74; cyan, pan-HLA-DR; green, oligo-HLA-D; yellow, red, CD31), Scale bars: 250 µm in overviews, 50 µm in zooms. Shown are 5 antibodies stained for (Supplementary Table 24). Arrowheads identify blood vessels ECs in close contact with immune cell expressing the indicated markers in the different tissues. Arrows identify blood vessel ECs not in close contact with immune cell expressing the indicated markers.

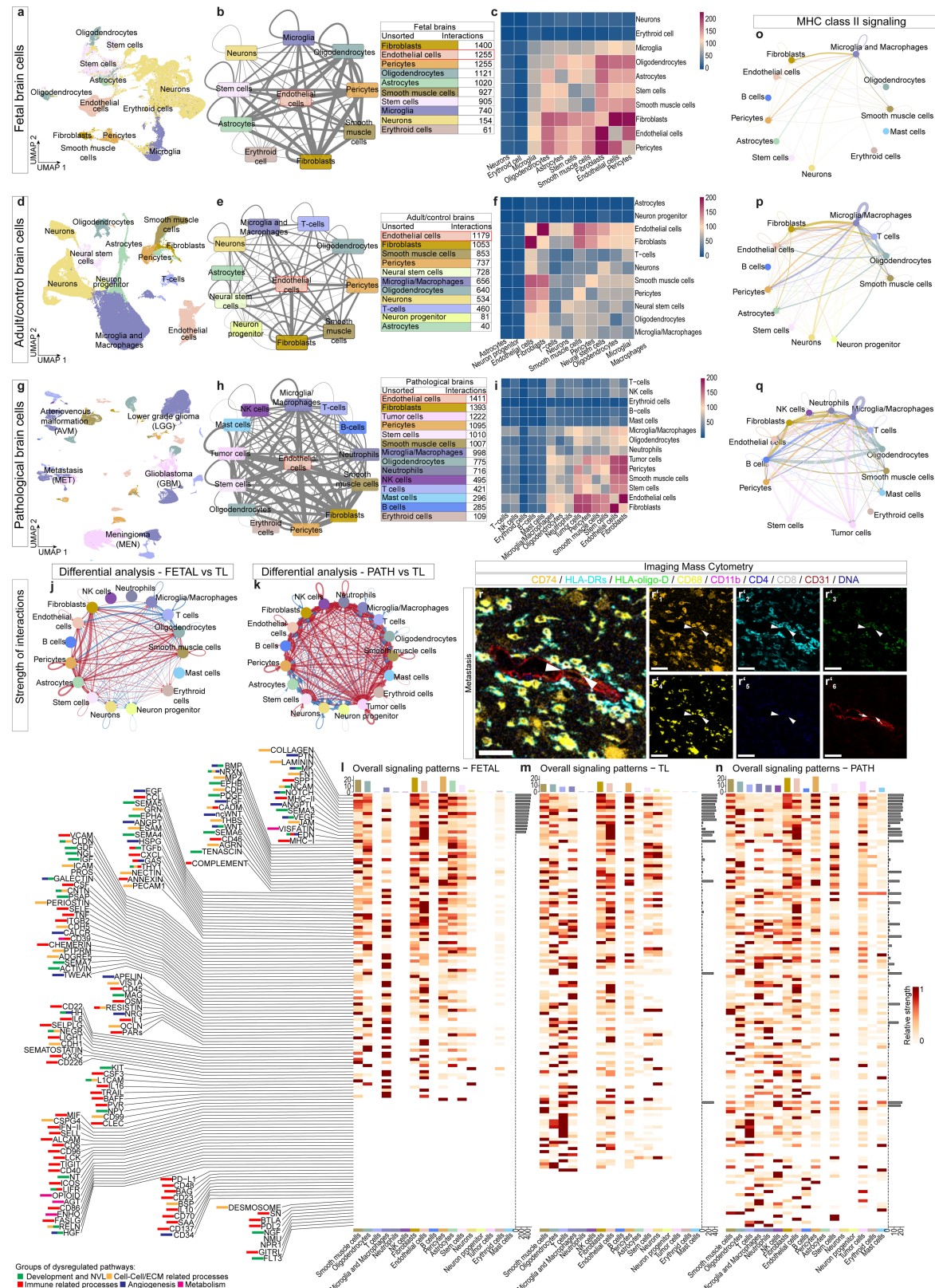

**Extended Data Fig. 12** | See next page for caption.

**Extended Data Fig. 12 | A key role for vascular ECs in the human brain neurovascular unit across development, adulthood and disease, highlighting MHC class II mediated interactions. a**,**d**,**g**, UMAP plots of endothelial and perivascular cells derived from fetal brains (40,557 cells from 7 individuals) (**a**), adult/control brains (80,515 cells from 6 individuals) (**d**) and pathological brains (Arteriovenous malformations: 12,013 cells from 3 individuals; Lower-grade glioma: 20,712 cells from 4 individuals; Glioblastoma: 22,297 cells from 5 individuals; Metastasis: 9,204 cells from 3 individuals; Meningioma: 26,916 cells from 3 individuals) (**g**). **b**,**e**,**h**, Ligand and receptor analysis of fetal brain (**b**), adult control brain (**e**), and brain pathology (**h**) cells using CellphoneDB. Line thickness indicates the number of interactions between cell types. Tables summarize the number of interactions for each cell type. **c**,**f**,**i**, Heatmaps showing the number of ligand-receptor interactions between the different cells of fetal brains (**c**), adult/control brains (**f**), and pathological brains (**i**) computed using CellphoneDB. **j**,**k**, Circle plot showing differential analysis of strength of ligand receptor interactions in fetal over adult/control brains (**j**) and pathology over adult/control brains (**k**), red indicating upregulation, while blue indicating downregulation. **l-n**, Heatmap showing overall signalling patterns of different cell types in adult control (**o**) and pathological (**p**) brains. **o**,**p**,**q**, Circle plots showing the strength of MHC class II signalling interactions between the different cell types of fetal brain (**o**), adult/control brain (**p**), and pathological brain (**q**). **r-r′₆** Mass cytometry (IMC) imaging of metastasis tissue sample visualizing metal-conjugated CD74, pan-HLA-DR, oligo-HLA-D, CD68, CD11b, CD4, CD8 and CD31 primary antibodies and metal-conjugated secondary stains, a subset of the antibody set (Supplementary Table 24) stains present in all images. Overlay of pseudocolor images as well as individual channels for CD74, pan-HLA-DR, oligo-HLA-D, CD68, CD11b and CD4, (white, overlap; orange, CD74; cyan, pan-HLA-DR; green, oligo-HLA-D; yellow, CD68; purple, CD11b; blue, CD4; grey, CD8; red, CD31), Scale bars = 50 μm. Shown are 8 antibodies stained for. Arrowheads identify blood vessel ECs in close contact with immune cell expressing the indicated markers in the different tissues. Arrows identify blood vessel ECs not in close contact with immune cell expressing the indicated markers.

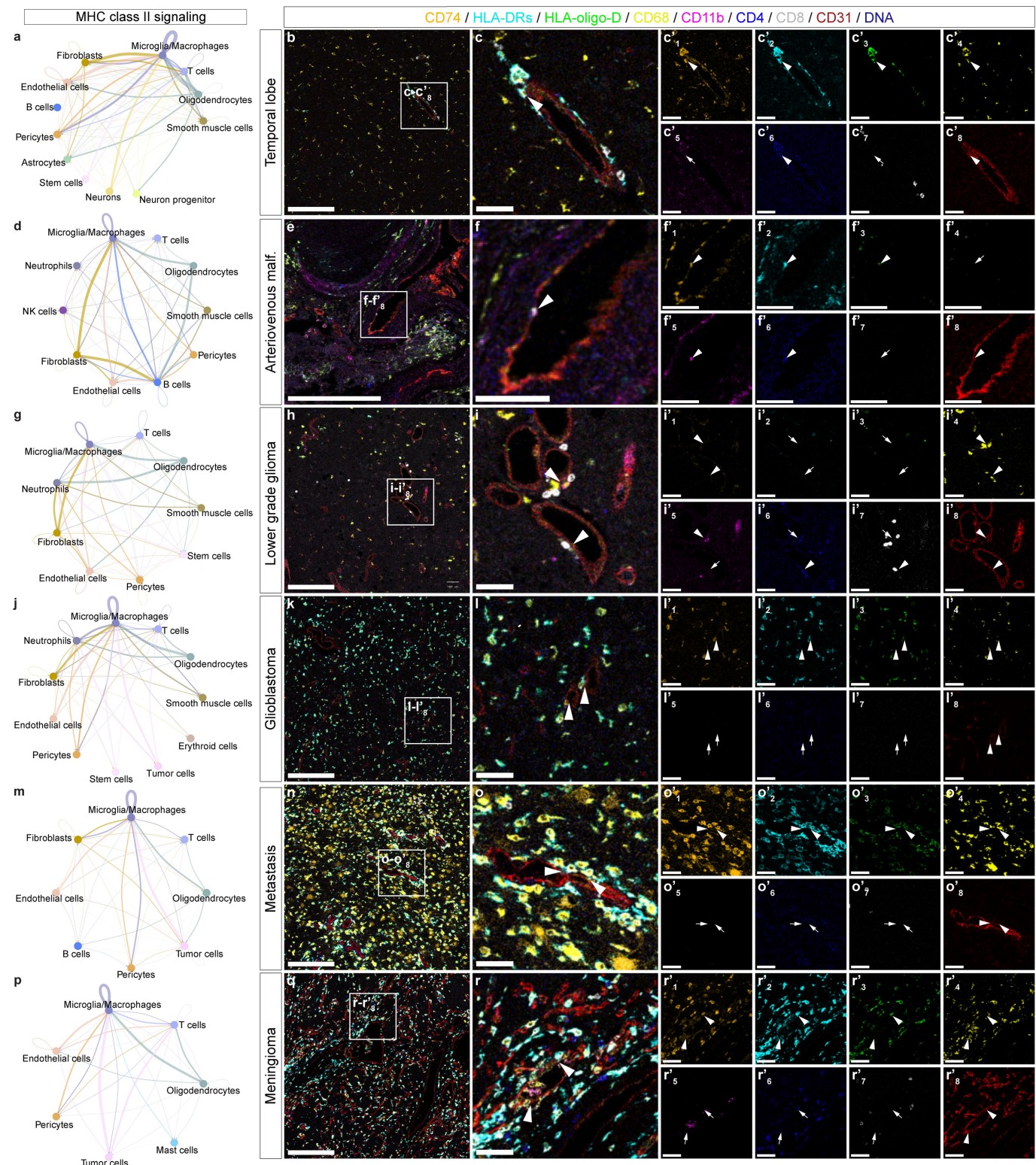

**Extended Data Fig. 13 | Validation of MHC class II mediated endothelial-immune cell interactions in the adult and pathological brain vasculature using IMC. a**,**d**,**g**,**j**,**m**,**p**, Circle plots showing the strength of MHC class II signalling interactions between the different cell types of adult/control brain (**a**) arteriovenous malformation (**d**), lower-grade glioma (**g**), glioblastoma (**j**), metastasis (**m**) and meningioma (**p**) cells. **b-c's**, **e-f's**, **h-i's**, **k-l's**, **n-o's**, **q-r's**, Mass cytometry (IMC) imaging of the indicated adult control and pathological brain tissue samples using metal-conjugated CD74, pan-HLA-DR, oligo-HLA-D, CD68, CD11b, CD4, CD8 and CD31 primary antibodies and metal-conjugated

secondary stains. Overlay of pseudocolor images as well as individual channels of the different markers (white, overlap; orange, CD74; cyan, pan-HLA-DR; green, oligo-HLA-D; yellow, CD68; purple, CD11b; blue, CD4; grey, CD8; red, CD31), Scale bars: 250 μm in overviews, 50 μm in zooms. Shown are 5 antibodies stained for (Supplementary Table 24). Arrowheads identify blood vessels ECs in close contact with immune cell expressing the indicated markers in the different tissues. Arrows identify blood vessel ECs not in close contact with immune cells expressing the indicated markers.

# Reporting Summary

## Statistics

For all statistical analyses, confirm that the following items are present in the figure legend, table legend, main text, or Methods section.

| n/a | Confirmed | |
|---|---|---|
| ☐ | ☒ | The exact sample size (*n*) for each experimental group/condition, given as a discrete number and unit of measurement |
| ☐ | ☒ | A statement on whether measurements were taken from distinct samples or whether the same sample was measured repeatedly |
| ☐ | ☒ | The statistical test(s) used AND whether they are one- or two-sided<br>*Only common tests should be described solely by name; describe more complex techniques in the Methods section.* |
| ☐ | ☒ | A description of all covariates tested |
| ☐ | ☒ | A description of any assumptions or corrections, such as tests of normality and adjustment for multiple comparisons |
| ☒ | ☐ | A full description of the statistical parameters including central tendency (e.g. means) or other basic estimates (e.g. regression coefficient) AND variation (e.g. standard deviation) or associated estimates of uncertainty (e.g. confidence intervals) |
| ☐ | ☒ | For null hypothesis testing, the test statistic (e.g. *F*, *t*, *r*) with confidence intervals, effect sizes, degrees of freedom and *P* value noted<br>*Give P values as exact values whenever suitable.* |
| ☒ | ☐ | For Bayesian analysis, information on the choice of priors and Markov chain Monte Carlo settings |
| ☒ | ☐ | For hierarchical and complex designs, identification of the appropriate level for tests and full reporting of outcomes |
| ☐ | ☒ | Estimates of effect sizes (e.g. Cohen's *d*, Pearson's *r*), indicating how they were calculated |

*Our web collection on statistics for biologists contains articles on many of the points above.*

## Software and code

Policy information about availability of computer code

| | |
|---|---|
| Data collection | FACS Aria III (BD Bioscience), Cellranger-5.0.0, Zeiss Zen 2.3 software |
| Data analysis | Single-cell RNA-seq libraries (chromium next GEM single cell 3' gene expression libraries) were obtained following the 10x Genomics recommended protocol, using the reagents included in the Chromium Single Cell v3 Reagent Kit. Quality control of cDNA and final libraries was done using 4200 TapeStation System (Agilent) and D5000 ScreenTape & Reagents. Libraries were sequenced on the NextSeq 500 (Illumina) instrument, aiming at 50k reads per cell. The 10x Genomics scRNA-seq data was processed using cellranger-5.0.0 with the Homo sapiens Gencode GRChm38.p13 genome Ensembl release. Based on filtered gene-cell count matrix by CellRanger's default cell calling algorithm, we performed the standard Seurat clustering (version 4.0.0 in R 4.2.2) workflow, as described below; raw expression values were normalized and log transformed (normalization.method = "LogNormalize"). In order to exclude low quality cells and doublets, cells with less than 500 or more than 3000 detected genes were filtered out. scDblFinder (v3.13) was used to validate that doublets were minimal. We also filtered cells with > 25% mitochondrial counts (Supplementary Tables 3,4). For integration/batch correction at the level of the overall merge of sorted endothelial cells of all entities Seurat's reciprocal PCA (RPCA) was applied as described in the "Methods" section. For integration/batch correction of sorted endothelial cells at the level of individual entities separately for every entity Seurat's reciprocal PCA (RPCA) was applied. For integration/batch correction of unsorted cells at the level of individual entities separately for every entity separately, Seurat's reciprocal PCA (RPCA) was applied. The integration/batch correction was not performed at the level of the overall merge of unsorted cells across all entities because we did not have biological questions that would required to do so. The detailed description of the integration/batch correction is present in the "Methods" section. In order to validate the integration results obtained by Seurat reciprocal PCA (RPCA), integration/batch correction was also performed with: 1) Harmony integration/batch correction(https://github.com/immunogenomics/harmony) (REF: https://www.nature.com/articles/s41592-019-0619-0), 2) Seurat's canonical correlation analysis (CCA) integration/batch correction, 3) scANVI (https://github.com/scverse/scvi-tools).<br>For cluster annotation, differential expression analysis for each cluster against all other clusters was computed using wilcoxauc function implemented in the github package presto (v1.0.0) (https://github.com/immunogenomics/presto) (FDR values were calculated using the Benjamini–Hochberg method). The resulting positive markers (that passed the threshold of p value < 0.05, and log2FC > 0.25) were used as |

March 2021

the top cluster markers for annotation.

Heatmap of the pairwise Jaccard distance between the 14 main EC clusters and 44 EC subclusters was computed using scclusteval package. BIOMEX (version 1.0.0) (https://carmelietlab.sites.vib.be/en/biomex) was using to perform the cluster similarity analysis (Figure 3m-p) as described in sections 33 and 37 of the package manual.

To predict the cell identity of the pathological brain endothelial cell clusters of the adult and fetal datasets as compared to the adult/control brain endothelial cells, the cell identity classification and label transfer was done using the standard Seurat workflow using the temporal lobe endothelial cells. as the reference dataset. Illustration of the results was generated using Seurat (v.4.0.0), Sankey plots were done using networkD3 (v.0.4).

For patient level analysis, DESeq2 package (v1.30.1) was used to perform pseudobulk differential expression analysis, volcano plots were done using EnhancedVolcano (v1.8.0) package and heatmaps were plotted using pheatmap (v1.0.12).

To statistically quantify the compositional changes between fetal, adult/control and pathological brain datasets, we used single-cell differential composition analysis (scCODA) package (version 0.1.7). Moreover, to validate scCODA findings while adjusting for covariates age and sex, we used: 1) tree-aggregated amplicon and single-cell compositional data analysis (tascCODA) (version 0.1.3) (https://github.com/bio-datascience/tascCODA), 2) Dirichlet regression was used via DirichletReg R package version 0.7.0 (https://github.com/maiermarco/DirichletReg), 3) The propeller method was used via speckle R package (0.99.1) (https://github.com/phipsonlab/speckle), 4) Single-Cell Interpretable Tensor Decomposition (scITD) (version 1.0.2) (https://github.com/kharchenkolab/scITD#walkthrough). As described in the methods section.

For bulk RNA-sequencing, the quality of the RNA and final libraries was determined using an Agilent 4200 TapeStation System. The libraries were pooled equimolarly and sequenced in an Illumina NovaSeq sequencer (single-end 100 bp) with a depth of around 20 Mio reads per sample. For mapping and trimming of FASTQ format sequences was performed using Trimmomatic v0.3.3, and sequence quality control was assessed using FastQC. Alignment to the Ensembl Homo_sapiens GRCh38.p10 reference genome (Release_91-2018-02-26) was performed using the STAR aligner. Gene expression values were computed with the function featureCounts from the R package Rsubread.

Sorted endothelial cells bulk RNA-seq deconvolution was performed using BayesPrism (version 2.0) (https://github.com/Danko-Lab/BayesPrism) and EPIC (version 1.1.5) (https://github.com/GfellerLab/EPIC) R packages.

Differential expression was computed using the wilcoxauc function implemented in the github package presto (https://github.com/immunogenomics/presto). FDR values were calculated using the Benjamini–Hochberg method. Pathway analysis was performed using the Gene Set Enrichment Analysis (GSEA) software from the Broad Institute (software.broadinstitute.org/GSEA) (version 4.0.1). A permutation-based P-value is computed and corrected for multiple testing to produce a permutation based Benjamini – Hochberg correction false-discovery rate q-value that ranges from 1 (not significant) to 0 (highly significant). The resulting pathways were ranked using NES and FDR q-value, P-values were reported in the GSEA output reports.

Human_GOBP_AllPathways_no_GO_iea_March_01_2021_symbol.gmt from [http://baderlab.org/GeneSets] was used to identify enriched pathways in GSEA analysis. Highly related pathways were grouped into a themes, labeled by AutoAnnotate (version 1.3) and plotted using Cytoscape (Version 3.7.0) and EnrichmentMap (version 3.3).

Pseudospace/pseudotime trajectory analysis was performed using Monocle 3 (version 1.0.0) (https://github.com/cole-trapnell-lab/monocle3) and Tools for Single Cell Analysis (TSCAN) (1.36.0) (https://github.com/zji90/TSCAN) in fetal, adult/control and pathological brain endothelial cells. Endothelial cells were clustered using the standard Seurat (version 4.0.0) clustering procedure and cluster markers were used to AV annotate those clusters, which were used as an input into Monocle to infer trajectory/lineage/arteriovenous relationships within endothelial cells. SeuratWrappers (v.0.3.0) was used to convert the Seurat objects to cell data set objects, while retaining the Seurat generated UMAP embeddings and cell clustering and then trajectory graph learning and pseudo-time measurement with Monocle3.

To further address trajectory inference, we performed RNA velocity and diffusion map analyses, which both address the pseudotime but not the pseudospace.  To compute RNA velocity of endothelial cells of the different entities, Velocyto package (version 0.17.17) (https://velocyto.org/velocyto.py/) was used on the CellRanger output BAM files and the genome annotation (.gtf file) from (https://support.10xgenomics.com/single-cell-gene-expression/software/pipelines/latest/advanced/references). For RNA velocity visualization, the generated loom file containing the spliced and unspliced RNA transcripts served as input into the scVelo package (version 0.2.4) (https://scvelo.readthedocs.io/en/stable/). Trajectory inference was also analyzed using destiny package (version 3.12.0) (https://bioconductor.org/packages/release/bioc/html/destiny.html), the method infers the low-dimensional manifold by estimating the eigenvalues and eigenvectors for the diffusion operator related to the data, In brief, the diffusion map was generated by applying the "DiffusionMap" function on the single cell experiment object.

Cell-cell (ligand receptor) interaction analysis between vascular cell types as well as endothelial and perivascular cells was performed using two published packages: CellPhoneDB (version 3.0.0) and Cellchat (version 1.6.1). First, using CellphoneDB ligand-receptor pairing matrix was constructed as follows; only ligands and receptors expressed in at least 10% of the cells in a particular cluster were considered, cluster labels were then permuted randomly 1,000 times to calculate the mean expression values of ligands and receptors, followed by pairwise comparisons between all cell types. The cut-off of expression was set to more that 0.1 and P-value to less than 0.05. The number of paired cell-cell interactions was based on the sum of the number of ligand-receptor interactions in each of the cell–cell pairs. Finally, Cytoscape was used to visualize the interaction network as a degree sorted circle layout.

Second, using CellChat (v.1.6.1) we followed the developers' suggested workflow, briefly applied the pre-processing functions identifyOverExpressedGenes, identifyOverExpressedInteractions, and projectData with standard parameters set. The CellChatDB including the Secreted Signaling pathways, ECM-receptor as well as Cell-Cell contact were analysed, in addition MHC class-II interactions reported in the CellphoneDB. Moreover, the gene expression data was projected onto experimentally validated protein-protein interaction. the standard package functions as computeCommunProb, computeCommunProbPathway, and aggregateNet were used with default parameters. Finally, to determine the ligand-receptor contributions and senders/receivers' roles in the network the functions netAnalysis_contribution and netAnalysis_signalingRole was applied on the netP data slot respectively.

We further compared cell–cell communication patterns by computing the Euclidean distance between ligand-receptor pairs of the shared signaling pathways (a measure of the difference between the signaling networks of datasets, see methods e.g. larger Euclidean distance implying larger difference of the communication networks between two datasets in terms of either functional or structure similarity, termed network architecture)57. We compared the information flow for each signaling pathway between, which is defined by the sum of communication probability among all pairs of cell groups for a given signaling pathway in the inferred network.

Software used include: R 4.2.2, Python 3.7, FV10-ASW 4.2 Viewer, Zeiss Zen 2.3 software, ImageJ 1.53e.

All data is accessible via the GEO accession number GSE186771 (will be made public as soon as upload is finished).

An interactive website is available at: Waelchli-lab-human-brain-vasculature-atlas.hest.ethz.ch

Our analyses are standard workflows that have been carried out using freely available software packages (detailed in the "Methods" section and also above), and we have now deposited the source code from the respective freely available software packages (indicated above and in the "Methods" section) that we used in:

 (https://github.com/Waelchli-lab/Single-cell-atlas-of-the-human-brain-vasculature-accross-development-adulthood-and-disease) to improve

reproducibility our results.

For manuscripts utilizing custom algorithms or software that are central to the research but not yet described in published literature, software must be made available to editors and reviewers. We strongly encourage code deposition in a community repository (e.g. GitHub). See the Nature Portfolio guidelines for submitting code & software for further information.

## Data

Policy information about availability of data

All manuscripts must include a data availability statement. This statement should provide the following information, where applicable:
- Accession codes, unique identifiers, or web links for publicly available datasets
- A description of any restrictions on data availability
- For clinical datasets or third party data, please ensure that the statement adheres to our policy

All data is accessible via the GEO accession number GSE256493 (access token: wzermaewdzernab). We have deposited our research data via the Nature Portfolio - figshare partnering on: https://doi.org/10.6084/m9.figshare.25151738

# Field-specific reporting

Please select the one below that is the best fit for your research. If you are not sure, read the appropriate sections before making your selection.

☒ Life sciences  ☐ Behavioural & social sciences  ☐ Ecological, evolutionary & environmental sciences

For a reference copy of the document with all sections, see nature.com/documents/nr-reporting-summary-flat.pdf

# Life sciences study design

All studies must disclose on these points even when the disclosure is negative.

| | |
|---|---|
| Sample size | We performed single cell RNAseq and bulk RNA-seq analysis on the number of fetuses/patients indicated in Supplementary Tables 1,2,3 and Extended data figure 1. We performed single-cell RNA sequencing of 606,380 freshly isolated endothelial, perivascular and other tissue-derived cells from 117 samples, from 68 human fetuses and adult patients to construct a molecular atlas of the developing fetal, adult control and diseased human brain vasculature. |
| Data exclusions | No data were excluded from the analysis. For the final count matrix, we excluded cells based on pre-established criteria for single-cells: we excluded low quality cells (i.e. - cells with low number of detected genes and high mitochondria content). In order to exclude low quality cells and doublets, cells with less than 500 or more than 3000 detected genes were filtered out. scDblFinder (v3.13) was used to validate that doublets were minimal. We also filtered cells with > 25% mitochondrial counts (Supplementary Tables 3,4). |
| Replication | We performed single cell RNAseq on the number of fetuses/patients indicated in Supplementary tables 1,2,3 and Extended data figure 1. The number of the different biological replicates are stated in the Supplementary Tables 1,2,3, and those attempts were fresh isolations and sequencing that are successful at the first and only attempt of the experiment. We only processed freshly operated and isolated samples. |
| Randomization | Different single cells were randomly captured before analysis. Human samples were not randomized due to practical constraints. |
| Blinding | We are blinded to analyzed cell types before single cell analyses. |

# Reporting for specific materials, systems and methods

We require information from authors about some types of materials, experimental systems and methods used in many studies. Here, indicate whether each material, system or method listed is relevant to your study. If you are not sure if a list item applies to your research, read the appropriate section before selecting a response.

## Materials & experimental systems

| n/a | Involved in the study |
|---|---|
| ☐ | ☒ Antibodies |
| ☒ | ☐ Eukaryotic cell lines |
| ☒ | ☐ Palaeontology and archaeology |
| ☒ | ☐ Animals and other organisms |
| ☐ | ☒ Human research participants |
| ☐ | ☒ Clinical data |
| ☒ | ☐ Dual use research of concern |

## Methods

| n/a | Involved in the study |
|---|---|
| ☒ | ☐ ChIP-seq |
| ☐ | ☒ Flow cytometry |
| ☒ | ☐ MRI-based neuroimaging |

# Antibodies

| Antibodies used | |
|---|---|
| | ZO1 Mouse Invitrogen 33 9100 |
| | Occludin Mouse Invitrogen 33 1500 |
| | GLUT1 Mouse Abcam ab40084 |
| | CNS-specificity markers |
| | SLC38A5 Rabbit Abcam ab72717 |
| | SPOCK3 Rabbit ThermoFischer PA531369 |
| | PPP1R14A Rabbit AVIVA Systems Biology OAAF01271 |
| | BSG (CD147) Mouse Abcam ab666 |
| | CD320 Rabbit Proteintech 10343-1-AP |
| | GPCPD1 Rabbit ThermoFischer PA5-65346 |
| | CD31 Guinea pig Synaptic Systems 351004 |
| | CD326(EPCAM) Mouse ThermoFisher 14-9326-82 |
| | SFTPB Rabbit Invitrogen PA5 42000 |
| | SOX2 Rabbit Abcam ab97959 |
| | PTPRZ1 Rabbit ThermoFisher PA5-53280 |
| | HLA DRB5 Rabbit Invitrogen PA5 60260 |
| | HLA DRA Rabbit Invitrogen PA5 27553 |
| | HLA DPA1 Rabbit Invitrogen PA5 28037 |
| | HLA DPA1 Rabbit Sigma HPA017967 |
| | CD74 Rabbit Sigma HPA010592 |
| | PLVAP Rabbit Sigma HPA002279 |
| | Alexa Fluor Goat anti-mouse 488 Invitrogen A28175 |
| | Alexa Fluor Goat anti-rabbit 568 Invitrogen A-11011 |
| | Alexa Fluor Donkey anti-mouse 488 Invitrogen A-21202 |
| | Alexa Fluor Donkey anti-rat 488 Invitrogen A-21208 |
| | Alexa Fluor Donkey anti-guinea pig 488 Jackson Immuno Research 706-545-148 |
| | Northern Light Donkey anti-sheep 493 R&D NL012 |
| | Northern Light Donkey anti-mouse 493 R&D NL009 |
| | Alexa Fluor Donkey anti-mouse 555 Invitrogen A-31570 |
| | Alexa Fluor Donkey anti-rabbit 555 Invitrogen A-31572 |
| | Northern Light Donkey anti-mouse 557 R&D NL007 |
| | Northern Light Donkey anti-rabbit 557 R&D NL004 |
| | Northern Light Donkey anti-mouse 637 R&D NL008 |
| | Northern Light Donkey anti-rabbit 637 R&D NL005 |
| | Northern Light Donkey anti-goat 637 R&D NL002 |
| | Alexa Fluor Donkey anti-rabbit 647 Invitrogen A-31573 |
| | CD68 KP1 Thermo Fisher 14-0688-82 2265228 https://www.thermofisher.com/antibody/product/CD68-Antibody-clone-KP1-Monoclonal/14-0688-82 |
| | CD31 EPR3094  abcam ab207090 GR3229164-11 https://www.abcam.com/cd31-antibody-epr3094-bsa-and-azide-free-ab207090.html |
| | CD11b EPR1344 abcam ab209970 GR3352581-4 https://www.abcam.com/cd11b-antibody-epr1344-bsa-and-azide-free-ab209970.html |
| | IMC Cell Segmentation Kit Protein 1 Protein1-unknown Fluidigm TIS-00001 1742007 |
| | IMC Cell Segmentation Kit Protein 2 Protein2-unknown Fluidigm TIS-00001 1742008 |
| | IMC Cell Segmentation Kit Protein 3 Protein3-unknown Fluidigm TIS-00001 1882005 |
| | CD8a C8/144B Thermo Fisher 14-0085-82 2247491 https://www.thermofisher.com/antibody/product/CD8a-Antibody-clone-C8-144B-Monoclonal/14-0085-82 |
| | HLA-DR TAL 1B5 abcam ab176408 GR3384096-1 https://www.abcam.com/hla-dr-antibody-tal-1b5-bsa-and-azide-free-ab176408.html |
| | CD4 EPR6855 abcam ab181724 GR3352909-4 https://www.abcam.com/cd4-antibody-epr6855-bsa-and-azide-free-ab181724.html |
| | SMA (ACTA2) 1A4 Thermo Fisher 14-9760-82 2288516 https://www.thermofisher.com/antibody/product/53-9760-82.html?ef_id=CjwKCAjwquWVBhBrEiwAt1Kmwg0mNovA4BDEYQrLlZsnBTvV6n6aMXnizAycM3_IK84aN4iQFe-8QRoCIcQQAvD_BwE:G:s&s_kwcid=AL!3652!3!459736943987!!!g!! &cid=bid_pca_aup_r01_co_cp1359_pjt0000_bid00000_0se_gaw_dy_pur_con&gclid=CjwKCAjwquWVBhBrEiwAt1Kmwg0mNovA4BDEYQrLlZsnBTvV6n6aMXnizAycM3_IK84aN4iQFe-8QRoCIcQQAvD_BwE |
| | pan Cytokeratin C11 Thermo Fisher MA1-12594 XB3490763 https://www.thermofisher.com/antibody/product/Cytokeratin-Pan-Antibody-clone-C11-Monoclonal/MA1-12594 |
| | pan Cytokeratin AE1 Sigma Aldrich MAB1612 3460341 https://www.sigmaaldrich.com/CA/en/product/mm/mab1612 |
| | Keratin Epithelial AE3 Sigma Aldrich MAB1611  3382323 https://www.sigmaaldrich.com/CA/en/product/mm/mab1611 |
| | E-Cadherin / P-Cadherin 36/E-Cadherin BD Biosciences 610182 2038668 https://www.bdbiosciences.com/content/bdb/paths/generate-tds-document.cn.610182.pdf |
| | BSG (CD147/Basigin) MEM-M6/1 abcam ab666 GR3344079-2 https://www.abcam.com/cd147-antibody-mem-m61-ab666.html |
| | CLDN5 EPR7583 abcam ab236066 GR3422726-1 https://www.abcam.com/claudin-5-antibody-epr7583-bsa-and-azide-free-ab236066.html |
| | CD74 LN2 abcam ab213104 GR3359511-2 https://www.abcam.com/cd74-antibody-ln-2-bsa-and-azide-free-ab213104.html |
| | PDGFRbeta Polyclonal R&D Systems AF385 B1W0821071 https://www.rndsystems.com/products/human-pdgf-rbeta-antibody_af385 |
| | GLUT1 (Glucose Transporter)  EPR3915 abcam ab252403 GR3374509-1 https://www.abcam.com/glucose-transporter-glut1-antibody-epr3915-bsa-and-azide-free-ab252403.html |
| | HLA- DRB5 Polyclonal Thermo Fisher PA5-60260 UL2898503A https://www.thermofisher.com/antibody/product/HLA-DRB5-Antibody-Polyclonal/PA5-60260 |
| | HLA- DRA Polyclonal Thermo Fisher PA5-27553 UL2898347B https://www.thermofisher.com/antibody/product/HLA-DRA-Antibody-Polyclonal/PA5-27553 |

HLA- DPA1 Polyclonal Thermo Fisher PA5-28037 UL2898341 https://www.thermofisher.com/antibody/product/HLA-DPA1-Antibody-Polyclonal/PA5-28037
HLA- DRB1 EPR6148 abcam ab133578 GR3300630-2 https://www.abcam.com/hla-class-ii-drb1-antibody-epr6148-ab133578.html
HLA- DQB1 Polyclonal abcam ab224600 GR3231742-9 https://www.abcam.com/hla-dqb1-antibody-ab224600.html
HLA- DMA Polyclonal Thermo Fisher PA5-22365 WB3188332C https://www.thermofisher.com/antibody/product/HLA-DMA-Antibody-Polyclonal/PA5-22365
Mouse IgG (H+L) Polyclonal Thermo Fisher A28174 2276480 https://www.thermofisher.com/antibody/product/Goat-anti-Mouse-IgG-H-L-Secondary-Antibody-Recombinant-Polyclonal/A28174
Rabbit IgG (H+L) Polyclonal Thermo Fisher A27033 RL246119A https://www.thermofisher.com/antibody/product/Goat-anti-Rabbit-IgG-Heavy-Chain-Secondary-Antibody-Recombinant-Polyclonal/A27033
Anti-Rat IgG (H+L) Polyclonal Thermo Fisher A18873 61-172-060320 https://www.thermofisher.com/antibody/product/Goat-anti-Rat-IgG-H-L-Cross-Adsorbed-Secondary-Antibody-Polyclonal/A18873

Validation | Validation are available for all antibodies from the manufacturer. Please refer to references contained in the provided links.

# Human research participants

Policy information about studies involving human research participants

Population characteristics | We performed single-cell RNA sequencing of 606,380 freshly isolated endothelial, perivascular and other tissue-derived cells from 117 samples, from 68 human fetuses and adult patients to construct amolecular atlas of the developing fetal, adult control and diseased human brain vasculature.
Tissues analyzed include fetal and adult samples covering fetal CNS and peripheral organs; as adult/control brains (temporal lobe (TL)), brain arteriovenous malformation (AVM), lower-grade glioma (LGG), high-grade gliomas/glioblastoma (GBM), lung cancer brain metastasis (MET) and meningioma (MEN).

Recruitment | Informed consent for fetal tissue collection and research was obtained from each patient after her decision to legally terminate her pregnancy but before the abortive procedure was performed. For adult tissue collection, informed consents for collection and research use of the surgically removed adult brain tissues was obtained from each patient before the operation.

Ethics oversight | The collection of human samples and research conducted in this study were approved by the institutional research ethics review boards of the University Hospital Zurich, the University Health Network Toronto and the Mount Sinai Hospital Toronto (approval numbers: BASEC 2016-00167, 13-6009, 20-0141-E). Informed consent for fetal tissue collection and research was obtained from each patient after her decision to legally terminate her pregnancy but before the abortive procedure was performed. For adult tissue collection, informed consents for collection and research use of the surgically removed adult brain tissues was obtained from each patient before the operation. Details on patient information and pathology reports are provided in Supplementary Tables 1,2. All the protocols used in this study were in strict compliance with the legal and ethical regulations of the University of Zurich, the University of Toronto and affiliated hospitals.

Note that full information on the approval of the study protocol must also be provided in the manuscript.

# Clinical data

Policy information about clinical studies

All manuscripts should comply with the ICMJE guidelines for publication of clinical research and a completed CONSORT checklist must be included with all submissions.

Clinical trial registration | NA

Study protocol | NA

Data collection | NA

Outcomes | NA

# Flow Cytometry

## Plots

Confirm that:

☒ The axis labels state the marker and fluorochrome used (e.g. CD4-FITC).

☒ The axis scales are clearly visible. Include numbers along axes only for bottom left plot of group (a 'group' is an analysis of identical markers).

☐ All plots are contour plots with outliers or pseudocolor plots.

☒ A numerical value for number of cells or percentage (with statistics) is provided.

## Methodology

Sample preparation | For human fetal brains/CNS and human fetal peripheral organ tissues, fresh fetal tissues were obtained from patients who

| Sample preparation | elected to terminate their pregnancies at fetal age (indicated by gestational weeks) 9 weeks – 21 weeks (GW 9 - 21) for reasons that are not genetic or medical conditions. Immediately following the termination of pregnancy procedure, fetal tissue samples were transferred to cold 0.01 M PBS / surgical physiological solution (Tis-U-Sol, Baxter) and transported on ice to the research facility in order to begin tissue processing and sample dissociation within 3 hours of collection.<br>For human adult/control brains (temporal lobe, TL), fresh normal cerebral cortex of the temporal lobe was obtained as part of a neurosurgical operation for epilepsy called temporal lobectomy1 for patients with pharmacoresistant epilepsy. Brain tissue of the neocortical resection of the temporal lobectomy corresponding to the normal cerebral cortex/neocortex overlying the hippocampus (to reach deep seated lesions in the amygdala and hippocampus causing epilepsy and removed during the amygdalohippocampectomy resection of the temporal lobectomy) which is thought to be uninvolved in the pathology2 (for details about the tissue samples, see Supplementary Tables 1,2) was harvested. We harvested the maximal safe amount of temporal neocortex after a piece of tissue was sent for histopathology (standard for every neurosurgical operation). All harvested brain specimens were >2 cm away from any radiographic abnormality on magnetic resonance imaging.<br>For human brain pathologies (brain tumors (LGG, GBM, MET, MEN) and brain vascular malformations (AVM), fresh samples were obtained during neurosurgical operations for either brain tumors or brain vascular malformations (for details about the tissue samples, see Supplementary Tables 1,2). In brief, we harvested the maximal safe amount of brain tumor/brain vascular malformation tissue after a piece of tissue was sent for histopathology (standard for every neurosurgical operation), (for details about the tissue samples, see Supplementary Tables 1,2).<br>All adult brain tissues (adult/control brain tissue, pathological brain tissues) were acquired by academic neurosurgeons familiar with routine sampling of surgical tissue for research purposes including cell isolations and sequencing experiments as well as tissue stainings. Generally, this involved i) resection "en bloc" (e.g. the entire brain tumor / brain vascular malformation, the entire neocortex overlying the hippocampal lesion) to maintain in situ tissue organization whenever possible ii) minimizing tissue damage/disruption by avoidance of electrocautery as much as possible, iii) washing the brain tumor or perfusing the brain vascular malformation tissue with cold 0.01 M PBS/surgical physiological solution (Tis-U-Sol, Baxter) to reduce intravascular blood/erythrocytes. In the case of brain arteriovenous malformations 1:9 heparin dilution was added to the physiological solution to perfuse the vessels. Immediately after neurosurgical resection, adult tissue samples were transferred to cold 0.01 M PBS / surgical physiological solution (Tis-U-Sol, Baxter) and transported on ice to the research facility in order to begin tissue processing and sample dissociation within 2-3 hours of resection. Fetal and adult tissue samples were then processed for either bulk- or sc-RNA sequencing (FACS-sorted or unsorted) or for immunofluorescence (IF), immunohistochemistry (IHC) or Imaging Mass Cytometry (IMC). Patient demographic information and details on brain tissue samples for all fetal brain and peripheral tissues as well as for all resected adult/control brain and pathological brain tissues utilized are summarized in Supplementary Tables 1,2).<br><br>Isolation of FACS-sorted human fetal and adult endothelial cells and of unsorted human fetal and adult endothelial and perivascular cells for single cell RNA-seq<br>Endothelial cells were isolated from human fetal and adult tissues using tissue digestion and subsequent CD31+ / CD45- FACS sorting whereas human endothelial and perivascular cells (all cells) were isolated from the unsorted fraction. Briefly, both fetal and adult tissues were quickly minced in a petri dish on ice, using two surgical blades. For CD31+ / CD45- FACS sorting, a cell suspension was obtained upon digesting the tissue in 2 mg/ml Dispase II (D4693, Sigma-Aldrich, Steinheim, Germany), 2 mg/ml Collagenase IV (#1710401, Thermo Fisher Scientific, Zurich, Switzerland) and 2 mM CaCl2 PBS solution for 40 min at 37°C with occasional shaking. The suspension was filtered sequentially through 100/70/40 μm cell strainers (#431751, Corning, New York, USA) to remove large cell debris. Cells were then centrifuged 500 RCF for 5 min at 4°C. In case of a visible myelin pellet 5 ml 25% Bovine Serum Albumin (BSA) (ice cold) was overlayed with 5 ml of the sample, centrifuged at 2000 RCF for 20 min (4°C). The supernatant was removed (including the lipid phase) and the pellet was resuspended in 9 ml PBS, followed by another round of centrifugation at 500 RCF for 5 min (4°C). Supernatant was subsequently discarded, while the cell pellets were resuspended in 3 ml of ACK hemolytic buffer at room temperature for 3 minutes. To stop the reaction, 30 ml of ice-cold PBS was added to the mixture and centrifuged at 500 RCF for 5 min at 4ºC. The cell pellets were resuspended in FACS buffer (PBS + 1% Bovine Serum Albumin), a volume is taken for unsorted scRNA-seq analysis. For CD31+ / CD45- FACS sorting, the cells were stained with anti-CD31 PE conjugated antibody in a concentration of 1:20 (#566125, clone MBC78.2, BD Pharmingen) and anti-CD45 APC conjugated antibody in a concentration of 1:20 (#17-0459-42, clone HI30, eBiosciences) for 30 min at 4º C, protected from light. Thereafter the cells were washed with 1ml of FACS buffer, centrifuged in a tabletop centrifuge at 500 RCF at 4ºC for 5 min. Finally, the cell pellets were resuspended in appropriate volumes of FACS buffer (PBS + 1% Bovine Serum Albumin) and the suspension was passed through a 35 μm cell strainer of a FACS sorting tube (#352235, Corning). Immediately before sorting, SYTOXTM blue was added in 1:1000 (Thermo Fisher Scientific, #S34857) to exclude dead cells from further analysis. Cell debris were excluded via a forward scatter-area/side scatter-area (FSC-A/SSC-A) gating, while singlets were selected for using a forward scatter-area (FSC-A)/ FSC-height (FSC-H) gating strategy. Viable (SYTOXTM blue negative) endothelial cells were FACS-sorted by endothelial marker CD31 positivity and negative selection for the brain microglia and macrophages marker CD45, whereas unsorted endothelial and perivascular cells were obtained from the SYTOXTM blue- fraction. Cells were sorted by a FACS Aria III (BD Bioscience) sorter using the four-way purity sorting mode directly in EGM2 medium (#CC-3162, Lonza, Basel, Switzerland). |
|---|---|
| Instrument | FACS Aria III |
| Software | FACS Aria III software |
| Cell population abundance | The abundance of endothelial cells (CD31+/CD45-) varied according to the patient sample, ranging from 5% to 15% of all viable cells. |

Gating strategy

Cell debris were excluded via a forward scatter-area/side scatter-area (FSC-A/SSC-A) gating, while singlets were selected for using a forward scatter-area (FSC-A)/ FSC-height (FSC-H) gating strategy. Viable (SYTOXTM blue negative) endothelial cells were FACS-sorted by endothelial marker CD31 positivity and negative selection for the brain microglia and macrophages marker CD45, whereas unsorted endothelial and perivascular cells were obtained from the SYTOXTM blue- fraction. Cells were sorted by a FACS Aria III (BD Bioscience) sorter using the four-way (Figure exemplifying the gating strategy is provided in Supplementary Figure 1p).

☒ Tick this box to confirm that a figure exemplifying the gating strategy is provided in the Supplementary Information.

