## [Peer Review File · Nature]

Manuscript Title: Single-cell atlas of the human brain vasculature across development, adulthood and disease

Reviewer Comments & Author Rebuttals

Reviewer Reports on the Initial Version:

Referees' comments:

Referee #1 (Remarks to the Author):

In this submitted work by Walchli et al., the authors provide a “molecular atlas” of the CNS endothelium using fresh surgical tissue from epilepsy surgeries in the adult age group, and rapid-autopsy in the fetal age group. In addition to describing the normal CNS endothelium in healthy controls, the authors also include samples from brain tumors and vascular malformations to demonstrate changes to the CNS endothelium in disease. The primary method used in this paper is single-cell RNA sequencing, with validation of key transcript expression using RNA-scope and immunofluorescence.

Although single-cell sequencing of brain tissue has been extensively published in the past, these authors provide a presumably more accurate approach using fresh surgical tissue. The focus on the CNS endothelium is additionally novel, as is the inclusion of fetal tissue. The inclusion of diseased tissue is interesting, but the overall contribution to the paper is less clear and may dilute the strength of the studies performed in non-diseased tissue. It is potentially interesting that alterations in arterio-venous differentiation patterns and changes in cell-cell interaction partners may be generally associated with pathology in the CNS vasculature. However, the evaluation of these diverse neurovascular and neoplastic disorders may have been overly simplified by clustering diseases. A more detailed evaluation of the individual transcriptomic profiles in these distinct disorders is limited by very small sample sizes per disease group. It is unclear whether the diseased tissue, when sorted into their respective pathologies, provide sufficient power to draw the conclusions when compared to normal tissue in this paper (for example, hemangioblastoma tissue only from one representative sample). Additionally, some of the validation of important transcriptional differences was lacking. For example, while MHC II components were described as expressed by CNS endothelial cells (an important finding), the immunofluorescence and RNA-scope is less convincing (Fig 5, Extended data figure 24).

Taken together, the main novelty provided by this paper is the fresh tissue source used in the adult samples, and the inclusion of fetal samples, to derive the CNS endothelial cell molecular atlas. On this point, it will be useful to compare some of the published NucSeq data to highlight the difference/similarity of the results between these two types of samples and methods. Major limitations are the number of fetal and diseased samples, the age diversity of both the adult and fetal samples, and the validation of key transcripts amongst the undiseased CNS tissue samples.

Major Points

- As noted above, overall inclusion of different diseased tissues include clustering of distinct tumor types. Analysis of individual diseases is limited by sample number.
- Given the different origins of the ECs across conditions, the authors should briefly describe the methods used to generate their merged dataset. CCA and RPCA were performed to integrate data across samples. Was ambient RNA correction also performed to minimize bias? Such an analysis may help eliminate the possibility that some observed differences are due to differences in brain regions where the samples are isolated, heterogeneity in the isolation method or differences among individuals.
- There are limitations on the generalizability of the results in the adult age group based on the age and sex distribution amongst the healthy control (temporal lobe) samples. An emphasis is made in the paper that the fetal brain is compared to the adult brain to model the “development” of the CNS endothelium, but the age range of the adults is 15-69. There is concern that 15 years-old is widely not considered to be “adult aged”, and that there are likely subtle differences between patient samples in their 20s and 40s compared to the aged adult sample that was 69. The age distribution should be addressed either as a significant limitation of the atlas, or the adult ages divided into age ranges and analyzed separately, which would require additional samples. Similarly, one of the fetal ages is 9 weeks while another is 18. There is concern that grouping two disparate ages during a period of rapid neurodevelopment would result in inaccurate representation of the fetal BBB.
- Gliomas are grouped by IDH1 status (wild-type), but also contain differences in MGMT status. This results in the clustering of what we know are phenotypically distinct tumors, that have ultimately distinct clinical features. This work is an oversimplification of single-cell sequencing that has already been described in GBMs (There are many publications, but Neftel et al 2019 Cell, for example). The same is true of the metastatic disease samples, which likely include distinct angiogenic features depending on the specific neoplasm.
- Figure 5: MHC II expression on human endothelial cells is a key finding in this paper. The immunofluorescence validation of CD74 is not convincing (fig 5i-p), and additional assays for validation would be needed to support this conclusion. The expression on AVMs appears cytoplasmic, however, in the healthy adult the immunofluorescence appears less convincing, and either nuclear or artifactual. Additionally, this figure should be inclusive of fetal tissue since this is an important comparator in the results section and discussion.

Minor Points

- Fig 1 a-c – this information is helpful but could be streamlined; numbers of sorted/unsorted endothelial cells are helpful and important, but overview could be condensed to make more room for data in the body of the manuscript
- Pathologic tissue is described as a “reactivated fetal” pathways in brain endothelium. It is likely more accurate to say that both pathways share a proliferation of less-differentiated cells, as there is no evidence to demonstrate that diseased cells start at with an adult transcriptional profile, and then activate fetal pathways to become less differentiated.
- It is unclear why meningiomas, a non-vascular derived tumor, were included in this paper.
- Justification should be provided on the value of including non-vascular tumors- do the authors hypothesize that the CNS endothelium is transcriptionally distinct in these tumors types due to intrinsic differences in tumor pathology?
- The authors write several times throughout the manuscript that the analysis includes 47 fetuses

and adult samples to create the molecular atlas. This is misleading as only 5 fetal brains and 8 normal temporal lobes in the adult were used to create the un-diseased atlas.

- There are several instances with portions of figures that appear to be redundant. Some examples include: Figure 1e,f appears redundant with figure 1g; Extended Data Fig 3 and 4; Extended Data Fig. 6 a-c.
- Fig 2g. – axis of bar chart or bars should be labeled with percentage values
- Extended Data Fig. 3 – many colors used to refer to different cell types; can be difficult to compare, for example, endothelial cells with stem cells and tumor cells in 3e and especially 3f.
- Figure 1m: unclear value of displaying pathologic transcript profiles compared to control. Clustering of distinct pathologies is an over-simplification of disease processes. Unclear value of Venn diagram in figure 1l, which demonstrates some overlap in transcriptional signature between the three cell sources, all of which are presumably endothelial in origin.
- Figure 2k: Is this pie chart derived from all samples pooled together (control, pathologic, and fetal, as in 2g) or just adult control? If only adult control, would be valuable to demonstrate shift in vascular compartments with pathology or age.
- Figure 2m: Signaling patterns by vascular compartments in the adult demonstrates interesting distribution of pathways. Again, in figure 2n, lumping distinct diseases raises concerns for differences in signaling between very different tumor types.
- Unclear added value of figure 3p,q,r,s.
- Figure 4 SPOCK3 and CD320 validation; unclear why these exact transcripts were chosen. Additionally, fetal tissue should be included in the IF validation since it is displayed as differentially regulated in part d.
- Extended Data Fig. 8 zii – EndoMT markers – ACTA2 is expressed in perivascular SMCs; perhaps an additional marker specific to mural cells (i.e. PDGFRB) would exclude the possibility that the signal is coming from a nearby mural cell?
- Extended Data Fig. 9 – immunofluorescence is not incredibly convincing to show co-expression of these markers in endothelial cells
- Figure 6g exemplifies the diversity of the tumor cell expression profiles, which were clustered together to form the interaction maps in part h onward. This oversimplifies the interactions of 10 or more tumor types into one interaction model.
- The authors compare their pathological endothelial cells with a BBB dysfunction module defined in the mouse. It would also be interesting to highlight certain genes that seem to be human specific in undiseased conditions (compare to an undiseased mouse BBB module) to determine if there are baseline differences in BBB regulation across species.

Referee #2 (Remarks to the Author):

A. Summary of the key results:

In this manuscript, Wälchli et al present a molecular atlas at the single-cell level of the human brain (and fetal peripheral organs) vasculature in physiological and pathological conditions. The authors originated a robust sample inventory of human tissue, and demonstrate along the whole paper different possible analysis to do with such resources.

B. Originality and significance:

While the creation of a single cell sequencing atlas of endothelial cells and the concept of inter heterogeneity of endothelial cells (ECs) in physiological and pathological conditions is not novel and has been done before in mouse, the work presented here is relevant, as it offers an important resource to the community that contains human endothelial cells from developmental stages and from relevant brain pathologies.

While there is some validation of the expression of some relevant genes in the different entities analyzed, there is no further analysis of the functionality or importance of those gene expression changes, or specific cell-cell communication pathways identified. Therefore, in this reviewer's eyes the manuscript is mainly seen as a resource paper that will provide the community an important dataset of single cell expression data in human ECs across different developmental stages and pathological situations.

C. Data & methodology; Appropriate use of statistics etc:

- The authors use as control ECs isolated from lobectomy in pharmaco-resistant epilepsy patients. Thus it might be that those ECs might already present significant changes with respect to a real control brain EC. This should be considered and fully addressed.

- The analysis of CNS-specific properties used bulk analysis of ECs from different peripheral organs. Thus, the authors are disregarding the EC differences between the different peripheral organs. This should be mentioned. Are brain ECs becoming more similar to a specific organ ECs?

- What developmental stages were the fetuses that are analyzed in this study? During developing brain angiogenesis is taking place and a such EC gene expression is changing depending on the developmental stage. Thus, a concern in their methodology is that merging all developmental stages might bias the results in the comparisons. This should be addressed.

- "CODE AVAILABILITY- the authors write: Specific code will be made available upon request to T.W.". As this article contains a large inventory of human samples sequenced, it is important as a resource paper. Therefore, the specific code used for the different analysis should be available upon publishing of the article.

- Fig1c: This is a composite of UMAP figures, please clarify this in the figure legend.
- At the main text part of inferred cell-cell interactions, it would be nice to indicate the program/database used, as that has a great impact on the results obtained in these analysis.
- Fig3 p,q,r What is exactly shown in these dotplots? The legend is really vague. Occurrence in what? Why are the y axis values all divisible by 10? What median rank is depicted on x? What is the selection criteria for the coloured boxes?
- The authors write: "we performed the standard Seurat clustering (version 4.0.0) workflow" - What were the parameters used for clustering? Do these parameters fit to the various dimensionality checks of the data?
- "Differential expression was computed using the wilcoxauc function implemented in the github package presto." Insert link to GitHub page, there are many GitHub repositories called presto.
- "Of note, the APP-CD74, COPA-CD74, and MIF-CD74 ligand-receptor pairs that were predicted to mediate MHC class II signaling in EC-EC interactions (Extended Data Figure 35a-c)" There is no EDFig35.
- "DESeq2 package (v1.30.1) was used to perform pseudobulk differential expression analysis" and "Differential expression was computed using the wilcoxauc function implemented in the github package presto" together: Which was used for what? One is a pseudobulk approach, the other is a sc method.

E. Conclusions: The authors mainly summarize their observations and validations of their scRNAseq and bioinformatic analysis.

F. Suggested improvements: The manuscript is overwhelming, there are too many figures, subfigures, comparisons, etc (6 main figures + 30 extended figures). This makes the paper very-very complicated to follow. It also greatly increases the chances for statistical errors and false positive hits. This should be considered and, while still providing all the information and datasets obtained, narrow the figures and presented data so that a more specific focus on the result analysis can be distilled. In this regards, the identification of MHC-class II gene expression in ECs in different pathologies and their role in immune surveillance could be a very interesting focus.

G. References: Ok, no concerns there.

H. Clarity and context: See comment in F.

Referee #3 (Remarks to the Author):

The manuscript by Walchli et al presents a large and comprehensive single cell transcriptomics atlas for human brain vasculature. The authors provide the atlas for healthy brain, for multiple disorders, and for development. In addition to single cell data, there are extensive validations using smFISH and immunohistochemistry. The atlas, without a doubt, will be a great resource for the scientific community. However, before accepting the manuscript, a number of issues should be addressed.

Major points:

1. Different ways for integration have been used for unsorted cells, CCA, and sorted cells, RPCA. CCA is used to integrate samples with conserved cell types, but with large expression differences, whereas RPCA has a much “softer” way of integration. Importantly, CCA is not the best way to integrate healthy with disease samples when the goal is to study differential expression, and in this case CCA correction could be too aggressive and result in overcorrection. Thus, CCA could help to identify similar cell types in control and disease, but certainly is not the best way to reveal gene expression differences between conditions. Therefore, differential expression analysis should be done using another algorithm for integration.

Finally, the authors are also recommended to read this manuscript on comparison of algorithms for integration

<https://www.biorxiv.org/content/10.1101/2020.05.22.111161v2>

where Seurat v3 performs well only on broader cell types, whereas its performance in a complex biological context is rather poor. The authors claim to provide a high-resolution atlas with >0.5 million brain vasculature cells in a complex biological context, including several disease conditions, to use for the scientific community. Thus, the data must be solid. The authors should confirm that integration with another method (that preserves biology better) shows similar results (including subtype annotation and differential expression) and if not then a better integration algorithm that allows to study complex biological systems and preserves biology should be used for this study. Additionally, the integration across the study should be done with the same algorithm in order to be able to cross-compare the datasets.

2. Too little metadata and QC are available. This is an atlas; thus, the authors should be very transparent about their data. For instance, outcomes of sorting (e.g. % out of total, % of cells for live/dead stain etc) should be provided. For QC, plot distribution of cells with high mitochondrial counts/per sample, plot number of genes per cell detected for each sample, plot genes per cell type, plot UMIs per cell type etc. In fact, I could not even find information about the numbers of cells sequenced per sample, the numbers of reads per sample, and an average number of genes/UMIs detected in a sample (maybe it is available, just difficult to find, then include it in metadata in Suppl table 1). Additionally, disease metadata is rather poor – for the temporal lobe, are all samples from patients with temporal lobe epilepsy? Any pathology in the cortex? For one sample, FCDIc is shown, anything for other samples? Was only the cortex used from these samples or also the hippocampus? If the hippocampus was used – a comparison of HS samples with non-HS should be done to either show that the vasculature is not affected by sclerosis, or there are some gene expression/cell type compositional changes that are related to sclerosis.

3. Although the authors provide some markers for potential transcriptomic subtypes in Figure 2 and Extended Figure 7, it is not very convincing. Additional subclustering according to the type of vasculature, e.g. venous compartment, arterial compartment, capillary compartment, should help to subdivide the large cloud in Figure 2a and hopefully could provide better evidence for the existence of subclusters. Such compartment-specific plots should be also implemented for follow-up analyses, instead of Figure 2a UMAP, where appropriate.

The same holds true for UMAPs in Extended Figure 9 and Figure 3, annotation is not convincing, and a similar approach as above could help to produce a more convincing annotation.

4. Related to the above, and even more importantly, such subclustering has to be done for identification of BBB signature, when comparing peripheral vs CNS. For instance, I am sure the authors know a hot debate about BBB in the fetal brain, which was more or less settled only recently. This dataset provides for a unique opportunity to study the fetal BBB at high resolution, and currently this opportunity is missed. Overall, for BBB, the authors should add:

4a. better resolution for BBB analysis, which could be done by subclustering according to venous, arterial, and capillary compartments

4b. same better resolution for BBB dysfunction module, and here also comparison for a BBB signature to a neurodegenerative disorder:

<https://www.biorxiv.org/content/10.1101/2021.04.26.441262v1.full>

4c. developmental resolution; even though it is only 5 samples, the trajectory of BBB from GW9 to GW18 would be great data to add. The period between GW8 and 18 is usually considered as the period for major development of the BBB.

4d. comparison of developmental trajectory per compartment with adult BBB markers – when the BBB is complete, based on adult marker expression, is it as early as GW9 or only at GW18?

5. Changes in cell type composition between control and pathological conditions should be statistically tested, where possible, given that enough samples are available, e.g. by scCODA (Buttner et al. 2021 Nat Commun) that so far shows the best performance controlling for false discoveries in abundancies.

6. Potential sex-specific expression should be analyzed.

7. Potential brain region-specificity in subtypes/expression should be analyzed (different brain regions are available e.g. in glioma).

8. Monocle lineage tracing often does not show a clear picture in this manuscript. Velocity should be better applicable here, see La Manno et al. Nature 2018, Bergen et al. Nat Biotech 2020, and Lange et al. bioRxiv 2020.

In addition, how the red line with arrows for trajectories was produced? Was it automatically generated by Monocle, or was it the author's interpretation, and was inserted manually? This should be stated in the figure legends of the corresponding figures.

9. Overall, a better description of computational methods should be provided.

Minor points:

1. For astrocytic marker – use Aqp4 instead of Gfap, Gfap expression is activation dependent.
2. It is not clear how endothelial and perivascular cells were obtained from unsorted samples. Did the authors sequence all cells, including neurons etc, and then filter out endothelial and perivascular cells computationally? The procedure how endothelial and perivascular cells were obtained from unsorted samples should be described in detail and necessary illustrations/plots should be provided.
3. When describing DE for figures 1k-m and Extended figures 5,6 – supplementary table for DE genes should be provided and referred to.
4. Better description of the annotation procedure will help. For instance, for Figure 2, did the authors just choose a clustering structure with a particular seed, or did they additionally manually verify their annotation?
5. Circle plots showing ligand-receptor interactions are often difficult to comprehend, due to too many lines. Having some threshold and showing only the most significant interactions will help to visualize the most important data. This is related to many plots in both main and supplementary figures.
6. The authors comment on the importance of compositional changes across different sample sets. While this data is indeed interesting, the authors should be clear that some of those changes could derive from batch effects due to too few samples per condition. In particular for cancer datasets – tissue sampling could affect cell type composition (unless the whole biopsy was used to generate each cancer sc dataset, and then it should be mentioned in the methods).
7. Some of the supplementary figures are not necessary.
8. “Notably, these “unassigned” ECs mainly belonged to small caliber vessels” – why it is notable? Please explain.

Author Rebuttals to Initial Comments:

Reviewers' Comments:

Reviewer #1:

Comments for the Author:

COMMENT:

In this submitted work by Walchli et al., the authors provide a “molecular atlas” of the CNS endothelium using fresh surgical tissue from epilepsy surgeries in the adult age group, and rapid-autopsy in the fetal age group. In addition to describing the normal CNS endothelium in healthy controls, the authors also include samples from brain tumors and vascular malformations to demonstrate changes to the CNS endothelium in disease. The primary method used in this paper is single-cell RNA sequencing, with validation of key transcript expression using RNA-scope and immunofluorescence.

ANSWER:

We thank the reviewer for their comments on our manuscript.

We now indeed provide additional experimental data in the revised manuscript that addresses all the concerns raised by the reviewers. The new data further validates the findings of our single-cell RNA-sequencing (scRNA-seq) atlas and includes:

- new bulk RNA-sequencing (bulk RNA-seq) data
- additional immunofluorescence (IF) stainings
- new Imaging Mass Cytometry (IMC) = Imaging Cytometry by Time of Flight (Imaging CyTOF) stainings
- new spatial transcriptomics data

thereby addressing key reviewer's concerns about additional validation experiments, additional patients, and inclusion of fetal brain tissues (for IF and IMC).

In addition, we have addressed all issues raised by the reviewers and by the editor.

COMMENT:

Although single-cell sequencing of brain tissue has been extensively published in the past, these authors provide a presumably more accurate approach using fresh surgical tissue. The focus on the CNS endothelium is additionally novel, as is the inclusion of fetal tissue. The inclusion of diseased tissue is interesting, but the overall contribution to the paper is less clear and may dilute the strength of the studies performed in non-diseased tissue. It is potentially interesting that alterations in arterio-venous differentiation patterns and changes in cell-cell interaction partners may be generally associated with pathology in the CNS vasculature. However, the evaluation of these diverse neurovascular and neoplastic disorders may have been overly simplified by clustering diseases. A more detailed evaluation of the individual transcriptomic profiles in these distinct disorders is limited by very small sample sizes per disease group.

ANSWER:

We thank the referee for these comments, the senior-editor made a similar comment regarding the different pathologies. We would like to clarify the raised issues.

According to our discussion with senior editor Dr. George Caputa on the 25-1-2022, we agreed to keep all of the different fetal, adult/control and diseased tissues in the paper. We are convinced that the comparison between the fetal brain endothelium, the adult/control brain endothelium, and the diseased/pathological brain endothelium is especially interesting and constitutes one of the key messages of this manuscript. For instance, the reactivation of fetal pathways in brain pathologies is very intriguing.

This being said, to address the reviewer's concerns, we have now further clarified which findings and transcriptomic changes are conserved among the different disease entities while also emphasizing findings that are more specific to the different neurovascular entities, namely brain vascular malformation versus brain tumor/neoplastic entities. The observation that we find commonly dysregulated pathways, alterations in AV-differentiation patterns and CNS-specificity, MHC class II upregulation and changes in cell-cell interactions in ECs (and cell-cell interactions with PVCs) across

different disease entities is novel and interesting in our view and thereby constitutes a “hallmark” of the diseased brain vascular endothelium.

To address the reviewer’s related concern about oversimplification to that regard, we would like to mention that we have provided the detailed results for the different scientific findings at multiple levels of analysis and comparison, namely:

- i) for differential expression analysis (for ECs), pathway analysis (for ECs) and ligand-receptor interaction analysis (for ECs and PVCs), we provide the following results:
 - for fetal brain versus adult/control brain (see Figure 1s,t, Figure 6a-c,j,o-p, Extended Data Figures 7d-f, 8d,17,18,34a,35,36, Supplementary Figures 17,20,23,26a,31,35-38,40).
 - for pathological brain versus adult/control brain (see Figure 1t,u, Figure 6g-i,k,s,t, Extended Data Figures 7g-l, 8d,17,18,31,34c,35,36, Supplementary Figures 16,18,22,26c,30,32-34,39).
 - for brain tumor versus adult/control brain (see Extended Data Figures, 33 a’-e’,34e, Supplementary Figure 3,24II,25II,26e,41II).
 - for brain vascular malformation versus adult/control brain (see Extended Data Figures 33a-e, 34d, Supplementary Figures 2,24I,25I,26d,41I).
 - for every single pathological/disease entity versus adult/control brain (see Extended Data Figures 9,33,34f-m, Supplementary Figures 7-14,24III-X,25III-X,29,41III-X)
- ii) EC and PVC clustering, we provide the following results:
 - for all fetal, adult, and pathological entities pooled together (see Figure 2, Extended Data Figure 4)
 - for every single brain entity (
 - for fetal brain
 - for adult/control brain (TL)
 - for every single pathological/disease entity separately (TLadjCAV, CAV, AVM, HEM, LGG, GBM, MET, MEN)
 - for every individual patient cell composition/AV annotation, please see Extended Data Figures 5u,11w,x).

Thus, while we have indeed clustered/merged the diseased samples together, we also provide the clustering/merge of every individual disease entity, and these different levels of comparisons complement each other to derive the respective scientific conclusions.

Regarding the limited sample size, we would like to point out that while the number of samples is reasonable (mostly n=5 per disease entity), the number of sorted ECs per sample and per entity is actually very high. We agreed in our discussion with senior editor Dr. George Caputa on the 25-1-2022 to not include further scRNA-seq samples in this study for reasons of feasibility and timeliness. However, to address the reviewer’s comment, we have mentioned the limitation of the relatively small sample number in the paragraph “Discussion”.

Moreover, to further support the key findings of our single-cell RNA-seq atlas, we have now also added bulk RNA-seq data of adult/control brain ECs and key diseased brain ECs (Extended Data Figure 9, Supplementary Figure 14) and these data confirm the main findings of the scRNA-seq analyses. The description of these newly obtained results as well as their description and comparison to the scRNA-seq data has been added throughout the text.

The results of the bulk RNA-seq experiments can be found in Extended Data Figure 9, Supplementary Figure 14 and this additional technique has now - together with IMC - also been added to the working model (see Figure 1a). Moreover, we added a new Supplementary Table 1 with all clinical / patient information of the bulk RNA-seq experiments.

Taken together, one purpose of the manuscript is to address common changes in various vascular-dependent brain diseases and we feel that the observed changes in transcriptomic profiles of endothelial cells across various vascular-dependent brain pathologies is a key finding of our study. Moreover, the precise description of endothelial cells transcriptomic profiles in the brain vasculature of the different brain diseases provides an important publicly available molecular atlas that can serve as a reference for future studies regarding the developing, adult, and diseased brain vasculature.

COMMENT:

It is unclear whether the diseased tissue, when sorted into their respective pathologies, provide sufficient power to draw the conclusions when compared to normal tissue in this paper (for example, hemangioblastoma tissue only from one representative sample).

ANSWER:

We thank the reviewer for this comment. While we agree on the relatively low number of samples for some entities, we would like to emphasize that the number of both endothelial- as well as perivascular cells per entity is very high and, to our knowledge, unmet as compared to any other currently published study. However, to address the reviewer's comment, we have performed additional bulk RNA-seq experiments of key diseased brain entities including the hemangioblastoma, which is an example of a typical vascular tumor. The key findings emanating of the scRNA-seq experiments described in the manuscript were confirmed by the bulk RNA-seq data (Extended Data Figure 9, Supplementary Figure 14), thereby addressing the reviewer's concerns. For instance, whilst hemangioblastoma is a very rare disease (they account for 1-2.5% of intracranial neoplasms, with overall incidence: 0.141 per 100,000 person-years Yin et al, 2020), by adding one more hemangioblastoma in the bulk RNA-seq experiments, we are quite confident that the statements made in our paper are representative even of this rare disease. Given that hemangioblastoma is a typical vascular tumor, we feel that we would like to keep it in our dataset.

Moreover, bulk RNA-sequencing confirmed that similar pathways were dysregulated in the different pathologies when compared to the temporal lobe (Extended Data Figure 9, Supplementary Figure 14), thereby confirming the common "hallmark" of regulated pathways across multiple pathologies. Thus, the conclusions drawn are not solely based on grouping various pathologies together.

The number of endothelial- and perivascular cells for every entity included in the analysis can be found in the Supplementary Tables 2,3. In addition, we now included the following sentence to the paragraph "Results":

"Bulk RNA-seq data of the adult/control brain (TL) and the different pathologies (CAV, AVM, HEM, LGG, GBM, MET, MEN) confirmed our single-cell RNA-seq findings including the common dysregulated pathways across pathologies (Figure 1, Extended Data Figure 9, Supplementary Figure 14)."

Moreover, we have further clarified the common findings/alterations across different disease entities as well as the specific findings for each disease in the manuscript.

COMMENT:

Additionally, some of the validation of important transcriptional differences was lacking. For example, while MHC II components were described as expressed by CNS endothelial cells (an important finding), the immunofluorescence and RNA-scope is less convincing (Fig 5, Extended data figure 24).

ANSWER:

We thank the reviewer for this comment. While we feel that the RNA-scope images demonstrate MHC II mRNA in CNS endothelial cells and IF images demonstrate MHC class II protein in CNS endothelial cells, we have performed additional validation experiments including more immunofluorescence as well as IMC experiments on the different brain entities. Importantly, we have also included the fetal brain tissues in these validation assays, as also requested by reviewer 1. Moreover, for TL and GBM, we now also provide spatial transcriptomic data from publicly available datasets (Extended data figure 32) showing co-localization of MHC class II receptors with MHC class II ligands APP, COPA, and MIF on ECs and microglia/macrophages in GBM (and in TL but with overall lower expression).

A brief summary of the key findings is provided here:

- Notably, these additional immunofluorescence and IMC validation experiments further confirmed the key findings regarding the increased expression of MHC class II components in pathological brain endothelial cells as compared to adult/control brain ECs (Figure 5, Extended Data Figure 30,37).
- In the fetal brain endothelium, MHC class II expression was low in IF and IMC experiments (Figure 5, Extended Data Figure 29).
- IMC showed upregulated expression of the MHC class II markers CD74, HLA-DRs, and a mix of HLA-Ds (referred to as HLA-oligo-D) in pathological brain ECs across various brain diseases (Figure 5, Extended Data Figure 30,37).

- IMC also revealed EC expressing MHC class II receptors physical proximity with immune cells (as microglia/macrophages) expressing various MHC class II receptors (Figure 6, Extended Data Figure 37).
- Spatial transcriptomic data for TL and GBM shows
 - co-localisation of MHC class II receptors (for instance CD74) and ligands (APP, COPA, and MIF) on ECs and microglia/macrophages in GBM and TL (Extended Data Figure 32).
 - higher expression of MHC class II receptors and ligands on both ECs and microglia/macrophages in GBM as compared to TL (Extended Data Figure 32).

Importantly, in addition to the validation experiments for MHC class II described here (main Figures 5 and 6, Extended Data Figures 29,30,32,37), we have now also added numerous validation experiments of important transcriptional differences for the other key biological findings (main Figures 1-6 and related Extended Data Figures and Supplementary Figures), notably:

For Figure 1 – Construction of atlas

- additional immunofluorescence stainings in fetal, adult and pathological tissues to validate the expression of the angiogenic markers ESM1 and PLVAP (Figure 1k-r', 2e-f, Extended Data Figure 13).

For Figure 2 – Inter-tissue heterogeneity

- additional immunofluorescence stainings in fetal, adult and pathological tissues to validate the expression of the angiogenic markers ESM1 and PLVAP (Figure 1k-r', 2e-f, Extended Data Figure 13).
- additional IMC and immunofluorescence stainings to validate the expression of the EndoMT markers ACTA2 in ECs and in PVCs (e.g. PDGFRB+) (Extended Data Figure 14)
- additional immunofluorescence stainings to validate the expression of the stem-to-EC markers: SOX2, PTPRZ1, EPCAM and SFTPB (Extended Data Figure 15)
- immunohistochemistry stainings of AV specification markers through the human protein atlas (HPA) (Extended Data Figure 16).

For Figure 4 – Alteration of CNS specificity

- additional IMC and immunofluorescence stainings in fetal, adult and pathological tissues to validate the expression of CNS signature and BBB markers (Figure 4, Extended Data Figures 22,27).

For Figure 5 – Upregulation of MHC class II receptors

- additional immunofluorescence and IMC stainings in fetal, adult and pathological tissues to validate the expression of MHC class II signature markers (Figure 5, Extended Data Figures 29,30).

For Figure 6 – A key role for ECs in the brain neurovascular unit

- additional IMC stainings in fetal, adult and pathological tissues to validate the co-localization/physical proximity of MHC class II receptors in ECs and PVCs (Figure 6, Extended Data Figures 37).

In summary, the additional validation experiments using IF and IMC confirm the transcriptional differences highlighted in main figures 1-6 and related Extended Data Figures and Supplementary Figures.

COMMENT:

Taken together, the main novelty provided by this paper is the fresh tissue source used in the adult samples, and the inclusion of fetal samples, to derive the CNS endothelial cell molecular atlas. On this point, it will be useful to compare some of the published NucSeq data to highlight the difference/similarity of the results between these two types of samples and methods. Major limitations are the number of fetal and diseased samples, the age diversity of both the adult and fetal samples, and the validation of key transcripts amongst the undiseased CNS tissue samples.

ANSWER:

We thank the reviewer for this comment. Reviewer 3 made a similar comment and specifically mentioned the paper Yang et al, *Nature*, 2022 (<https://www.biorxiv.org/content/10.1101/2021.04.26.441262v1.full>).

Moreover, based on our e-mail exchange with Dr. George Caputa of the 27-1-2022 and the 28-1-2022 after the paper Winkler et al., was published in *Science* on the 27-1-2022³, we agreed that it would be interesting to compare our AVM and temporal lobe datasets with the AVM and temporal lobe datasets presented in Winkler et al., *Science*, 2022 .

Accordingly, in order to account for the reviewer's and editor's requests regarding published sc/nuc-seq data, we have now compared both our adult/control dataset (undiseased atlas) as well as our pathological dataset (diseased atlas) to recently published datasets.

For every comparison, we performed two analyses:

- i) first, to compare different cell types (ECs and PVCs), we first compared the different cell type clusters (EC and PVC clusters) in our unsorted dataset with the EC and PVC clusters in the unsorted published datasets
- ii) second, to compare EC clusters, we compared our sorted EC dataset with EC clusters of the unsorted published datasets:

Comparison of our adult/control EC datasets (undiseased atlas) to published Nuc-seq datasets:**Comparison of cell types (ECs and PVCs) in our unsorted adult/control brain/temporal lobe dataset with the EC and PVC clusters in the unsorted adult/control brain published datasets**

We have compared our adult/control EC datasets (undiseased atlas) to the following recently published datasets:

i) Winkler et al. A single-cell atlas of the normal and malformed human brain vasculature. *Science*, 2022³

DOI: 10.1126/science.abi7377

<https://www.science.org/doi/10.1126/science.abi7377>

The paper by Winkler et al., described the brain vasculature in AVMs and in adult/control brains (no sorting of brain ECs, no fetal brain ECs, no tumor ECs).

ii) Yang et al. A human brain vascular atlas reveals diverse mediators of Alzheimer's risk. *Nature*, 2022⁶

<https://doi.org/10.1038/s41586-021-04369-3>

The paper by Yang et al⁶ addressed the brain vasculature in Alzheimer's Disease and in post-mortem hippocampus and superior frontal cortex (no sorted ECs, no fetal brain ECs, no tumor ECs).

iii) Garcia et al. Single-cell dissection of the human brain vasculature. *Nature* 2022⁸

<https://doi.org/10.1038/s41586-022-04521-7>

The paper by Garcia et al.⁸ examined the brain vasculature in Huntington's disease, in post-mortem human prefrontal cortex and in fresh ex-vivo temporal lobes (no sorted ECs, no fetal brain ECs, no tumor ECs).

Comparison of PVC and EC cell types in our unsorted datasets with PVC and EC cell types/clusters in the unsorted datasets of the three papers revealed an overall similarity between the different studies, e.g. the corresponding PVC and EC cell types mapped to each other.

Notably, the mapping revealed that (Extended Data Figure 6):

- Our neurons mapped to neurons in the Garcia et al and Yang et al datasets, whereas our neurons mapped to “unassigned” in the Winkler et al. dataset and the neurons in the Winkler et al dataset (only very few cells, around 895 cells = 1.2 % of cells, while our adult/control brain dataset has 20,040 cells = 42.35 % of cells) mapped to fibroblasts in our dataset.
- Our oligodendrocytes mapped to oligodendrocytes in the three datasets.
- Our pericytes mapped to pericytes/mural/vascular cells in the three datasets (according to their annotation).
- Our smooth muscle cells were mapped to the smooth muscle cells/mural/vascular cells in the three datasets (according to their annotation).
- Our fibroblasts were mapped to fibroblasts
- Our endothelial cells mapped to endothelial cells in the three datasets
- Our immune cells mapped to immune cells in the three datasets (immune cells include microglia/macrophages and T-cells)
- Astrocytes/neural stem cells/neural progenitors in our dataset didn't map to cells in the three datasets.

In summary, whereas corresponding cells mapped to each other, we only observed minor discrepancies regarding neurons when comparing to the Winkler et al paper. (see Extended Data Figure 6), thereby validating our unsorted EC and PVC dataset as well as the mapping technique using Seurat⁹.

Comparison of EC clusters in our sorted adult/control brain/temporal lobe dataset with the EC clusters of the unsorted adult/control brain published datasets

Comparison of EC clusters in our sorted EC dataset with the EC clusters in the unsorted datasets of the three papers revealed an overlap of the main EC clusters along the AV-axis, namely arteries, capillaries, and veins (Extended Data Figure 6).

Notably, the mapping revealed that (Extended Data Figure 6):

- Our arterial ECs mapped to arterial ECs in the three datasets
- Our venous ECs mapped to venous ECs in the three datasets
- Our small caliber vessel ECs (including arterioles, capillaries, angiogenic capillaries, venules) mapped to capillary ECs in the three datasets
- Other EC clusters including (proliferating) stem-to-ECs, (proliferating) EndoMT mapped to capillary ECs

In summary, corresponding EC clusters mapped to each other (see Extended Data Figure 6), thereby validating our sorted EC dataset as well as the mapping technique using Seurat⁹.

Importantly, however, we would like to point out that our dataset of sorted ECs offers a higher resolution (with significantly more clusters / subclusters as compared to the three papers) and thus a more in-depth insight into human brain EC biology. This enhanced resolution of human brain EC biology is due to:

- i) the significant difference in the number of ECs analyzed (47,652 ECs from adult/control brains, all sorted ECs analyzed: 296,208) in our paper as compared to the three published papers
- ii) the process of CD31⁺ / CD45⁻ FACS sorting and thus enrichment of the rare EC cell population

Please find below an overview of differences in the number of ECs analyzed between our dataset and these three datasets / temporal lobes .

- Our dataset
 - 47,652 sorted adult/control brain ECs, all sorted ECs analyzed: 296,208.
 - 12 EC clusters for the temporal lobe (overall 14 EC clusters)
- Yang et al., *Nature*, 2022
 - 24,982 non-sorted adult/control brain ECs (dextran-based removal of the myelin from cortex and hippocampus)
 - post-mortem samples (cortex and hippocampus)
 - 3 EC clusters

- Garcia et al., *Nature*, 2022
 - 4,345 non-sorted adult/control brain ECs
 - post-mortem samples (only cortex)
 - 3 EC clusters
 - 2,213 non-sorted adult/control brain ECs
 - From human temporal lobes
 - 3 EC clusters
- Winkler et al., *Science*, 2022
 - 5,018 non-sorted adult/control brain ECs
 - fresh surgical samples (neocortical samples overlying temporal lobectomies)
 - 6 EC clusters

In summary, we would like to emphasize that our dataset provides by far the highest number of adult/control brain endothelial cells, resulting in a more adequate description of the different EC clusters due to the higher resolution. This is illustrated by the fact that our dataset provides numerous EC clusters for every group of ECs (e.g. arteries, capillaries, veins) along the arteriovenous tree (Figure 2, Extended Data Figure 11).

Moreover, we utilized fresh surgical and fetal samples and performed CD31⁺/CD45⁻ FACS sorting for endothelial cells to enrich this rare cell population, which is the most appropriate method to address the underlying EC biology in our opinion. Accordingly, whereas all three groups have analyzed significantly less ECs than we did, the comparisons to Yang et al. and Garcia et al. are additionally limited because both groups did not use sorting of ECs with Yang et al using only post-mortem samples and Garcia et al using post-mortem and fresh samples. Meanwhile, the comparison to Winkler et al is limited given that this group did not perform EC sorting.

Finally, while the comparison to the published datasets validated our unsorted EC and PVC datasets as well as our sorted EC datasets, the advantage coming from the sorting of ECs (and thus the enrichment of this rare cell population) and the significant differences in the number of analyzed ECs between our and the above-mentioned papers emphasizes the importance of our methodological approach and large dataset.

Comparison of our pathological EC datasets (diseased atlas) to published sc/nucseq datasets:

We have compared our pathological EC datasets (diseased atlas) to the following recently published datasets:

For brain AVMs

- iv) Winkler et al. A single-cell atlas of the normal and malformed human brain vasculature. *Science*, 2022¹⁰

<https://www.science.org/doi/10.1126/science.abi7377>

As mentioned above, the paper by Winkler et al.¹⁰, described the brain vasculature in AVMs and in adult/control brains (no sorting of brain ECs, no fetal brain ECs, no tumor ECs).

Comparison of cell types (Ecs and PVCs) in our unsorted brain AVM dataset with the EC and PVC clusters in the unsorted brain AVM published datasets

Comparison of PVC and EC cell types in our unsorted datasets with PVC and EC cell types in the unsorted datasets of the Winkler et al paper revealed an overall similarity between the different studies, e.g. the corresponding PVC and EC cell types mapped to each other.

Notably, the mapping revealed that (Extended Data Figure 6):

- Our pericytes mapped to fibromyocytes (FBMCs)
- Our endothelial cells mapped to endothelial cells
- Our immune cells mapped to immune cells as follows:
 - Microglia/macrophages mapped to myeloid cells
 - Neutrophils mapped to myeloid cells
 - T-cells mapped to T-cells
 - B-cells mapped to B-cells
 - NK-cells mapped to T-cells
- Smooth muscle cells mapped to smooth muscle cells (SMCs) and fibromyocytes (FBMCs)
- Fibroblasts mapped to fibroblasts

- Oligodendrocytes mapped to oligodendrocytes

In summary, whereas corresponding cells mapped to each other, we only observed minor discrepancies (see Extended Data Figure 6), thereby validating our unsorted EC and PVC dataset as well as the mapping technique using Seurat⁹.

Comparison of EC clusters in our sorted brain AVM dataset with the EC clusters of the unsorted brain AVM published datasets

Comparison of EC clusters in our sorted EC dataset with the EC clusters in the unsorted dataset of the Winkler et al paper revealed an overlap of the main brain AVM EC clusters along the AV-axis, namely arteries, capillaries, and veins (Extended Data Figure 6).

Notably, the mapping revealed that (Extended Data Figure 6):

- Our large artery and arterial ECs mapped to arterial ECs
- Our arteriole ECs mapped to various cells as nidus/venule/venous ECs
- Our venous (large veins, veins, venules) ECs mapped to venous ECs
- Our angiogenic capillaries/capillaries mapped to various cells as nidus/venule/venous ECs
- Other EC clusters including (proliferating) stem-to-ECs, (proliferating) EndoMT were not mapped (unassigned).
- Regarding the pathological nidus cluster in brain AVMs in the paper Winkler et al.
 - We have identified MHC class II expression in large>small caliber vessel ECs.
 - In our dataset however, these MHC class II expressing ECs clustered according to the AV-specification
 - These MHC class II expressing ECs are likely attributed to the pathological nidus of arteriovenous malformations.
 - We, however, prefer not to label them as "nidus ECs" as the "nidus" is an angiographically/anatomically defined macroscopic entity and it's not possible to conclusively attribute scRNA-seq EC clusters to the anatomical "nidus".

In summary, whereas corresponding EC clusters mapped to each other, we only observed minor discrepancies (see Extended Data Figure 6).

Importantly, however – and similar to what we mentioned above for the adult/control brain/ temporal lobes (see above) – we would like to point out that our dataset of sorted ECs offers a higher resolution (with significantly more clusters / subclusters as compared to the Winkler et al paper) and thus a more in-depth insight into human brain EC biology. This enhanced resolution of human brain EC biology is due to:

- i) the significant difference in the number of ECs analyzed (20,305 in our paper as compared to the Winkler et al paper)
- ii) the process of CD31⁺/CD45⁻ FACS sorting and thus enrichment of the rare EC cell population

Please find below an overview of differences in the number of ECs analyzed between our dataset and the Winkler et al brain AVM dataset.

- Our dataset
 - 20,305 sorted brain AVM ECs
 - 12 EC clusters for the AVM ECs (overall 14 EC clusters)
- Winkler et al., *Science*, 2022
 - 4,601 non-sorted brain AVM ECs
 - fresh surgical samples
 - 6 EC clusters

Similar to the comparison of adult/control brain ECs (see above), we would like to point out that given the significant difference in the number of brain AVM ECs analyzed (20,305 sorted ECs in our paper (Supplementary Table 2) versus 4,601 ECs in the Winkler et al paper), our dataset on sorted ECs offers a higher resolution and thus more in-depth insight into human brain AVM EC biology.

In summary, we would like to emphasize that our dataset provides a higher number of brain AVM endothelial cells, resulting in a more thorough description of the different EC clusters due to the higher resolution. This is illustrated by the fact that our dataset provides numerous EC clusters for every

group of ECs (e.g. Large arteries, arteries, arterioles, angiogenic capillaries, capillaries, venules, veins, large veins, EndoMT, proliferating cells) along the arteriovenous tree (Figure 2, Extended Data Figure 11).

Moreover, we referred to fresh surgical and fetal samples and performed CD31⁺/CD45⁻ FACS sorting for endothelial cells to enrich this rare cell population, which is the most appropriate method to address the underlying EC biology in our eyes. Accordingly, whereas Winkler and colleagues have analyzed significantly less ECs than we did, the comparison to Winkler et al is also limited given that this group did not refer to EC sorting.

Finally, while the comparison to the published datasets validated our unsorted EC and PVC datasets as well as our sorted EC datasets, the advantage coming from the sorting of ECs (and thus the enrichment of this rare cell population) and the significant differences in the number of analyzed ECs between our and the Winkler and colleagues' paper emphasizes the importance of our methodological approach and large dataset.

For GBMs

i) Neftel et al. An Integrative Model of Cellular States, Plasticity, and Genetics for Glioblastoma. *Cell*, 2019¹¹

10.1016/j.cell.2019.06.024

The paper by Neftel et al.¹¹ addressed glioblastoma cellular states, plasticity and genetics (no sorted ECs, no fetal brain ECs, no tumor ECs).

- Our dataset
 - 38,257 sorted GBM ECs
 - 14 EC clusters for GBMs (overall 14 EC clusters)
- Neftel et al., *Cell*, 2019
 - no ECs identified

Comparison of cell types (ECs and PVCs) in our unsorted GBM dataset with the EC and PVC clusters in GBM published datasets

Comparison of cell types (ECs and PVCs) in our unsorted GBM dataset with the EC and PVC clusters in the unsorted brain GBM published dataset of Neftel et al., *Cell*, 2019 revealed that (Extended Data Figure 6):

- Our pericytes mapped to malignant cells
- Our endothelial cells mapped to malignant cells
- Our smooth muscle cells mapped to malignant cells
- Our fibroblasts mapped to malignant cells
- Our immune cells mapped to immune cells as follows:
 - Microglia/macrophages mapped to macrophages
 - Neutrophils mapped to macrophages
 - T-cells mapped to T-cells
- Fibroblasts mapped to malignant cells
- Tumor cells and tumor stem cells mapped to malignant cells
- Oligodendrocytes mapped to oligodendrocytes

In summary, whereas oligodendrocytes, immune cells and tumor cells mapped to each other, we observed discrepancies as vascular cells were not identified in the Neftel et al., *Cell* 2019 dataset (see Extended Data Figure 6), thereby validating our unsorted EC and PVC dataset as well as the mapping technique using Seurat⁹.

ii) Richards et al., Gradient of Developmental and Injury Response transcriptional states defines functional vulnerabilities underpinning glioblastoma heterogeneity. *Nat Cancer*, 2021

DOI: 10.1038/s43018-020-00154-9

The paper by Richards et al.¹¹ addressed glioblastoma transcriptional heterogeneity (no sorted ECs, no fetal brain ECs, no tumor ECs).

- Our dataset

- 38,257sorted GBM ECs
- 14 EC clusters for GBMs (overall 14 EC clusters)
- Richards et al., *Nat Cancer*, 2021
 - no ECs identified in the sc-RNAseq
 - 316 endothelial nuclei in the snuc-RNAseq dataset

Comparison of cell types (ECs and PVCs) in our unsorted GBM dataset with the EC and PVC clusters in the unsorted brain GBM published dataset of Richards et al., *Nat Cancer*, 2021_revealed that (Extended Data Figure 6):

- Our pericytes mapped to malignant cells
- Our endothelial cells mapped to malignant cells
- Our smooth muscle cells mapped to malignant cells
- Our fibroblasts mapped to malignant cells
- our immune cells mapped to immune cells as follows:
 - Microglia/macrophages mapped to macrophages
 - Neutrophils mapped to macrophages
 - T-cells mapped to T-cells
- Fibroblasts mapped to malignant cells
- Tumor cells and tumor stem cells mapped to malignant cells
- Oligodendrocytes mapped to oligodendrocytes

In summary, whereas oligodendrocytes, immune cells and tumor cells mapped to each other, we observed discrepancies as vascular cells were not identified in the Richards et al., *Nat Cancer*, 2021 dataset (see Extended Data Figure 6), thereby validating our unsorted EC and PVC dataset as well as the mapping technique using Seurat⁹.

iii) Darmanis et al., Single-Cell RNA-Seq analysis of infiltrating neoplastic cells at the migrating front of human glioblastoma. *Cell Rep*, 2017¹¹

DOI: 10.1016/j.celrep.2017.10.030The paper by Darmanis et al.¹¹ addressed the migration front in human glioblastoma (no sorted ECs, no fetal brain ECs, no tumor ECs).

- Our dataset
 - 38,257sorted GBM ECs
 - 14 EC clusters for GBMs (overall 14 EC clusters)
- Darmanis et al., *Cell Rep*, 2017
 - identification of vascular cell,
 - however, no specific identification of ECs pericytes, fibroblasts and smooth muscle cells

Comparison of cell types (ECs and PVCs) in our unsorted GBM dataset with the EC and PVC clusters in the unsorted brain GBM published dataset of Darmanis et al., *Cell Rep*, 2017_revealed that (Extended Data Figure 6):

- Our pericytes mapped to vascular cells
- Our endothelial cells mapped to vascular cells
- Our smooth muscle cells mapped to vascular cells
- Our fibroblasts mapped to vascular cells
- Our immune cells mapped to immune cells as follows:
 - Microglia/macrophages mapped to myeloid cells
 - Neutrophils mapped to macrophages
 - T-cells were unassigned
- Fibroblasts mapped to malignant cells
- Tumor cells and tumor stem cells mapped to neoplastic cells
- Oligodendrocytes mapped to oligodendrocytes

In summary, whereas oligodendrocytes, immune cells and tumor cells mapped to each other, we observed discrepancies as our endothelial cells, pericytes, fibroblasts and smooth muscle cells all mapped to vascular cells as the resolution in Darmanis et al., *Cell Rep*, 2017 dataset is less than ours (see Extended Data Figure 6), thereby validating our unsorted EC and PVC dataset as well as the mapping technique using Seurat⁹.

For lung cancer brain metastasis (METs)

iv) Gonzalez et al. Cellular architecture of human brain metastases, *Cell*, 2022¹¹

[DOI: 10.1016/j.cell.2021.12.043](https://doi.org/10.1016/j.cell.2021.12.043)

The paper by Gonzalez et al.¹¹ addressed the cellular architecture of human brain metastases (no sorted ECs, no fetal brain ECs, no tumor ECs).

- Our dataset
 - 23,962 sorted MET ECs
 - 14 EC clusters for METs (overall 14 EC clusters)
- Gonzalez et al., *Cell*, 2022
 - 462 non-sorted ECs for lung cancer brain metastases

Comparison of cell types (ECs and PVCs) in our unsorted MET dataset with the EC and PVC clusters in MET published datasets

Comparison of cell types (ECs and PVCs) in our unsorted MET dataset with the EC and PVC clusters in the unsorted brain MET published dataset of Gonzalez et al, *Cell*, 2022 revealed that (Extended Data Figure 6):

- Our pericytes mapped to pericytes
- Our endothelial cells mapped to endothelial cells
- Our smooth muscle cells mapped to smooth muscle cells
- Our immune cells mapped to immune cells as follows:
 - Microglia/macrophages mapped to macrophages
 - B-cells mapped to B-cells
 - T-cells mapped to T-cells
- Fibroblasts mapped to malignant cells
- Tumor cells and tumor stem cells mapped to neoplastic cells
- Oligodendrocytes mapped to astrocytes

In summary, whereas, immune cells and tumor cells, endothelial cells and other vascular cells mapped to each other, we observed minor discrepancies as our oligodendrocytes mapped to astrocytes as Gonzalez et al, *Cell*, 2022¹² dataset didn't identify oligodendrocytes (see Extended Data Figure 6), thereby validating our unsorted EC and PVC dataset as well as the mapping technique using Seurat⁹.

Finally, we would like to emphasize that for all other entities except the adult/control ECs, the AVM ECs (Winkler et al., *Science*, 2022 and the MET ECs (Gonzalez et al., *Cell*, 2022) mentioned above, it is not possible to thoroughly compare our dataset of sorted ECs to what is currently published in the literature. The main reason is the absence of – computationally extracted ECs / vascular cells or the lack of identification of ECs in the vascular compartment (for instance in GBMs in Richards et al, *Nature Cancer*, 2021¹³, Neftel et al., *Cell*, 2019¹⁴), limiting the scientific value of such comparisons given the significant difference in cell number and techniques (which is sorted ECs versus computationally extracted ECs).

Additional validation experiments

Furthermore, we have addressed all major limitations mentioned by the reviewer by

- i) performing bulk RNA-seq of key adult/control and diseased entities (fetal brain samples were included in all additional analyses and validation experiments) to address the concern about the limited sample number
- ii) analyzing the age and sex diversity amongst the adult samples and age diversity for the fetal samples (see Extended Data Figures 12,22,29 for fetal brain and Figures to the Reviewer 1,3, Supplementary table 10 for age and sex for the adult/fetal samples).
- iii) providing additional validation experiments of key transcripts amongst the undiseased, diseased, and fetal CNS tissue samples using bulk RNA-sequencing, additional immunofluorescence stainings, IMC and spatial transcriptomics (Extended Data Figures 9,13,14,15,22,27,29,30,32,37).

According to our discussion with Dr. George Caputa on the 25-1-2022, we agreed that in light of the overall very significant number of samples, the high number of single-cells and the considerable amount of additional costs and time involved, we would not perform additional single-cell RNA-sequencing experiments.

Major Points

COMMENT:

- As noted above, overall inclusion of different diseased tissues include clustering of distinct tumor types. Analysis of individual diseases is limited by sample number.

ANSWER:

We thank the reviewer for this comment. As mentioned above, we have performed computational analyses at various levels

for the overall merge including the fetal brain, the adult brain, and all disease types. We have also analyzed all the individual entities (fetal brain, adult/control brain, including all diseases) separately (Figures 2b-d, 2g-j, Extended Data Figures 5,11,19,21,24,25,26,28,33, Supplementary Figures 24-27,41). Moreover, we have performed computational analyses for every single patient (Extended Data Figures 5u,11w,x).

Therefore, one key message in the six main figures of our paper is to describe the transcriptional changes that are common across different disease entities (thereby identifying "hallmarks" of diseases). However, of equal importance are the transcriptional characteristics of every individual disease, which we display in Extended Data, and Supplementary, Reviewer's Figures providing easy accessibility for readers who are interested in particular disease properties.

According to our discussion with senior editor Dr. George Caputa on the 25-1-2022, we agreed to keep all the different fetal, adult, and diseased tissue in the paper. We feel that the comparison between the fetal brain endothelium, the adult/control brain endothelium, and the diseased/pathological brain endothelium is especially interesting and constitutes one of the key messages of this manuscript. For instance, the reactivation of fetal pathways in brain pathologies is very interesting. This being said, to address the editor's and reviewer's concerns, we have now further clarified which findings and transcriptomic changes are conserved among the different disease entities while also emphasizing findings that are more specific to neurovascular entities, namely brain vascular malformations versus brain tumor/neoplastic entities. The observation that we find commonly dysregulated pathways, alterations in AV-differentiation patterns and CNS-specificity, MHC class II upregulation and changes in cell-cell interactions in ECs (and PVCs for the cell-cell interaction patterns) across different disease entities is novel and interesting in our eyes and thereby constitutes a "hallmark" of the diseased brain vascular endothelium.

To address the reviewer's and editor's related concern about oversimplification in that regard, we would like to mention that we have provided the detailed results for the different scientific findings at multiple levels of analysis and comparison, namely:

- i) for differential expression analysis (for ECs), pathway analysis (for ECs) and ligand-receptor interaction analysis (for ECs and PVCs), we provide the following results:

- for fetal brain versus adult/control brain (see Figure 1s,t, Figure 6a-c,j,o-p, Extended Data Figures 7d-f, 8d,17,18,34a,35,36, Supplementary Figures 17,20,23,26a,31,35-38,40).
- for pathological brain versus adult/control brain (see Figure 1t,u, Figure 6g-i,k,s,t, Extended Data Figures 7g-l, 8d,17,18,31,34c,35,36, Supplementary Figures 16,18,22,26c,30,32-34,39).
- for brain tumor versus adult/control brain (see Extended Data Figures, 33 a'-e',34e, Supplementary Figure 3,24II,25II,26e,41II).
- for brain vascular malformation versus adult/control brain (see Extended Data Figures 33a-e, 34d, Supplementary Figures 2,24I,25I,26d,41I).
- for every single pathological/disease entity versus adult/control brain (Extended Data Figures 9,33,34f-m, Supplementary Figures 7-14,24III-X,25III-X,29,41III-X)

- ii) EC and PVC clustering, we provide the following results:

- for all fetal, adult, and pathological entities pooled together (see Figure 2, Extended Data Figure 4)
- for every single brain entity
 - for fetal brain
 - for adult/control brain (TL)

for every single pathological/disease entity separately (TLadjCAV, CAV, AVM, HEM, LGG, GBM, MET, MEN), see Extended Data Figures 5,11.

- for every individual patient cell composition/AV annotation, please see Extended Data Figures 5u,11w,x.

Thus, while we have indeed clustered/merged the diseased samples together, we also provide the clustering/merge of every individual disease entity, and these different levels of comparisons complement each other to derive the respective scientific conclusions (Figure 2, Extended Data Figures 5,11).

Regarding the limited sample size, we would like to point out that while the number of samples is reasonable (mostly $n=5$ per disease entity), the number of sorted ECs per sample and per entity is actually very high. We agreed in our discussion with senior editor Dr. George Caputa on the 25-1-2022 to not include further scRNA-seq samples in this study for reasons of feasibility and timeliness. However, to address the reviewer's and editor's comment, we have mentioned the limitation of the relatively small sample number in the paragraph "Discussion". Moreover, to further support the key findings of our single-cell RNA-seq atlas, we have now also added bulk RNA-seq data of adult/control brain ECs and various diseased brain ECs (Extended Data Figure 9, Supplementary Figure 14) and these data confirm the main findings of the scRNA-seq analyses. The description of these newly obtained results as well as their description and comparison to the scRNA-seq data has been added throughout the text.

The results of the bulk RNA-seq experiments can be found in Extended Data Figures 9 / Supplementary Figures 14 and this additional technique has now -together with IMC also been added to the working model (see Figure). Moreover, we added a new Supplementary Table 1 with all clinical / patient information of the bulk RNA-seq experiments.

Taken together, one purpose of the manuscript is to address common transcriptomic pathological changes in various vascular-dependent brain diseases (in addition to describing the disease-specific pathological changes) and we feel that the observed changes in transcriptomic profiles of endothelial cells across various vascular-dependent brain pathologies is a key finding of our study. Moreover, the precise description of the transcriptomic profiles of endothelial cells in the brain vasculature of the different brain diseases provides an important publicly available molecular atlas that can serve as a reference for future studies regarding the developing, adult, and diseased brain vasculature.

Moreover, we would like to emphasize that despite the relative low number of samples by entity, the strength of our atlas lies in the overall number of fetal, healthy, and diseased samples as well as in the number of sorted endothelial as well as unsorted endothelial and perivascular cells overall as well as per disease entity. This assembly of various fetal, healthy and diseased brain entities allows for comparisons which are impossible to make without such a dedicated comparative atlas.

We have also added these aspects outlined above to the paragraph "Discussions".

COMMENT:

- Given the different origins of the ECs across conditions, the authors should briefly describe the methods used to generate their merged dataset. CCA and RPCA were performed to integrate data across samples. Was ambient RNA correction also performed to minimize bias? Such an analysis may help eliminate the possibility that some observed differences are due to differences in brain regions where the samples are isolated, heterogeneity in the isolation method or differences among individuals.

ANSWER:

We have described the methodology regarding the computational algorithms used to generate the merged datasets that include CCA, RPCA, scANVI and harmony in the paragraph "methods – Chromium 10X library preparation, single cell RNA-sequencing and data analysis". Reviewer 3 also commented about the different methods of integration, and we have compared the different integration methods (Figure to the Reviewer 2). RPCA, harmony, scANVI and CCA showed very comparable integration. We also plotted the RPCA annotated clusters onto the harmony, scANVI and CCA integrated objects and the results show that cells indeed cluster according to their AV specification (AV-annotation were very consistent) as we have observed with the RPCA integrated cells in the submitted manuscript (Figure 2). Additionally, as per the reviewer's request, we performed ambient RNA analysis for both the sorted and unsorted samples using the SoupX method

(<https://academic.oup.com/gigascience/article/9/12/gjaa151/6049831>) and found that for all samples, the level of ambient RNA across fetal, adult, and diseased brain entities was below the acceptable threshold of 10% (Extended Data Figure 3s). Accordingly, we conclude that the likelihood that the observed differences are due to differences in brain regions from where the samples were isolated or due to differences among individuals is very low and that the results we show therefore represent real biological differences/findings. The brain regions from which the samples were isolated from are displayed in Supplementary Table 1).

COMMENT:

- There are limitations on the generalizability of the results in the adult age group based on the age and sex distribution amongst the healthy control (temporal lobe) samples. An emphasis is made in the paper that the fetal brain is compared to the adult brain to model the "development" of the CNS endothelium, but the age range of the adults is 15-69. There is concern that 15 years-old is widely not considered to be "adult aged", and that there are likely subtle differences between patient samples in their 20s and 40s compared to the aged adult sample that was 69. The age distribution should be addressed either as a significant limitation of the atlas, or the adult ages divided into age ranges and analyzed separately, which would require additional samples. Similarly, one of the fetal ages is 9 weeks while another is 18. There is concern that grouping two disparate ages during a period of rapid neurodevelopment would result in inaccurate representation of the fetal BBB.

ANSWER:

We fully agree that the parameters "age" and "sex" are important in the fetal, adult/control and diseased brain entities.

Accordingly, we have performed the analysis addressing the parameters age and sex distribution by computing differentially expressed genes (DEGs) as well as running scCODA analysis to check if there are significant cell type compositional changes. In summary, the results indicate no significant differences in the differential expression or cell compositional changes between neither age nor sex distribution and are displayed in Figures to the Reviewer 1, 3, and supplementary tables 10,20. We would like to emphasize that for both parameters and scientific questions, future dedicated studies will be needed to thoroughly address them (see below).

Regarding the age range of the adult/control temporal lobe samples, we have specifically examined the youngest age (15 years) and the oldest age (69 years) and the results show that these two are not outliers (Figure to Reviewers 1,3 and supplementary table 10,20). In addition, we have found that there are no statistical differences in the patient samples in their age range between their 20's and 40's, indicating that age doesn't have an effect in our dataset (Figure to Reviewers 1,3 and supplementary table 10,20).

Accordingly, we conclude that we do not have enough samples to address the question regarding age distribution and thus a dedicated future study to address this question is required.

Regarding the sex distribution, we have performed differential expression analysis between ECs of female and male patients for every single entity including, adult, and diseased entities/brains. Overall, the number of significant DEGs between the two sexes are low and is comparable to random shuffles, also scCODA analysis showed no significant compositional changes. These results are displayed in Figure to Reviewers 1,3 and supplementary table 10). As mentioned previously, we conclude that we do not have enough samples to address the question regarding sex differences and thus a dedicated future study to address this question is required.

In agreement with the senior editor George Caputa in our call on the 25-1-2022, we decided that whilst the questions regarding age and sex raised by the reviewer are scientifically interesting, we will not add further samples in this study as a thorough investigation of age and sex differences in the different entities merits stand-alone studies for every single entity. Therefore, the questions regarding age and sex cannot be conclusively addressed in this study. Accordingly, we have mentioned in the manuscript (paragraph "Discussion") that the age distribution of the samples in the different entities consists in a significant limitation of the atlas and needs to be addressed in the future.

In summary, when addressing the parameters age in TLs and sex in all entities we found few DEGs that did however not result in biologically meaningful pathways. We are well aware of the limited sample size and have mentioned this in the paragraph "Discussion".

Regarding the human fetal BBB, a similar question was asked by reviewers 3, who also required a

more thorough analysis according to the timeline of the human fetal BBB. To address the reviewer's concern regarding the development of the fetal brain vasculature, we have thus performed additional analyses for the fetal brain samples at the different Gestational Weeks (GWs) 9 to 18.

Notably, we addressed arteriovenous specification, CNS-specificity and blood-brain-barrier (BBB) markers as well as MHC class II markers (Extended Data Figures 12,22,29), moreover scCODA analysis showed there is no significant compositional changes between the different GSWs analyzed (Supplementary table 10). The results show that the key conclusions we make when comparing to other brain entities (adult/control and disease entities) are valid and are not mainly influenced by the age of the different fetal brain samples. For instance, the conclusions made regarding the arteriovenous specification and the cellular/cluster composition including the significantly increased amount of angiogenic capillaries in the fetal brain as compared to the temporal lobe hold true for all the different fetal brains at different GWs (Extended data figure 12), illustrated also by the scCODA analyses, (see Figure to Reviewer 3, supplementary table 10). Thus, the key conclusions made for the AV-specification when comparing the pooled fetal and pooled adult brain ECs reflect the AV-specification at the different gestational ages in an accurate manner.

To further address the development of the fetal BBB, we have indicated EC expression levels of the BBB markers Cadherin 5 (CDH5), Tight Junction Protein 1 (TJP1) = Zona occludens 1 (ZO1), Occludin (OCLN) and Claudin 5 (CLDN5) which show an increased expression with increased gestational age, indicating the temporal development of the BBB in the human fetal brain (see Extended Data Figure 22). Furthermore, we have added the expression of the fetal CNS and BBB signatures at the different gestational ages GWs (see Extended Data Figure 22) indicating the establishment of the CNS signature with increasing gestational age. Moreover, according to a request made by reviewer 3, we have also included the expression levels of the BBB markers and the CNS and peripheral signature according to the AV-specification (sub)clusters. The results show the gradual increase in expression of BBB genes with gestational age (see Extended Data Figure 22).

While the conclusions drawn in our manuscript are valid, we agree with the reviewer that the precise examination of the different gestational ages during neurodevelopment are important and need to be addressed in future studies to most accurately describe the human fetal BBB. This scientific question can, however, not be addressed in this atlas but requires a dedicated follow-up study. We have also mentioned the limitations of the sample number and the need for future investigations of the human fetal brain vasculature and blood-brain-barrier and described these findings in the paragraph "Discussion".

In conclusion and in agreement with the senior editor George Caputa in our call on the 25-1-2022, while we provide these additional analyses as Figures to the Reviewer's we would like to emphasize that the questions regarding age and sex distribution in the different physiological and pathological brain entities as well as regarding the human fetal brain vasculature need to be further addressed in depth in dedicated separate follow-up studies.

COMMENT:

- Gliomas are grouped by IDH1 status (wild-type), but also contain differences in MGMT status. This results in the clustering of what we know are phenotypically distinct tumors, that have ultimately distinct clinical features. This work is an oversimplification of single-cell sequencing that has already been described in GBMs (There are many publications, but Neftel et al 2019 Cell, for example). The same is true of the metastatic disease samples, which likely include distinct angiogenic features depending on the specific neoplasm.

ANSWER:

We thank the reviewer for this comment and we agree that an objective analysis on the IDH1 and MGMT status is of added value. We would like to emphasize that one aim of our study was to find commonalities across pathologies (brain tumors and brain vascular malformations) rather than dissecting differences among different types of brain tumors. The detailed comparison of the vasculature of different types of brain tumors warrants future studies dedicated to this specific question. We have now also mentioned this limitation and the request for future follow-up studies in the paragraph "Discussion".

Nevertheless, to respond to the reviewer's comment regarding the MGMT status of gliomas, we have performed scCODA and differential expression analysis of endothelial cells derived from MGMT methylated versus MGMT non-methylated glioblastomas samples. We do not find any differentially expressed genes significantly different from what is expected based on random change.

(see Supplementary Table 10,20 and Figures to the Reviewer 1,3).

- i) analysis using pseudobulking, shuffling and differential expression using DE-seq (converting single-cell RNA-seq to bulk RNA-seq) data
- ii) scCODA to address the compositional changes.

The results of the differential expression analysis show that the differences observed are no different than random shuffles (differences) (see Supplementary Table 20 and Figures to the Reviewer 1). Moreover, scCODA analysis showed that there is no significant cell type compositional differences between the MGMT methylated vs non-methylated samples. We would like to mention that the interpretation of these results is limited by the small sample number and would need to be addressed in a separate, dedicated follow-up study comparing MGMT status in high-grade gliomas. In summary, when addressing the MGMT status in GBMs, and specific neoplasms in lung cancer METs, we found few DEGs that did however not result in biologically meaningful pathways. We are well aware of the limited sample size and have mentioned this in the paragraph "Discussion". Regarding the IDH1 mutation status of gliomas, we would like to point out that all low(er)-grade glioma samples are IDH1 mutant, except one sample that is IDH2 mutant (IDH2 mutations are mutually exclusive with IDH1 mutations¹⁵), see Supplementary Table 1) whereas all high-grade glioma samples are IDH1 wild-type (see Supplementary Table 1) and thus our analyses of low(er)-versus high-grade gliomas (Supplementary Figure 6) already account for this IDH1 mutation status comparison. Moreover, according to the new WHO classification¹⁶ published in August 2021, the presence of IDH1 mutation equals/implies the diagnosis of WHO grade II or III glioma whereas the absence of IDH1 mutation indicates a WHO grade IV glioblastoma, which further validates the comparisons between IDH1 mutant low(er) grade and IDH1 wild-type high-grade / glioblastomas described above. Finally, IDH1 mutant high-grade gliomas / glioblastomas are very rare and account for only approximately 3.7%-5% of glioblastomas (GBMs)¹⁵. Thus, in order to address a comparison of IDH1 mutant versus IDH1 wild-type glioblastomas (according to the old WHO classification of 2016), a much higher sample size would be required that can only be achieved in a separate dedicated follow-up study.

Therefore, given the necessity for future follow-up studies dedicated to answer the questions regarding the IDH and MGMT mutation status, and in agreement with senior editor Dr. George Caputa (in our discussion on the 25-1-2022), we decided to not collect more glioma samples.

We would like to point out that the paper mentioned by the reviewer, Neftel et al., *Cell*, 2019¹⁴ is indeed a great scientific paper. However, the authors did not address the glioma vasculature, and thus this paper is very different from our story. Nevertheless, we have analyzed the dataset in Neftel et al., *Cell*, 2019¹⁴ and were not able to identify/find endothelial cells in the dataset (https://portals.broadinstitute.org/single_cell/study/SCP393/single-cell-rna-seq-of-adult-and-pediatric-glioblastoma).

To additionally point out the value and novelty of our study that focuses on FACS sorted ECs, we have further analyzed another recently published GBM scRNA-sequencing dataset¹³. Similarly to what we observed in Neftel et al., *Cell*, 2019¹⁴, we were not able to identify/find endothelial cells in the dataset (https://singlecell.broadinstitute.org/single_cell/study/SCP503).

We further analyzed the publication Darmanis et al., *Cell Rep*, 2017¹⁷ and the mapping of our data to this dataset showed that our endothelial cells, pericytes, smooth muscle cells and fibroblasts mapped to their "vascular cells" however the resolution in this paper didn't segregate between the different vascular cells, thus validating the mapping methodology used for all three papers.

The mapping of our dataset to the Neftel et al., *Cell*, 2019¹⁴, Richards et al., *Nat Cancer*, 2021¹³ and Darmanis et al., *Cell Rep*, 2017¹⁷ is shown in Extended Data Figure 6.

In summary, we would like to emphasize that the brain tumor vasculature (including gliomas, metastases, meningiomas, and other brain tumors) including ECs and PVCs has – to the best of our knowledge – never been specifically addressed at the single-cell level so far.

We would like to point out that we are the first to look at the brain vasculature and at sorted ECs at the single-cell level in the fetal brain, in adult/control brain, and in brain tumors (including GBMs and metastases) and brain vascular malformations.

Regarding the lung cancer metastases samples and the question about differences between specific neoplasms, we would like to clarify again that we exclusively analyzed lung cancer brain metastases.

We agree that an objective analysis on the specific neoplasms is of added value. Accordingly, regarding the specific neoplasm of lung cancer brain metastases, we have performed differential analysis of lung adenocarcinomas (n=3), small-cell lung cancers (n=1), and poorly differentiated lung cancers of unknown subtype (n=1) samples in the lung cancer brain metastases (see Supplementary Table 10,20 and Figures to the Reviewer 1,3).

We have performed the following analysis of data with low n numbers:

- i) analysis using shuffling and pseudobulking (converting single-cell RNA-seq to bulk RNA-seq) data
- ii) scCODA

The differential expression analysis results show that the differences observed are no different than random differences, moreover the scCODA analysis shows no significant compositional differences (see Supplementary Tables 10,20 and Figures to the Reviewer 1,3). We would like to mention that the interpretation of these results is limited by the small sample number and would need to be addressed in a separate, dedicated follow-up study comparing the different specific neoplasms in lung cancer brain metastases.

In our call with senior editor Dr. George Caputa on the 25-1-2022, we agreed that addressing the questions regarding

- the MGMT status in glioblastomas
 - as well as the specific neoplasms in lung cancer metastases
- goes beyond the scope of this paper and will require dedicated follow-up studies. We thus agreed that within the frame of this paper, we will not to add more samples for scRNA-sequencing.

COMMENT:

- Figure 5: MHC II expression on human endothelial cells is a key finding in this paper. The immunofluorescence validation of CD74 is not convincing (fig 5i-p), and additional assays for validation would be needed to support this conclusion. The expression on AVMs appears cytoplasmic, however, in the healthy adult the immunofluorescence appears less convincing, and either nuclear or artifactual. Additionally, this figure should be inclusive of fetal tissue since this is an important comparator in the results section and discussion.

ANSWER:

We agree with the reviewer that the MHC class II expression on human brain endothelial cells is a key finding of our study.

We have defined an MHC class II signature (Figure 5d, see methods) and have validated multiple transcripts of this MHC class II signature including CD74 (Figure 5, Extended Data Figures 29,30,37), which is displayed in Main Figure 5 and in Extended Data Figures 29,30. In addition, we have validated various other proteins of the MHC class II signature (Figure 5d), namely HLA-DPA1 (Figure 5, Extended Data Figures 29,30,37), HLA-DRA (Figure 5, Extended Data Figures 29,30), as well as HLA-DRB5 (Figure 5, Extended Data Figures 29,30,37).

Regarding the CD74 immunofluorescence, we now provide additional immunofluorescence as well as IMC stainings in temporal lobe (TL) and the different disease entities (Extended Data Figures 29,30). The expression of MHC class II proteins including CD74 was overall low in TL in both immunofluorescence (Figure 5j,j', Extended Data Figure 30l k₁-l₁') and RNAscope (Extended Data Figure 30l y₁-z₁), but was clearly present. The RNA scope stainings of CD74 in the TL show low expression in some endothelial cells which is also cytoplasmic.

Accordingly, to respond to the reviewer's request to include fetal tissues, we have now performed immunofluorescence as well as IMC validation for MHC class II signature markers/transcripts that are expressed in fetal brain ECs, namely CD74, HLA-DRs and HLA-oligo-Ds (Figure 29,30). Moreover, we have performed IF staining on two peripheral tissues (GI tract and kidney) in fetal tissue.

We observed the following expression patterns

- The MHC class II signature markers HLA-DRA and HLA-DPA1 showed some expression in fetal brain ECs and higher expression in peripheral ECs (fetal GI tract and kidney ECs, with strongest expression in GI tract villi ECs, Extended Data Figure 30II)
- CD74 and HLA-DRB5 only showed rare expression in fetal brain ECs but were expressed in peripheral ECs (fetal GI tract and kidney ECs, with strongest expression in GI tract villi ECs, Extended Data Figure 30II).

- In summary, we observe that MHC class II signature transcripts are rarely expressed in fetal brain ECs (most fetal brain ECs don't express MHC class II transcripts) but are expressed in peripheral ECs. We also observe high MHC class II expression in non-ECs in peripheral tissues, for instance in the GI tract, indicating the antigen-expressing capacity/function of the GI-tract^{18,19}. The precise expression of MHC class II signature transcripts across different stages of fetal brain development needs to be addressed further in dedicated follow-up studies.

Minor Points

COMMENT:

- Fig 1 a-c – this information is helpful but could be streamlined; numbers of sorted/unordered endothelial cells are helpful and important, but overview could be condensed to make more room for data in the body of the manuscript

ANSWER:

We thank the reviewer for this comment and agree that the information presented in Figures 1a-c is helpful to outline the overview of the different samples and the methodological approach in this manuscript. The numbers of sorted endothelial cells and of unsorted endothelial and perivascular cells can also be found in Supplementary Tables 2,3.

While we feel that it is difficult to streamline the information displayed in Figure 1a-c (given that it already shows the essential aspects that cannot be further shortened, we have responded to the reviewer's request for making more room for data by moving Figure 1e-g,i to Extended Data Figure 1d-g while keeping Figure 1g in the main figure 1. (see "Minor Point" regarding Figure 1e,f,g and Extended Data Figures 3,4, 6a-c).

Thus, via adaptation of Figures 1e,f,g we have responded to both comments of this reviewer by:

- making more room for data
- removing anything that appears redundant

Based on the reviewer's request, we have also added the following data to Figure 1

- the different angiogenic signatures including
 - Human fetal/developmental CNS/brain signature
 - Human pathological signature
 - Human brain tumor signature
 - Human brain vascular malformation signature
- immunofluorescence validation of the two key angiogenic markers PLVAP and ESM-1.

COMMENT:

- Pathologic tissue is described as a "reactivated fetal" pathways in brain endothelium. It is likely more accurate to say that both pathways share a proliferation of less-differentiated cells, as there is no evidence to demonstrate that diseased cells start at with an adult transcriptional profile, and then activate fetal pathways to become less differentiated.

ANSWER:

We would like to clarify our definition of "reactivated fetal pathways".

Angiogenesis, the formation of new blood vessels, is highly dynamic during fetal and postnatal brain development, quiescent in the adult healthy brain, and again highly dynamic in the vascular-dependent brain pathologies such as brain tumors and brain vascular malformations²⁰. The similarity between fetal pathways and pathological pathways is referred to as "onco-fetal" axis and has been described previously

- in various brain and peripheral tumors^{13,21-25}
- as well as in endothelial cells in liver cancer²⁴, where onco-fetal reprogramming of endothelial cells drives immunosuppressive macrophages in hepatocellular carcinoma.
- for instance, Couturier and colleagues described that glioblastoma recapitulates a normal neurodevelopmental hierarchy and that normal brain development reconciles glioblastoma development²⁵.
- however, the similarity of activated pathways between the fetal brain and the pathological brain has never been described in ECs so far

Notably, in Figure 1s-u, we show that more than 50% of the pathways that are dysregulated in pathological brain ECs vs adult/control brain ECs are also upregulated in fetal brain ECs vs. adult/control brain ECs.

With regard to the proliferation of less-differentiated cells, we would like to refer to the bar graphs in Figures 2g-j that indicate the shared significant upregulation of the "angiogenic capillary" EC cluster in the fetal brain, the pathological brain, and in brain tumors. Moreover, we would like to point out that the mentioned "reactivated fetal pathways" were also observed in the EC-EC ligand-receptor interaction analyses in the angiogenic capillaries (and all other EC clusters) when comparing

- fetal brain vs adult/control brain (Figure 2o-q, Extended Data Figure 17, Supplementary figures 16,17,22,23)
- pathological brain vs adult/control brain (Figure 2n-q, Extended Data Figure 17, Supplementary figures 16,17,22,23)

However, we would like to emphasize that the "reactivated fetal pathways" or shared pathways between the fetal brain and the adult brain are present across different EC clusters and not restricted to a certain EC cluster (Supplementary Figure 22,23). While we observed indeed the highest regulation of these pathways in "angiogenic capillaries", the same group of pathways was also dysregulated across most other EC clusters (Supplementary Figure 22,23).

Moreover, we have analyzed the potency of endothelial cells using SCENT package (Supplementary Figure 15, Supplementary table 12), (<https://github.com/aet21/SCENT>) from: i, fetal, adult/control and pathological brains. ii, fetal brains, adult/control, brain vascular malformations and brain tumors. iii, the indicated entities and iv, Entropy analysis comparing the potency of endothelial cell subtypes (14 different AV clusters), the results indicate that the pathological brain endothelial cells acquire potency levels comparable to fetal brain ECs.

Thus, our additional analysis speaks against a model in which shared angiogenic pathways between the fetal and the pathological brain are restricted to/is exclusive to a given EC cluster.

We do agree with the reviewer that there is currently no evidence to demonstrate that diseased cells start at with an adult transcriptional profile, and then activate fetal pathways to become less differentiated.

In summary, for the reasons outlined above and taking into account this reviewer's comment, we have adapted the formulation of "reactivated fetal pathways" to "shared pathways in the fetal and pathological brain vasculature" throughout the manuscript.

COMMENT:

- It is unclear why meningiomas, a non-vascular derived tumor, were included in this paper.

ANSWER:

Meningiomas are a vascular-rich type of tumor that are angiogenesis-dependent for their growth²⁶.

The reason for inclusion of meningiomas in this manuscript is because we aimed to compare different brain tumors (see Figures 1a,c, 2j, Extended Figure 2j, Extended Data Figures 4,5,11, Supplementary Figures 4,5,6) including:

- Primary intraaxial brain tumors
 - Hemangioblastoma
 - Low-grade gliomas
 - Glioblastomas
- Primary extraaxial brain tumors
 - Meningiomas
- Secondary brain tumors
 - Lung cancer metastases

The critical role for the vasculature in brain tumor growth has been described for all the included brain tumors including hemangioblastomas²⁷, low- and high-grade gliomas²⁸⁻³⁰, brain metastases^{28,31}, meningiomas²⁶, and thus the creation of a molecular single-cell atlas including all primary and secondary, intra- and extraaxial brain tumors is well justified.

COMMENT:

- *Justification should be provided on the value of including non-vascular tumors- do the authors hypothesize that the CNS endothelium is transcriptionally distinct in these tumor types due to intrinsic differences in tumor pathology?*

ANSWER:

We thank the reviewer for this comment and would like to explain the different reasons why meningiomas were included in this study.

First, the vasculature is important for the growth of every brain tumor (see references above) but poorly described at the single-cell level.

Second, the creation of an atlas of the human brain vasculature including endothelial and perivascular cells - including the following brain tumors:

- Primary intraaxial brain tumors
 - Hemangioblastoma (HEM)
 - Low-grade gliomas (LGGs)
 - Glioblastomas (GBMs)
- Primary extraaxial brain tumors
 - Meningiomas (MENs)
- Secondary brain tumors
 - Lung cancer metastases (METs) -

to compare the vasculature (endothelial and perivascular) cells between the different types of brain tumors is of fundamental importance (see Figure 2j, Extended Data Figures 4,5,11,19,21).

Third, we indeed hypothesized that the endothelium of the different brain tumors shows transcriptional differences with regard to key characteristics due the different nature of these brain tumors.

For instance, the CNS-signature characterizing typical CNS-specific properties (see Figure 4d,f,h) of endothelial cells showed key differences between the different brain tumor entities and was higher for intraaxial primary brain tumors (coming from the brain parenchyma) as compared to intraaxial secondary brain tumors (coming from peripheral tissues, in this case the lung) and as compared to extraaxial brain tumors (coming from the leptomeninges, namely the arachnoidea mater), (Figure 4f,h, Extended Data Figure 24,25,27):

- CNS-specificity / CNS signature
 - TL > LGG > GBM > MET > MEN/HEM

On the other hand, the MHC class II signature characterizing immunological properties of endothelial cells showed an opposing trend, namely for (Figure 5h, Extended Data Figures 28,30):

- MHC class II signature
 - MEN > MET > HEM > LGG > TL > GBM

Moreover, differences between the different brain tumor types were also observed in the ligand-receptor interactions signaling pathways (Figure 6, Extended Data Figures 33,34, Supplementary Figure 24,25,26,29,41).

Thus, the transcriptome of brain tumor endothelial cells indeed differs according to the different brain tumor types and revealed very important, fundamental differences.

In addition to the results shown, we have also emphasized this further in the paragraph "Results" as well as in the paragraph "Discussion" throughout the manuscript.

COMMENT:

- *The authors write several times throughout the manuscript that the analysis includes 47 fetuses and adult samples to create the molecular atlas. This is misleading as only 5 fetal brains and 8 normal temporal lobes in the adult were used to create the un-diseased atlas.*

ANSWER:

We included 47 samples, which included fetal, adult healthy and adult pathological samples. We have now clarified this in the paragraph "Results" and "Methods" as well as in the figure legend of Figure 1, Extended Data Figure 1 and in the Supplementary Figures and Table 1.

In addition, we have also added the samples for bulk RNA-sequencing and have clarified in both the Figures as well as in the text (paragraphs "Results", "Discussion" and "Methods") how many fetal,

adult/control, and adult/diseased samples were used in the manuscript for both scRNA-sequencing and bulk RNA-sequencing.

COMMENT:

- *There are several instances with portions of figures that appear to be redundant. Some examples include: Figure 1e,f appears redundant with figure 1g; Extended Data Fig 3 and 4; Extended Data Fig. 6 a-c.*

ANSWER:

We thank the reviewer for this comment and have addressed the point raised for Figures 1e,f,g. Accordingly, we have now moved Figure 1 e-h to Extended Data Figure 1d-g (as we feel that the direct comparisons between fetal brain ECs vs fetal periphery ECs (previous Figure 1e), between fetal brain ECs and adult brain ECs (previous Figure 1f), as well as between fetal brain ECs, fetal peripheral ECs, adult brain ECs, brain tumor ECs and brain vascular malformation ECs (previous Figure 1g) are of value and we would like to display this information), while keeping Figure 1g (current Figure 1f) in the main figure 1. Thereby, anything that appeared redundant in main Figure 1 has been removed. In addition, this adaptation has also made more space for data, as requested by the reviewer in their first of the “Minor Points” regarding Figures 1a-c (see above). We have now added the different angiogenic signatures including

- Human fetal/developmental CNS/brain signature
- Human pathological signature
- Human brain tumor signature
- Human brain vascular malformation signature

To address the reviewer's concern, regarding the Extended Data Figure 3 and 4, we would like to emphasize that the point about not merging all pathologies but more precisely addressing/displaying the different entities, for instance:

- brain tumors
- brain vascular malformations
- all individual entities

Moreover, we have added immunofluorescence stainings of the two key angiogenic markers PLVAP and ESM-1 in main Figure 1.

This issue has been raised several times by this and the two other reviewers. Therefore, we would like to clarify what is displayed in the different figures that were mentioned:

- Extended Data Figure 4 shows the important data for unsorted ECs and PVCs, with the corresponding tissue proportions (Extended Data Figure 4a-d), the respective cluster compositions (Extended Data Figure 4e,f), and the clustering of cell types within and across entities/tissues (Extended Data Figure 4g).
- Extended Data Figure 5 displays the cluster composition for the different entities for both the unsorted ECs and PVCs (Extended Data Figure 5a-t) as well as for the sorted ECs (Extended Data Figure 11a-v)
 - based on the reviewer's requests, we have now also added the cluster composition for both unsorted ECs and PVCs (Extended Data Figure 5u) and for sorted ECs (Extended Data Figure 11w,x) for each patient.
- Extended Data Figure 7 displays the differentially expressed genes (DEGs) and dysregulated pathways between:
 - fetal brain vs fetal periphery (Extended Data Figure 7a-c)
 - fetal brain vs adult/control brain (Extended Data Figure 7d-f)
 - adult/control brain vs pathological brain (Extended Data Figure 7g-i)
- Extended Data Figure 8 shows the differentially expressed genes (DEGs) for different comparisons, namely:
 - adult/control brain vs brain tumors vs brain vascular malformations (Extended Data Figure 8a)
 - fetal brain vs periphery vs adult/control brain vs brain tumors vs brain vascular malformations (Extended Data Figure 8b)
 - fetal brain vs adult/control brain vs brain tumors vs brain vascular malformations (Extended Data Figure 8c)

- as well as the pathway analysis of the commonly dysregulated pathways between fetal brain ECs vs adult/control brain ECs and pathological brain ECs vs adult/control brain ECs (Extended Data Figure 8d, referring to Main Figure 1t).

Therefore, we would like to clarify that the information displayed in the different figures is not redundant but answers the reviewer's requests to clearly display the analyses at the different levels of comparisons, namely

- level of all pathologies
- level of all brain tumors
- level of all brain vascular malformations
- level of all individual entities/pathologies
- level of all individual patients

In summary, we would like to state that we feel that it is important to include very detailed analyses of our data at the different levels outlined above (and to display this data in figures) as this atlas will be useful for researchers from a variety of research areas including neuroscience, vascular biology, developmental biology, neuro-oncology, neurovascular biology, single-cell genomics and others. With that aim in mind, we have attempted to balance requests for additional validation and analysis with requests to remove redundant data. Since this is an atlas, we would like to make all of the data that we have generated to be accessible depending on the interests of the reader. Accordingly, we have tried to make this data as accessible and comprehensible as possible and have displayed it in figures accordingly.

COMMENT:

- Fig 2g. – axis of bar chart or bars should be labeled with percentage values

ANSWER:

We thank the reviewer for this input and have now added the percentage values to the Figures 2g,h,i,j accordingly. We have added the cell composition numbers/fraction/percentages for all entities in Supplementary table 9.

See also below answer to "Figure 2k."

COMMENT:

- Extended Data Fig. 3 – many colors used to refer to different cell types; can be difficult to compare, for example, endothelial cells with stem cells and tumor cells in 3e and especially 3f.

ANSWER:

We thank the reviewer for this comment. We would like to point out that we selected a color code designed to maximize distinction between colors to best display/visualize the different cluster/subcluster given the number of clusters that we define. While we are not able to change the color code, we now

- add dotted lines on the UMAPs for each cluster and annotate the clusters
- provide additional tables with the percentage of each cluster/subcluster for every individual entity (Supplementary Tables 9)

which allows for better visualization of clusters and direct comparison of clusters between different entities.

COMMENT:

- Figure 1m: unclear value of displaying pathologic transcript profiles compared to control. Clustering of distinct pathologies is an over-simplification of disease processes. Unclear value of Venn diagram in figure 1l, which demonstrates some overlap in transcriptional signature between the three cell sources, all of which are presumably endothelial in origin.

ANSWER:

Regarding the reviewer's concerns about the clustering of distinct pathologies, we would like to refer to our detailed response to the reviewer's similar comment above.

Regarding Figure 1m, we aimed to identify transcriptomic profiles across various vascular-dependent brain pathologies including brain tumors and brain vascular malformations. Interestingly, we thereby identified a common “hallmark” of transcriptomic changes across different disease states. (Figure 1, Extended Data Figures 7,8,9, and Supplementary Figure 2-14). This being said, while Figure 1s-u displays dysregulated pathways in pathological brain ECs and fetal brain ECs as compared to adult/control brains, we performed the very same pathway analyses also for:

- i) brain tumor ECs versus adult/control brain ECs (Supplementary Figure 3)
- ii) brain vascular malformation ECs versus adult/control brain ECs (Supplementary Figure 2)
- iii) every single brain disease entity ECs versus adult/control brain ECs (Supplementary Figure 7-14)

Regarding the Venn diagram in Figure 1l, we would like to clarify that it aims to compare two different gene set enrichment analyses in endothelial cells, namely:

- i) pathways that are dysregulated in fetal brain ECs versus temporal lobe adult/control ECs (1s) and
- ii) pathways that are dysregulated in pathological brain ECs versus temporal lobe adult/control ECs (1u)

The result is one of the key messages of Figure 1, namely that more than half of the pathways that are dysregulated in pathological brain ECs versus adult/control brain ECs are also differentially regulated in fetal brain ECs versus adult/control brain ECs, namely 357/612 (=58.33%).

Accordingly, the key message derived from this venn diagram is that there is considerable overlap of pathways that are differentially regulated in the fetal brain ECs as compared to the adult/control brain ECs and of pathways that are dysregulated pathological brain ECs as compared to the adult/control brain ECs, indicating a similarity between fetal and pathological pathways in brain ECs and a possible “reactivation” of fetal angiogenic pathways in brain pathologies.

Again, while Figure 1t displays the overlap of dysregulated pathways' profiles in fetal brain and pathological brain ECs dysregulated as compared to adult/control brain ECs, we performed the very same pathway analyses also for:

- i) brain tumor ECs versus adult/control brain ECs (Supplementary Figure 3)
- ii) brain vascular malformation ECs versus adult/control brain ECs (Supplementary Figure 2)
- iii) every single brain disease entity ECs versus adult/control brain ECs (Supplementary Figures 7-14)

COMMENT:

- *Figure 2k: Is this pie chart derived from all samples pooled together (control, pathologic, and fetal, as in 2g) or just adult control? If only adult control, would be valuable to demonstrate shift in vascular compartments with pathology or age.*

ANSWER:

The pie chart in Figure 2k is indeed derived from all samples pooled together (fetal brain ECs, adult/control brain ECs, pathological brain ECs of all different brain entities) - identical as in Figure 2g - and is not just derived from the adult control brain.

Figure 2g and Figure 2k are showing the same information.

The further breakdown of the EC compartments / EC clusters according to the different pathological brain entities is shown in Figures 2h,i,j and Extended data Figures 5,11 as well as Supplementary tables 9,10.

The question regarding the shift in vascular compartments between the different brain entities and whether these shifts are significant or not was also raised by reviewer 3. Accordingly, we have performed statistical testing using the scCODA method according to Buttner et al., *Nat Comm*, 2021³² as suggested by reviewer 3 (Supplementary table 10, reviewer figure 3).

The results show the following significant changes in EC compartments / EC clusters:

- i) a significant upregulation of angiogenic capillaries
 - in fetal brain ECs versus adult/control brain ECs
 - in brain tumor ECs versus adult/control brain ECs

- ii) a significant upregulation of venous compartments
 - in brain vascular malformation ECs versus adult/control brain ECs

iii) For individual entities:

- Fetal brains vs adult/control brains:
 - > Significant up-regulation of:
 - Angiogenic capillary
 - Proliferating cell
 - > Significant down-regulation of:
 - Arterioles
 - Capillary

- TL adjacent to cavernomas vs adult/control brains:
 - > Significant up-regulation of:
 - non
 - > Significant down-regulation of:
 - non

- Cavernoma vs adult/control brains:
 - > Significant up-regulation of:
 - Venule
 - > Significant down-regulation of:
 - Arteriole
 - Capillary

- Arteriovenous malformations vs adult/control brains:
 - > Significant up-regulation of:
 - Large artery
 - Artery
 - Angiogenic capillary
 - Venule
 - Vein
 - Large vein
 - > Significant down-regulation of:
 - non

- Hemangioblastomas vs adult/control brains:
 - > Significant up-regulation of:
 - Angiogenic capillary
 - > Significant down-regulation of:
 - Arteriole
 - Capillary

- Low grade gliomas vs adult/control brains:
 - > Significant up-regulation of:
 - Angiogenic capillary
 - > Significant down-regulation of:
 - non

- Glioblastomas vs adult/control brains:
 - > Significant up-regulation of:
 - Angiogenic capillary
 - Stem-to-EC
 - > Significant down-regulation of:
 - Artery
 - Arteriole
 - Capillary

- Metastases vs adult/control brains:
 - > Significant up-regulation of:

- Angiogenic capillary
- Stem-to-EC
- > Significant down-regulation of:
 - Artery
 - Arteriole
 - Capillary
- Meningiomas vs adult/control brains:
 - > Significant up-regulation of:
 - Angiogenic capillary
 - Vein
 - > Significant down-regulation of:
 - Arteriole
 - Capillary

Moreover, ligand-receptor signaling patterns in pathological brain ECs, fetal brain ECs and adult/control brain ECs in Figures 2 o-q (and Extended Data Figures 17,18 show that there are not only considerably more LR interactions involving the angiogenic EC cluster in the pathological and fetal brain as compared to the adult/control brain but also significantly more signaling strength.

Regarding the age distribution in the adult/control brains, we have performed, scCODA analysis as well as pseudobulking/shuffling to address this question and found that there was no significant difference according to age.

In summary, when addressing the parameter age in adult/control brains (TLs), we found few DEGs that did however not result in biologically meaningful pathways (Figure to the Reviewer 1, see also below), as well as no significant compositional change as tested by scCODA. We are well aware of the limited sample size and have mentioned this in the paragraph "Discussion".

COMMENT:

- *Figure 2m: Signaling patterns by vascular compartments in the adult demonstrates interesting distribution of pathways. Again, in figure 2n, lumping distinct diseases raises concerns for differences in signaling between very different tumor types.*

ANSWER:

We agree with the reviewer that the signaling patterns in the angiogenic capillaries in pathological brain ECs versus adult/control brain ECs demonstrate a very interesting distribution of pathways (Figure 2m) and thank the reviewer for this comment.

Figure 2n shows the overall signaling patterns in the temporal lobe (adult/control brain) ECs, so we understand that the reviewer probably referred to Figure 2o, which displays the overall signaling patterns in the pathological brain ECs.

As mentioned above, we would like to emphasize that one aim of our study is to find common transcriptomic changes across pathologies (brain tumors and brain vascular malformations), whereas another aim is to describe the disease-specific pathological changes. Accordingly, in addition to having displayed the hallmark of transcriptional changes across pathologies, we have now provided the analysis for the individual entities in Supplementary Figures 24,25,41. The detailed comparison of the vasculature of different types of brain tumors warrants future studies dedicated to this specific question. We have now also mentioned this limitation and the request for future follow-up studies in the paragraph "Discussion".

To address the reviewer's concerns regarding the lumping together of different disease types, we would also like to refer to our detailed response to this reviewer's second comment above.

First, we have now further clarified which findings and transcriptomic changes are conserved among the different disease entities while also emphasizing findings that are more specific to neurovascular (malformations) versus neoplastic (tumor) entities. As outlined above, the observations regarding commonly dysregulated pathways, common alterations in AV-differentiation patterns and CNS-specificity, MHC class II upregulation and changes in cell-cell interactions in ECs (and PVCs for the cell-cell interaction patterns) across different disease entities is novel and interesting in our eyes and thereby constitutes a "hallmark" of the diseased brain vascular endothelium.

However, to further address the reviewer's related concern about differences in signaling pathways in the different brain diseases/pathologies, brain vascular malformations, and brain tumors, we have now provided the signaling pathway patterns in the angiogenic capillary cluster (as shown in Figures 2n-q) at multiple/different levels of comparison, namely:

- for fetal brain ECs versus adult/control brain ECs (see Figure 2m,o, Extended Data Figures 17,18, Supplementary Figures 17,26a).
- for pathological brain ECs versus adult/control brain ECs (see Figure 2m,n,q, Extended Data Figures 17,18, Supplementary Figures 16,26c).
- for brain tumor ECs versus adult/control brain ECs (see Supplementary Figure 24II, 25II, 26e).
- for brain vascular malformation ECs versus adult/control brain ECs (see Supplementary Figures 24I, 25I, 26d).
- for every single brain disease entity ECs versus adult/control brain ECs (see Supplementary Figures 24III-X, 25III-X, 26b,f-m)

In summary, we can conclude that the signaling pathways that are upregulated across various disease entities including different brain tumors and different vascular malformations belong to the five groups of pathways that we described in the manuscript, namely angiogenesis, neurovascular link (NVL) and development, metabolism, cell-cell/ECM related processes, immune-related processes.

To further respond to the reviewer's request, we have now additionally displayed the regulated signaling pathways in different EC clusters including the angiogenic capillaries for the different levels of comparison mentioned above (including the different brain tumor types) and have added this in the paragraph "Results" and "Discussion" in the manuscript (Figure 2n, Extended Data Figure 17j-m, Supplementary Figures 25).

COMMENT:

- Unclear added value of figure 3p,q,r,s.

ANSWER:

We are sorry that this was not explained well enough and led to confusion. Reviewer 2 raised a similar concern. We would like to clarify the results displayed in the figures mentioned by the reviewer.

The Figures 3p,q,r,s address the heterogeneity of ECs according to the vessel size respectively according to the main groups of EC clusters. Accordingly, these figure panels display ECs belonging to large arteries, capillaries, and large veins.

Overall, the Figures 3p,q,r,s illustrate that the ECs are most heterogeneous at the level of the small-caliber vessels being the capillaries. While figure 3p,q,r highlight this finding by displaying common and specific markers across all brain entities showing that large caliber vessels (large arteries and large veins) have more common markers and less specific markers whereas for small caliber vessels it is the other way round, namely fewer common markers and more specific markers.

Figure 3s further supports this statement by displaying the Principal Component Analysis (PCA) showing that the capillaries are the most different EC type across the different vascular beds being fetal brain, adult/control brain, brain tumors and brain vascular malformations while the large arteries and large veins cluster of these vascular beds cluster together. This indicates that small caliber ECs (capillary ECs) are the most susceptible to their local microenvironment whereas large caliber ECs (large artery ECs and large vein ECs) seem to be more conserved across different brain tissue types. Importantly, while these findings are in agreement with what has previously been reported in a mouse EC atlas across different tissues⁴, this has never been shown before in human brain vascular ECs. We are convinced that these are important findings and hope that we hereby clarified these findings for the reviewer.

COMMENT:

- Figure 4 SPOCK3 and CD320 validation; unclear why these exact transcripts were chosen. Additionally, fetal tissue should be included in the IF validation since it is displayed as differentially regulated in part d.

ANSWER:

We have defined a CNS signature (Figure 4d, see methods) and have validated multiple proteins of this CNS signature including SPOCK3 (Extended Data Figures 27I a'-n') and CD320 (Figures 4 o-t', Extended Data Figures 27I c'-p'), which are displayed in Main Figure 4 and in Extended Data Figure

27I). In addition, we have validated various other transcripts of the CNS signature (Figure 4d), namely BSG (Figures 4 i-n', Extended Data Figures 27 o-bi'), GPCPD1 (Extended Data Figures 27I qi-dii'), PPP1R14A (Extended Data Figures 27II a-n') as well as SLC38A5 (Extended Data Figures 27II o-bi').

Accordingly, to respond to these reviewer's requests, we have now performed additional immunofluorescence as well as IMC validation for CNS signature genes that are highly expressed in fetal brain ECs, namely SLC38A5, CD320, BSG, and GPCPD1 (Figure 4d). We did not stain for SPOCK3 and PPP1R14A in the fetal tissues as they show low transcriptional expression in the fetal brain in Figure 4d. Moreover, we have added two fetal peripheral tissues (GI tract and kidney, both known to be highly vascularized^{33,34} as negative controls for these markers (see Extended Data Figure 27IV).

The CNS signature markers BSG, CD320 and SLC38A5 showed expression in fetal brain ECs and lower expression in peripheral ECs (fetal GI tract and kidney ECs, Extended Data Figure 27 IV). GPCPD1 showed lower expression levels in both fetal brain ECs as well as in peripheral ECs (fetal GI tract and kidney ECs, Extended Data Figure 27IV).

Because SPOCK3 is not highly expressed in fetal brain (Figure 4d) and because we decided to include fetal brain tissue in the main and Extended Data figures to respond to the reviewer's request, we replaced SPOCK3 with BSG in the main figure 4.

In addition to these CNS signature transcripts, we also stained for the BBB markers Occludin (OCLN), Zona occludens 1 (ZO-1/TJP1), and GLUT1 (= SLC2A1), which is also a CNS signature marker, (see Figure 4d) in the fetal tissues in order to complete what we have already shown for the adult/control and diseased brain tissues (Extended Data Figure 27 III).

In the fetus, GLUT1 (= SLC2A1) showed high expression in fetal brain ECs and was absent in peripheral ECs (fetal GI tract and kidney ECs, Extended Data Figure 27IV y-fi'). Occludin and ZO-1 showed expression in brain ECs and some expression in the peripheral ECs (fetal GI tract and kidney ECs, Extended Data Figure 27IV g-vi'), in agreement with what was previously reported³⁵⁻³⁷.

In summary, we observe that CNS signature genes are expressed in fetal brain ECs and expressed to a lower level in peripheral ECs (whereas most peripheral ECs don't express the CNS signature transcripts). The precise expression of CNS signature and BBB transcripts and proteins across different stages of fetal brain development needs to be addressed further in dedicated follow-up studies, and we have added this limitation in the paragraph "Discussion".

COMMENT:

- Extended Data Fig. 8 zii – EndoMT markers – ACTA2 is expressed in perivascular SMCs; perhaps an additional marker specific to mural cells (i.e. PDGFRB) would exclude the possibility that the signal is coming from a nearby mural cell?

ANSWER:

We thank the reviewer for the comment. First, we would like to briefly remind the reviewer that there is a difference in RNAscope as compared to immunofluorescence data. A colocalization in RNAscope is seen as puncta within the same cell but doesn't necessarily result in an overlap of the two colors (as it is the case in immunofluorescence stainings) since it stains single mRNA molecules.

To further address the reviewer's request, we now provide additional immunofluorescence stainings using the EndoMT marker ACTA2 and the endothelial cell marker CD31 in the fetal brain, temporal lobe, glioblastoma, and lung cancer metastasis. Although we were unable to add PDGFRB stainings, it is clear from the additional images that here are ECs that express ACTA2 and that these cells are not pericytes based on EC markers and location (Extended Data Figure 13ti-aiii').

These additional stainings thereby provide more clarity regarding endothelial and perivascular cell expression.

In glioblastoma and in lung cancer brain metastasis, we observed ACTA2 expression in both CD31⁻ perivascular cells as well as in CD31⁺ endothelial cells (Extended Data Figures 13xi-aiii'). In the temporal lobe, we observed ACTA2 expression mainly/exclusively in CD31⁻ perivascular cells but not in endothelial cells (Extended Data Figure 13 vii-wii'), whereas in the fetal brain, we observed rare ACTA2 expression in fetal brain CD31⁺ endothelial cells (Extended Data Figure 13 ti-uii'), in accordance with our scRNA-sequencing data.

In addition to these immunofluorescence stainings, we now also provide IMC experiments across fetal, adult and pathological entities and observed expression of ACTA2 in CD31⁺ / CLDN5⁺ / PDGFRB⁻ ECs (Extended Data Figure 14), thereby further indicating that ACTA2 can indeed be expressed in CD31⁺ / CLDN5⁺ ECs that are PDGFRB⁻.

Moreover, in response to this reviewer and to other reviewers regarding the inclusion of fetal tissue, we would like to mention that we have now also performed further immunofluorescence for the angiogenic marker ESM-1 in the fetal brain, temporal lobe, glioblastoma, and lung cancer brain metastasis. ESM-1 was highly expressed in fetal brain and pathological GBM and metastasis ECs but showed low expression levels in temporal lobe ECs (Extended Data Figure 13^{lii-sii}), in agreement with our scRNA-sequencing data and with what was previously reported.

COMMENT:

- *Extended Data Fig. 9 – immunofluorescence is not incredibly convincing to show co-expression of these markers in endothelial cells*

ANSWER:

Regarding the immunofluorescence for the validation of the stem-to-EC clusters in glioblastomas and in lung brain metastases, we now provide additional immunofluorescence stainings in both disease entities as well as in TL (Extended Data Figure 15).

Regarding the stem cell markers for the glioblastomas, we have added additional images for both SOX2 and PTPRZ1 in the temporal lobe and in the glioblastoma (Extended Data Figure 15).

- SOX2 was expressed high in both GBM ECs and in MET ECs
- In GBMs non-ECs we see nuclear SOX2 but not in the lung cancer metastases non-ECs

Regarding the stem cell markers for the lung cancer metastases, we have added additional images for both EPCAM and SFTPB in the temporal lobe and in the lung cancer brain metastasis (Extended Data Figure 15).

In summary, the glioblastoma stem cell marker PTPRZ1 was expressed in glioblastoma ECs but not in lung cancer metastasis ECs while the lung cancer stem cell markers EPCAM and SFTBP were expressed in lung cancer metastasis ECs but not in glioblastoma ECs (Extended Data Figure 15). Interestingly, however, SOX2 showed – in addition to its expression in glioblastoma ECs (see above) – is expressed in lung cancer brain metastasis ECs, in agreement with its expression in the scRNA-sequencing data (Extended Data Figure 15).

COMMENT:

- *Figure 6g exemplifies the diversity of the tumor cell expression profiles, which were clustered together to form the interaction maps in part h onward. This oversimplifies the interactions of 10 or more tumor types into one interaction model.*

ANSWER:

We thank the reviewer for this comment and would like to explain the rationale why in Figure 6g and onwards, we show the interaction maps of all the different pathological ECs and PVCs including brain tumor ECs/PVCs and brain vascular malformation ECs/PVCs.

Similar to our response to the reviewer's comment to Figure 1m (see above), we aimed to identify transcriptomic profiles and cellular interaction maps across various vascular-dependent brain pathologies including brain tumors and brain vascular malformations. As mentioned in the paragraph "results" in the manuscript and in our response to this reviewer's comment to Figure 1m (see above), we thereby identified a common "hallmark" of transcriptomic changes and cellular interaction maps across different disease states. (Figure 6, Extended Data Figure 33). Importantly, while Figure 6g displays interaction maps in pathological brain transcriptomic ECs/PVCs profiles as compared to pathways in adult/control brain transcriptomic ECs/PVCs profiles, we performed the very same analyses also for:

- i) brain tumor ECs versus adult/control brain ECs (Extended Data Figure 33a'-e', 34, Supplementary Figure 41II)
- ii) brain vascular malformation ECs versus adult/control brain ECs (Extended Data Figure 33a-e, Supplementary Figure 41I)

- iii) every single brain disease entity ECs versus adult/control brain ECs (Extended Data Figure 33f-t,f'-ziv', Supplementary Figure 41 III-X)
- iv) Moreover we have provided the interaction model (as in figure 6h) for each individual entity (Extended Data Figure 33f-t,f'-ziv')

Finally, as mentioned above, we would like to emphasize that one aim of our study was to find commonalities across pathologies - brain tumors and brain vascular malformations - in addition to describing the disease specific pathological changes rather than dissecting differences among different types of brain tumors. The detailed comparison of the vasculature of different types of brain tumors warrants future studies dedicated to this specific question. We have now also mentioned this limitation and the request for future follow-up studies in the paragraph "Discussion".

COMMENT:

- *The authors compare their pathological endothelial cells with a BBB dysfunction module defined in the mouse. It would also be interesting to highlight certain genes that seem to be human specific in undiseased conditions (compare to an undiseased mouse BBB module) to determine if there are baseline differences in BBB regulation across species.*

ANSWER:

We thank the reviewer for this comment and agree with this very valuable suggestion. In order to address the reviewer's comment regarding the comparison of undiseased BBB modules between human and mouse, we have now compared our dataset of sorted ECs in the temporal lobe (undiseased conditions) to the two following recently published datasets that describe the BBB module/BBB in the mouse brain:

Comparison of human adult/control brain endothelial cells to publicly available mouse adult control brain endothelial cells

i) Vanlandewijck et al., A molecular atlas of cell types and zonation in the brain vasculature, *Nature*, 2018
<https://doi.org/10.1038/nature25739>

The paper by Vanlandewijck et al.^{5,10} described the brain vasculature (ECs) in the adult healthy mouse brain (CD31⁺/ GFP⁺ sorting of brain ECs).

ii) Kalucka et al., Single-Cell Transcriptome Atlas of Murine Endothelial Cells. *Cell*, 2020
10.1016/j.cell.2020.01.015
DOI: 10.1016/j.cell.2020.01.015

The paper by Kalucka et al.⁴ addressed the vasculature (ECs) in various adult healthy peripheral mouse tissues including the adult healthy mouse brain (CD31⁺/CD45⁻ sorting) of peripheral and brain ECs.

Accordingly, we compared the results of our sorted EC dataset for the adult/control human brains (temporal lobes) with the sorted EC cluster for the adult/control mouse brains (entire mouse brains) of the Vanlandewijck et al. and Kalucka et al. papers^{4,5}.

The results indeed reveal human-specific genes in undiseased conditions (= in adult/control brain ECs) when comparing to an undiseased mouse BBB module (Figure 2I, Extended Data Figure 6zxx-zxxii, 6zxxvi-zxxviii, 6xxix-zxxx, Supplementary table 11, and as detailed below).

Comparison of sorted EC datasets

Comparison of EC clusters in our sorted adult/control brain dataset with the EC clusters of the mouse brain published datasets of Vanlandewijck et al., *Nature*, 2018 and Kalucka et al., *Cell*, 2020 revealed that most human AV-clusters did not map to mouse AV-clusters, with the exception of large arteries and arterioles (Figure 2I, Extended Data Figure 6zxx-zxxii, 6zxxvi-zxxviii, 6xxix-zxxx, Supplementary table 11).

We observe that while the overall structure of AV-zonation was conserved between human and mouse, the number of conserved/common AV-zonation genes in the different AV-compartments was low (when comparing our dataset to Kalucka et al.⁴; 2.1%, 3.3% and 7.8% for venules, capillaries and

arterioles, and 10.1% and 14.8% for large veins and large arteries, respectively, (Figure 2I, Extended Data Figures 6zxxvi-zxxviii), indicating a higher conservation of AV-markers between species for larger>small caliber and for arterial>venous vessel ECs with more pronounced species-specific differences at the level of small>large caliber and venous>arterial vessel ECs (Figure 2I, Extended Data Figure 6zxxix,zxxx). Accordingly, we found the highest proportion of human-specific AV-zonation markers in small>large caliber and venous>arterial vessel ECs (Figure 2I and Extended Data Figure 6zxxix,zxxx).

Notably, we found the following novel human-specific AV-zonation markers which we validated through immunostaining referring to the Human Protein Atlas HPA^{6,38}, confirming their expression in the human brain vasculature (Figure 2e,f, Extended Data Figure 16):

- i) for large arteries
 - *COL8A1* and *S100A6*
- ii) for arteries
 - *PLPP1* and *CXCL12*
- iii) for arterioles
 - *HERPUD1* and *AIF1L*
- iv) for capillaries
 - *ADIRF* and *CAVIN2*
- v) for venules
 - *HYAL2* and *S100A10*
- vi) for veins
 - *PDLIM1* and *PLAT*
- vii) and for large veins
 - *SOD2* and *GADD45B*

We found the following human–mouse conserved AV-zonation markers which we also validated through immunostaining referring to the Human Protein Atlas HPA^{6,38} (Figure 2e,f,l, Extended Data Figure 6zxxix,zxxx, Extended Data Figure 16):

- i) for large arteries
 - *SAT1* and *S100A11*
- ii) for arteries
 - *TM4SF1* and *GJA4*
- iii) for arterioles
 - *CYB5R3* and *CP*
- iv) for capillaries
 - *SLC7A5* and *MFSD2A*
- v) for venules
 - *TSHZ2* and *FN1*
- vi) for veins
 - *VWF* and *ANXA2*
- vii) and for large veins
 - *NFKBIA* and *ICAM1*

Comparison of the unsorted EC and PVC datasets

To further address species-specificity in the brain NVU (ECs and PVCs, that constitute the BBB³), we performed the comparison of PVC and EC cell types in our unsorted datasets with PVC and EC cell types in the unsorted datasets of the *Tabula muris*³⁹. This analysis revealed that corresponding PVC and EC cell types mapped to each other.

Notably, the mapping showed that (Extended Data Figure 6zxvii-zxvix):

- Our endothelial cells mapped to endothelial cells
- Our pericytes mapped to pericytes
- Our smooth muscle cells mapped to pericytes (as smooth muscle cells weren't identified in the *tabula muris* dataset)
- Our fibroblasts didn't map to a specific cell type
- Our oligodendrocytes mapped to oligodendrocytes.
- Our astrocytes mapped to oligodendrocytes (as astrocytes were too few cells in the *tabula muris* dataset)
- Our neurons mapped to neurons and oligodendrocytes
- Our neural stem cells mapped to neurons
- Our neuronal progenitors mapped to oligodendrocytes

- Our microglia/macrophages mapped to microglia

In summary, whereas most corresponding cells mapped to each other, we observed some discrepancies regarding the mapping of certain perivascular cell (see Extended Data Figure 6zxvii-zxvix), most likely due to species-specific differences⁴⁰.

Moreover, we observed that whereas neurons and astrocytes showed the lowest percentage of common markers/greatest transcriptional divergence, endothelial cells together with oligodendrocytes displayed the highest percentage of human-specific markers, in agreement with previous reports^{3,6,7,40}.

Comparison to the literature

The comparison of the human and mouse brain vasculature for ECs and for PVCs has also been addressed in the papers Yang et al., *Nature*, 2022 and Garcia et al., *Nature*, 2022 and has revealed similar findings.

Yang and colleagues found that a significant number of markers in each of the different vessel segments (arteries, capillaries, veins) for both ECs and PVCs lost their zonation specificity between species, indicating species-specific differences in endothelial and mural/perivascular cells that reflect fundamental differences in properties of the human and mouse brain vasculature⁶.

Similarly, Garcia and colleagues reported that although the functional organization of AV-zonation is a conserved characteristic between human and mouse, the set of zoned genes highly diverged, with only a small fraction of zoned genes being conserved between species ($\approx 10\%$)⁷. While they found some conservation between markers of human and mouse arterioles (VEGFC, BMX and EFNB2) and capillaries (MFSD2A and TFRC), they observed that venule markers differed between human and mouse.

Both studies observed that cell-type identity markers for both endothelial and mural cell markers highly varied between species, indicating that the human cerebrovasculature exhibits a species-specific gene expression pattern. Notably, this combination of conserved and species-specific markers in the human and mouse brains was also found for neurons and other PVCs⁴⁰.

Conclusion

Taken together, our findings support the key observations made by Yang and Garcia^{6,7} major species-specific endothelial and mural cell differences that are likely to contribute to fundamental differences in brain vascular properties between mice and humans. However, We would like to point out that the comparison between our dataset of sorted CD31⁺/GFP⁺ ECs to the sorted CD31⁺/CD45⁻ EC datasets of Vanlandenwijck et al., and to the sorted CD31⁺/CD45⁻ EC datasets of Kalucka et al.⁴, represents probably the most relevant comparison (and thus offers a more in-depth insight into human and mouse brain EC and PVC biology) given that the similar strategies for enrichment of ECs was used for all three papers (this being the CD31⁺/CD45⁻ and CD31⁺ / GFP⁺ FACS-sorting).

Importantly, we have also validated novel human AV-markers as well human-specific AV markers found in our scRNA-sequencing analyses by referring to the Human Protein Atlas (Extended Data Figure 16). For every AV-compartment (large arteries, arteries, arterioles, capillaries, venules, veins, large veins) we now display novel human-specific AV-markers in the human brain vasculature (Extended Data Figure 16, Supplementary table 14), thereby providing additional evidence for these important findings emanating from our scRNA-sequencing atlas.

In summary, whereas there are some conserved properties of the mouse and human brain vasculature, our findings reveal species-specific differences of endothelial (ECs) and mural cells (PVCs) and for EC clusters along the AV-axis with numerous human-specific genes in the undiseased brain, indicating baseline differences in BBB regulation across species.

Comparison of human pathological brain endothelial cells to publicly available human and mouse dysfunction modules

Moreover, we would like to mention that, in addition to the comparison of the different pathological endothelial cells in our dataset to the BBB dysfunction module of Munji et al, *Nat Neurosci*, 2019⁴¹, we have now – based on the reviewer's request - compared our dataset of sorted pathological ECs also to additional publicly available datasets of the diseased brain (Extended Data Figure 25), (see also response regarding the BBB dysfunctional modules), namely:

- Yang et al., *Nature*, 2022⁶ (neurodegenerative disease, Alzheimer's disease)

- Garcia et al., *Nature*, 2022⁷ (neurodegenerative disease, Huntington's disease)
- Winkler et al., *Science*, 2022³ (brain vascular malformation, AVM)

Reviewer #2:

A. Summary of the key results:

COMMENT:

In this manuscript, Wälchli et al present a molecular atlas at the single-cell level of the human brain (and fetal peripheral organs) vasculature in physiological and pathological conditions. The authors originated a robust sample inventory of human tissue, and demonstrate along the whole paper different possible analysis to do with such resources.

ANSWER:

We thank the reviewer for their comments on our manuscript.

We now indeed provide additional experimental data in the revised manuscript that addresses all the concerns raised by the reviewers. The new data further validates the findings of our scRNA-sequencing atlas and includes:

- new bulk RNA-sequencing data
- additional immunofluorescence (IF) stainings
- new Imaging Mass Cytometry (IMC) = Imaging Cytometry by Time of Flight (Imaging CyTOF) stainings
- new spatial transcriptomics data

thereby addressing key reviewer's concerns about additional validation experiments, additional patients, and inclusion of fetal brain tissues (for IF and IMC).

In addition, we have addressed all issues raised by the reviewers and by the editor.

B. Originality and significance:

COMMENT:

While the creation of a single cell sequencing atlas of endothelial cells and the concept of inter heterogeneity of endothelial cells (ECs) in physiological and pathological conditions is not novel and has been done before in mouse, the work presented here is relevant, as it offers an important resource to the community that contains human endothelial cells from developmental stages and from relevant brain pathologies.

While there is some validation of the expression of some relevant genes in the different entities analyzed, there is no further analysis of the functionality or importance of those gene expression changes, or specific cell-cell communication pathways identified. Therefore, in this reviewer's eyes the manuscript is mainly seen as a resource paper that will provide the community an important dataset of single cell expression data in human ECs across different developmental stages and pathological situations.

ANSWER:

We thank the reviewer for his/her comment and agree with the statements made.

It was indeed our main aim to create a resource paper that provides the scientific community with an important set of single cell expression data in human ECs and PVCs across different developmental stages, in the adult healthy brain, and in brain pathologies including brain tumors and brain vascular malformations.

In this context, the IMC stainings, as shown in Figures 4,5,6 and Extended Data Figures 14,22,27,28,29,30,37, provide an additional validation step. Moreover, the combination of multiple markers of interest in the same brain section across different entities enables to further confirm the important cellular and molecular biological insights into vascular- and perivascular cell types and endothelial clusters of the brain neurovascular unit.

We believe that this resource will allow others to further functional mechanisms and therapeutic relevance in future studies.

C. Data & methodology; Appropriate use of statistics etc:

COMMENT:

- The authors use as control ECs isolated from lobectomy in pharmacoresistant epilepsy patients. Thus it might be that those ECs might already present significant changes with respect to a real control brain EC. This should be considered and fully addressed.

ANSWER:

We thank the reviewer for this comment, a similar question was also raised by reviewer 3. We would like to clarify that the temporal lobe tissue from which adult/control brain ECs are derived from the neocortical part of temporal lobectomies. The neocortical part of temporal lobectomy surgeries is the temporal lobe that overlies the epileptic focus in the hippocampus (see Supplementary Table 1). The hippocampal tissue was not used in our experiments and analyses. While the neocortical part harbors no known epileptic lesion, we cannot fully exclude the possibility of very slight/minor pathological changes in these brain samples but literature indicates that this tissue is "normal"^{3,6,7,40}.

In addition, we would like to emphasize that this is the most "normal" brain tissue available from fresh surgical samples and is the "gold standard" for human control brain tissue from fresh surgical samples (there is no better human control brain EC available). This question was also addressed previously in the landmark paper Hodge et al., *Nature*, 2019⁴⁰, where the authors compared scRNA-seq data from epilepsy patients to post-mortem brain tissue. Notably, they found that the transcriptomic profiles of neurons in the fresh surgical epilepsy samples and the post-mortem tissue looked very similar arguing against the existence of "pathological" brain tissue. Importantly, the authors used neocortical brain tissue from the middle temporal gyrus for both the fresh epileptic samples as well as the postmortem tissue, which is exactly the anatomical location where we got temporal lobe tissue from (the neocortical part of the temporal lobectomies = middle temporal gyrus).

Moreover, as requested by reviewer 1 (see above), we have compared our temporal lobe EC transcriptomic profiles to the control brain tissue of three recently published papers^{3,6,7}. Even though these authors did not use sorting strategies to enrich for brain endothelial cells, the transcriptomic profiles of the adult/control brain ECs in our study and the adult/control brain ECs in these three papers are very similar (see Extended Data Figure 6).

Taken together, we thus conclude that the adult/control brain ECs in our manuscript show similar transcriptomic profiles when compared to the ones used in the recently published literature^{3,6,7,40}.

COMMENT:

- The analysis of CNS-specific properties used bulk analysis of ECs from different peripheral organs. Thus, the authors are disregarding the EC differences between the different peripheral organs. This should be mentioned. Are brain ECs becoming more similar to a specific organ ECs?

ANSWER:

We thank the reviewer for this comment and agree that the differences between the peripheral organs are not addressed in detail in our study. We would like to mention that we utilized ECs of peripheral organs by means of comparing to brain ECs, at both the fetal and the adult stages. The ECs of the peripheral organs were thus pooled to understand what the difference between a CNS signature of ECs and a peripheral signature of ECs looks like. We observed that ECs are partially losing their CNS signature/properties in pathologies (brain tumors and brain vascular malformations). We are not claiming that pathological ECs acquire properties of a particular peripheral organ, we just observed that pathological brain ECs do not maintain their CNS signature/characteristics.

However, in order to address the reviewer's request, we have analyzed whether pathological brain ECs become similar to ECs of a specific peripheral organ. Our data show that this is not the case and that pathological brain ECs are not acquiring EC properties of a specific peripheral organ (Reviewer figure 4).

Methodologically, while we were able to isolate ECs from peripheral organs of fresh fetal tissues, we didn't (and don't) have this possibility for peripheral adult tissues. Therefore, to define the fetal CNS signature, we referred to our own data by comparing ECs isolated from freshly collected fetal brains to ECs isolated from freshly collected peripheral tissues. On the other hand, to define the adult CNS

signature, we compared computationally extracted ECs from adult brain and peripheral tissues referring to publicly available datasets⁴².

To address this reviewer's question regarding the different peripheral organs, we have plotted both the CNS and peripheral signatures on the different fetal organs separately, indicating that the pathological brain endothelial cells didn't gain a particular peripheral organ signature, but rather a general peripheral endothelial signature was observed (Reviewer figure 4).

A thorough investigation of the precise differences in EC transcriptomics in different peripheral human fetal and adult organs will, however, require a follow-up study in our eyes.

Thus, in agreement with the senior editor, Dr. George Caputa (in our discussion on the 25-1-2022), we decided that addressing vascular beds and ECs of other organs besides the brain are out of scope for this manuscript and would require a separate, dedicated study.

As requested by the reviewer, we have now mentioned this limitation as well as the related need for additional studies in the paragraph "Discussion".

COMMENT:

- What developmental stages were the fetuses that are analyzed in this study? During developing brain angiogenesis is taking place and a such EC gene expression is changing depending on the developmental stage. Thus, a concern in their methodology is that merging all developmental stages might bias the results in the comparisons. This should be addressed.

ANSWER:

We thank the reviewer for this comment. The developmental stages used in our study were of the following gestational ages/gestational weeks (GWs):

- GW 9
- GW 14.4 – 16.4 (pooled samples as received by the fetal biobank, see methods)
- GW 15.5
- GW 18

We also agree with reviewer 3 that our fetal brain dataset presents a unique opportunity to address the development of the human fetal blood-brain-barrier (BBB). Accordingly, we have now added multiple analyses, experiments and figures that address the different developmental timepoints of the human fetal brain vasculature including:

- additional computational analyses
- additional immunofluorescence (IF) stainings
- new Imaging Mass Cytometry (IMC) stainings

Despite these additional analyses and figures, we would like to mention that the reason to merge the different developmental/fetal timepoints was to compare the developmental/fetal human brain vasculature to the adult human brain vasculature and to the vasculature in various adult brain pathologies.

While we have provided additional analyses, experiments, and figures, we also agreed with Dr. George Caputa in our call of the 25-1-2022 that in order to address the precise changes of the human brain vasculature at different development stages, a separate, dedicated study/atlas will be required.

To further address the development of the fetal BBB, we have indicated EC expression levels of the BBB markers Cadherin 5 (CDH5), Tight Junction Protein 1 (TJP1) = Zona occludens 1 (ZO1), Occludin (OCLN) and Claudin 5 (CLDN5) which show an increased expression with increased gestational age, indicating the temporal development of the BBB in the human fetal brain (see Extended data figure 22). Furthermore, we have added the development of the fetal CNS signature and BBB signature at the different gestational ages (see Extended data figure 22) indicating the establishment of the CNS/BBB signature with increasing gestational age. Moreover, according to a request made by reviewer 3, we have also included the expression levels of the BBB markers and the CNS and peripheral signature according to the AV-specification (sub)clusters. The results show that CNS/BBB signature is highest expressed at the level of small caliber>large caliber EC clusters (see Extended data figure 22).

While the conclusions drawn in our manuscript are valid, we agree with the reviewer that the precise examination of the different gestational ages during neurodevelopment are important and need to be addressed in future studies but can, however, not be addressed in this atlas. We have also mentioned the limitations of the sample number and the need for future investigations of the human fetal brain vasculature and blood-brain-barrier in the "Discussion".

In conclusion and in agreement with the senior editor George Caputa in our call on the 25-1-2022, while we provide these additional analyses as Figures to the Reviewer's we would like to emphasize that the questions regarding age and sex distribution in the different physiological and pathological brain entities as well as regarding the human fetal brain vasculature need to be further addressed in depth in dedicated separate follow-up studies.

COMMENT:

- "CODE AVAILABILITY- the authors write: *Specific code will be made available upon request to T.W.*". As this article contains a large inventory of human samples sequenced, it is important as a resource paper. Therefore, the specific code used for the different analysis should be available upon publishing of the article.

ANSWER:

We thank the reviewer for pointing this out and now provide the specific codes for all different analyses made in the paper in the "Code availability" section, providing the following github link: (<https://github.com/GhobrialMoheb/Molecular-atlas-of-the-human-brain-vasculature-at-the-single-cell-level>).

All data is now accessible via the accession number GSE186771. Additional supplementary information including the supplementary figures are available at 10.6084/m9.figshare.16926574.

COMMENT:

- Fig1c: *This is a composite of UMAP figures, please clarify this in the figure legend.*

ANSWER:

We have adapted the figure legend of Figure 1c and have clarified that the UMAP shown in Figure 1c is a composite of different UMAP figures.

COMMENT:

- *At the main text part of inferred cell-cell interactions, it would be nice to indicate the program/database used, as that has a great impact on the results obtained in these analysis.*

ANSWER:

For inferred cell-cell interactions, we have used two different computational packages, namely CellPhoneDB and CellChat. We have now clarified this in the paragraphs "results" and "methods" and have added the information in the figure legends where applicable.

COMMENT:

- *Fig3 p,q,r What is exactly shown in these dotplots? The legend is really vague. Occurrence in what? Why are the y axis values all divisible by 10? What median rank is depicted on x? What is the selection criteria for the coloured boxes?*

ANSWER:

We are sorry that this was not explained well enough and led to confusion. Reviewer 1 raised a similar concern. We would like to clarify the results displayed in the figures mentioned by the reviewer.

The Figures 3p,q,r,s address the heterogeneity of ECs according to the vessel size respectively according to the main groups of EC clusters. Accordingly, these Figure panels display ECs belonging to large arteries, capillaries, and large veins.

Overall, the Figures 3p,q,r,s illustrate that the ECs are most heterogeneous at the level of the small-caliber vessels being the capillaries. While Figures 3p,q,r highlight this finding by displaying common and specific markers across all brain entities showing that large caliber vessels (large arteries and large veins) have more common markers and less specific markers whereas for small caliber vessels it is the other way round, namely less common markers and more specific markers.

Figure 3s further supports this statement by displaying the Principal Component Analysis (PCA) showing that the capillaries are the most different EC type across the different vascular beds being fetal brain, adult/control brain, brain tumors and brain vascular malformations while the large arteries

and large veins cluster of these vascular beds cluster together. This indicates that small caliber ECs (capillary ECs) are the most susceptible to their local microenvironment whereas large caliber ECs (large artery ECs and large vein ECs) seem to be more conserved across different brain tissue types. Importantly, while these findings are in agreement with what has previously been reported in a mouse EC atlas across different tissues⁴, this has never been shown before in human brain vascular ECs. We are convinced that these are important findings and hope that we hereby clarified these findings for the reviewer.

We have also adapted the figure legend of Figures 3 p,q,r and have clarified the questions raised by the reviewer. The occurrence means the occurrence of a marker gene among the top 50 top cluster markers of an AV cluster in the entities below namely:

- Fetal brain ECs
- Adult/control brain ECs (temporal lobe ECs)
- Temporal lobe adjacent to cavernoma
- Cavernoma
- Arteriovenous malformations
- Hemangioblastoma
- Low grade glioma
- Glioblastoma
- Metastasis
- Meningioma

For example, for large arteries, if a marker gene is a top cluster marker in one of the above mentioned entities it will get an occurrence value of 10%, if a marker is top cluster marker in nine of the above mentioned entities, it will get an occurrence value of 90%. Thus the y-axis is divisible by 10, as there are 10 entities compared.

The rank depicted on the x-axis is the median rank of the top 50 markers genes according to significance and fold change of expression.

The red colored boxes (with red colored dots) are genes that have a median rank of top 15 and are present in more than 70% (> 7/10) of the ten studied brain EC entities listed above. Those genes are thus depicted as "common markers" for a particular AV cluster analyzed.

The blue colored boxes (with blue colored dots) are genes that have a median rank of top 10 and are present in less than 30% (> 3/10) of the ten studied brain EC entities listed above. Those genes are thus depicted as "specific markers" for a particular AV cluster analyzed.

BIOMEX (<https://carmelietlab.sites.vib.be/en/biomex>) was using to perform the cluster similarity analysis (Figure 3p-s) as described in sections 33 and 37 of the package manual.

COMMENT:

- *The authors write: "we performed the standard Seurat clustering (version 4.0.0) workflow" - What were the parameters used for clustering? Do these parameters fit to the various dimensionality checks of the data?*

ANSWER:

We thank the reviewer for pointing this out and have added the parameters for clustering in the paragraph "methods" to clarify this. Also, we can indeed confirm that the parameters used fit to the various dimensionality checks for the data (Extended Data Figure 10).

COMMENT:

- *"Differential expression was computed using the wilcoxauc function implemented in the github package presto." Insert link to GitHub page, there are many GitHub repositories called presto.*

ANSWER:

We thank the reviewer for pointing this out and have inserted the link to the GitHub package (<https://github.com/immunogenomics/presto>) as described in <https://www.biorxiv.org/content/10.1101/653253v1>. We have also adapted the text in the paragraph "methods" to clarify this.

COMMENT:

- "Of note, the APP-CD74, COPA-CD74, and MIF-CD74 ligand-receptor pairs that were predicted to mediate MHC class II signaling in EC-EC interactions (Extended Data Figure 35a-c)" There is no EDFig35.

ANSWER:

We apologize for this mistake. The APP-CD74, COPA-CD74, and MIF-CD74 ligand-receptor pairs that were predicted to mediate MHC class II signaling in EC-EC interactions are displayed in (Extended Data Figure 31a-c) and not in Extended Data Figure 35a-c. We have corrected this mistake in the paragraph "results" accordingly.

COMMENT:

- "DESeq2 package (v1.30.1) was used to perform pseudobulk differential expression analysis" and "Differential expression was computed using the wilcoxauc function implemented in the github package presto" together: Which was used for what? One is a pseudobulk approach, the other is a sc method.

ANSWER:

We apologize for this confusion and have clarified this by adapting the text in the paragraphs "methods".

E. Conclusions:

The authors mainly summarize their observations and validations of their scRNAseq and bioinformatic analysis.

F. Suggested improvements:**COMMENT:**

The manuscript is overwhelming, there are too many figures, subfigures, comparisons, etc (6 main figures + 30 extended figures). This makes the paper very-very complicated to follow. It also greatly increases the chances for statistical errors and false positive hits. This should be considered and, while still providing all the information and datasets obtained, narrow the figures and presented data so that a more specific focus on the result analysis can be distilled. In this regards, the identification of MHC-class II gene expression in ECs in different pathologies and their role in immune surveillance could be a very interesting focus.

ANSWER:

We thank the reviewer for this comment and have addressed it accordingly. First, we would like to mention that in agreement with senior editor Dr. George Caputa (in our discussion on the 25-1-2022), we decided to keep the six main figures in the paper as they address the six key messages of this manuscript, namely:

- Construction of a molecular single-cell atlas of the human brain vasculature (Figure 1)
- Inter-tissue heterogeneity of brain vascular endothelial cells (Figure 2)
- Alteration of AV-specification in pathological brain vascular endothelial cells (Figure 3)
- Alteration of CNS-specificity in pathological brain vascular endothelial cells (Figure 4)
- Upregulation of MHC class II receptors in pathological brain vascular endothelial cells (Figure 5)
- A key role for endothelial cells in the human brain neurovascular unit (Figure 6)

Accordingly, we would also like to briefly mention that the focus of this manuscript is really to construct an atlas of ECs and PVCs across development and disease (Figure 1), to then compare the transcriptomic profiles of brain ECs across these entities (Figure 2), and to subsequently focus on key biological characteristics of brain ECs across development, adulthood and disease including AV-specification (Figure 3), CNS-specificity (Figure 4), and the unexpected finding of MHC class II receptors upregulation (Figure 5). Finally, we address the interaction of ECs with PVCs in the brain neurovascular unit across development, adulthood, and disease by describing cellular signaling networks/ ligand-receptor interactions where we focus on MHC class II signaling indeed (Figure 6).

In addition to the focus provided on MHC class II signaling in cell-cell signaling networks/ligand-receptor interactions (Figure 6), this focus on MHC class II has also further been manifested in additional validation experiments including IMC. IMC revealed physical proximity between MHC class II expressing ECs and immunological cells including CD68⁺ and/or CD11b⁺ microglia and macrophages, CD4⁺ and/or CD8⁺ other immunological cells as well as other PVCs (Figure 6, Extended Data Figures 37). The inferred cell-cell communication/ligand-receptor networks as well as the physical proximity between MHC class II expressing ECs and perivascular cells including immunological cells indicate the relevance of MHC class II signaling in the neurovascular unit in different brain diseases and suggest a possible role in immune surveillance. IMC images have been added for both Figure 5 and Figure 6, thereby providing a stronger focus on MHC class II genes.

Moreover, we have emphasized the potential role of MHC-class II genes in ECs in different pathologies in immune surveillance in the paragraph “Discussion”, also stating that the functional investigation of MHC class II genes in immune surveillance could be a very interesting focus for future studies.

With regard to narrowing the figures while still providing all information and datasets obtained, we have made a suggestion of what can be moved from the Extended Data Figures to either Supplementary Figures or Figures to the Reviewer. However, we prioritized to provide all the data to answer the reviewer's requests and therefore we were not able to narrow down the number of Extended Data Figures at this stage. Once the paper hopefully accepted, we would of course be happy to reduce the number of Figures.

Finally, regarding the reviewer's concern about statistical errors and false positive hits, we would like to mention that we have reported the False Discovery Rate (FDR) for both pathway analyses and Differentially Expressed Genes (DEG) throughout all figures. In addition, we have also provided detailed statistical analyses and output in the provided Supplementary Tables. Most of our analyses are standard workflows and we have now published the source code to improve reproducibility and allowing detailed checks of our results.

G. References:

COMMENT:

Ok, no concerns there.

ANSWER:

We thank the reviewer for this comment.

H. Clarity and context:

COMMENT:

See comment in F.

ANSWER:

We have adapted the manuscript, figures and Extended Data Figures accordingly, see comment in F.

Reviewer #3:**COMMENT:**

The manuscript by Walchli et al presents a large and comprehensive single cell transcriptomics atlas for human brain vasculature. The authors provide the atlas for healthy brain, for multiple disorders, and for development. In addition to single cell data, there are extensive validations using smFISH and immunohistochemistry. The atlas, without a doubt, will be a great resource for the scientific community. However, before accepting the manuscript, a number of issues should be addressed.

ANSWER:

We thank the reviewer for their comments on our manuscript.

We now indeed provide additional experimental data in the revised manuscript that addresses all the concerns raised by the reviewers. The new data further validates the findings of our scRNA-sequencing atlas and includes:

- new bulk RNA-sequencing data
- additional immunofluorescence (IF) stainings
- new Imaging Mass Cytometry (IMC) stainings
- new spatial transcriptomics data

thereby addressing key reviewer's concerns about additional validation experiments, additional patients' samples, and inclusion of fetal brain tissues (for IF and IMC).

In addition, we have addressed all issues raised by the reviewers and by the editor.

Major points:**COMMENT:**

1. Different ways for integration have been used for unsorted cells, CCA, and sorted cells, RPCA. CCA is used to integrate samples with conserved cell types, but with large expression differences, whereas RPCA has a much "softer" way of integration. Importantly, CCA is not the best way to integrate healthy with disease samples when the goal is to study differential expression, and in this case CCA correction could be too aggressive and result in overcorrection. Thus, CCA could help to identify similar cell types in control and disease, but certainly is not the best way to reveal gene expression differences between conditions. Therefore, differential expression analysis should be done using another algorithm for integration.

Finally, the authors are also recommended to read this manuscript on comparison of algorithms for integration

[https://www.biorxiv.org/content/10.1101/20](https://www.biorxiv.org/content/10.1101/2020.05.22.111161v2)

20.05.22.111161v2

where Seurat v3 performs well only on broader cell types, whereas its performance in a complex biological context is rather poor. The authors claim to provide a high-resolution atlas with >0.5 million brain vasculature cells in a complex biological context, including several disease conditions, to use for the scientific community. Thus, the data must be solid. The authors should confirm that integration with another method (that preserves biology better) shows similar results (including subtype annotation and differential expression) and if not then a better integration algorithm that allows to study complex biological systems and preserves biology should be used for this study. Additionally, the integration across the study should be done with the same algorithm in order to be able to cross-compare the datasets.

ANSWER:

We thank the reviewer for this comment and the very helpful inputs and feedback regarding the different methods of integration. Reviewer 1 has made a similar comment. We would like to clarify a few aspects regarding the integration methods that we have used for the different sorted (ECs) and unsorted (ECs and PVCs) datasets:

- i) we have used CCA for integration of:
 - individual patients within a given brain entity (e.g. -Integration of the five fetal brain samples' EC together.

- Integration of the six TL patients' ECs together.
 - Integration of the five GBM patients' ECs together.
 - Integration of the five AVM patients' ECs together, etc. – and the same for every individual entity) for sorted ECs (sorted samples).
- individual patients within a given brain entity (e.g.
 - Integration of the five fetal brain samples' cells together.
 - Integration of the six TL patients' cells together.
 - Integration of the five GBM patients' cells together.
 - Integration of the five AVM patients' cells together, etc.... – and the same for every other individual entity) for unsorted ECs and PVCs (unsorted samples).
- ii) we have used RPCA for integration of:
 - different brain entities for sorted ECs (sorted samples)
 - iii) we did not do the integration/batch correction of the different brain entities for the unsorted ECs and PVCs (unsorted samples)
 - the reason is that the different cell types in the unsorted samples (ECs and PVCs) are not the same across different the brain entities (fetal brain, TL, all brain diseases) (and the cell types cluster by cell type) and thus it does not make sense to integrate/batch correct the different entities.
 - this is obviously different for the sorted samples which contain only ECs and thus integration and batch correction across the different brain entities makes sense (see ii)).

Notably, differential gene expression that is used for cell annotation/AV-annotation and pathway analysis is always done on the non-integrated/non batch corrected counts (= the “RNA assay of Seurat”) and thus does not involve batch corrected counts – which is also the recommendation of SEURAT package developers.

Regarding the comparison of algorithms for integration, we have plotted the RPCA annotation onto the Harmony-, scANVI-integrated and CCA-integrated datasets and the results regarding the AV-annotation were very consistent (Reviewer Figure 2).

In addition, we have also read and accounted for the paper mentioned by the reviewer:

<https://www.biorxiv.org/content/10.1101/2020.05.22.111161v2>

which was meanwhile published in *Nature Methods*:

<https://www.nature.com/articles/s41592-021-01336-8>

Accordingly, we have performed the additional integration methods scANVI which is the top ranked method in this paper. The comparison to the methods of integration that we used, namely Harmony, Seurat v4 RPCA and Seurat V4 CCA showed very similar results (Reviewer Figure 2) and we thus conclude that. integration with other methods show similar results.

COMMENT:

2. Too little metadata and QC are available. This is an atlas; thus, the authors should be very transparent about their data. For instance, outcomes of sorting (e.g. % out of total, % of cells for live/dead stain etc) should be provided. For QC, plot distribution of cells with high mitochondrial counts/per sample, plot number of genes per cell detected for each sample, plot genes per cell type, plot UMIs per cell type etc. In fact, I could not even find information about the numbers of cells sequenced per sample, the numbers of reads per sample, and an average number of genes/UMIs detected in a sample (maybe it is available, just difficult to find, then include it in metadata in Suppl table 1). Additionally, disease metadata is rather poor – for the temporal lobe, are all samples from patients with temporal lobe epilepsy? Any pathology in the cortex? For one sample, FCD1c is shown, anything for other samples? Was only the cortex used from these samples or also the hippocampus? If the hippocampus was used – a comparison of HS samples with non-HS should be done to either show that the vasculature is not affected by sclerosis, or there are some gene expression/cell type compositional changes that are related to sclerosis.

ANSWER:

We apologize and have now provided additional metadata and QC in the manuscript (Extended Data Figure 3 and Supplementary Tables 3,4). We have addressed all parameters mentioned by the reviewer in order to clarify all details regarding metadata and QC and to assure fully transparency of the data in our atlas. The respective information can be found in Extended Data Figure 3 and Supplementary Tables 3,4. In fact, we have addressed/displayed the numbers of cells sequenced per entity that have passed the QC in Supplementary Table 2 but apologize for the fact that it wasn't displayed clearly enough and thus difficult to find. We have now additionally displayed the number of genes per cell detected for each sample/entity, the number of genes per cell type in each entity, and the number of UMIs/reads per cell type in each entity in Supplementary Table 2,4. Regarding cells with high mitochondrial counts/sample, we excluded cells with mitochondrial counts/sample > 25% of all.

Regarding the temporal lobe tissue used, reviewer 2 had a similar comment (see comment C "The authors use as control ECs isolated from lobectomy in pharmacoresistant epilepsy patients. Thus it might be that those ECs might already present significant changes with respect to a real control brain EC. This should be considered and fully addressed." there).

We have only analyzed the neocortical resection of the temporal lobectomies and did not analyze the hippocampus.

Regarding the disease metadata about the temporal lobe samples, all available patient data is displayed in Supplementary Table 1.

We would like to clarify that the temporal lobe tissue from which adult/control brain ECs are derived is taken from the neocortical part of temporal lobectomies. The neocortical part of temporal lobectomy surgeries is the temporal lobe that overlies the epileptic focus in the hippocampus (see Supplementary Table 1). The hippocampus tissue was not used in our experiments and analyses. While the neocortical part harbors no known epileptic lesion, we cannot fully exclude the possibility of very slight/minor pathological changes in these brain samples.

However, we would like to emphasize that this is the most "normal" brain tissue available from fresh surgical samples and consist in the "gold standard" for human control brain tissue from fresh surgical samples (as there are no better human control brain ECs available). This question was also addressed previously in the landmark paper Hodge et al., *Nature*, 2019⁴⁰, where the authors compared scRNA-seq data from epilepsy patients to post-mortem brain tissue. Notably, they found that the transcriptomic profiles of neurons in the fresh surgical epilepsy samples and the post-mortem tissue looked very similar arguing against the "pathological" brain issue. Importantly, the authors used neocortical brain tissue from the middle temporal gyrus for both the fresh epileptic samples as well as the postmortem tissue, which is exactly the anatomical location where we got temporal lobe tissue from (the neocortical part of the temporal lobectomies = middle temporal gyrus).

Moreover, as requested by reviewer 1 (see above), we have compared our temporal lobe EC transcriptomic profiles to the control brain tissue of three recently published papers^{3,6,7}. Even though these authors did not use sorting strategies to enrich for brain endothelial cells, the transcriptomic profiles of the adult/control brain ECs in our study and the adult/control brain ECs in these three papers are very similar (see Extended Data Figure 6).

Taken together, we thus conclude that the adult/control brain ECs in our manuscript show similar transcriptomic profiles when compared to the ones used in the recently published literature^{3,6,7,40}.

COMMENT:

3. Although the authors provide some markers for potential transcriptomic subtypes in Figure 2 and Extended Figure 7, it is not very convincing. Additional subclustering according to the type of vasculature, e.g. venous compartment, arterial compartment, capillary compartment, should help to subdivide the large cloud in Figure 2a and hopefully could provide better evidence for the existence of subclusters. Such compartment-specific plots should be also implemented for follow-up analyses, instead of Figure 2a UMAP, where appropriate.

The same holds true for UMAPs in Extended Figure 9 and Figure 3, annotation is not convincing, and a similar approach as above could help to produce a more convincing annotation.

ANSWER:

We thank the reviewer for this comment and would like to mention that we show the subcluster analysis in Extended Data Figures 10a,f-h. As mentioned in the paragraph "Results", we pooled EC

clusters of similar identity (e.g. “arterial”, “venous”, “capillary”) for further downstream analyses, which results in the EC clusters shown in Figure 2a-d, Figure 3a,e,i and Extended data figure a,n. We would like to emphasize that even after pooling we still provide 14 EC clusters and that every compartment is subdivided even in the pooled analysis. The arterial compartment is composed of the “large artery”, “artery”, “arterioles” clusters, the venous compartment is composed of the “veins”, “veins” “venules” clusters, and the capillary compartment is composed of the “capillary” and “angiogenic capillary” clusters (Figure 2a). These 14 EC clusters provide a more accurate representation of the underlying biology as compared to other recent publications which display between 3-6 clusters^{3,6,7} :

However, to address this question and to clarify the sub-clustering and to further support our final choice for the EC clusters, we provide a Jaccard stability analysis on the different clusters and subclusters (Extended Data Figure 10). Regarding the stability analysis on the clusters, we referred to recent publication Tang et al., *Bioinformatics*, 2021⁴³. (<https://pubmed.ncbi.nlm.nih.gov/33165513/>) which evaluates cluster stability by subsampling and by using the Jaccard similarity index. The results justify our final choice of clusters as the subclusters that were pooled for further downstream analysis very adequately mapped to the 14 pooled clusters are shown in (Extended Data Figure 10).

COMMENT:

4. Related to the above, and even more importantly, such subclustering has to be done for identification of BBB signature, when comparing peripheral vs CNS. For instance, I am sure the authors know a hot debate about BBB in the fetal brain, which was more or less settled only recently. This dataset provides for a unique opportunity to study the fetal BBB at high resolution, and currently this opportunity is missed. Overall, for BBB, the authors should add:

ANSWER:

We thank the reviewer for this comment and agree that our fetal brain dataset presents a unique opportunity to address the development of the human fetal blood-brain-barrier (BBB). A similar comment was also made by reviewers 1 and 2, who also required a more thorough analysis according to the timeline of the human fetal BBB.

Because the development of the human fetal BBB (which occurs between Gestational Week (GW) 8 to 18^{44, 45} and peaks at GW35⁴⁶) has important biological and medical implications and has been a matter/subject of intense debate and controversy^{47,48}, we took advantage of our single-cell dataset to study human fetal BBB development at high resolution and defined a human BBB signature including the four BBB marker genes CDH5, TJP1 (= ZO1), OCLN and CLDN5 (Extended Data Figure 22, Supplementary Table 15, Supplementary figure 28).

- We observed an upregulated BBB signature with increasing gestational age (GW9, GW14.4-16.4, GW15 and GW18) and a corresponding gradual increase of expression levels of the individual BBB markers (Extended Data Figure 22b,e-j), in agreement with what was previously reported for Claudin-5 and Occludin expression in the human fetal brain³⁵.
- At GW18, the BBB signature expression levels were below those in the adult/control brain (Extended Data Figure Xa,x), in line with the known peak of angiogenesis and BBB development at GW35⁴⁶.
- When evaluating the BBB signature in the different arteriovenous compartments, we observed a higher expression in small > large caliber vessels across the different developmental stages (Extended Data Figure 22d).

These observations provide indications of the temporal development of the BBB in the human fetal brain (Extended Data Figure 22).

Furthermore, we have added the development of the fetal CNS signature at the different gestational ages (Extended Data Figure 22a-c) indicating the establishment of the CNS signature with increasing gestational age.

Moreover, according to a request made by reviewer 3, we have also included the expression levels of the BBB markers and the CNS and peripheral signature according to the AV-specification (sub)clusters.

Importantly

- the BBB signature showed higher expression in small > large caliber vessels (Extended Data Figure 22d)

- the CNS signature showed higher expression in small > large caliber vessels (Extended Data Figure 22d)

While the conclusions drawn in our manuscript are valid, we agree with the reviewer that the precise examination of the different gestational ages during neurodevelopment are important and need to be addressed in future studies but can, however, not be addressed in this atlas. We have also mentioned the limitations of the sample number and the need for future investigations of the human fetal brain vasculature and blood-brain-barrier in the described these findings in the paragraph "Discussion". In conclusion and in agreement with the senior editor George Caputa in our call on the 25-1-2022, while we provide these additional analyses, we would like to emphasize that the questions regarding human fetal brain vasculature need to be further addressed in depth in dedicated separate follow-up studies.

COMMENT:

4a. better resolution for BBB analysis, which could be done by subclustering according to venous, arterial, and capillary compartments

ANSWER:

See response above regarding the fetal brain ECs and regarding the subclustering that we now did for Figure 2 and 3 and Extended Data Figure 10.

When evaluating the BBB signature in the different arteriovenous compartments, we observed a higher expression in small > large caliber vessels across the different developmental stages (Extended Data Figure 22d).

In addition to our response to comment 4c. we have also plotted the expression levels of both the different BBB markers Cadherin 5 (CDH5), Tight Junction Protein 1 (TJP1) = Zona occludens 1 (ZO1), Occludin (OCLN) and Claudin 5 (CLDH5) individually as well as of the BBB signature comprising these four BBB markers according to the AV-specification (AV compartments) (see Extended Data Figure 22d, Supplementary Figure 28). The results show that those BBB markers are expressed across all AV clusters, with higher expression in small > large caliber vessels.

COMMENT:

4b. same better resolution for BBB dysfunction module, and here also comparison for a BBB signature to a neurodegenerative disorder:

<https://www.biorxiv.org/content/10.1101/2021.04.26.441262v1.full>

ANSWER:

We thank the reviewer for this comment and the reference to this very interesting paper by Yang et al., which was meanwhile published in *Nature* early 2022⁴⁹.

Accordingly, and as also suggested by reviewer 1, we have now compared the following BBB dysfunction modules and BBB signatures (Extended data figure 25, Supplementary table 16):

- i) BBB dysfunction module (derived from mouse brain Munji et al., *Nat Neurosci*, 2019⁴¹).
- ii) BBB signature in Alzheimer's disease⁶.
- iii) BBB signature in Huntington's disease^{6,7}.
- iv) BBB signature in AVMs³.
- v) BBB signature of the pathological brain ECs in our dataset.
- vi) BBB signature of the brain tumor ECs in our dataset.
- vii) BBB signature of the brain vascular malformation ECs in our dataset.

The results show specific differentially regulated genes in ECs in Alzheimer's disease versus temporal lobe in humans⁶, in Huntington's disease versus temporal lobe in humans⁷, in various brain pathologies (multiple sclerosis, epilepsy, stroke, traumatic brain injury) versus adult control brain in mice and in vascular-dependent brain pathologies (various brain tumors, various brain vascular malformations) versus temporal lobe in humans (our dataset) (see Extended Data Figure 25a'-e').

Interestingly, we also found 14 differentially expressed genes (DEGs) that are upregulated in ECs of the different pathologies in our human dataset (human brain tumor ECs, human brain vascular malformation ECs) and the ECs of the different pathologies in the Munji et al., *Nat Neurosci*, 2019 mouse dataset (multiple sclerosis, epilepsy, stroke, traumatic brain injury) (see Supplementary Table

16). Notably, 13 DEGs were overlapping between human brain tumor ECs and the mouse ECs in Munji et al and 17 DEGs between the human brain vascular malformation ECs and the mouse ECs in Munji et al (see Supplementary Table 16).

In summary, the results show differentially regulated as well as common genes among the BBB signatures of pathological brain ECs in our dataset with the BBB signatures in human Alzheimer's⁶ and human Huntington's diseases⁷, human brain AVMs³ (Extended Data Figure 25a'-e').

Pathological brain AVM signature

Comparison of the pathological brain AVM signature from our dataset (brain AVM ECs versus temporal lobe ECs, see Extended Figure 25c') to the pathological brain AVM signature in the Winkler et al dataset³ (brain AVM ECs versus temporal lobe ECs revealed common differentially expressed genes (DEGs)).

The results of the comparison show 292 different and 72 common differentially expressed genes (DEGs) (see Extended Data Figure 25c'). The observed differences might be due to the difference in cell numbers analysed, to different techniques of isolation used, due the relatively low sample number in both datasets (n=5) or linked to the rupture status of brain AVMs.

COMMENT:

4c. developmental resolution; even though it is only 5 samples, the trajectory of BBB from GW9 to GW18 would be great data to add. The period between GW8 and 18 is usually considered as the period for major development of the BBB.

ANSWER:

This is indeed a very interesting aspect that has also been raised by reviewers 1 and 2, who also required a more thorough analysis according to the timeline of the human fetal BBB.

Regarding the human fetal BBB, a similar question was asked by reviewer 1. To address the reviewer's concern regarding the development of the fetal brain vasculature, we have thus performed additional analyses for the fetal brain samples at the different Gestational Weeks (GWs) 9 to 18. Notably, we addressed arteriovenous specification, CNS-specificity, and blood-brain-barrier (BBB) markers (Extended Data Figures 12,22,29). The results show that the key conclusions we make when comparing to other brain entities (adult/control and disease entities) are valid and are not mainly influenced by the age of the different fetal brain samples. For instance, the conclusions made regarding the arteriovenous specification and the cellular/cluster composition including the significantly increased amount of angiogenic capillaries in the fetal brain as compared to the temporal lobe hold true for all the different fetal brains at different GWs (illustrated also by the scCODA analyses, see Extended Data Figure 12 and Figure to Reviewer 3, Supplementary table 10), moreover there was no significant compositional changes among the different GW analyzed (Supplementary table 10). Thus, the key conclusions made for the AV-specification when comparing the pooled fetal and pooled adult brain ECs reflect the AV-specification at the different gestational ages in an accurate manner. To further address the development of the fetal BBB, we have indicated EC expression levels of the BBB markers Cadherin 5 (CDH5), Tight Junction Protein 1 (TJP1) = Zona occludens 1 (ZO1), Occludin (OCLN) and Claudin 5 (CLDN5) which show an increased expression with increased gestational age, indicating the temporal development of the BBB in the human fetal brain (see Extended Data Figure 22, Supplementary Figure 28). Furthermore, we have added the development of the fetal CNS and BBB signature at the different gestational ages GWs (see Extended Data Figure 22 and Supplementary Figure 28) indicating the establishment of the CNS signature with increasing gestational age. Moreover, according to a request made by reviewer 3, we have also included the expression levels of the BBB markers and the CNS and peripheral signature according to the AV-specification (sub)clusters. The results show that the expression of the CNS and BB signatures are higher at the level of small caliber vessels (capillaries, arterioles and venules) than at the level of larger caliber vessels, (see Extended Data Figure 22, Supplementary Figure 28).

While the conclusions drawn in our manuscript are valid, we agree with the reviewer that the precise examination of the different gestational ages during neurodevelopment are important and need to be addressed in future studies to most accurately describe the human fetal BBB. This scientific question can, however, not be addressed in this atlas but requires a dedicated follow-up study. We have also mentioned the limitations of the sample number and the need for future investigations of the human fetal brain vasculature and blood-brain-barrier in the described these findings in the paragraph "Discussion".

In conclusion and in agreement with the senior editor George Caputa in our call on the 25-1-2022, while we provide these additional analyses as Figures to the Reviewer's we would like to emphasize that the questions regarding the human fetal brain vasculature need to be further addressed in depth in dedicated separate follow-up studies.

COMMENT:

4d. comparison of developmental trajectory per compartment with adult BBB markers – when the BBB is complete, based on adult marker expression, is it as early as GW9 or only at GW18?

ANSWER:

This question has also been raised by the reviewer 1. When evaluating the BBB signature in the different arteriovenous compartments, we observed that the expression is higher at the level of small>large caliber vessels across the different developmental stages (Extended Data Figure 22d). In addition to our response to comment "4c" of this reviewer (see above), we have also plotted the expression levels of the different BBB markers Cadherin 5 (CDH5), Tight Junction Protein 1 (TJP1) = Zona occludens 1 (ZO1), Occludin (OCLN) and Claudin 5 (CLDN5) individually as well as of the BBB signature comprising these four BBB markers according to the AV-specification (AV compartments) (see Extended Data Figure 22d and Supplementary Figure 28). The results show that those BBB markers are expressed across all AV clusters, with higher expression at the level of small>large caliber vessels.

These observations are supported by the results of the CNS signature resolved according to the AV-specification (AV compartments) (Figure 4h, Extended Data Figure 22d).

COMMENT:

5. Changes in cell type composition between control and pathological conditions should be statistically tested, where possible, given that enough samples are available, e.g. by scCODA (Buttner et al. 2021 Nat Commun) that so far shows the best performance controlling for false discoveries in abundancies.

ANSWER:

We have addressed the changes in cell type/cluster composition between control and pathological conditions according to the paper by Buttner et al⁴⁹.

The analyses shows significant upregulation of the "angiogenic capillary" cluster in fetal brain vs adult/control brain as well as in brain tumors vs adult/control brains, while the "venous" clusters (Large veins and veins clusters) were upregulated in brain vascular malformations vs adult/control brain (Supplementary table 10, Figure to Reviewer 3), in agreement with what we described in the paragraph "results".

The results show the following significant changes in EC compartments/EC clusters:

- i) a significant upregulation of angiogenic capillaries
 - in fetal brain ECs versus adult/control brain ECs
 - in brain tumor ECs versus adult/control brain ECs
- ii) a significant upregulation of venous compartments
 - in brain vascular malformation ECs versus adult/control brain ECs
- iii) For individual entities:
 - Fetal brains vs adult/control brains:
 - > Significant up-regulation of:
 - Angiogenic capillary
 - Proliferating cell
 - > Significant down-regulation of:
 - Arterioles
 - Capillary
 - TL adjacent to cavernomas vs adult/control brains:
 - > Significant up-regulation of:
 - non

- > Significant down-regulation of:
 - non

- Cavernoma vs adult/control brains:
 - > Significant up-regulation of:
 - Venule
 - > Significant down-regulation of:
 - Arteriole
 - Capillary

- Arteriovenous malformations vs adult/control brains:
 - > Significant up-regulation of:
 - Large artery
 - Artery
 - Angiogenic capillary
 - Venule
 - Vein
 - Large vein
 - > Significant down-regulation of:
 - non

- Hemangioblastomas vs adult/control brains:
 - > Significant up-regulation of:
 - Angiogenic capillary
 - > Significant down-regulation of:
 - Arteriole
 - Capillary

- Low grade gliomas vs adult/control brains:
 - > Significant up-regulation of:
 - Angiogenic capillary
 - > Significant down-regulation of:
 - non

- Glioblastomas vs adult/control brains:
 - > Significant up-regulation of:
 - Angiogenic capillary
 - Stem-to-EC
 - > Significant down-regulation of:
 - Artery
 - Arteriole
 - Capillary

- Metastases vs adult/control brains:
 - > Significant up-regulation of:
 - Angiogenic capillary
 - Stem-to-EC
 - > Significant down-regulation of:
 - Artery
 - Arteriole
 - Capillary

- Meningiomas vs adult/control brains:
 - > Significant up-regulation of:
 - Angiogenic capillary
 - Vein
 - > Significant down-regulation of:
 - Arteriole
 - Capillary

The results show the following significant changes in cell compositions:

- All malformations vs adult/control brains:
 - > Significant up-regulation of:
 - Smooth muscle cells
 - Oligodendrocytes
 - B cells
 - Neutrophils
 - > Significant down-regulation of:
 - Non

- All tumors vs adult/control brains:
 - > Significant up-regulation of:
 - Tumor cells
 - > Significant down-regulation of:
 - Microglia and Macrophages

- Fetal brains vs adult/control brains:
 - > Significant up-regulation of:
 - Astrocytes
 - Stem cells
 - Neurons
 - > Significant down-regulation of:
 - T cells
 - Microglia and Macrophages

- TL adjacent to cavernomas vs adult/control brains:
 - > Significant up-regulation of:
 - non
 - > Significant down-regulation of:
 - non

- Cavernoma vs adult/control brains:
 - > Significant up-regulation of:
 - non
 - > Significant down-regulation of:
 - Astrocytes

- Arteriovenous malformations vs adult/control brains:
 - > Significant up-regulation of:
 - Smooth muscle cells
 - Oligodendrocytes
 - T cells
 - NK cells
 - B cells
 - > Significant down-regulation of:
 - non

- Hemangioblastomas vs adult/control brains:
 - > Significant up-regulation of:
 - Endothelial cells
 - > Significant down-regulation of:
 - Oligodendrocytes
 - Microglia and Macrophages

- Low grade gliomas vs adult/control brains:
 - > Significant up-regulation of:
 - non
 - > Significant down-regulation of:
 - non

- Glioblastomas vs adult/control brains:
 - > Significant up-regulation of:
 - T cells

- Neutrophils
- Stem cells
- Tumor cells
- > Significant down-regulation of:
 - non

- Metastases vs adult/control brains:
 - > Significant up-regulation of:
 - T cells
 - Tumor cells
 - > Significant down-regulation of:
 - non

- Meningiomas vs adult/control brains:
 - > Significant up-regulation of:
 - Tumor cells
 - > Significant down-regulation of:
 - Smooth muscle cells
 - Oligodendrocytes
 - Fibroblasts

We have added the scCODA results in the Figure to Supplementary table 10, Reviewer 3 and have mentioned the statistically significant compositional changes in the paragraph “results”. In agreement with the scCODA analyses, we can therefore exclude the likelihood of false discoveries in abundancies.

COMMENT:

6. Potential sex-specific expression should be analyzed.

ANSWER:

A similar point was raised by reviewer 1.

These data are displayed as the Figures to the Reviewer 1 and 3 and Supplementary Tables 10,20. The results show that the number of significant differentially expressed genes (DEGs) comparing the sexes is minimal (and no biologically meaningful pathways were enriched) and is no different from random shuffles, moreover scCODA analysis showed that there was no significant compositional changes in comparing the sexes.

It is crucial to mention that addressing sex-specific expression differences in the different brain entities in a conclusive manner would however require a separate dedicated study. This is also in agreement with our discussion with senior editor George Caputa on the 10-1-2022.

In summary, when addressing the parameter sex in all entities, we found few DEGs (that weren't different from random shuffles) that did however not result in biologically meaningful pathways (Figure to the Reviewer 3, see also above). We are well aware of the limited sample size and have mentioned this in the paragraph "Discussion".

COMMENT:

7. Potential brain region-specificity in subtypes/expression should be analyzed (different brain regions are available e.g. in glioma).

ANSWER:

We thank the reviewer for this interesting question regarding brain region-specific subtypes and expression patterns. We have performed differential expression analysis between the low(er) grade gliomas and the glioblastomas (Supplementary Figure 6). However, the number of gliomas in the different brain lobes is too low to perform an adequate comparison. Please find below the overview of the samples in the different brain lobes for both low(er) and high-grade gliomas (see also Supplementary Tables 1):

- Frontal lobe
 - Two low(er) grade gliomas

- Three glioblastomas
- Temporal lobe
 - one low(er) grade glioma
 - two glioblastomas
- Parietal lobe
 - one low(er) grade glioma
 - zero glioblastoma
- Occipital lobe
 - zero low(er) grade gliomas
 - zero glioblastomas

Notably, the frequency of gliomas in the different brain lobes is in agreement with what is reported in the literature, namely frontal > temporal > parietal > occipital⁵⁰.

However, given the very low sample number by brain region and the fact that a pool of low(er) grade gliomas and glioblastomas represents a crucial confounding factor, we decided that such an analyses would not be feasible.

We would like to emphasize that the sample number of brain regions/brain lobes per disease entity is not high enough in order to draw final conclusions. Therefore, addressing brain-region-specific expression differences in the different brain entities (including gliomas) in a conclusive manner requires a separate dedicated study. While this is a very interesting questions without any doubt, answering it goes beyond the scope of this atlas. This is also in agreement with your discussion with senior editor George Caputa on the 25-1-2022. We have nevertheless included information about brain lobe in our sample annotation to enable future studies to make use of our data to help answer this question (Supplementary table 1).

COMMENT:

8. Monocle lineage tracing often does not show a clear picture in this manuscript. Velocity should be better applicable here, see La Manno et al. Nature 2018, Bergen et al. Nat Biotech 2020, and Lange et al. bioRxiv 2020.

In addition, how the red line with arrows for trajectories was produced? Was it automatically generated by Monocle, or was it the author's interpretation, and was inserted manually? This should be stated in the figure legends of the corresponding figures.

ANSWER:

We thank the reviewer for this comment. To address lineage tracing, we have performed different additional analyses.

To address the pseudospace (AV-specification) and the pseudotime, we performed computational analyses using Tools for Single Cell Analysis (TSCAN) pseudotime⁵¹ and the results are very similar to and in agreement with what we show using Monocle pseudotime (Extended Data Figure 21, Figure 3). Notably, both Monocle pseudotime and TSCAN pseudotime reproduce a lineage tracing along the AV-axis, from arteries to capillaries to veins (Figure 3, Extended Data Figure 19, Supplementary Figure 27).

In addition, based on the reviewer's suggestion, we have applied RNA velocity^{52,53} and diffusion map⁵⁴, which both address the pseudotime but not the pseudospace (Extended Data Figure 21). Both these methods did not reveal a clear trajectory according to the AV-axis (in opposition to Monocle- and TSCAN pseudotime) but revealed vectors from multiple EC clusters (including arteries, capillaries, and veins) towards angiogenic capillaries (RNA velocity PAGA plot for brain tumors), while for vascular malformations the PAGA vectors converge on the venous compartments. Moreover, the stem-to-EC and EndoMT clusters were "outside" of the AV-zonation (at the edge of the velocity pseudotime/diffusion map) These data are displayed as Extended Data Figure 21.

Regarding the red line, this was indeed inserted manually, similarly as it was done in Buechler et al., *Nature*, 2021⁵⁵. This is now described in the figure legends of Figure 3 and Extended Data Figure 19 as well as in the paragraph "methods".

COMMENT:

9. Overall, a better description of computational methods should be provided.

ANSWER:

Accordingly, we have now more thoroughly described the computational methods in the paragraph “methods”.

Minor points:**COMMENT:**

1. For astrocytic marker – use *Aqp4* instead of *Gfap*, *Gfap* expression is activation dependent.

ANSWER:

In response to this reviewer’s comment, we have now displayed the expression for both astrocytic markers AQP4 and GFAP (see Extended Data Figure 2m,n).

COMMENT:

2. It is not clear how endothelial and perivascular cells were obtained from unsorted samples. Did the authors sequence all cells, including neurons etc, and then filter out endothelial and perivascular cells computationally? The procedure how endothelial and perivascular cells were obtained from unsorted samples should be described in detail and necessary illustrations/plots should be provided.

ANSWER:

We apologize that this methodology wasn’t described clearly enough in the manuscript. To clarify the reviewer’s comment, we have now adapted the formulation about how the sorted and unsorted samples were processed in the paragraphs “results” and “methods” as well as in the figure legend of Figures 1a,b, which describe the working model and have provided a detailed description.

First, we would like to mention again that with “perivascular cells (= PVCs)” we describe all non-endothelial cells (= ECs) throughout this manuscript. This was already described in the paragraphs “Introduction” and “Results” but we have now further clarified this in the text.

In brief, from an experimental point of view, the tissue samples were enzymatically digested using collagenase and dispase immediately after tissue harvesting (see Figure 1a,b, for technical details see paragraph “methods”).

- For the unsorted EC and PVCs samples, the digested tissue was then FACS sorted to exclude debris (via forward- and side scatter plots, see methods) and dead cells (via cytox blue, see methods), and then immediately single-cell sequenced using the 10x droplet protocol (see Figure 1a,b, for technical details see paragraph “methods”).
- For the sorted EC samples, the digested tissue was digested, FACS sorted to exclude debris and dead cells and then processed using CD31⁺/CD45⁻ FACS sorting to gate for ECs and subsequently single-cell sequenced using the 10x droplet protocol (see Figure 1a,b, for technical details see paragraph “methods”).

From a computational point of view, in the unsorted (ECs/PVCs) samples, we indeed sequenced all cells including neurons, disease-specific cell types (e.g. tumor cells in the tumor entities) and any/all other cell types. The annotation of the different cell types was then performed manually based on top cluster markers and comparison to multiple published papers (see methods).

For the unsorted and sorted samples, EC and PVC cell types (unsorted samples) and EC clusters (sorted samples) were annotated referring to different published papers as well as to canonical PVC and EC markers to derive marker genes (for details, see methods) including:

- Vanlandewijck et al, *Nature*, 2018⁵
- Goveia et al., *Cancer Cell*, 2020⁵⁶
- Kalucka et al., *Cell*, 2020⁴

In addition to the further detailed description about the experimental and computational methodology regarding the unsorted and sorted samples, we have also provided additional illustrations/plots regarding the cell annotation and top cluster markers that can be found in Figure 2, Extended Data Figure 10 and Supplementary Table 13,14.

COMMENT:

3. When describing DE for figures 1k-m and Extended figures 5,6 – supplementary table for DE genes should be provided and referred to.

ANSWER:

We thank the reviewer for this comment and have now provided the differentially expressed Genes (DEG) in the Supplementary Tables 6,21 and have as well referred to in the paragraph “results” in the manuscript.

COMMENT:

4. Better description of the annotation procedure will help. For instance, for Figure 2, did the authors just choose a clustering structure with a particular seed, or did they additionally manually verify their annotation?

ANSWER:

We have further clarified the annotation procedure in the paragraph “Methods”. In addition, we indeed additionally verified the annotation manually and mapped it to/compared it with different published manuscripts including

- Vanlandewijck et al, *Nature*, 2018⁵
- Goveia et al., *Cancer Cell*, 2020⁵⁶
- Kalucka et al., *Cell*, 2020⁴

And other datasets as stated above

COMMENT:

5. Circle plots showing ligand-receptor interactions are often difficult to comprehend, due to too many lines. Having some threshold and showing only the most significant interactions will help to visualize the most important data. This is related to many plots in both main and supplementary figures.

ANSWER:

According to the reviewer’s request, we have made additional illustrations including scatter plots, heatmaps, dotplots, chord plots and bar graphs to show the most significant interactions and thus highlighting the most important data. We have applied this to all related figures, namely Figure 2,6, Extended Figures 17,18,31,33-37 and Supplementary Data Figures 16-26,29-41.

COMMENT:

6. The authors comment on the importance of compositional changes across different sample sets. While this data is indeed interesting, the authors should be clear that some of those changes could derive from batch effects due to too few samples per condition. In particular for cancer datasets – tissue sampling could affect cell type composition (unless the whole biopsy was used to generate each cancer sc dataset, and then it should be mentioned in the methods).

ANSWER:

We thank the reviewer for this comment. Similar to what this reviewer asked for in major point 5. about the compositional changes across different sample sets where he suggested us to perform statistical testing using scCODA³². The results from this analysis can be found in Figure 2, Figure to the Reviewer 3, Supplementary tables 9,10.

Regarding tissue sampling, we have now explained in further detail how all fetal and adult tissues including the cancers were collected in the paragraph “methods”. In brief, for all pathological samples including the cancers, we asservated the maximum amount of tissue that could be resected safely (see paragraph “methods”). We have also mentioned this limitation as well as the limitation linked to the relatively small number of samples per condition in the paragraph “Discussion”.

Regarding the batch effects, we refer this reviewer to the response for reviewer 1.

We have compared the different batch correction/integration methods (Figure to the Reviewer 2). RPCA, harmony, scANVI and CCA showed very comparable integration. We also plotted the RPCA

annotated clusters onto the harmony, scANVI and CCA integrated objects and the results show that cells indeed cluster according to their AV specification (AV-annotation were very consistent) as we have observed with the RPCA integrated cells in the submitted manuscript.

COMMENT:

7. *Some of the supplementary figures are not necessary.*

ANSWER:

We have tried to adapt and reduce the number of “Supplementary Figures” accordingly and have converted some of the “Supplementary Figures” to “Figures to the Reviewers”.

COMMENT:

8. “Notably, these “unassigned” ECs mainly belonged to small caliber vessels” – why it is notable? Please explain.

ANSWER:

We thank the reviewer for this comment and have clarified in the text that the small caliber vessels in the different pathological brain entities (most clearly observable in the glioblastoma and in the lung cancer metastasis) map to “unassigned” ECs of the adult/control brain (temporal lobe) (see Extended Data Figures 19u-x), indicating that the transcriptomic profile of pathological versus adult/control brain ECs is most different at the level of small caliber vessels, most likely indicating their susceptibility to the local microenvironment⁴

REFERENCES

- 1 Yang, A. C. *et al.* A human brain vascular atlas reveals diverse cell mediators of Alzheimer's disease risk. *bioRxiv*, 2021.2004.2026.441262, doi:10.1101/2021.04.26.441262 (2021).
- 2 Garcia, F. J. *et al.* Single-cell dissection of the human cerebrovasculature in health and disease. *bioRxiv*, 2021.2004.2026.440975, doi:10.1101/2021.04.26.440975 (2021).
- 3 Winkler, E. A. *et al.* A single-cell atlas of the normal and malformed human brain vasculature. *Science* **375**, eabi7377, doi:10.1126/science.abi7377 (2022).
- 4 Kalucka, J. *et al.* Single-Cell Transcriptome Atlas of Murine Endothelial Cells. *Cell* **180**, 764-779.e720, doi:10.1016/j.cell.2020.01.015 (2020).
- 5 Vanlandewijck, M. *et al.* A molecular atlas of cell types and zonation in the brain vasculature. *Nature* **554**, 475-480, doi:10.1038/nature25739 (2018).
- 6 Yang, A. C. *et al.* A human brain vascular atlas reveals diverse mediators of Alzheimer's risk. *Nature* **603**, 885-892, doi:10.1038/s41586-021-04369-3 (2022).
- 7 Garcia, F. J. *et al.* Single-cell dissection of the human brain vasculature. *Nature* **603**, 893-899, doi:10.1038/s41586-022-04521-7 (2022).
- 8 Wälchli, T. *et al.* Nogo-A regulates vascular network architecture in the postnatal brain. *J Cereb Blood Flow Metab* **37**, 614-631 (2017).
- 9 Hao, Y. *et al.* Integrated analysis of multimodal single-cell data. *Cell* **184**, 3573-3587 e3529, doi:10.1016/j.cell.2021.04.048 (2021).
- 10 Heinzer, S. *et al.* Novel three-dimensional analysis tool for vascular trees indicates complete micro-networks, not single capillaries, as the angiogenic endpoint in mice overexpressing human VEGF(165) in the brain. *Neuroimage* **39**, 1549-1558, doi:10.1016/j.neuroimage.2007.10.054 (2008).
- 11 Meyer, E. P., Ulmann-Schuler, A., Staufenbiel, M. & Krucker, T. Altered morphology and 3D architecture of brain vasculature in a mouse model for Alzheimer's disease. *Proc Natl Acad Sci U S A* **105**, 3587-3592, doi:10.1073/pnas.0709788105 (2008).
- 12 Gonzalez, H. *et al.* Cellular architecture of human brain metastases. *Cell* **185**, 729-745 e720, doi:10.1016/j.cell.2021.12.043 (2022).
- 13 Richards, L. M. *et al.* Gradient of Developmental and Injury Response transcriptional states defines functional vulnerabilities underpinning glioblastoma heterogeneity. *Nature Cancer* **2**, 157-173, doi:10.1038/s43018-020-00154-9 (2021).
- 14 Neftel, C. *et al.* An Integrative Model of Cellular States, Plasticity, and Genetics for Glioblastoma. *Cell* **178**, 835-849 e821, doi:10.1016/j.cell.2019.06.024 (2019).
- 15 Cohen, A. L., Holmen, S. L. & Colman, H. IDH1 and IDH2 mutations in gliomas. *Curr Neurol Neurosci Rep* **13**, 345, doi:10.1007/s11910-013-0345-4 (2013).
- 16 Louis, D. N. *et al.* The 2021 WHO Classification of Tumors of the Central Nervous System: a summary. *Neuro Oncol* **23**, 1231-1251, doi:10.1093/neuonc/noab106 (2021).
- 17 Darmanis, S. *et al.* Single-Cell RNA-Seq Analysis of Infiltrating Neoplastic Cells at the Migrating Front of Human Glioblastoma. *Cell Rep* **21**, 1399-1410, doi:10.1016/j.celrep.2017.10.030 (2017).
- 18 Wosen, J. E., Mukhopadhyay, D., Macaubas, C. & Mellins, E. D. Epithelial MHC Class II Expression and Its Role in Antigen Presentation in the Gastrointestinal and Respiratory Tracts. *Front Immunol* **9**, 2144, doi:10.3389/fimmu.2018.02144 (2018).
- 19 MacDonald, T. T., Weinel, A. & Spencer, J. HLA-DR expression in human fetal intestinal epithelium. *Gut* **29**, 1342-1348, doi:10.1136/gut.29.10.1342 (1988).
- 20 Wälchli, T. *et al.* Quantitative assessment of angiogenesis, perfused blood vessels and endothelial tip cells in the postnatal mouse brain. *Nat Protoc* **10**, 53-74, doi:10.1038/nprot.2015.002 (2015).
- 21 Huijbers, E. J. M., Khan, K. A., Kerbel, R. S. & Griffioen, A. W. Tumors resurrect an embryonic vascular program to escape immunity. *Sci Immunol* **7**, eabm6388, doi:10.1126/sciimmunol.abm6388 (2022).
- 22 Vladoiu, M. C. *et al.* Childhood cerebellar tumours mirror conserved fetal transcriptional programs. *Nature* **572**, 67-73, doi:10.1038/s41586-019-1158-7 (2019).

- 23 Hsu, Y. L. *et al.* Hypoxic lung cancer-secreted exosomal miR-23a increased angiogenesis and vascular permeability by targeting prolyl hydroxylase and tight junction protein ZO-1. *Oncogene* **36**, 4929-4942, doi:10.1038/ncr.2017.105 (2017).
- 24 Sharma, A. *et al.* Onco-fetal Reprogramming of Endothelial Cells Drives Immunosuppressive Macrophages in Hepatocellular Carcinoma. *Cell* **183**, 377-394 e321, doi:10.1016/j.cell.2020.08.040 (2020).
- 25 Couturier, C. P. *et al.* Single-cell RNA-seq reveals that glioblastoma recapitulates a normal neurodevelopmental hierarchy. *Nat Commun* **11**, 3406, doi:10.1038/s41467-020-17186-5 (2020).
- 26 Barresi, V. Angiogenesis in meningiomas. *Brain Tumor Pathol* **28**, 99-106, doi:10.1007/s10014-010-0012-2 (2011).
- 27 Pierscianek, D. *et al.* Study of angiogenic signaling pathways in hemangioblastoma. *Neuropathology* **37**, 3-11, doi:10.1111/neup.12316 (2017).
- 28 Arvanitis, C. D., Ferraro, G. B. & Jain, R. K. The blood-brain barrier and blood-tumour barrier in brain tumours and metastases. *Nat Rev Cancer* **20**, 26-41, doi:10.1038/s41568-019-0205-x (2020).
- 29 Aldape, K., Zadeh, G., Mansouri, S., Reifenberger, G. & von Deimling, A. Glioblastoma: pathology, molecular mechanisms and markers. *Acta Neuropathol* **129**, 829-848, doi:10.1007/s00401-015-1432-1 (2015).
- 30 Weller, M. *et al.* Glioma. *Nat Rev Dis Primers* **1**, 15017, doi:10.1038/nrdp.2015.17 (2015).
- 31 Achrol, A. S. *et al.* Brain metastases. *Nat Rev Dis Primers* **5**, 5, doi:10.1038/s41572-018-0055-y (2019).
- 32 Buttner, M., Ostner, J., Muller, C. L., Theis, F. J. & Schubert, B. scCODA is a Bayesian model for compositional single-cell data analysis. *Nat Commun* **12**, 6876, doi:10.1038/s41467-021-27150-6 (2021).
- 33 Suranyi, A., Nogrady, M., Altorjay, A., Nyari, T. & Nemeth, G. Examination of the vascularization of fetal kidney with three-dimensional power Doppler technique in pregnancies complicated by increased maternal blood pressure. *Interv Med Appl Sci* **10**, 7-12, doi:10.1556/1646.10.2018.15 (2018).
- 34 Senra, J. C. *et al.* Kidney impairment in fetal growth restriction: three-dimensional evaluation of volume and vascularization. *Prenat Diagn* **40**, 1408-1417, doi:10.1002/pd.5778 (2020).
- 35 Virgintino, D. *et al.* Immunolocalization of tight junction proteins in the adult and developing human brain. *Histochem Cell Biol* **122**, 51-59, doi:10.1007/s00418-004-0665-1 (2004).
- 36 Wolfs, T. G. *et al.* Chorioamnionitis-induced fetal gut injury is mediated by direct gut exposure of inflammatory mediators or by lung inflammation. *Am J Physiol Gastrointest Liver Physiol* **306**, G382-393, doi:10.1152/ajpgi.00260.2013 (2014).
- 37 Marzioni, D. *et al.* Expression of ZO-1 and occludin in normal human placenta and in hydatidiform moles. *Mol Hum Reprod* **7**, 279-285, doi:10.1093/molehr/7.3.279 (2001).
- 38 Uhlen, M. *et al.* Proteomics. Tissue-based map of the human proteome. *Science* **347**, 1260419, doi:10.1126/science.1260419 (2015).
- 39 Schaum, N. *et al.* Single-cell transcriptomics of 20 mouse organs creates a Tabula Muris. *Nature* **562**, 367-372, doi:10.1038/s41586-018-0590-4 (2018).
- 40 Hodge, R. D. *et al.* Conserved cell types with divergent features in human versus mouse cortex. *Nature* **573**, 61-68, doi:10.1038/s41586-019-1506-7 (2019).
- 41 Munji, R. N. *et al.* Profiling the mouse brain endothelial transcriptome in health and disease models reveals a core blood-brain barrier dysfunction module. *Nature Neuroscience* **22**, 1892-1902, doi:10.1038/s41593-019-0497-x (2019).
- 42 Han, X. *et al.* Construction of a human cell landscape at single-cell level. *Nature* **581**, 303-309, doi:10.1038/s41586-020-2157-4 (2020).
- 43 Tang, M. *et al.* Evaluating single-cell cluster stability using the Jaccard similarity index. *Bioinformatics* **37**, 2212-2214, doi:10.1093/bioinformatics/btaa956 (2021).
- 44 Saunders, N. R., Liddelow, S. A. & Dziegielewska, K. M. Barrier mechanisms in the developing brain. *Front Pharmacol* **3**, 46, doi:10.3389/fphar.2012.00046 (2012).
- 45 Marin-Padilla, M. The human brain intracerebral microvascular system: development and structure. *Front Neuroanat* **6**, 38, doi:10.3389/fnana.2012.00038 (2012).
- 46 Saili, K. S. *et al.* Blood-brain barrier development: Systems modeling and predictive toxicology. *Birth Defects Res* **109**, 1680-1710, doi:10.1002/bdr2.1180 (2017).
- 47 Saunders, N. R. *et al.* The rights and wrongs of blood-brain barrier permeability studies: a walk through 100 years of history. *Front Neurosci* **8**, 404, doi:10.3389/fnins.2014.00404 (2014).
- 48 Saunders, N. R., Dziegielewska, K. M., Mollgard, K. & Habgood, M. D. Physiology and molecular biology of barrier mechanisms in the fetal and neonatal brain. *J Physiol* **596**, 5723-5756, doi:10.1113/JP275376 (2018).
- 49 Zhang, X. *et al.* High-resolution mapping of brain vasculature and its impairment in the hippocampus of Alzheimer's disease mice. *National Science Review* **6**, 1223-1238, doi:10.1093/nsr/nwz124 (2019).

- 50 Gould, J. Breaking down the epidemiology of brain cancer. *Nature* **561**, S40-S41, doi:10.1038/d41586-018-06704-7 (2018).
- 51 Zhicheng Ji, H. J. TSCAN: Tools for Single-Cell Analysis. R package version 1.34.0. (2022).
- 52 La Manno, G. *et al.* RNA velocity of single cells. *Nature* **560**, 494-498, doi:10.1038/s41586-018-0414-6 (2018).
- 53 Lange, M. *et al.* CellRank for directed single-cell fate mapping. *Nat Methods* **19**, 159-170, doi:10.1038/s41592-021-01346-6 (2022).
- 54 Angerer, P. *et al.* destiny: diffusion maps for large-scale single-cell data in R. *Bioinformatics* **32**, 1241-1243, doi:10.1093/bioinformatics/btv715 (2016).
- 55 Buechler, M. B. *et al.* Cross-tissue organization of the fibroblast lineage. *Nature* **593**, 575-579, doi:10.1038/s41586-021-03549-5 (2021).
- 56 Goveia, J. *et al.* An Integrated Gene Expression Landscape Profiling Approach to Identify Lung Tumor Endothelial Cell Heterogeneity and Angiogenic Candidates. *Cancer Cell* **37**, 21-36.e13, doi:10.1016/j.ccell.2019.12.001 (2020).

Reviewer Reports on the First Revision:

Referee expertise:

Referees' comments:

Referee #1 (Remarks to the Author):

In this revised manuscript by Walchli et al., the authors build upon the “molecular atlas” of the CNS endothelium in the fetal and adult brain from their original submission to now include additional validation at the transcript and protein level. Importantly, additional validation is performed for MHCII expression at the brain EC. The authors now include additional analysis of non-diseased brain tissue across ages, and comparators amongst their diseased samples. The transcriptomics data across ages in humans presented in this manuscript will likely serve as an important reference, and the authors now include comparisons between their database and published datasets in this revision. Overall, the author has included a great deal of additional analysis in response to reviewer comments, however, concerns raised by this reviewer and others regarding the overwhelming amount of content in the manuscript, and the inclusion of diseased tissue with very low sample numbers, remain largely unaddressed. This manuscript and associated figures could benefit from significant editing to be more concise.

Comments on specific author responses:

- In regards to the author's comments on small sample sizes and generalizability of transcriptomics data across an entire disease state: Bulk-seq data is included as validation for scRNA-seq data, however, there is unclear benefit in validating transcript data with additional transcript data from the same sample. The sample numbers remain low in some cases, which is most appropriately interpreted as a study of that particular patient, without enough data to support generalizability to a common disease state. A high number of isolated endothelial and perivascular cells in a sample may improve intrasample power, but still does not address the concern of low or no overall biological replicates.
- The authors now include improved validation of MHCII expression in ECs.
- Regarding concerns for variation within the adult age group (15 years to 69) the authors provide reviewer figures 1 and 3, and supplemental table 10. The authors report that the two ages on either end of the range are “not outliers”, but there does appear to be differences in the volcano plots between these two ages in reviewer figure 1. The same is true of sex differences. It is difficult to analyze these differences further given the small size of the graphs with absent labeling, and supplemental table 10 could not be located as it was also not clearly labelled. There remains concern that 15 years-old is not an adult, and that there are likely differences in ECs across such a broad age range.
- Language regarding fetal pathways is refined.
- Concern for grouping of genetically distinct GBM tissues was noted by the authors, and it is acknowledged that there are not sufficient sample numbers to compare tumors with different genetic profiles. Although the authors acknowledge that future studies would be needed, a recurrent theme amongst the diseases discussed in this paper are insufficient numbers and the

inclusion of several distinct entities rather than a more focused, better powered study.

- Several instances throughout the author response, an off-the-record discussion with editorial staff was cited as rationale for dismissal of a reviewer comment. The author should consider a more thorough rationale for either exclusion of under-powered disease comparisons (and thus appropriately make the overall manuscript more concise) or include more samples where comparisons are made and include them in the paper.

Referee #2 (Remarks to the Author):

The authors have addressed substantially the concerns raised by this reviewer. The manuscript and the data that they provide will be a valuable resource for the scientific community.

Referee #3 (Remarks to the Author):

The authors have invested a large amount of work to address reviewer's concerns. Nevertheless, my previous critique was only partially addressed. Below are several major points to be addressed.

1. Integration of single-cell data is still the major issue. While in the rebuttal there is a structured description of when and what type of integrations were used, in the manuscript, it is difficult to understand. This approach to have some analyses with CCA integration, some with RPCA integration, some without integration does not help the reader to grasp fast whether an analysis is robust or not.

For instance, lines 175-177:

To broadly map the neurovasculature and to address the NVU across different brain entities, 44,116 fetal, 47,305 adult/control and 121,436 pathological unsorted EC and PVC transcriptomes from 33 patients were pooled, clustered, annotated using known marker genes

Was it done with integration or not? The methods part for single-cell experiments is all together under umbrella of one chapter "Chromium 10X library preparation, single cell RNA-sequencing and data analysis" and it is difficult to work through. This chapter should be split into subchapters for different analyses.

To address the issue with integration of single-cell data, the authors should either use the same method for integration across their analyses or make it clear in the Results text, with further details in the Methods, where what type of integration was used. Arguments for justification why CCA was used should be provided.

Finally, the Reviewer figure 2 in fact does show that CCA integration is far more noisy in comparison to eg scANVI. Obviously, by integrating using CCA, the authors miss many cell type-relevant markers that are removed by CCA during integration process. This needs to be admitted in the manuscript (unless it is shown that it is not the case). Comparison of integration by different methods needs to be included in the manuscript (not Reviewer figure, proper Extended data figure) and this should

include how integration with RPCA and an alternative method (Harmony or scANVI) affects differential gene expression. Jaccard index that was already used for stability analysis by the authors is a good tool to implement here.

2. Addition of metadata and QC plots helped a lot in understanding where the data is robust and where it is not. Here also a question whether it is better to focus on something more specific arises, since some sample sets provide only limited knowledge.

To illustrate this, I will walk through TL adjacent CAV samples using the QC and metadata that are now available. These samples have by far fewer genes than other sample categories. When looking in metadata, Supplementary table 3, a very small number of cells from TL adjacent CAV samples were filtered – out of ~10,000 cells, ~200 cells were filtered. However, as standard QC, cells with <500 genes were filtered, and based on Extended data figure 3d, it looks like many cells should be filtered. Thus, first more specific question arise – can the authors double check their QC plots and tables to confirm that there are no mistakes.

Second more general question, why TL adjacent CAV samples are needed at all? They are low quality, only 2 samples, not enough to draw any conclusions for comparison to TL or CAV samples. Why to keep such samples in the atlas?

Finally, most general comment – the effect of covariates can be huge, eg it was already shown that in single-cell analyses age, sex, sequencing batch, processing site etc can have a large impact on single-cell data. The authors did try to address sex and age covariate effects on cell type composition using scCODA. However, unfortunately, scCODA does not have integrated covariate option. Tools to study covariates are only being developed now, with most advanced likely scITD and Cacao, which are still pre-prints. Nevertheless, even if implementing the most advanced tools, the impact of covariates on the current atlas can be only superficially addressed, since in the current version, there are simply too many conditions. Focusing the atlas on those analyses where there are enough samples, and where robust analyses can be made – this is what the users of the atlas need.

3. Based on Jaccard index (Extended data figure 10h), capillary subclusters are very unstable, thus subclustering of capillary cells should be reduced since these subclusters can be just due to noise.

4. Overall, the figures contain too many panels squeezed too tight and it is difficult to go through. The authors should consider different format for some plots and removing of some plots from the main figures to Extended data figures.

Author Rebuttals to First Revision:

Reviewers' Comments:

Reviewer #1:

Comments for the Author:

COMMENT:

In this revised manuscript by Walchli et al., the authors build upon the “molecular atlas” of the CNS endothelium in the fetal and adult brain from their original submission to now include additional validation at the transcript and protein level. Importantly, additional validation is performed for MHCII expression at the brain EC. The authors now include additional analysis of non-diseased brain tissue across ages, and comparators amongst their diseased samples. The transcriptomics data across ages in humans presented in this manuscript will likely serve as an important reference, and the authors now include comparisons between their database and published datasets in this revision. Overall, the author has included a great deal of additional analysis in response to reviewer comments, however, concerns raised by this reviewer and others regarding the overwhelming amount of content in the manuscript, and the inclusion of diseased tissue with very low sample numbers, remain largely unaddressed. This manuscript and associated figures could benefit from significant editing to be more concise.

ANSWER:

We thank the reviewer for their comments on our manuscript.

Provided that some of the questions and comments of reviewer 1, of reviewer 3 and of the editor are overlapping, we feel it is adequate to repeat the overview of what we have done during this second round of revisions (REVISIONS II) for both reviewers and for the editor, for reasons of clarity and convenience.

We have been working extensively in the past months to address all the editors' and reviewers' comments, notably:

- i) we have improved the readability of our manuscript by streamlining the manuscript text, main Figures, Extended Data Figures, and Supplementary Figures
 - as a result, the manuscript is now shorter, much more focused and concise, thus easier to read
- ii) we have addressed the covariates age and sex using state of the art statistical methods in our own datasets and also referring to and analyzing multiple publicly available datasets
 - In summary, the biological findings in our atlas do not depend on the covariates age and sex
- iii) we have compared multiple methods of integration/batch correction for both the sorted and the unsorted samples and based on these results have homogenized and streamlined the computational analyses throughout the paper
 - In summary, the biological conclusions in our atlas do not depend on the method of integration/batch correction
- iv) we have removed the very rare samples from the atlas (2 cavernomas, 2 temporal lobes adjacent to cavernomas, 2 hemangioblastomas)
 - In summary, the biological conclusions in our atlas do not depend on the integration or exclusion of these very rare samples. The very rare samples serve as an additional validation of the findings in the atlas - for instance supporting the concepts of common "hallmarks" of the diseased brain vasculature across various brain pathologies and of reactivated fetal pathways in brain diseases - and will be published in a future follow-up study.
- v) we have also added numerous samples of sorted endothelial cells (ECs) (3 adult/control brains, 2 lower-grade gliomas, 3 high-grade gliomas/glioblastomas) as well as additional samples of unsorted endothelial (EC)- and perivascular cells (PVCs) (2 fetal brains, 2 adult/control brains), analyzed via scRNA-seq to further strengthen and validate our findings in the atlas
 - In summary, the biological conclusions in our atlas are further confirmed by these additional samples, both:
 - i) at the level of individual disease entities
 - ii) as well as across diseases
 - thereby further strengthening
 - i) the signature of diseased brain ECs (and PVCs) for every entity
 - ii) as well as the common signature of activated brain ECs (and PVCs) across entities
 - and also further illustrating the robustness of our biological findings and the usefulness of our atlas as a resource
- vi) we have provided additional computational analyses taking into account the updated set of all 117 samples derived from 68 different patients including:
 - new computational analysis of the scRNA-sequencing data on CD31⁺/CD45⁻ sorted endothelial cells (ECs) and on unsorted endothelial (EC)- and perivascular cells (PVCs) which further confirms the findings of the scRNA-seq atlas
 - additional computational analysis of the bulk RNA-sequencing data on CD31⁺/CD45⁻ sorted endothelial cells (ECs) utilizing bulk RNA-seq deconvolution which confirms the findings of the scRNA-seq atlas
 - an adapted working model in Figure 1a clarifying both the experimental and computational workflows
 - In summary, the new computational analyses further validate/strengthen the computational results and biological findings of our scRNA-sequencing atlas

According to the suggestion of the reviewer (and as also mentioned by reviewer 3, see below) we have now excluded the diseased samples with low numbers (including 2 cavernomas, 2 temporal lobes adjacent to cavernomas, 2 hemangioblastomas) from the atlas.

Moreover, we have also added various additional samples to further confirm our findings in the atlas. In our latest version of our manuscript, we have analyzed 117 samples derived from 68 patients.

Thereby, we would like to emphasize that:

- we avoided conclusions that are based on only a few samples
- we further strengthened and confirmed the findings of our scRNA-seq atlas based on a very solid amount of samples
- we more clearly pointed out the commonalities (“hallmarks”) of the pathological brain vasculature across diseases.

The analyzed samples are derived from the following entities:

For sorted brain ECs

- 5 fetal brain samples coming from 5 fetuses
- 5 fetal periphery samples coming from 5 fetuses
- 12 adult/control brain (temporal lobe) samples coming from 11 patients
- 57 diseased samples coming from 50 patients

For unsorted brain ECs and PVCs

- 7 fetal brain samples coming from 7 fetuses
- 7 fetal periphery samples coming from 7 fetuses
- 6 adult/control brain (temporal lobe) samples coming from 6 patients
- 18 diseased samples coming from 18 patients

Furthermore, we would like to highlight that the number of both endothelial- as well as perivascular cells per entity is very high and, to our knowledge, unmet as compared to any other currently published study. Moreover, to address the reviewer's comment and further validate our findings, we have performed further bulk RNA-seq analyses. The key findings emanating of the scRNA-seq experiments described in the manuscript were confirmed by the bulk RNA-seq data (Supplementary Figure 6, Supplementary Tables 1-part2, 21^{1,2}).

In summary, we have thereby addressed all key reviewers' and editors' concerns regarding:

- covariates age and sex
- computational methods/overall analytical approach
- sample numbers amongst the different pathologies
- lengthiness of the manuscript and number of figures

In order to fully address the reviewer's request (and also based on the feedback of reviewer 3, see below) regarding the diseased tissues with very low numbers, we have now performed the analysis by excluding these rare diseased tissues. Importantly, the key biological findings are not affected by the exclusion or inclusion of these very rare samples.

Regarding editing of the manuscript and the associated figures

We have now edited the manuscript and associated figures in a significant manner in order to make the manuscript and figures more concise. As also suggested by reviewer 3, we also moved some figure panels and plots from the main Figures to the Extended Data Figures. In summary, we have made the following adaptations:

i) manuscript

- substantial shortening of the manuscript, resulting in a word count of 4440 words (reduction by 3446 words).

ii) figures

- iia) substantial simplification of the five main Figures resulting in none of them extending more than half an A4 page
- notably, reduction of the number of panels in the main Figures by removing panels, simplifying existing panels and moving panels and plots to Extended Data Figures, as follows:
- ii ai) reduction of the number of panels in Figure 1
 - we have moved the former panels Figure 1b,c,d,e to Extended Data Figure 1c,d,e,f and Figure 1s,u to Supplementary Figure 5d,e
- ii aii) reduction of the number of panels in Figure 2
 - we have removed the former panels Figure 2b,c,d,k and have moved the former panel Figure 2f to Supplementary Figure 7i
- ii aiii) reduction of the number of panels in Figure 3
 - we have moved the former panels Figure 3n,o to Extended Data Figure 5zv, zvii
- ii aiv) reduction of the number of panels in Figure 4

- we have simplified the panels of Figure 4a-c by removing 6 UMAP plots, and we removed former panel Figure 4h
- iiav) reduction of the number of panels in Figure 5
 - we have simplified the panels of Figure 5a-c by removing 6 UMAP plots, and we removed former panel Figure 5h
- iiavi) conversion of former Figure 6 to Extended Data Figure 11
- iib) substantial reduction of the number of Extended Data Figures and Supplementary Figures, as follows:
 - iibi) reduction of the number of Extended Data Figures from 37 to 15
 - iibii) reduction of the number of Supplementary Figures from 41 to 16

Please find below a graphical illustration of the simplifications and shortenings during this round of revisions

Thereby, we now provide an edited, more concise and easily readable version of the manuscript and figures while still providing all additional data and analyses addressing all editors' and the reviewers' comments and requests in a satisfactory and comprehensive manner. Please note that we also accounted for the reviewer's requests for additional experimental and computational data.

Comments on specific author responses:

COMMENT:

- In regards to the author's comments on small sample sizes and generalizability of transcriptomics data across an entire disease state: Bulk-seq data is included as validation for scRNA-seq data, however, there is unclear benefit in validating transcript data with additional transcript data from the same sample. The sample numbers remain low in some cases, which is most appropriately interpreted as a study of that particular patient, without enough data to support generalizability to a common disease state. A high number of isolated endothelial and perivascular cells in a sample may improve intrasample power, but still does not address the concern of low or no overall biological replicates.

ANSWER:

We thank the reviewer for these comments. We would like to clarify the raised issues.

Regarding the low sample numbers, the addition of samples and the generalizability of the findings

To address the reviewer's comments regarding the low sample numbers, we have now removed the very rare samples (both their sorted as well as its unsorted datasets) from the atlas, notably:

- 2 cavernomas
- 2 temporal lobes adjacent to cavernomas
- 2 hemangioblastomas

We agree with the reviewer that these very low sample numbers may be insufficient to support generalizability to a common disease state. We initially included those very rare samples because:

- these very rare samples display many of the common hallmarks found across the more frequent samples
- and also because we wanted to make the data publicly available

Importantly, even after having removed these very rare samples from the atlas, we would like to emphasize that these very rare samples serve as a validation of our findings in the atlas.

- the summary is that the biological conclusions in our atlas do not depend on the integration or exclusion of these very rare samples.

In addition to the exclusion of the very rare samples and to further address the reviewer's concern about the number of biological replicates, we have now added multiple additional samples of:

- i) sorted endothelial cells analyzed via scRNA-seq
 - 3 adult/control brains (temporal lobes)
 - 2 lower-grade gliomas
 - 3 high-grade gliomas/glioblastomas
- ii) unsorted cells analyzed via scRNA-seq
 - 2 fetal brains
 - 2 adult/control brains (temporal lobes)
- iii) sorted endothelial cells analyzed via bulk RNA-seq (22 additional samples)
 - 1 adult/control brain (temporal lobe)
 - 7 arteriovenous malformations
 - 3 lower-grade gliomas
 - 4 high-grade gliomas/glioblastomas
 - 4 metastases
 - 3 meningiomas

Thus we further validate the findings of our atlas

- the summary is that the biological conclusions in our atlas can be validated by these additional samples, further illustrating the usefulness of our atlas as a resource.

Accordingly, we have increased the number of isolated endothelial and perivascular cells in key entities and have thus improved inter-sample power (in addition to the high intra-sample power), thereby addressing the concern of low or no overall biological replicates stated by the reviewer.

Regarding the bulk RNA-seq data and its independence of the sc RNA-seq data

Regarding the bulk RNA-seq data of sorted endothelial cells, we would like to clarify that the bulk RNA-seq samples predominantly derived from patients that are different from the scRNA-seq patients:

Accordingly, and based on the summary for additional samples provided above, please find an overview of the sorted endothelial cell samples that were analyzed via bulk RNA-sequencing (Extended Data Figure 1, Supplementary Tables 1-part2, 21^{1,2}) in the paper and their independence of the single-cell RNA-sequencing:

- Adult/control brains (temporal lobes)
 - 3 patients in total
 - 1 of which is different from the scRNA-seq samples
- Arteriovenous malformations
 - 7 patients in total
 - All of them are different from the scRNA-seq samples
- High grade gliomas
 - 6 patients in total
 - 4 of which are different from the scRNA-seq samples
- Lower grade gliomas
 - 4 patients in total
 - 3 of which are different from the scRNA-seq samples
- Metastases
 - 5 patients in total
 - 4 of which are different from the scRNA-seq samples

- Meningiomas
 - 6 patients in total
 - 3 of which are different from the scRNA-seq samples

In summary, the endothelial cells that were analyzed via bulk RNA-sequencing (Extended Data Figure 1, Supplementary Table 1-part2) in the atlas present as follows:

- 22 out of 31 bulk RNA-seq samples of sorted endothelial cells are patient samples that are different from the single-cell RNA-seq samples of sorted endothelial cells and thus serve as an independent cohort of samples that constitute an additional validation of our findings in the scRNA-seq atlas (in addition to all the other validation experiments, see Figure 1a).
- 9 out of 31 bulk RNA-seq samples of sorted endothelial cells are patient samples that are same (generated from the same patients) as the single-cell RNA-seq samples. These samples further validate the findings within a given patient by using two different sequencing techniques.

The overview of the patient samples analyzed in our manuscript with both scRNA-seq as well as with bulk RNA-seq is displayed in Extended Data Figure 1, and Supplementary Tables 1-part1, 1-part2.

Thus, most bulk RNA-seq data does not stem from the same patients as the scRNA-seq data, and therefore, the bulk RNA-seq data indeed provides an independent verification of the biological findings observed in our scRNA-seq data atlas, because:

- it refers predominantly to a different cohort of patients (see above)
- it refers to a different protocol of a similar technique
- it provides deeper sequencing (thereby providing further confirmation)

Therefore, we believe that the bulk RNA-seq samples are indeed serving as a further validation for the scRNA-seq analysis confirming the major findings of the single-cell RNA-seq atlas, in particular the characteristic cellular expression patterns in brain tumors and brain vascular malformations.

To that regard, and most importantly, we would also like to clearly mention and emphasize that:

- i) we have removed the rare samples from the atlas
- ii) we have added additional samples to the atlas to increase the number of biological replicates thereby supporting the findings of common disease states across pathologies.

Furthermore, we would like to mention that the validation of the findings of our scRNA-seq data referred to various experimental data utilizing different techniques (including but not exclusive to bulk RNA-seq data) that complement the bulk RNA-seq data (Figure 1a) including:

- Immunofluorescence (IF) stainings
- Imaging Mass Cytometry (IMC) = Imaging Cytometry by Time of Flight (Imaging CyTOF) stainings
- Spatial transcriptomics data

thereby addressing key reviewer's concerns about additional validation experiments, additional patients, and inclusion of the fetal brain tissues (for IF and IMC).

Finally, we would also like to mention that validation of scRNA-seq data with bulkRNA-seq data has been widely utilized in various recent high-impact scRNA-seq publications such as for example:

- Travaglini et al., *Nature*, 2021
- Buechler et al., *Nature*, 2021
- Winkler et al., *Science*, 2022
- Goveia et al., *Cancer Cell*, 2020

COMMENT:

- *The authors now include improved validation of MHCII expression in ECs.*

ANSWER:

We thank the reviewer for their comment.

COMMENT:

- *Regarding concerns for variation within the adult age group (15 years to 69) the authors provide reviewer figures 1 and 3, and supplemental table 10. The authors report that the two ages on either*

end of the range are “not outliers”, but there does appear to be differences in the volcano plots between these two ages in reviewer figure 1. The same is true of sex differences. It is difficult to analyze these differences further given the small size of the graphs with absent labeling, and supplemental table 10 could not be located as it was also not clearly labelled. There remains concern that 15 years-old is not an adult, and that there are likely differences in ECs across such a broad age range.

ANSWER:

We thank the reviewer for this comment.

Regarding the concerns for the variation within the adult age group (15 years to 69 years), we indeed report that the two ages 15 years and 69 years at either end of the range are “not outliers” for the reasons outlined below.

Importantly, to further address this reviewer’s comment, we have now added several additional samples for sorted endothelial cells scRNA-seq samples, notably:

- 3 adult/control brains (temporal lobes)
 - 2 lower-grade gliomas
 - 3 high-grade gliomas/glioblastomas
- as well as additional samples of unsorted cells analyzed via scRNA-seq
- 2 fetal brains
 - 2 adult/control brains

Accordingly, the addition of three adult/control brains (temporal lobes) further addresses the concern of the reviewer regarding the age range within the adult/control brain (temporal lobe) samples.

Provided that some of the questions and comments of reviewer 1 and of reviewer 3 are overlapping, parts of the following text can also be found in response to reviewer 3 (see below). Accordingly, wherever we feel it is adequate, we repeat part of these answers for reasons of clarity and convenience for the two reviewers.

Regarding the potential influence of the covariates age and sex on the biological findings in the atlas

To comprehensively address the covariates age and sex in our own datasets, we have performed various analyses during the two rounds of REVISIONS.

- i) Upon the first round of REVISIONS (REVISIONS I), we had performed the following analyses:
 - ia) comparison of the parameters “age” and “sex” against random shuffles of the parameters “age” and “sex”, which revealed:
 - that the number of significant DEGs between the different ages and the two sexes are low (Reviewer Figure 1).
 - and that these differences in terms of the number of significant DEGs are comparable to or even lower than (and thus not significantly different from) random shuffles of the parameters “age” and “sex” (Supplementary Table 19).
 - ib) pathway analysis comparing the different ages/sexes on the DEGs for both parameters “age” and “sex” revealing:
 - that no biologically meaningful pathways were significantly enriched.
 - ic) scCODA analysis showed no significant compositional changes between the different sexes and ages in the different entities (Reviewer Figure 2, Supplementary Table 10-part1).
- ii) In addition, during this second round of REVISIONS (REVISIONS II), we have now performed the following additional analyses:
 - iia) to regress the covariates age and sex, we have utilized state-of-the-art statistical methodologies which have integrated covariate options³⁻⁵ while performing cell compositional analysis (Supplementary Tables 10-part2, 10-part3, 10-part4). Notably, we referred to three independent computational packages/algorithms, namely:
 - Tree-aggregated amplicon and single-cell compositional data analysis (tascCODA) which is a fully Bayesian model for tree-aggregated modeling of count data and is an extension of the scCODA³.

- Dirichlet regression analysis (DirichletReg R package), dirichlet regression models can be used to analyze a set of variables lying in a bounded interval that sum up to a constant (e.g., proportions, compositions, etc.) exhibiting skewness and heteroscedasticity, without having to transform the data^{4,6}.
- Propeller method (speckle R package), which accommodates complex experimental designs and features the possibility to model additional covariates of interest, for instance age and sex⁵.

Importantly, we can conclude that the key findings related to cell compositional changes in sorted ECs of brain tumors and of brain vascular malformations as compared to sorted ECs from adult/control brains are not affected by the covariates age and sex utilizing the three above-mentioned methods (Supplementary Table 10). These analyses also verify the scCODA findings during the first round of REVISIONS (REVISIONS I).

- iib) to further address possible changes in endothelial cell cluster composition, we have performed compositional analysis using Cacoa⁷, as mentioned by reviewer 3 (Supplementary Table 10 part5), which showed:
 - that there is no significant difference in EC cluster composition between the adult/control brains (temporal lobe) samples by age.
 - that there is no significant difference in EC cluster composition between male and female samples for the different entities studied, notably: TL, AVM, LGG, GBM, MET, and MEN.
 - that there is no significant difference in EC cluster composition according to the MGMT methylation status for the GBM samples.
 - that there is no significant difference in EC cluster composition according to the histopathological subtype for the brain MET samples.
- iic) additionally we have performed analysis using the scITD package⁸, as also suggested by reviewer 3 (Review Figure 3n-v). The results show:
 - that there are no significant age-associated factors in sorted ECs of temporal lobe (adult/control brain) samples.
 - that there are no significant sex-associated factors in sorted ECs of all the studied entities.
 - that there are no significant MGMT methylation status associated factors in sorted ECs of GBM samples.
 - that there are no significant metastasis subtype associated factors in sorted ECs of brain metastasis samples.
- iid) we have performed differential expression analysis with regression of age, sex using the DEseq package (DOI: 10.18129/B9.bioc.DESeq2) (Supplementary Table 23), and showed that the results are not affected at the gene and pathway level (Jaccard index = 0.988).
- iie) integration/batch correction with regression of sex/age covariates, and showed that the results are not affected (Reviewer Figure 3w-z).
- iif) To further validate these findings in other cohorts and to further address the reviewer's comments, we performed additional analyses using publicly available datasets for age and sex related genes. By compiling multiple publicly available brain single cell/single nucleus RNA-seq (sc/snRNA-seq) datasets that have "age" and "sex" annotation of endothelial cells.
- regarding the sex comparison:
 - we have compiled 9 human sn-RNAseq datasets from 7 publications⁹⁻¹⁵, that included 278 patients and computationally extracted/subset 34,322 endothelial cells
 - then did differential expression analysis comparing male vs female endothelial cells, the resulting genes were filtered based on a threshold of being $p\text{-value} < 0.05$, $\log_2FC \geq 0.25$, resulting in a set of brain ECs sex regulated genes (Reviewer Figure 3g-i).

- X- and Y-chromosome genes obtained from the C1 positional geneset (http://www.gsea-msigdb.org/gsea/msigdb/human/collection_details.jsp#C1), (Reviewer Figure 3j-m).
- regarding the age comparison
 - we have analyzed the endothelial cell subset of Ximerakis et al., *Nature Neuroscience* 2019¹⁶, that did single-cell transcriptomic profiling of the aging mouse brain.
 - performing differential expression analysis between 8 young (2–3 months) and 8 old (21–23 months) mice's endothelial cells (the resulting genes were filtered based on a threshold of being $p\text{-value} < 0.05$, $\log_2FC \geq 0.25$), followed my homology mapping to human genes resulting in a set of brain ECs age regulated genes (Reviewer Figure 3a-f).

The gene lists generated in the aforementioned points were compared to the pathological signatures. we defined in our paper using:

- Venn diagrams
- Correlation analysis using Jaccard test and Fisher's exact test
- GSEA analysis

(Reviewer Figure 3a-m, Supplementary Table 23).

In summary, the additional analyses show that in endothelial cells i) differentially expressed genes, ii) resulting signaling pathways, iii) endothelial cell cluster composition, and iv) results of integration/batch correction in the adult control brains and in the different diseases are not influenced by age and sex, further underlining that the covariates age and sex do not influence our biological findings across adulthood diseases.

Regarding the Supplementary Table 10 and Supplementary Table 20 (currently is Supplementary Table 19), we would like to mention that upon the first round of REVISIONS, (REVISIONS I) those tables were provided as excel files with the corresponding labeling and submitted through the *Nature* portal. We are not exactly sure why the reviewer wasn't able to locate the Supplementary Table 10 but apologize for any possible confusion. We have made sure again that the labeling of the Supplementary Tables is correct and are confident that the Supplementary Tables 10 and 20 (currently is Supplementary Table 19) can be found and read.

We have increased the plot sizes and improved the labelling of Reviewer Figure 1 to make it easier to read and analyze. The plots provided in Reviewer Figure 1 are labeled with the compared conditions as well as with "Real comparison" to indicate the correctly labeled samples' comparison or labeled as "Shuffled comparison" to indicate the randomly shuffled sample comparisons.

Regarding the concerns for the variation within the adult age group (15 years to 69 years)

Regarding the age range of the adult/control temporal lobe samples, we have specifically examined the youngest age (15 years) and the oldest age (69 years) and the results indeed show that these two are not outliers (Reviewer Figures 1, 2, 3 and Supplementary Tables 10, 19).

We would like to clarify to the reviewer why these two ages on either end of the range are indeed “not outliers” and why the parameter “age” is not a confounder:

The volcano plot in the Reviewer Figure 1c displays the comparison of the 15 years old adult/control brain (temporal lobe) sample to all other adult/control brain (temporal lobe) samples.

In order to address whether the observed differences (that the reviewer is commenting on) are due to the parameter “age” of the adult/control brain (temporal lobe) sample, we then performed shuffling of the parameter “age” of all adult/control brain (temporal lobe) samples.

- comparison of the “real age comparison” with the “shuffled age comparison” revealed that the number of significant DEGs observed in the “real age comparison” were not statistically different to the ones observed in the “shuffled age comparison” (Reviewer Figure 1c, Supplementary Table 19)

Indicating that the parameter “age” (including the two ages on either end of the age range) is not a confounder in the adult/control brain (temporal lobe) samples.

Moreover, to further validate that "age" is not a confounder affecting our adult/control brain (temporal lobe) endothelial datasets, we have utilized publicly available datasets addressing the effect of the parameter "age" in the mouse brain.

- i) Namely, we have analyzed the endothelial cell subset of Ximerakis et al., *Nature Neuroscience* 2019¹⁶- that did single-cell transcriptomic profiling of the aging mouse brain -. performing differential expression analysis between 8 young (2–3 months) and 8 old (21–23 months) mice's endothelial cells (the resulting genes were filtered based on a threshold of being $p\text{-value} < 0.05$, $\log_2\text{FC} \geq 0.25$), followed my homology mapping to human genes resulting in a set of brain ECs age regulated genes (Reviewer Figure 3a-f).
- ii) Next, we have performed differential expression analysis comparing the 15 years old adult/control brain (temporal lobe) sample to all other adult/control brain (temporal lobe) samples. As, well as, differential expression analysis comparing the 69 years old adult/control brain (temporal lobe) sample to all other adult/control brain (temporal lobe) samples.

The gene list generated in i) was compared to the gene lists generated in ii) using:

- Venn diagrams
- Correlation analysis using Jaccard test and Fisher's exact test

As shown in Reviewer Figure 3a-f, Supplementary Table 23.

The results indicate that the brain ECs age regulated genes are not significantly differentially regulated in the 15 and 69 years old adult/control brain (temporal lobe) EC samples thus further validating that those samples are not outliers to the dataset.

Finally, we have performed analysis using scITD package (Reviewer Figure 3n-v), as mentioned by reviewer 2. The results show that there are no significant age associated factors in temporal lobe (adult/control brain) ECs confirming our aforementioned conclusions.

We have confirmed the compositional analysis results of scCODA with other methods of compositional analysis such as DirichReg, Propeller, tascCODA, and Cacoa indeed verifying that there is no significant cell type compositional changes between the 15 and 69 years old temporal lobe EC samples and other patients in the dataset (Supplementary Table 10).

Regarding the concerns regarding the sex distribution

Regarding the sex distribution, we have performed differential expression analysis between ECs of female and male patients for every single entity including adult, and diseased entities/brains.

We would also like to clarify to the reviewer why the parameter "sex" is not a confounder:

The volcano plot in the Reviewer Figure 1d-i displays the comparison of the male versus female samples for the adult/control brain (temporal lobe) as well as for all pathological samples (including all pathological brain samples, the brain tumor samples, and the brain vascular malformation samples).

In order to address whether the observed differences (that the reviewer is commenting on) are due to the parameter "sex" of the adult/control brain (temporal lobe) samples and the pathological brain samples, we then performed shuffling of the parameter "sex" of all the adult/control brain (temporal lobe) samples and the pathological brain samples.

- Comparison of the "real sex comparison" with the "shuffled sex comparison" revealed that the number of significant DEGs observed in the "real sex comparison" were not statistically different to the ones observed in the "shuffled sex comparison" (Reviewer Figure 1d-i, Supplementary Table 19).

indicating that the parameter "sex" is not a confounder in the adult/control brain (temporal lobe) as well as in the pathological brain samples analyzed.

In summary, we conclude that the parameters "age" and "sex" are not confounders for the observed findings in this manuscript.

However, we would like to mention that for both parameters and related scientific questions, future dedicated studies will be needed to thoroughly address them.

COMMENT:

- *Language regarding fetal pathways is refined.*

ANSWER:

We thank the reviewer for this comment.

COMMENT:

- Concern for grouping of genetically distinct GBM tissues was noted by the authors, and it is acknowledged that there are not sufficient sample numbers to compare tumors with different genetic profiles. Although the authors acknowledge that future studies would be needed, a recurrent theme amongst the diseases discussed in this paper are insufficient numbers and the inclusion of several distinct entities rather than a more focused, better powered study.

ANSWER:

We thank the reviewer for this comment.

Regarding the genetically distinct GBM tissues.

Regarding the genetic signatures of the GBM tissues, we would like to emphasize that we actually do not group GBM tumors with different genetic profiles as the two groups that are compared are genetically homogeneous with regard to the IDH1/2 mutational status.

Regarding the IDH1/2 mutation status of gliomas

Regarding the IDH1/2 mutation status of gliomas, we would like to point out that:

- all lower-grade glioma samples are IDH1 mutant except one sample which is IDH2 mutant (Supplementary Table 1)
 - of note, IDH2 mutations are mutually exclusive with IDH1 mutations¹⁷
- whereas all high-grade glioma/glioblastoma samples are IDH1 wild-type (Supplementary Table 1)

accordingly, with regard to the IDH1/2 mutation status of gliomas, our analyses of lower-versus high-grade gliomas/glioblastomas already account for their genetic IDH1/2 profile.

Notably, according to the new WHO classification¹⁸ which was published in August 2021:

- the presence of IDH1 mutation defines the diagnosis of WHO grade II or III glioma
 - whereas the absence of IDH1 mutation defines a WHO grade IV glioblastoma
- which further validates our comparisons between IDH1/2 mutant lower grade and IDH1 wild-type high-grade gliomas/glioblastomas.

Finally, we would also like to mention that – according to the previous WHO classification¹⁹ IDH1 mutant high-grade gliomas/glioblastomas considered a very rare entity accounting for only approximately 3.7%-5% of glioblastomas (GBMs)¹⁷.

Moreover, to further address this reviewer comment, we have also added additional samples of sorted ECs, notably:

We have analyzed 13 GBMs and 8 LGG samples (Extended Data Figure 1, Supplementary Table 1). Moreover, for the diseased entities for which we have low sample numbers, for instance hemangioblastoma and cavernoma, we have removed them from the atlas while more clearly pointing out the commonalities across diseases including:

- 57 diseased samples coming from 50 patients for the sorted analysis
 - 18 diseased samples coming from 18 patients for the unsorted analysis
- thereby avoiding conclusions that are based on only a few samples.

Finally, we would also like to add that we aimed to address the brain vasculature at different levels, notably:

- level of all pathologies
- level of individual entities
- level of individual patients

And that one key finding of our paper is the identification of common hallmarks of disease across different pathologies (Figures 1, 2, Supplementary Figure 5).

COMMENT:

- Several instances throughout the author response, an off-the-record discussion with editorial staff was cited as rationale for dismissal of a reviewer comment. The author should consider a more

thorough rationale for either exclusion of under-powered disease comparisons (and thus appropriately make the overall manuscript more concise) or include more samples where comparisons are made and include them in the paper.

ANSWER:

Regarding the off-the-record discussion with editorial staff

We would really like to apologize to the reviewer for this impression. We would also like to clarify that we never dismissed any reviewers' comments but tried to answer to all points raised. When discussing the strategy for revisions with the senior editor Dr. George Caputa on the 25-1-2022 and on 19-01-2023, we emphasized that we aim to respond to all editors' and reviewer's requests.

Importantly, we would like to emphasize that we performed all experimental and computational tasks asked by the reviewers and the editors. During this second round of REVISIONS (REVISIONS II), we have further very thoroughly addressed all raised points by the reviewers.

Regarding the exclusion of under-powered disease comparisons

Regarding the reviewer's comment about the under-powered disease comparisons, we have now:

- i) removed the very rare samples of both sorted ECs and of unsorted ECs and PVCs (2 cavernomas, 2 temporal lobes adjacent to cavernomas, 2 hemangioblastomas) from the atlas.
- ii) added various additional samples, notably:
 - sorted ECs (3 adult/control brains, 2 lower-grade gliomas, 3 high-grade gliomas/glioblastomas)
 - unsorted ECs and PVCs (2 fetal brains, 2 adult/control brains)

to further validate our findings in the atlas

This being said, we now present a collection of scRNA-seq samples of sorted brain ECs and of unsorted brain ECs and PVCs that is – to the best of our knowledge – unprecedented when compared to the existing literature. Please find below the overview of the number of samples (Extended Data Figure 1).

For sorted brain ECs

- 5 fetal brain samples coming from 5 fetuses
- 5 fetal periphery samples coming from 5 fetuses
- 12 adult/control brain (temporal lobe) samples coming from 11 patients
- 57 diseased samples coming from 50 patients

For unsorted brain ECs and PVCs

- 7 fetal brain samples coming from 7 fetuses
- 7 fetal periphery samples coming from 7 fetuses
- 6 adult/control brain (temporal lobe) samples coming from 6 patients
- 18 diseased samples coming from 18 patients

In summary, taking into account the additional samples that we included in this second round of REVISIONS (REVISIONS II), 117 sorted and unsorted samples derived from 68 patients were analyzed in our manuscript. We would thus like to emphasize that the conclusions that are made in our manuscript are not based on only a few samples, but are rather based on an unprecedented amount of samples:

- both at the level of the overall merge of all entities
- as well as at the level of individual entities.

thereby focusing the atlas on the analyses where there are enough samples and where robust analyses can be made (while also having excluded the diseased entities with very low sample numbers).

Importantly, the exclusion of very rare samples and the inclusion of additional samples has also clearly made the manuscript and the corresponding figures more concise, as suggested by the reviewer.

We have improved the readability of our manuscript by streamlining the manuscript text, the main Figures, the Extended Data Figures, and the Supplementary Figures, as a result, the manuscript is now shorter, much more focused and thus easier to read.

Regarding the content in the manuscript, we have now edited the manuscript and associated figures in a significant manner in order to make the manuscript and figures more concise. As also suggested by

reviewer 3, we have also moved some panels and plots from the main Figures to the Extended Data Figures. In summary, we have made the following adaptations:

i) manuscript

- substantial shortening of the manuscript, resulting in a word count of 4440 (reduction by 3446 words).

ii) figures

- ii a) substantial simplification of the five main Figures resulting in none of them extending more than half an A4 page
- notably, reduction of the number of panels in the main Figures by removing panels, simplifying existing panels and moving panels and plots to Extended Data Figures, as follows:
- ii a) reduction of the number of panels in the main Figures by moving panels and plots to Extended Data Figures, notably:
 - ii ai) reduction of the number of panels in Figure 1
 - we have moved the former panels Figure 1b,c,d,e to Extended Data Figure 1c,d,e,f and Figure 1s,u to Supplementary Figure 5d,e
 - ii aii) reduction of the number of panels in Figure 2
 - we have removed former panels Figure 2b,c,d,k and moved the former panel Figure 2f to Supplementary Figure 7i
 - ii aiii) reduction of the number of panels in Figure 3
 - we have moved the former panels Figure 3n,o to Extended Data Figure 5zv,zvii
 - ii biv) reduction of the number of panels in Figure 4
 - we have simplified the panels of Figure 4a-c by removing 6 UMAP plots, and we removed former panel Figure 4h
 - ii bv) reduction of the number of panels in Figure 5
 - we have simplified the panels of Figure 5a-c by removing 6 UMAP plots, and we removed former panel Figure 5h
 - ii bivi) conversion of former Figure 6 to Extended Data Figure 11
- ii b) substantial reduction of the number of Extended Data and Supplementary figures, as follows:
 - ii bi) reduction of the number of Extended Data Figures from 37 to 15
 - ii bii) reduction of the number of Supplementary Figures from 41 to 16

Please find below a graphical illustration of the simplifications and shortenings during this round of revisions

Thereby, we now provide an edited, more concise and easily readable version of the manuscript and the figures while still providing all additional data addressing all editors' and the reviewers' comments/requests in a satisfactory and most comprehensive manner. Please note that we also accounted for the reviewer's requests for additional experimental and computational data.

Reviewer #2:

COMMENT:

The authors have addressed substantially the concerns raised by this reviewer. The manuscript and the data that they provide will be a valuable resource for the scientific community.

ANSWER:

We thank the reviewer for their comments on our manuscript.

Reviewer #3:

COMMENT:

The authors have invested a large amount of work to address reviewer's concerns. Nevertheless, my previous critique was only partially addressed. Below are several major points to be addressed.

ANSWER:

We thank the reviewer for their comments on our manuscript.

Provided that some of the questions and comments of reviewer 3, of reviewer 1 and of the editor are overlapping, we feel it is adequate to repeat the overview of what we have done during this second round of revisions (REVISIONS II) for both reviewers and for the editor, for reasons of clarity and convenience.

We have been working extensively in the past months to address all the editors' and reviewers' comments, notably:

- i) we have improved the readability of our manuscript by streamlining the manuscript text, main Figures, Extended Data Figures, and Supplementary Figures
 - as a result, the manuscript is now shorter, much more focused and concise, thus easier to read
- ii) we have addressed the covariates age and sex using state of the art statistical methods in our own datasets and also referring to and analyzing multiple publicly available datasets
 - In summary, the biological findings in our atlas do not depend on the covariates age and sex
- iii) we have compared multiple methods of integration/batch correction for both the sorted and the unsorted samples and based on these results have homogenized and streamlined the computational analyses throughout the paper
 - In summary, the biological conclusions in our atlas do not depend on the method of integration/batch correction
- iv) we have removed the very rare samples from the atlas (2 cavernomas, 2 temporal lobes adjacent to cavernomas, 2 hemangioblastomas),
 - In summary, the biological conclusions in our atlas do not depend on the integration or exclusion of these very rare samples. The very rare samples serve as an additional validation of the findings in the atlas - for instance supporting the concepts of common "hallmarks" of the diseased brain vasculature across various brain pathologies and of reactivated fetal pathways in brain diseases - and will be published in a future follow-up study.
- v) we have also added numerous samples of sorted endothelial cells (ECs) (3 adult/control brains, 2 lower-grade gliomas, 3 high-grade gliomas/glioblastomas) as well as additional samples of unsorted endothelial (EC) and perivascular- cells (PVCs) (2 fetal brains, 2 adult/control brains), analyzed via scRNA-seq to further strengthen and validate our findings in the atlas
 - In summary, the biological conclusions in our atlas are further confirmed by these additional samples, both:
 - i) at the level of individual disease entities
 - ii) as well as across diseases
 - thereby further strengthening
 - i) the signature of diseased brain ECs (and PVCs) for every entity
 - ii) as well as the common signature of activated brain ECs (and PVCs) across entities
 - and also further illustrating the robustness of our biological findings and the usefulness of our atlas as a resource

- vi) we have provided additional computational analyses taking into account the updated set of all 117 samples derived from 68 different patients including:
 - new computational analysis of the scRNA-sequencing data on CD31⁺/CD45⁻ sorted endothelial cells (ECs) and on unsorted endothelial (EC)- and perivascular cells (PVCs) which further confirms the findings of the scRNA-seq atlas
 - additional computational analysis of the bulk RNA-sequencing data on CD31⁺/CD45⁻ sorted endothelial cells (ECs) utilizing bulk RNA-seq deconvolution which confirms the findings of the scRNA-seq atlas
 - an adapted working model in Figure 1a clarifying both the experimental and computational workflows
 - In summary, the new computational analyses further validate/strengthen the computational results and biological findings of our scRNA-sequencing atlas

According to the suggestion of the reviewer (and as also mentioned by reviewer 1, see above) we have now excluded the diseased samples with low numbers (including 2 cavernomas, 2 temporal lobes adjacent to cavernomas, 2 hemangioblastomas) from the atlas.

Moreover, we have also added various additional samples to further confirm our findings in the atlas. In our latest version of our manuscript, we have analyzed 117 samples derived from 68 patients.

Thereby we would like to emphasize that:

- we avoided conclusions that are based on only a few samples
- we further strengthened and confirmed the findings of our scRNA-seq atlas based on a very solid amount of samples
- we more clearly pointed out the commonalities (“hallmarks”) of the pathological brain vasculature across diseases

The analyzed samples are derived from the following entities:

For sorted brain ECs

- 5 fetal brain samples coming from 5 fetuses
- 5 fetal periphery samples coming from 5 fetuses
- 12 adult/control brain (temporal lobe) samples coming from 11 patients
- 57 diseased samples coming from 50 patients

For unsorted brain ECs and PVCs

- 7 fetal brain samples coming from 7 fetuses
- 7 fetal periphery samples coming from 7 fetuses
- 6 adult/control brain (temporal lobe) samples coming from 6 patients
- 18 diseased samples coming from 18 patients

Furthermore, we would like to highlight that the number of both endothelial- as well as perivascular cells per entity is very high and, to our knowledge, unmet as compared to any other currently published study. Moreover, to address the reviewers' comments and further validate our findings, we have performed further bulk RNA-seq analyses. The key findings emanating of the scRNA-seq experiments described in the manuscript were confirmed by the bulk RNA-seq data (Supplementary Figure 6, Supplementary Tables 1-part2, 21^{1,2}).

In summary, we have thereby addressed all key reviewers' and editors' concerns regarding:

- covariates age and sex
- computational methods/overall analytical approach
- sample numbers amongst the different pathologies
- lengthiness of the manuscript and number of figures

COMMENT:

1. Integration of single-cell data is still the major issue. While in the rebuttal there is a structured description of when and what type of integrations were used, in the manuscript, it is difficult to understand. This approach to have some analyses with CCA integration, some with RPCA integration, some without integration does not help the reader to grasp fast whether an analysis is robust or not.

ANSWER:

We thank the reviewer for this comment and the very helpful inputs and feedback regarding the different methods of integration/batch correction. We would like to apologize that the description of the methods of integration/batch correction wasn't clear enough in the manuscript.

Regarding the different methods of integration/batch correction

Accordingly, regarding the different methods of integration/batch correction, we have now further clarified the computational methods of integration/batch correction used, in the:

- i) “Methods” section
- ii) updated working model in Figure 1a, which streamlines the workflow

This now helps the reader to quickly grasp the method of integration/batch correction used.

Moreover, we would like to clarify below a few aspects regarding the integration/batch correction methods that we have used for the different sorted (ECs) and unsorted (ECs and PVCs) datasets.

COMMENT:

For instance, lines 175-177:

To broadly map the neurovasculature and to address the NVU across different brain entities, 44,116 fetal, 47,305 adult/control and 121,436 pathological unsorted EC and PVC transcriptomes from 33 patients were pooled, clustered, annotated using known marker genes

Was it done with integration or not? The methods part for single-cell experiments is all together under umbrella of one chapter “Chromium 10X library preparation, single cell RNA-sequencing and data analysis” and it is difficult to work through. This chapter should be split into subchapters for different analyses.

To address the issue with integration of single-cell data, the authors should either use the same method for integration across their analyses or make it clear in the Results text, with further details in the Methods, where what type of integration was used. Arguments for justification why CCA was used should be provided.

Finally, the Reviewer figure 2 in fact does show that CCA integration is far more noisy in comparison to eg scANVI. Obviously, by integrating using CCA, the authors miss many cell type-relevant markers that are removed by CCA during integration process. This needs to be admitted in the manuscript (unless it is shown that it is not the case). Comparison of integration by different methods needs to be included in the manuscript (not Reviewer figure, proper Extended data figure) and this should include how integration with RPCA and an alternative method (Harmony or scANVI) affects differential gene expression. Jaccard index that was already used for stability analysis by the authors is a good tool to implement here

ANSWER:

We thank the reviewer for this comment and would like to clarify what we meant and explain the different methods of integration/batch correction used.

Regarding the different experiments and datasets in our manuscript

As outlined in the adapted working model Figure 1a and explained in both paragraphs “Results” and “Methods”, we performed two different set of experiments and subsequent analysis of corresponding datasets, notably:

- i) experiments and datasets of CD31⁺/CD45⁻ FACS-sorted ECs
- ii) experiments and datasets of unsorted ECs and PVCs

and the text in the manuscript:

“For instance, lines 175-177:

To broadly map the neurovasculature and to address the NVU across different brain entities, 44,116 fetal, 47,305 adult/control and 121,436 pathological unsorted EC and PVC transcriptomes from 33 patients were pooled, clustered, annotated using known marker genes”

refers to the datasets of unsorted ECs and PVCs, which was – as opposed to the dataset of sorted ECs – not integrated/batch corrected at the level of the overall merge of all entities (fetal, adult/control, and pathological brains) together.

To further clarify the different methods of integration/batch correction, we would like to mention that in summary, the two key questions of the reviewer are:

- i) why were there different methods of integration/batch correction:

- ia) RPCA for integration/batch correction of CD31⁺/CD45⁻ FACS-sorted endothelial cells at the level of the overall merge of all entities (fetal, adult, and pathological brains)
- ib) CCA for the integration/batch correction of the overall merge of unsorted endothelial- and perivascular cells at the level of all entities separately (e.g. in fetal brains)
- ii) why was there no integration/batch correction for the unsorted endothelial- and perivascular cells at the level of the overall merge of all entities (fetal, adult/control, and pathological brains)

Our explanations regarding these two key questions:

We would like to clarify the computational analysis and that there seems to be a certain amount of confusion.

Regarding the different methods of integration/batch correction

- i) We find it appropriate to:
 - ia) integrate/batch-correct the endothelial cells from all entities (fetal, adult and pathological brain entities) together in an overall merge using RPCA
 - ib) integrate/batch correct the unsorted endothelial- and perivascular cells at the level of the individual entity.
 - ibi) for both the initial and first revised version (REVISIONS I) of the paper, we integrated/batch corrected the unsorted endothelial- and perivascular cells at the level of the individual entity using CCA (this integration/batch correction was already in the revised version of the paper and was approved by this reviewer in the first round of REVISIONS (REVISIONS I) as quoted below.
 - based on the reviewer's comments in the first round of REVISIONS (REVISIONS I), CCA performed well and could be used to identify conserved cell types
 - ibii) based on the reviewer's comments in the second round of REVISIONS (REVISIONS II), we meanwhile performed additional computational analyses comparing the different methods of integration/batch correction (namely CCA with RPCA and Harmony) for the unsorted endothelial- and perivascular cells at the level of the individual entity
 - we have added a figure that shows the comparison of the different integration/batch correction methods for the unsorted endothelial- and perivascular cells at the level of every individual entity (Reviewer Figure 4)₂ and reveals that the cell clustering is conserved across/between the three integration/batch correction methods.

Regarding point ibi), we would also like to briefly mention that during the first round of REVISIONS (REVISIONS I) of our manuscript, we were trying to take into account the reviewer's comment stating the following:

“CCA is used to integrate samples with conserved cell types, but with large expression differences, whereas RPCA has a much “softer” way of integration” and “Thus, CCA could help to identify similar cell types in control and disease, but certainly is not the best way to reveal gene expression differences between conditions.”

and therefore decided back then to utilize both methods of integration/batch correction, notably CCA and RPCA.

Importantly, however, in order to be consistent with our analyses throughout the manuscript as the reviewer advised during the second round of REVISIONS (REVISIONS II), we have now performed all integration/batch correction using RPCA, which performed very well for both the sorted EC and unsorted EC and PVC datasets (Figure 2a, Extended Data Figure 4, Reviewer Figure 4).

Regarding the question of integration/batch correction of the unsorted dataset

- ii) we don't find it suitable to:
 - integrate/batch-correct the unsorted endothelial- and perivascular cells from all entities (fetal, adult and pathological brain entities) as their cellular composition is very different (e.g. it is not suitable to integrate glial brain tumor cells with brain AVM fibroblasts or fetal pericytes as it would result in wrongly annotated and noisy clusters, Reviewer Figure 5).

So, in order to address the reviewers' comments and to streamline the method of integration/batch correction used, we have clarified in the manuscript the following analysis done:

- i) we have used RPCA for integration/batch correction of:
 - sorted ECs of individual patients within a given brain entity, e.g:
 - Integration of ECs of the five fetuses' brains together.
 - Integration of ECs of the nine adult/control brain (temporal lobe) patients together.
 - Integration of ECs of the eight GBM patients together.
 - Integration of ECs of the five AVM patients together, etc.
 - and the same for every individual entity for the sorted endothelial cells' samples.
 - unsorted cells (ECs and PVCs) of individual patients within a given brain entity, e.g:
 - Integration of unsorted cells of the seven fetuses' brains together.
 - Integration of unsorted cells of the six adult/control brain (temporal lobe) patients together.
 - Integration of unsorted cells of the five GBM patients together.
 - Integration of unsorted cells of the five AVM patients together, etc.
 - and the same for every other individual entity for unsorted ECs and PVCs samples.
- ii) we have used RPCA for integration/batch correction of:
 - the overall merge of CD31⁺/CD45⁻ FACS sorted endothelial cells from different brain entities (fetal, adult/control, and pathological brains).
- iii) we did not do the integration/batch correction at the level of the overall merge of different brain entities for the unsorted cells (ECs and PVCs).
 - the reason is that the different cell types in the unsorted samples (ECs and PVCs) are not the same across different the brain entities (fetal brains, adult/control brains, brain tumors and brain vascular malformations), and as the cell types cluster by cell type, is not effective to integrate/batch correct the different entities, as it would result in incorrect annotation and noisy clusters (Reviewer Figure 5).
 - this is obviously different for the sorted samples which contain only ECs and thus integration and batch correction across the different brain entities is effective (see ii)).

In summary

- i) integration/batch correction at the level of the overall merge of all entities
 - ia) for FACS-sorted CD31⁺/CD45⁻ endothelial cells
 - method of choice: RPCA
 - scANVI and Harmony, CCA are equally good, (Extended Data Figure 4), also shown by applying the scIB diagnostics (Modified average silhouette width (ASW)) (Supplementary Table 22)²⁰. As the reviewer advised we have now included this figure in manuscript by making it an Extended Data Figure (it was formerly Reviewer Figure 2), in which we have also integrated the Jaccard index analysis as per the reviewer's recommendation.

importantly, independent of the method of integration/batch correction, the biological findings are not affected

 - ib) for unsorted endothelial- and perivascular cells
 - integration/batch correction was not done/used at the level of the overall merge of all entities
- ii) integration/batch correction at the level of the individual entities
 - iia) for FACS-sorted CD31⁺/CD45⁻ endothelial cells
 - method of choice: RPCA
 - iib) for unsorted endothelial- and perivascular cells
 - method of choice: RPCA
 - CCA and harmony are equally good (Reviewer Figure 4), also proven by Jaccard index comparison.

Importantly, independent of the method of integration/batch correction, the biological conclusions are not affected. Thus, as the reviewer advised, we have applied the same method of integration/batch

correction (RPCA) across the different analyses we performed in the manuscript, and we have clarified the details in the “Methods” section.

As outlined above, we have also provided newly performed computational analyses based on the reviewer’s comments comparing the different algorithms for integration/batch correction, the cell cluster annotation was very consistent (Extended Data Figure 4, Reviewer Figure 4). The comparison to the methods of integration/batch correction that we used, showed very similar results (Extended Data Figure 4, Reviewer Figure 4).

Notably, differential gene expression that is used for cell annotation/AV-annotation and pathway analysis is always done on the non-integrated/non batch corrected counts (i.e. the “RNA assay of Seurat”) and thus does not involve integrated/batch corrected counts – which is also the recommendation of SEURAT package developers.

Importantly, we have performed the integration/batch correction analysis taking into account key benchmarking articles comparing methods of integration/batch correction, as mentioned by the reviewer:

<https://www.biorxiv.org/content/10.1101/2020.05.22.111161v2>

which was meanwhile published in *Nature Methods*²⁰:

<https://www.nature.com/articles/s41592-021-01336-8>

as well as: "Tran, H. T. N. et al. A benchmark of batch-effect correction methods for single- cell RNA sequencing data. *Genome Biol* 21, 12, doi:10.1186/s13059-019-1850- 9 (2020)."

COMMENT:

2. Addition of metadata and QC plots helped a lot in understanding where the data is robust and where it is not. Here also a question whether it is better to focus on something more specific arises, since some sample sets provide only limited knowledge.

To illustrate this, I will walk through TL adjacent CAV samples using the QC and metadata that are now available. These samples have by far fewer genes than other sample categories. When looking in metadata, Supplementary table 3, a very small number of cells from TL adjacent CAV samples were filtered – out of ~10,000 cells, ~200 cells were filtered. However, as standard QC, cells with <500 genes were filtered, and based on Extended data figure 3d, it looks like many cells should be filtered. Thus, first more specific question arise – can the authors double check their QC plots and tables to confirm that there are no mistakes.

Second more general question, why TL adjacent CAV samples are needed at all? They are low quality, only 2 samples, not enough to draw any conclusions for comparison to TL or CAV samples. Why to keep such samples in the atlas?

ANSWER:

We thank the reviewer for their comment, we are happy to hear that the additional QC plots were helpful for the understanding.

We have double checked the QC plots and tables and we can confirm that there are no mistakes.

Moreover, as advised by this reviewer and by reviewer 1, we have removed the very rare samples from the atlas (2 cavernomas, 2 temporal lobes adjacent to cavernomas, 2 hemangioblastomas). Importantly the biological conclusions in our atlas do not depend on the integration or exclusion of those very rare samples.

COMMENT:

Finally, most general comment – the effect of covariates can be huge, eg it was already shown that in single-cell analyses age, sex, sequencing batch, processing site etc can have a large impact on single-cell data. The authors did try to address sex and age covariate effects on cell type composition using scCODA. However, unfortunately, scCODA does not have integrated covariate option. Tools to study covariates are only being developed now, with most advanced likely scITD and Cacao, which are still pre-prints. Nevertheless, even if implementing the most advanced tools, the impact of covariates on the current atlas can be only superficially addressed, since in the current version, there are simply too many conditions. Focusing the atlas on those analyses where there are enough samples, and where robust analyses can be made – this is what the users of the atlas need.

ANSWER:

We thank the reviewer for this comment. We appreciate the reviewer's suggestions and have in consequence:

- removed the samples with very low numbers
- increased the number of samples
- and thoroughly addressed the influence of all covariates on the biological findings in our atlas which we outline point-by-point below.

Regarding the number of different conditions in the atlas, the exclusion of rare samples and of under-powered disease comparisons, and the addition of samples

To address the reviewer's comment regarding the number of different conditions in the atlas and the under-powered disease comparisons, we have now:

- removed the very rare samples (2 cavernomas, 2 temporal lobes adjacent to cavernomas, 2 hemangioblastomas) from the atlas.
- added various additional samples (3 adult/control brains 2 lower-grade gliomas, 3 high-grade gliomas/glioblastomas) to the atlas.

thereby further strengthening and validating our findings in the atlas by focusing the atlas on the conditions with enough samples.

This being said, we now present a number of collected scRNA-seq samples of both sorted brain ECs and of unsorted brain ECs and PVCs that is – to the best of our knowledge – unprecedented when compared to the existing literature. Accordingly, please find below the overview of the number of samples (Extended Data Figure 1).

Provided that some of the questions and comments of reviewer 3 and of reviewer 1 are overlapping, parts of the following text can also be found in response to reviewer 1 (see above). Accordingly, wherever we feel it is adequate, we repeat part of these answers for reasons of clarity and convenience for the two reviewers.

The analyzed samples are derived from the following entities:

For sorted brain ECs

- 5 fetal brain samples coming from 5 fetuses
- 5 fetal periphery samples coming from 5 fetuses
- 12 adult/control brains (temporal lobes) samples coming from 11 patients
- 57 diseased samples coming from 50 patients

For unsorted brain ECs and PVCs

- 7 fetal brain samples coming from 7 fetuses
- 7 fetal periphery samples coming from 7 fetuses
- 6 adult/control brains (temporal lobes) samples coming from 6 patients
- 18 diseased samples coming from 18 patients

In summary, taking into account the additional samples that we included in this second round of REVISIONS (REVISIONS II), 117 sorted and unsorted samples derived from 68 patients were analyzed in our manuscript. We would thus like to emphasize that the conclusions that are made in our manuscript are not based on only a few samples, but are rather based on an unprecedented amount of samples:

- both at the level of the overall merge of all entities
- as well as at the level of individual entities.

thereby focusing the atlas on the analyses where there are enough samples and where robust analyses can be made (while also having excluded the diseased entities with very low sample numbers).

Regarding the potential influence of the covariates age and sex on the biological findings in the atlas

To comprehensively address the covariates age and sex in our own datasets, we have performed various analysis during the two rounds of REVISIONS.

- i) Upon the first round of REVISIONS, we had performed the following analyses:
 - ia) comparison of the parameters “age” and “sex” against random shuffles of the parameters “age” and “sex”, which revealed:
 - that the number of significant DEGs between the different ages and the two sexes are low (Reviewer Figure 1).

- and that these differences in terms of the number of significant DEGs are comparable to or even lower than (and thus not significantly different) random shuffles of the parameters “age” and “sex” (Supplementary Table 19).
- ib) pathway analysis comparing the different ages/sexes, on the DEGs for both parameters “age” and “sex”
 - revealing that no biologically meaningful pathways were significantly enriched
- ic) scCODA analysis showed no significant compositional changes between the different sexes and ages in the different entities (Reviewer Figure 2, Supplementary Table 10-part1)
- ii) In addition, for this second round of REVISIONS (REVISIONS II), we have now performed the following additional analyses:
 - iia) to regress the covariates age and sex, we have utilized we have applied the state-of-the-art statistical methodologies which have integrated covariate options³⁻⁵ while performing cell compositional analysis (Supplementary Tables 10-part2, 10-part3, 10-part4). Notably, we referred to three independent computational packages/algorithms, namely:
 - Tree-aggregated amplicon and single-cell compositional data analysis (tascCODA) which is a fully Bayesian model for tree-aggregated modeling of count data and is an extension of the scCODA³.
 - Dirichlet regression analysis (DirichletReg R package) dirichlet regression models can be used to analyze a set of variables lying in a bounded interval that sum up to a constant (e.g., proportions, compositions, etc.) exhibiting skewness and heteroscedasticity, without having to transform the data^{4,6}.
 - Propeller method (speckle R package), which accommodates complex experimental designs and has the flexibility to model additional covariates of interest⁵.

Importantly, we can conclude that the key findings related to cell compositional changes in sorted ECs of brain tumors' and brain vascular malformations' ECs as compared to sorted ECs from adult/control brain ECs, are not affected by the covariates age and sex utilizing the three above-mentioned methods. These analyses verify the scCODA findings during the first round of REVISIONS (REVISIONS I).

- iib) to further address possible changes in endothelial cell cluster composition, we have performed compositional analysis using Cacoa⁷ - based on the reviewer's comment (Supplementary Table 10 part5), which showed:
 - that there is no significant difference in EC cluster composition between the adult/control brains (temporal lobe) samples by age
 - that there is no significant difference in EC cluster composition between the male and female samples for the different entities studied, notably: TL, AVM, LGG, GBM, MET and MEN.
 - that there is no significant difference in EC cluster composition according to the MGMT methylation status for the GBM samples
 - that there is no significant difference in EC cluster composition according to the histopathological subtype for the brain MET samples
- iic) we have performed analysis using scITD package - based on the reviewer's comment (Review Figure 3n-v), results show that:
 - that there are no significant age-associated factors in sorted ECs of temporal lobe (adult/control brain) samples
 - that there are no significant sex-associated factors in sorted ECs of all the studied entities
 - that there are no significant MGMT methylation status-associated factors in sorted ECs of GBM samples
 - that there are no significant metastasis subtype-associated factors in sorted ECs of brain metastasis samples
- iid) we have performed differential expression analysis with regression of age and sex using DEseq package (DOI: 10.18129/B9.bioc.DESeq2) (Supplementary Table 23), and the results revealed that the findings are not affected at the gene and pathway level (Jaccard index = 0.988).

- iie) we have performed integration/batch correction with regression of sex/age covariates, and showed that the results are not affected (Reviewer Figure 3w-z)

- iif) To further validate these findings in other cohorts and to further address the reviewer's comments, we have performed additional analyses using publicly available datasets for age and sex related genes. By compiling multiple publicly available brain sc/snRNA-seq datasets that have “age” and “sex” annotation of endothelial cells.
- regarding the sex comparison
 - we have compiled 9 human snRNA-seq datasets from 7 publications⁹⁻¹⁵, that included 278 patients and computationally extracted/subset 34,322 endothelial cells
 - then did differential expression analysis comparing male vs female endothelial cells, the resulting genes were filtered based on a threshold of being p-value < 0.05, log₂FC ≥ 0.25), resulting in a set of brain ECs sex regulated genes (Reviewer Figure 3g-i).
 - X- and Y- chromosome genes obtained from the C1 positional geneset (http://www.gsea-msigdb.org/gsea/msigdb/human/collection_details.jsp#C1), (Reviewer Figure 3j-m).
- regarding the age comparison
 - we have analyzed the endothelial cell subset of Ximerakis et al., *Nature Neuroscience* 2019¹⁶- that did single-cell transcriptomic profiling of the aging mouse brain
 - performing differential expression analysis between 8 young (2–3 months) and 8 old (21–23 months) mice's endothelial cells (the resulting genes were filtered based on a threshold of being p-value < 0.05, log₂FC ≥ 0.25), followed my homology mapping to human genes resulting in a set of brain ECs age regulated genes (Reviewer Figure 3a-f).

The gene lists generated in the aforementioned points were compared to the pathological signatures that we defined in our paper using:

- Venn diagrams
- Correlation analysis using Jaccard test and Fisher's exact test
- GSEA analysis

(Reviewer Figure 3a-m, Supplementary Table 23).

In summary, the additional analyses show that in endothelial cells:

- i) differentially expressed genes,
 - ii) resulting signaling pathways,
 - iii) endothelial cell cluster composition
 - iv) results of integration in the adult control brains and in the different diseases
- are not influenced by age and sex, further underlining that the covariates age and sex do not influence our biological findings across adulthood diseases.

Regarding sample collection and the potential influence of the covariates sequencing batch and processing site on the biological findings in the atlas

Finally, we would like to highlight that we have followed a very stringent methodology in sample collection, preparation and sequencing thereby minimizing the effects of sequencing batch and processing site. In consequence, we are very confident that technical introduced variability is minimal.

Regarding the experimental methodology of sample harvesting:

In brief, from an experimental point of view, the tissue samples were enzymatically digested using collagenase and dispase immediately after tissue harvesting (see Figure 1a,b, for technical details see the “Methods” section).

- For the unsorted EC and PVCs samples, the digested tissue was then FACS sorted to exclude debris (via forward- and side scatter plots, see the “Methods” section) and dead cells (via cytox blue, see the “Methods” section), and then immediately single-cell sequenced using the 10x droplet protocol (see Figure 1a,b, for technical details the “Methods” section).
- For the sorted EC samples, the harvested tissue was digested, FACS sorted to exclude debris and dead cells and then processed using CD31⁺/CD45⁻ FACS sorting to gate for ECs and

subsequently single-cell sequenced using the 10x droplet protocol (see Figure 1a, for technical details see the “Methods” section).

Regarding the computational methodology of integration/batch correction:

- Regarding sequencing batch and processing sites, we would like to further emphasize that we have compared multiple methods of integration/batch correction, and that – importantly – we found that independent of the method of integration/batch correction, the biological findings are not affected as referred to above.

COMMENT:

3. *Based on Jaccard index (Extended data figure 10h), capillary subclusters are very unstable, thus subclustering of capillary cells should be reduced since these subclusters can be just due to noise.*

ANSWER:

We thank the reviewer for this comment.

Regarding the Jaccard index and the capillary subclusters, we would like to mention that we show the subcluster analysis in Supplementary Figure 7a,f,g,h.

As mentioned in the paragraph “Results”, we pooled EC (sub)clusters of similar identity, notably:

- arterial subclusters
- venous subclusters
- capillary subclusters

for further downstream analyses. This resulted in the 14 EC clusters displayed in Figure 2a, Figure 3a,e,i and Supplementary Figure 7a,f,g,h.

Accordingly, these 14 EC clusters are subdivided as follows (Figure 2a):

- i) the arterial compartment is composed of the EC clusters
 - large artery
 - artery
 - arteriole
- ii) The venous compartment is composed of the EC clusters
 - large vein
 - vein
 - venule
- iii) and the capillary compartment is composed of the EC clusters
 - capillary
 - angiogenic capillary
- iv) EC clusters outside arteriovenous specification
 - (proliferating) EndoMT
 - (proliferating) stem-to-EC
 - proliferating cells

Importantly, we would like to emphasize that we provide 14 EC clusters in which every compartment (i.e. arterial, venous, capillary) is subdivided into multiple EC sub-clusters, highlighting the unprecedented resolution of EC clusters provided here. Accordingly, these 14 EC clusters provide a more accurate representation of the underlying EC biology when compared to recent publications, which display between 3-6 EC clusters^{9,10,21}

As per the reviewers advice, we have reduced the number of capillary subclusters as shown in the Jaccard index analysis (Supplementary Figure 7a,f,g,h). This has resulted in a very high degree of congruity, thereby strongly suggesting that these EC subclusters are actually not just due to noise but in contrast, are biological meaningful.

COMMENT:

4. *Overall, the figures contain too many panels squeezed too tight and it is difficult to go through. The authors should consider different format for some plots and removing of some plots from the main figures to Extended data figures.*

ANSWER:

Regarding the content in the manuscript, we have now edited the manuscript and associated figures in a significant manner in order to make the manuscript and figures more concise. As also suggested by reviewer 1, we have moved some panels and plots from the main Figures to the Extended Data Figures. In summary, we have made the following adaptations:

i) manuscript

- substantial shortening of the manuscript, resulting in a word count of 4440 words (reduction by 3446 words).

ii) figures

- ii a) substantial simplification of the five main Figures resulting in none of them extending more than half an A4 page
- notably, reduction of the number of panels in the main Figure by removing panels, simplifying existing panels and moving panels and plots to Extended Data Figures, as follows:
- ii ai) reduction of the number of panels in Figure 1
 - we have moved the former panels Figure 1b,c,d,e to Extended Data Figure 1c,d,e,f and Figure 1s,u to Supplementary Figure 5d,e
- ii aii) reduction of the number of panels in Figure 2
 - we have removed former panels Figure 2b,c,d,k and moved the former panel Figure 2f to Supplementary Figure 7i
- ii aiii) reduction of the number of panels in Figure 3
 - we have moved the former panels Figure 3n,o to Extended Data Figure 5zv, zvii
- ii biv) reduction of the number of panels in Figure 4
 - we have simplified the panels of Figure 4a-c by removing 6 UMAP plots, and we removed former panel Figure 4h
- ii bv) reduction of the number of panels in Figure 5
 - we have simplified the panels of Figure 5a-c by removing 6 UMAP plots, and we removed former panel Figure 5h
- ii bvi) conversion of former Figure 6 to Extended Data Figure 11
- ii b) substantial reduction of the number of Extended Data Figures and Supplementary Figures, as follows:
 - ii bi) reduction of the number of Extended Data Figures from 37 to 15
 - ii bii) reduction of the number of Supplementary Figures from 41 to 16

Accordingly, to fully respond to the reviewer's request and in order to facilitate readability of the manuscript and figures, we now provide:

- 5 main Figures
 - simplified as compared to the previous round of REVISIONS (REVISIONS I)
 - reduced in number from 6 Figures during the previous round of REVISIONS (REVISIONS I)
- 15 Extended Data Figures
 - massively reduced in number from 37 Extended Data Figures during the previous round of REVISIONS (REVISIONS I)
- 16 Supplementary Figures
 - massively reduced in number from 41 Supplementary Figures during the previous round of REVISIONS (REVISIONS I)

Please find below a graphical illustration of the simplifications and shortenings during this round of revisions

Thereby, we now provide an edited, more concise, and easily readable version of the manuscript and of the figures while still providing all additional addressing all editors' and the reviewers' comments/requests in a satisfactory and most comprehensive manner.

In the previous round of REVISIONS (REVISIONS I), our main goal was to address the reviewers' comments to the greatest extent possible within the provided time frame, with the thought to thereafter adapt the figures for publication in case the paper would be accepted by performing the necessary arrangements.

REFERENCES

- 1 Chu, T., Wang, Z., Pe'er, D. & Danko, C. G. Cell type and gene expression deconvolution with BayesPrism enables Bayesian integrative analysis across bulk and single-cell RNA sequencing in oncology. *Nat Cancer* **3**, 505-517, doi:10.1038/s43018-022-00356-3 (2022).
- 2 Racle, J. & Gfeller, D. EPIC: A Tool to Estimate the Proportions of Different Cell Types from Bulk Gene Expression Data. *Methods Mol Biol* **2120**, 233-248, doi:10.1007/978-1-0716-0327-7_17 (2020).
- 3 Ostner, J., Carcy, S. & Muller, C. L. tascCODA: Bayesian Tree-Aggregated Analysis of Compositional Amplicon and Single-Cell Data. *Front Genet* **12**, 766405, doi:10.3389/fgene.2021.766405 (2021).
- 4 Maier, M. J. DirichletReg: Dirichlet Regression for Compositional Data in R. *Research Report Series* (2014).
- 5 Phipson, B. *et al.* propeller: testing for differences in cell type proportions in single cell data. *Bioinformatics* **38**, 4720-4726, doi:10.1093/bioinformatics/btac582 (2022).
- 6 Eraslan, G. *et al.* Single-nucleus cross-tissue molecular reference maps toward understanding disease gene function. *Science* **376**, eabl4290, doi:10.1126/science.abl4290 (2022).
- 7 Viktor Petukhov, A. I., Rasmus Rydbirk, Shenglin Mei, Lars Christoffersen, Konstantin Khodosevich, Peter V. Kharchenko. Case-control analysis of single-cell RNA-seq studies. *bioRxiv* (2022).
- 8 Jonathan Mitchel, M. G. G., Richard K. Perez, Evan Biederstedt, Raymund Bueno, View ORCID ProfileChun Jimmie Ye, View ORCID ProfilePeter V. Kharchenko. Tensor decomposition reveals coordinated multicellular patterns of transcriptional variation that distinguish and stratify disease individuals. *bioRxiv* (2022).
- 9 Garcia, F. J. *et al.* Single-cell dissection of the human brain vasculature. *Nature* **603**, 893-899, doi:10.1038/s41586-022-04521-7 (2022).
- 10 Yang, A. C. *et al.* A human brain vascular atlas reveals diverse mediators of Alzheimer's risk. *Nature* **603**, 885-892, doi:10.1038/s41586-021-04369-3 (2022).
- 11 Yang, A. C. *et al.* Dysregulation of brain and choroid plexus cell types in severe COVID-19. *Nature* **595**, 565-571, doi:10.1038/s41586-021-03710-0 (2021).
- 12 Grubman, A. *et al.* A single-cell atlas of entorhinal cortex from individuals with Alzheimer's disease reveals cell-type-specific gene expression regulation. *Nat Neurosci* **22**, 2087-2097, doi:10.1038/s41593-019-0539-4 (2019).
- 13 Velmeshev, D. *et al.* Single-cell genomics identifies cell type-specific molecular changes in autism. *Science* **364**, 685-689, doi:10.1126/science.aav8130 (2019).
- 14 Ayhan, F. *et al.* Resolving cellular and molecular diversity along the hippocampal anterior-to-posterior axis in humans. *Neuron* **109**, 2091-2105 e2096, doi:10.1016/j.neuron.2021.05.003 (2021).
- 15 Bennett, D. A. *et al.* Religious Orders Study and Rush Memory and Aging Project. *J Alzheimers Dis* **64**, S161-S189, doi:10.3233/JAD-179939 (2018).
- 16 Ximerakis, M. *et al.* Single-cell transcriptomic profiling of the aging mouse brain. *Nat Neurosci* **22**, 1696-1708, doi:10.1038/s41593-019-0491-3 (2019).
- 17 Cohen, A. L., Holmen, S. L. & Colman, H. IDH1 and IDH2 mutations in gliomas. *Curr Neurol Neurosci Rep* **13**, 345, doi:10.1007/s11910-013-0345-4 (2013).
- 18 Louis, D. N. *et al.* The 2021 WHO Classification of Tumors of the Central Nervous System: a summary. *Neuro Oncol* **23**, 1231-1251, doi:10.1093/neuonc/noab106 (2021).
- 19 Louis, D. N. *et al.* The 2016 World Health Organization Classification of Tumors of the Central Nervous System: a summary. *Acta Neuropathol* **131**, 803-820, doi:10.1007/s00401-016-1545-1 (2016).
- 20 Luecken, M. D. *et al.* Benchmarking atlas-level data integration in single-cell genomics. *Nat Methods* **19**, 41-50, doi:10.1038/s41592-021-01336-8 (2022).
- 21 Winkler, E. A. *et al.* A single-cell atlas of the normal and malformed human brain vasculature. *Science* **375**, eabi7377, doi:10.1126/science.abi7377 (2022).

Reviewer Reports on the Second Revision:

Referees' comments:

Referee #1 (Remarks to the Author):

In this second resubmission by Walchli et al, the authors present a molecular atlas of the human brain across several age groups and disease conditions. Overall, this second-revised manuscript has improved readability and is more concise compared to the prior revisions. Both major concerns previously raised were addressed. The authors address age variation through additional analysis and external validation with published databases. Low sample numbers have been removed (cavernoma, hemangioma) which addresses both the concern for low biologic replicates and refines the scope of the paper. Additional samples were added to GBM atlas. EC transcriptional profiles from scRNA seq database again validated with further bulk RNA seq, but clarified that this validation is derived from different samples (internal validation across sequencing platforms). Clarified reference to select IF, CyCIF, and spatial transcriptomics validation. Although there have been prior published large datasets of the adult brain vasculature, the fetal vasculature, and select disease states, this manuscript does offer a more direct comparison between these atlases with a relatively large sample number. Authors also have comparison to peripheral tissues, which it seems makes this identification of markers more clear than previous work.

One minor concern is that given the current conclusions from authors' analysis and their discussion it is not clear whether this work is broadly applicable beyond the cerebrovascular field. I would suggest that authors elaborate their discussion to describe how they envision their data can be applied more broadly. In addition, language regarding "reactivated fetal pathways" should also be refined in the abstract, as the authors have done in the manuscript body. Significant editing and reduction in the manuscript size has improved the readability. If authors can do some more will make the manuscript clearer.

Referee #3 (Remarks to the Author):

The authors made great efforts to have their manuscript more reader friendly. The data presentation is better organised, and it is easier to read and understand. As for the other two major points that were raised before, data integration and covariate (e.g. sex and age) impact, I have mixed feelings regarding the reply. The replies for each point are very complex, some information is repeated several times, in some instances the sentences are cryptic, and sometimes it was difficult to go through. Overall, I have feelings, although I might be mistaken, that the authors try to use multiple computational tools just to use them and show some data, without really looking deeply in these major issues – whether the data integration went well or whether covariates have an impact on their data. Below I provide detailed explanations.

(1) Integration.

The summary of integration QC is in Extended Data Figure 4. Similar as in the previous round of the revision, it is rather obvious that different ways of integration have different impact on classification (e.g. please look at very different subclustering of arteriole, artery, capillary, mitochondrial at panel b and panel c). As recommended, the authors used Jaccard index to quantify stability of their clusters, and they used `scclusteval` package. Based on the results, the authors claim that the integration was similar between the methods. However, there is some confusion in interpretation of Extended Data Figure 4g-i – such diagonal line of the highest Jaccard indices is not what is the most important, the exact values of Jaccard indices are important. Based in the scale, there are quite some clusters that have maximum Jaccard indices of ~ 0.5 or below, which is highly unstable. In fact, there are only few clusters of >0.85 , which is stable. Thus, the interpretation of Extended Data Figure 4g-i is opposite to what the authors write.

The authors can argue that the stability is much higher in region-wise comparison, which is shown in Reviewer figure 4. This is true, however, comparison in Reviewer figure 4 is done on cell types, where endothelial cells are all in one category “endothelial cells”, whereas in Extended Data Figure 4 endothelial cells are subclustered into 14 subtypes. Subclustering and identification of specific markers for endothelial subtype is the main point of the manuscript and this has to be robust.

(2) Covariate control.

The authors did a lot of additional analyses to demonstrate (potential) lack of age and sex impact on compositional changes. However, it looks as if the authors do not take it serious that covariates can have an impact on their data, since all analyses are either in Reviewer figures (thus not included in the published manuscript) or in complicated Suppl Tables (there are no explanations for the reader how to interpret highly complicated data in Suppl Table 10, which means that the reader will just skip it). Finally, as I also stated in the previous round of revision, it might be that the number of samples per disease type is too low to perform robust analysis for covariate impact, which is the case after going through the revised data (see below).

Coming to the analyses themselves, and the claim that sex and age (and these are only 2 covariates considered, however, there are others) do not have an impact on the data:

- scCODA does show compositional differences in young vs old, e.g. Reviewer figure 2 panel v shows twice higher proportion of arteriole in old, and panel t – twice higher proportion of venule in old
- for propeller method, Phipson et al. *Bioinformatics* 2022, it is a robust method with enough replicates, and the number of replicates is rather small in the compared datasets, so the method is not applicable here

- scITD does not work well on 5-7 samples, most comparisons in Reviewer figure 3n-v; this analysis has to be done on overall integrated dataset control + all diseases to see potential impact and identify covariate specific gene expression signatures
- the authors claimed that they did DESeq2 regression for sex and age, which showed little impact on differential expression and refer to Suppl Table 23; I could not find these data in Suppl Table 23, e.g. all data for Age comparison is for comparison with Ximerakis et al. paper, not for the current dataset (and I apologize if the data is there and I simply was not able to find it)

Importantly, the comparison with sex and age signatures extracted from publicly available data is elegant and is the best proof the authors have so far for low impact of sex and age.

Overall, having written all above, for (1 Integration) I think that bringing all integration under umbrella of RPCA is fine, and even though some clusters are unstable, RPCA is a good enough integration algorithm for such dataset, and for (2 Covariate) I think that the impact of sex/age specific signatures can be evaluated by comparing with published datasets and by larger scITD analysis. Data presentation is much better in the revised manuscript. Nevertheless, the approach that was chosen by the authors, which looks like “to throw a lot of data without looking deep into data interpretation and just hiding it somewhere in complex and hard to go through supplementary tables or reviewer figure files”, is worrisome (again my feelings after digging through the data, it was difficult). They should be also more careful in claiming 14 subtypes of endothelial cells, unless they validate all subtypes by immunohistochemistry.

Author Rebuttals to Second Revision:

Reviewers' Comments:

Reviewer #1:

Comments for the Author:

COMMENT:

In this second resubmission by Walchli et al, the authors present a molecular atlas of the human brain across several age groups and disease conditions. Overall, this second-revised manuscript has improved readability and is more concise compared to the prior revisions. Both major concerns previously raised were addressed. The authors address age variation through additional analysis and external validation with published databases. Low sample numbers have been removed (cavernoma, hemangioma) which addresses both the concern for lowbiologic replicates and refines the scope of the paper. Additional samples were added to GBM atlas. EC transcriptional profiles from scRNA seq database again validated with further bulk RNA seq, but clarified that this validation is derived from different samples (internal validation across sequencing platforms). Clarified reference to select IF, CyCIF, and spatial transcriptomics validation. Although there have been prior published large datasets of the adult brain vasculature, the fetal vasculature, and select disease states, this manuscript does offer a more direct comparison between these atlases with a relatively large sample number. Authors also have comparison to peripheral tissues, which it seems makes this identification of markers more clear than previous work.

ANSWER:

We thank the reviewer for their very positive comments on our manuscript. For your convenience, please find below a comprehensive overview of what we have done to satisfy all the remaining questions/comments of the editor and of reviewer 3.

- i) with regard to the computational analyses
 - we ensured that we use state-of-the-art analysis methods (methods of integration, computational analyses, etc.) for the scRNA-seq analyses by having everything reviewed, discussed, improved accordingly, and approved by numerous leading international single-cell experts
 - according to the different scRNA-seq experts (with additional backgrounds in vascular biology, neuroscience, tumor biology, developmental biology, computational biology, etc.) we consulted, the computational analyses as well as the different methods of integration satisfied – after integration of all their input and advice – the highest standards in the field
- ii) with regard to the derived data and biological findings
 - independent of the methods of integration used – and even after the addition of new samples as requested by the referees – the core findings of the study were not affected and were consistent throughout
- iii) with regard to the reproducibility of our analyses and the underlying code
 - we have created a fully reproducible data and code repository to help the community build on our work and to assure that all our data and figures can be reproduced by anyone

We have thoroughly addressed every aspect of the scRNA-seq referees' concerns in the previous round of revisions and are confident with the computational analysis and biological interpretations based on our numerous interactions with the above-mentioned experts.

COMMENT:

One minor concern is that given the current conclusions from authors' analysis and their discussion it is not clear whether this work is broadly applicable beyond the cerebrovascular field. I would suggest that authors elaborate their discussion to describe how they envision their data can be applied more broadly.

ANSWER:

We thank the reviewer for this comment. While we had already stated corresponding sentences regarding “the organizing principles and single-cell heterogeneity of universal, specialized and activated endothelial and perivascular cells” in the “Discussion” section, we have now added additional

sentences in the “Supplementary Discussion” section to further discuss the potential broad applicability of our work beyond the cerebrovascular field.

Notably, to more specifically assess possible commonalities and differences between the cerebrovasculature and individual peripheral vascular beds, dedicated future studies are required.

COMMENT:

In addition, language regarding “reactivated fetal pathways” should also be refined in the abstract, as the authors have done in the manuscript body.

ANSWER:

We thank the reviewer for this comment and are fully aware that the hypothesized reactivation of developmentally active angiogenic signaling pathways in vascular-dependent CNS pathologies – although very intriguing – will need functional validation in future studies.

Identified pathways and genes that we found to be upregulated at the fetal stage of brain development, subsequently entering quiescence in the adult healthy brain vasculature, and becoming reactivated in various vascular-dependent CNS pathologies, will need to be tested using several animal models and confirmed both *in vitro* as well as *in vivo*.

Due to word limitation and required further shortening of the manuscript, we have not adapted/nuanced the wording in the “Abstract” section. However, we have adapted/nuanced the wording when discussing this concept in the “Supplementary Discussion” section and have added additional clarifications to explain to the reader that this assumption is based mainly on transcriptomic similarities between fetal and diseased cells and is in need of future validation in animal and/or human organoid models.

Notably, we have mentioned in the “Supplementary Discussion” section:

- that “reactivated developmental signaling pathways in disease are shared signaling pathways between development and disease whereas silenced in the adult^{1,2}; an interesting concept requiring further future validation^{1,2}.”
- that it remains to be clarified whether the pathways observed in brain pathologies are reactivated developmental pathways or rather reflect the persistence (for example, the presence since development) of a less differentiated cell type (or even a combination of these two)¹.

COMMENT:

Significant editing and reduction in the manuscript size has improved the readability. If authors can do some more will make the manuscript clearer.

ANSWER:

We thank the reviewer for this comment. We have further adapted the manuscript accordingly to respect all formatting guidelines, notably:

- the manuscript title
- the manuscript length
- the length of subheadings

Reviewer #3:

Comments for the Author:

COMMENT:

The authors made great efforts to have their manuscript more reader friendly. The data presentation is better organised, and it is easier to read and understand. As for the other two major points that were raised before, data integration and covariate (e.g. sex and age) impact, I have mixed feelings regarding the reply. The replies for each point are very complex, some information is repeated several times, in some instances the sentences are cryptic, and sometimes it was difficult to go through. Overall, I have feelings, although I might be mistaken, that the authors try to use multiple computational tools just to use them and show some data, without really looking deeply in these major issues – whether the data integration went well or whether covariates have an impact on their data. Below I provide detailed explanations.

ANSWER:

We thank the reviewer for their positive comments on our manuscript. We would like to share a few important aspects:

- i) with regard to the computational analyses
 - we ensured that we use state-of-the-art analysis methods (methods of integration, computational analyses, etc.) for the scRNA-seq analyses by having everything reviewed, discussed, improved accordingly, and approved by numerous leading international single-cell experts
 - according to the different scRNA-seq experts (with additional backgrounds in vascular biology, neuroscience, tumor biology, developmental biology, computational biology, etc.) we consulted, the computational analyses as well as the different methods of integration satisfied – after integration of all their inputs and advice – the highest standards in the field
- ii) with regard to the derived data and biological findings
 - independent of the methods of integration used – and even after the addition of new samples as requested by the referees – the core findings of the study were not affected and were consistent throughout
- iii) with regard to the reproducibility of our analyses and the underlying code
 - we have created a fully reproducible data and code repository to help the community build on our work and to assure that all our data and figures can be reproduced by anyone

We have thoroughly addressed every aspect of the scRNA-seq referee's concerns in the previous round of revisions and are confident with the computational analysis and biological interpretations based on our numerous interactions with the above-mentioned experts

In summary, we have thereby addressed all key reviewers' and editors' concerns regarding:

- covariates age and sex
- computational methods/overall analytical approach
- sample numbers amongst the different pathologies
- lengthiness of the manuscript and number of figures

COMMENT:

(1) Integration.

The summary of integration QC is in Extended Data Figure 4. Similar as in the previous round of the revision, it is rather obvious that different ways of integration have different impact on classification (e.g. please look at very different subclustering of arteriole, artery, capillary, mitochondrial at panel b and panel c). As recommended, the authors used Jaccard index to quantify stability of their clusters, and they used scclusteval package. Based on the results, the authors claim that the integration was similar between the methods. However, there is some confusion in interpretation of Extended Data Figure 4g-i – such diagonal line of the highest Jaccard indices is not what is the most important, the exact values of Jaccard indices are important. Based in the scale, there are quite some clusters that have maximum Jaccard indices of ~0.5 or below, which is highly unstable. In fact, there are only few clusters of >0.85, which is stable. Thus, the interpretation of Extended Data Figure 4g-i is opposite to what the authors write.

The authors can argue that the stability is much higher in region-wise comparison, which is shown in Reviewer figure 4. This is true, however, comparison in Reviewer figure 4 is done on cell types, where endothelial cells are all in one category “endothelial cells”, whereas in Extended Data Figure 4 endothelial cells are subclustered into 14 subtypes. Subclustering and identification of specific markers for endothelial subtype is the main point of the manuscript and this has to be robust.

ANSWER:

We thank the reviewer for the feedback regarding the computational integration/batch correction.

We would like to clarify two aspects:

- i) regarding the methods of integration, we did the comparison of different methods of integration (RPCA, CCA and Harmony) based on the reviewer's comments and advice during the second round of revisions (REVISIONS II)
 - ia) we would like to emphasize that we came to agreement with the reviewer and all the leading international single-cell experts we consulted that RPCA is the most adequate method to use in the manuscript for our datasets.
 - This is based on the fact that RPCA performed best in benchmarking, reduced batch effects in our data, and resulted in an endothelial map that manifested the best clustering according to AV specification (Supplementary Figure 9).
 - Moreover, the top cluster markers emanating from RPCA were also experimentally validated by immunohistochemistry (IHC) and immunofluorescence (IF) stainings (Figure 1g-n', Extended Data Figure 2u-di', c_{ii} - j_{ii} ', Extended Data Figure 3, Supplementary Figure 8, imaging mass cytometry (IMC) (Extended Data Figure 2k_{ii}-q_{ii}⁵) and RNA scope analysis (Extended Data Figure 2i-t, e_i - b_{ii}), thereby underlining the biological relevance of the RPCA-derived computational findings.
 - ib) The other methods of integration that were compared to RPCA – notably CCA and Harmony – can obviously not perfectly match in every aspect.
 - scRNA-seq integration/batch correction and clustering are unsolved and actively research areas, thus we expect our data to be reprocessed in the future by others who will build on our results regarding the comparison of different methods of scRNA-seq integration/batch correction.
- ii) regarding the assessment of the stability of the clusters between the different methods of integration for the overall merge of sorted endothelial cells, we have utilized the Jaccard index based on the reviewer's comments and advice during the second round of revisions (REVISIONS II)
 - iia) based on the choice of RPCA for the method of integration in our paper (i), see above), we performed the Jaccard index comparison based on the reviewer's comments (Reviewer Figure 4)
 - iib) we selected RPCA by the above-mentioned criteria (see (i)). Knowing that the other integration methods did not perform equally well as judged by these criteria, we are well aware that there are some EC clusters that compare with a low Jaccard index, indicating that the other methods of integration don't perfectly match to RPCA
 - we addressed the clusters resulting from RPCA experimentally by validating cluster markers at the protein level (see (i)) above), and interpreting the biological results in the context of prior knowledge.
 - iic) We make our data openly accessible so that others may build on our results and reprocess the results with other methods in the future. We would like to emphasize that based on our discussion with numerous leading international single-cell experts, that this is an issue the computational field is occupied with at the moment and which we therefore can't fully resolve at the time being.
 - moreover, it is not the topic of the paper to compare the different methods of integration.

In addition to everything mentioned above, based on the agreement to use of RPCA as the method of integration in our paper, what matters the most is the cluster definition of the ECs by RPCA, to which we have provided already extensive computational and experimental validation, notably:

- i) computational validation:
 - ia) comparison of the different methods of integration using UMAPs (Supplementary Figure 9b-e), showing that all methods of integration clustered the endothelial cells according to the arteriovenous specification in a comparable way

- ib) comparison of the proportion of the different AV clusters (Supplementary Figure 9f), showing that the proportion of the different AV clusters is comparable across the different methods of integration
- ii) experimental validation:
 - iia) at the RNA level using:
 - RNAscope
 - iib) at the protein level using:
 - immunofluorescence (IF)
 - immunohistochemistry (IHC)
 - imaging mass cytometry (IMC)

Importantly, we thank the reviewer for approving that we brought all integration/batch correction under the umbrella of RPCA and that this method is suitable for our datasets. This way we have ensured to be consistent with our analyses throughout the manuscript as the reviewer advised during the second round of revisions (REVISIONS II). We have indeed performed all integration/batch correction using RPCA, which performed very well for both the sorted EC and unsorted EC and PVC datasets (Figure 2a, Supplementary Figure 9, Reviewer Figure 4).

We have clarified in the manuscript the following analysis done:

- i) we have used RPCA for integration/batch correction of:
 - ia) CD31⁺/CD45⁻ FACS sorted ECs of individual patients within a given brain entity, e.g:
 - Integration of ECs of the five fetuses' brains together.
 - Integration of ECs of the nine adult/control brain (temporal lobe) patients together
 - Integration of ECs of the eight GBM patients together
 - Integration of ECs of the five AVM patients together, etc
 - and the same for every other individual entity for the sorted endothelial cells' samples
 - ib) unsorted cells (ECs and PVCs) of individual patients within a given brain entity, e.g:
 - Integration of unsorted cells of the seven fetuses' brains together
 - Integration of unsorted cells of the six adult/control brain (temporal lobe) patients together
 - Integration of unsorted cells of the five GBM patients together
 - Integration of unsorted cells of the five AVM patients together, etc
 - and the same for every other individual entity for unsorted ECs and PVCs samples
- ii) we have used RPCA for integration/batch correction of:
 - the overall merge of CD31⁺/CD45⁻ FACS sorted endothelial cells from different brain entities (fetal, adult/control, and pathological brains)
- iii) we did not do the integration/batch correction at the level of the overall merge of different brain entities for the unsorted cells (ECs and PVCs)
 - the reason is that the different cell types in the unsorted samples (ECs and PVCs) are not the same across different the brain entities (fetal brains, adult/control brains, brain tumors and brain vascular malformations), and as the cell types cluster by cell type, is not effective to integrate/batch correct the different entities, as it would result in incorrect annotation and noisy clusters (Reviewer Figure 5).
 - this is obviously different for the sorted samples which contain only ECs and thus integration and batch correction across the different brain entities is effective (see ii))

In summary

- i) integration/batch correction at the level of the overall merge of all entities
 - ia) for CD31⁺/CD45⁻ FACS-sorted endothelial cells
 - integration/batch correction was done using four different methods of integration, notably RPCA, CCA, Harmony, and scANVI
 - method of choice: RPCA
 - CCA, Harmony, and scANVI are comparable in aspects (as outlined above), (Supplementary Figure 9), also shown by applying the scIB diagnostics (Modified average silhouette width (ASW)) (Supplementary Table 28)³.

importantly, independent of the method of integration/batch correction, the biological findings are not affected, as evidenced by the clustering of the endothelial cells according to the arteriovenous specification (Supplementary Figure 9b-e) and the similar proportions of the different endothelial cell clusters annotated across the four different methods of integration compared (Supplementary Figure 9f).

- ib) for unsorted endothelial- and perivascular cells
 - integration/batch correction was not done/used at the level of the overall merge of all entities
- ii) integration/batch correction at the level of the individual entities
 - iia) for CD31⁺/CD45⁻ FACS-sorted endothelial cells
 - method of choice: RPCA
 - iib) for unsorted endothelial- and perivascular cells
 - method of choice: RPCA
 - CCA and Harmony are equally good (Reviewer Figure 4), also proven by Jaccard index comparison.

Importantly, independent of the method of integration/batch correction, the biological conclusions are not affected. Thus, as the reviewer advised, we have applied the same method of integration/batch correction (RPCA) across the different analyses we performed in the manuscript, and we have clarified the details in the “Methods” section.

Notably, differential gene expression that is used for cell annotation/AV-annotation and pathway analysis is always done on the non-integrated/non batch corrected counts (i.e. the “RNA assay of Seurat”) and thus does not involve integrated/batch corrected counts – which is also the recommendation of SEURAT package developers.

Importantly, we have performed the integration/batch correction analysis taking into account key benchmarking articles comparing methods of integration/batch correction, as mentioned by the reviewer:

-

- Luecken, M.D., Büttner, M., Chaichoompu, K. et al. Benchmarking atlas-level data integration in single-cell genomics. *Nat Methods* 19, 41–50 (2022). <https://doi.org/10.1038/s41592-021-01336-8> (<https://www.biorxiv.org/content/10.1101/2020.05.22.111161v2> which was meanwhile published in *Nature Methods*³)

- Tran, H.T.N., Ang, K.S., Chevrier, M. et al. A benchmark of batch-effect correction methods for single-cell RNA sequencing data. *Genome Biol* 21, 12 (2020). <https://doi.org/10.1186/s13059-019-1850-9>

Regarding the Jaccard index - which we have performed as the reviewer advised - and the endothelial subclusters, we would like to mention that we show the subcluster analysis in Supplementary Figure 7a,f,g,h.

As mentioned in the “Results” section, we pooled EC (sub)clusters of similar identity, notably:

- i) arterial subclusters
- ii) venous subclusters
- iii) capillary subclusters
- iv) EC subclusters outside the arteriovenous compartment

for further downstream analyses. This resulted in the 14 EC clusters displayed in Figure 2a, Figure 3a,e,i and Supplementary Figure 7a,f,g,h.

Accordingly, these 14 EC clusters are subdivided as follows (Figure 2a):

- i) the arterial compartment is composed of the EC clusters:
 - large artery
 - artery
 - arteriole
- ii) the venous compartment is composed of the EC clusters:

- large vein
- vein
- venule
- iii) and the capillary compartment is composed of the EC clusters:
 - capillary
 - angiogenic capillary
- iv) the EC clusters outside the arteriovenous specification are:
 - (proliferating) EndoMT
 - (proliferating) stem-to-EC
 - proliferating cells

Importantly, we would like to emphasize that we provide 14 EC clusters in which every compartment (i.e. arterial, venous, capillary) is subdivided into multiple EC subclusters, highlighting the unprecedented resolution of EC clusters provided here. Accordingly, these 14 EC clusters provide a more accurate representation of the underlying EC biology when compared to recent publications, which display between 3-6 EC clusters⁴⁻⁶.

As per the reviewers advice, we have reduced the number of capillary subclusters from 11 to 2 as shown in the Jaccard index analysis (Supplementary Figure 7a,f,g,h). This has resulted in a very high degree of congruity, thereby strongly suggesting that these EC subclusters are actually not just due to noise but in contrast, are biologically meaningful.

COMMENT:

(2) Covariate control.

The authors did a lot of additional analyses to demonstrate (potential) lack of age and sex impact on compositional changes. However, it looks as if the authors do not take it serious that covariates can have an impact on their data, since all analyses are either in Reviewer figures (thus not included in the published manuscript) or in complicated Suppl Tables (there are no explanations for the reader how to interpret highly complicated data in Suppl Table 10, which means that the reader will just skip it). Finally, as I also stated in the previous round of revision, it might be that the number of samples per disease type is too low to perform robust analysis for covariate impact, which is the case after going through the revised data (see below).

Coming to the analyses themselves, and the claim that sex and age (and these are only 2 covariates considered, however, there are others) do not have an impact on the data:

- scCODA does show compositional differences in young vs old, e.g. Reviewer figure 2 panel v shows twice higher proportion of arteriole in old, and panel t – twice higher proportion of venule in old

- for propeller method, Phipson et al. Bioinformatics 2022, it is a robust method with enough replicates, and the number of replicates is rather small in the compared datasets, so the method is not applicable here

- scITD does not work well on 5-7 samples, most comparisons in Reviewer figure 3n-v; this analysis has to be done on overall integrated dataset control + all diseases to see potential impact and identify covariate specific gene expression signatures

- the authors claimed that they did DESeq2 regression for sex and age, which showed little impact on differential expression and refer to Suppl Table 23; I could not find these data in Suppl Table 23, e.g. all data for Age comparison is for comparison with Ximerakis et al. paper, not for the current dataset (and I apologize if the data is there and I simply was not able to find it)

Importantly, the comparison with sex and age signatures extracted from publicly available data is elegant and is the best proof the authors have so far for low impact of sex and age.

ANSWER:

We thank the reviewer for this comment.

Regarding the supplementary tables provided

We have provided a detailed Supplementary Information (SI) guide with detailed explanation of the provided Supplementary Tables to make it easy for the reader to interpret and analyze.

Regarding the number of different conditions in the atlas, the exclusion of rare samples and of under-powered disease comparisons, and the addition of samples

To address the reviewer's comment regarding the number of different conditions in the atlas and the under-powered disease comparisons, we have:

- i) removed the very rare samples (2 cavernomas, 2 temporal lobes adjacent to cavernomas, 2 hemangioblastomas) from the atlas.
- ii) added various additional samples (3 adult/control brains 2 lower-grade gliomas, 3 high-grade gliomas/glioblastomas) to the atlas.

thereby further strengthening and validating our findings in the atlas by focusing the atlas on the conditions with enough samples.

This being said, we now present a number of collected scRNA-seq samples of both sorted brain ECs and of unsorted brain ECs and PVCs that is – to the best of our knowledge – unprecedented when compared to the existing literature. Accordingly, please find below the overview of the number of samples (Extended Data Figure 1).

The analyzed samples are derived from the following entities:

For sorted brain ECs

- 5 fetal brain samples coming from 5 fetuses
- 5 fetal periphery samples coming from 5 fetuses
- 12 adult/control brains (temporal lobes) samples coming from 11 patients
- 57 diseased samples coming from 50 patients

For unsorted brain ECs and PVCs

- 7 fetal brain samples coming from 7 fetuses
- 7 fetal periphery samples coming from 7 fetuses
- 6 adult/control brains (temporal lobes) samples coming from 6 patients
- 18 diseased samples coming from 18 patients

In summary, taking into account the additional samples that we included in this second round of REVISIONS (REVISIONS II), 117 sorted and unsorted samples derived from 68 patients were analyzed in our manuscript. We would thus like to emphasize that the conclusions that are made in our manuscript are not based on only a few samples, but are rather based on an unprecedented amount of samples:

- both at the level of the overall merge of all entities
- as well as at the level of individual entities.

thereby focusing the atlas on the analyses where there are enough samples and where robust analyses can be made (while also having excluded the diseased entities with very low sample numbers).

Regarding the potential influence of the covariates age and sex on the biological findings in the atlas

We have worked extensively to comprehensively address the covariates age and sex in our own datasets, we have performed various analysis during the previous two rounds of revisions (REVISIONS I and REVISIONS II):

- i) Upon the first round of revisions (REVISIONS I), we had performed the following analyses:
 - ia) comparison of the parameters “age” and “sex” against random shuffles of the parameters “age” and “sex”, which revealed:
 - that the number of significant DEGs between the different ages and the two sexes are low (Reviewer Figure 1)
 - and that these differences in terms of the number of significant DEGs are comparable to or even lower than (and thus not significantly different) random shuffles of the parameters “age” and “sex” (Supplementary Table 25)
 - ib) pathway analysis comparing the different ages/sexes, on the DEGs for both parameters “age” and “sex”
 - revealing that no biologically meaningful pathways were significantly enriched
 - ic) scCODA analysis showed no significant compositional changes between the different sexes and ages in the different entities (Reviewer Figure 2, Supplementary Table 12)
 - we would like to confirm that based on our scCODA analysis that there were not statistically significant differences comparing the old vs young endothelial cells as indicated in the corresponding tabs of the excel table provided (Supplementary Table 12)
- ii) In addition, for this second round of revisions (REVISIONS II), we have now performed the following additional analyses:

- iia) to regress the covariates age and sex, we have utilized/applied the state-of-the-art statistical methodologies which have integrated covariate options⁷⁻⁹ while performing cell compositional analysis (Supplementary Tables 13, 14, 15). Notably, we referred to three independent computational packages/algorithms, namely:
 - i) Tree-aggregated amplicon and single-cell compositional data analysis (tascCODA), which is a fully Bayesian model for tree-aggregated modeling of count data and is an extension of the scCODA⁷
 - ii) Dirichlet regression analysis (DirichletReg R package) dirichlet regression models can be used to analyze a set of variables lying in a bounded interval that sum up to a constant (e.g., proportions, compositions, etc.) exhibiting skewness and heteroscedasticity, without having to transform the data^{8,10}
 - iii) Propeller method (speckle R package), which accommodates complex experimental designs and has the flexibility to model additional covariates of interest⁹.
 - The input number of replicates was compatible with the developer's requirements

Importantly, we can conclude that the key findings related to cell compositional changes in sorted ECs of brain tumors' and brain vascular malformations' ECs as compared to sorted ECs from adult/control brain, are not affected by the covariates age and sex utilizing the three above-mentioned methods (tascCODA, Dirichlet regression and Propeller). These analyses verify the scCODA findings during the first round of revisions (REVISIONS I).

- iib) to further address possible changes in endothelial cell cluster composition, we have performed compositional analysis using Cacoa¹¹ - based on the reviewer's comment - which showed (Supplementary Table 16):
 - that there is no significant difference in EC cluster composition between the adult/control brains (temporal lobe) samples by age
 - that there is no significant difference in EC cluster composition between the male and female samples for the different entities studied, notably: TL, AVM, LGG, GBM, MET and MEN
 - that there is no significant difference in EC cluster composition according to the MGMT methylation status for the GBM samples
 - that there is no significant difference in EC cluster composition according to the histopathological subtype for the brain MET samples
- iic) we have performed analysis using scITD package - based on the reviewer's comment - which showed (Review Figure 3n-v):
 - that there are no significant age-associated factors in sorted ECs of temporal lobe (adult/control brain) samples
 - that there are no significant sex-associated factors in sorted ECs of all the studied entities
 - that there are no significant MGMT methylation status-associated factors in sorted ECs of GBM samples
 - that there are no significant metastasis subtype-associated factors in sorted ECs of brain metastasis samples
- iid) we have performed differential expression analysis with regression of sex using DESeq package (DOI: 10.18129/B9.bioc.DESeq2) (Supplementary Table 29 in the corresponding tab of the excel file), and the results revealed that the findings are not affected at the gene (Jaccard index = 0.988) and pathway level
- iie) we have performed integration/batch correction with regression of sex/age covariates, and showed that the results are not affected (Reviewer Figure 3w-z)
- iif) To further validate these findings in other cohorts and to further address the reviewer's comments, we have performed additional analyses using publicly available datasets for age and sex related genes. By compiling multiple publicly available brain sc/snRNA-seq datasets that have "age" and "sex" annotation of endothelial cells.
 - regarding the sex comparison

- we have compiled 9 human snRNA-seq datasets from 7 publications^{5,6,12-16}, that included 278 patients and computationally extracted/subset 34,322 endothelial cells
- then did differential expression analysis comparing male vs female endothelial cells, for which the resulting genes were filtered based on a threshold of being p-value < 0.05, log₂FC ≥ 0.25), resulting in a set of brain ECs sex regulated genes (Reviewer Figure 3g-i).
- X- and Y- chromosome genes obtained from the C1 positional geneset (http://www.gsea-msigdb.org/gsea/msigdb/human/collection_details.jsp#C1), (Reviewer Figure 3j-m).
- regarding the age comparison
 - we have analyzed the endothelial cell subset of Ximerakis et al., *Nature Neuroscience*, 2019¹⁷- that did single-cell transcriptomic profiling of the aging mouse brain
 - performing differential expression analysis between 8 young (2–3 months) and 8 old (21–23 months) mice's endothelial cells (the resulting genes were filtered based on a threshold of being p-value < 0.05, log₂FC ≥ 0.25), followed my homology mapping to human genes resulting in a set of brain ECs age regulated genes (Reviewer Figure 3a-f).

The gene lists generated in the aforementioned points were then compared to the pathological signatures that we defined in our paper using:

- Venn diagrams
- Correlation analysis using Jaccard test and Fisher's exact test
- GSEA analysis

(Reviewer Figure 3a-m, Supplementary Table 29).

We thank the reviewer for complementing this analysis and describing it as elegant, and as the best proof for the low impact of covariates age and sex. We believe that in addition to it, the utilization of the state-of-the-art statistical methodologies which have integrated covariate options⁷⁻⁹ have strengthened our conclusions.

In summary, the additional analyses show that in endothelial cells:

- i) differentially expressed genes
- ii) resulting signaling pathways
- iii) endothelial cell clustering
- iv) results of integration in the adult control brains and in the different diseases

are not influenced by age and sex, further underlining that the covariates age and sex do not influence our biological findings across adulthood and adult diseases.

Regarding other potential covariates related to sample collection and the potential influence of the covariates sequencing batch and processing site on the biological findings in the atlas

Finally, we would like to highlight that we have followed a very stringent methodology in sample collection, preparation and sequencing thereby minimizing the effects of sequencing batch and processing site. In consequence, we are very confident that technical introduced variability is minimal. Regarding the experimental methodology of sample harvesting:

In brief, from an experimental point of view, the tissue samples were enzymatically digested using collagenase and dispase immediately after tissue harvesting (see Figure 1a,b, for technical details see the “Methods” section).

- For the unsorted EC and PVCs samples, the digested tissue was then FACS sorted to exclude debris (via forward- and side scatter plots, see the “Methods” section) and dead cells (via cytox blue, see the “Methods” section), and then immediately single-cell sequenced using the 10x droplet protocol (see Figure 1a,b, for technical details the “Methods” section).
- For the sorted EC samples, the harvested tissue was digested, FACS sorted to exclude debris and dead cells and then processed using CD31⁺/CD45⁻ FACS sorting to gate for ECs and subsequently single-cell sequenced using the 10x droplet protocol (see Figure 1a, for technical details see the “Methods” section).

COMMENT:

Overall, having written all above, for (1 Integration) I think that bringing all integration under umbrella of RPCA is fine, and even though some clusters are unstable, RPCA is a good enough integration algorithm for such dataset, and for (2 Covariate) I think that the impact of sex/age specific signatures can be evaluated by comparing with published datasets and by larger scITD analysis. Data presentation is much better in the revised manuscript. Nevertheless, the approach that was chosen by the authors, which looks like “to throw a lot of data without looking deep into data interpretation and just hiding it somewhere in complex and hard to go through supplementary tables or reviewer figure files”, is worrisome (again my feelings after digging through the data, it was difficult). They should be also more careful in claiming 14 subtypes of endothelial cells, unless they validate all subtypes by immunohistochemistry.

ANSWER:

We thank the reviewer for approving that we brought all integration/batch correction under the umbrella of RPCA and that this method is suitable for our datasets. This way we have ensured to be consistent with our analyses throughout the manuscript as the reviewer advised during the second round of revisions (REVISIONS II).

We have indeed performed all integration/batch correction using RPCA, which performed very well for both the sorted EC and unsorted EC and PVC datasets (Figure 2a, Supplementary Figure 9, Reviewer Figure 4).

We thank the reviewer for complementing the covariate analysis by comparing to public datasets and describing it as elegant, and as the best proof for the low impact of covariates age and sex.

We believe that in addition to this analysis, the utilization of the state-of-the-art statistical methodologies which have integrated covariate options⁷⁻⁹ have strengthened our conclusions related to the impact of covariates. We have worked extensively to comprehensively address the covariates age and sex in our own datasets, for which we have performed various analysis using independent algorithms during the two rounds of revisions (REVISIONS I and REVISIONS II) including methods advised by the reviewer to address the concerns related to covariates in the most holistic way. Moreover, we have meticulously scrutinized the outcomes to ensure our conclusions are correct and accurate.

Furthermore, regarding the Supplementary Tables, we have provided a Supplementary Information (SI) guide with detailed explanation of the provided Supplementary Tables to make it easy for the reader to interpret and analyze.

In addition to everything mentioned above, based on the agreement to use of RPCA as the method of integration in our paper, what matters the most is the cluster definition of the ECs by RPCA, to which we have provided already extensive computational and experimental validation, notably:

- i) computational validation:
 - ia) comparison of the different methods of integration using UMAPs (Supplementary Figure 9b-e), showing that all methods of integration clustered the endothelial cells according to the arteriovenous specification in a comparable way
 - ib) comparison of the proportion of the different AV clusters (Supplementary Figure 8f), showing that the proportion of the different AV clusters is comparable across the different methods of integration
- ii) experimental validation:
 - iia) at the RNA level using:
 - RNAscope (Extended Data Figure 2i-t, e_i-b_{ii})
 - iib) at the protein level using:
 - immunofluorescence (IF) (Figure 1g-n', Extended Data Figure 2u-d', c_{ii}-j_{ii}', Extended Data Figure 3)
 - immunohistochemistry (IHC) (Supplementary Figure 8)
 - imaging mass cytometry (IMC) (Extended Data Figure 2k_{ii}-q_{ii}'⁵)

REFERENCES

- 1 Wälchli, T. *et al.* Shaping the brain vasculature in development and disease in the single-cell era. *Nat Rev Neurosci* **24**, 271-298, doi:10.1038/s41583-023-00684-y (2023).
- 2 Schwab, M. *et al.* Nucleolin promotes angiogenesis and endothelial metabolism along the oncofetal axis in the human brain vasculature. *JCI Insight* **8**, doi:10.1172/jci.insight.143071 (2023).
- 3 Luecken, M. D. *et al.* Benchmarking atlas-level data integration in single-cell genomics. *Nat Methods* **19**, 41-50, doi:10.1038/s41592-021-01336-8 (2022).
- 4 Winkler, E. A. *et al.* A single-cell atlas of the normal and malformed human brain vasculature. *Science* **375**, eabi7377, doi:10.1126/science.abi7377 (2022).
- 5 Garcia, F. J. *et al.* Single-cell dissection of the human brain vasculature. *Nature* **603**, 893-899, doi:10.1038/s41586-022-04521-7 (2022).
- 6 Yang, A. C. *et al.* A human brain vascular atlas reveals diverse mediators of Alzheimer's risk. *Nature* **603**, 885-892, doi:10.1038/s41586-021-04369-3 (2022).
- 7 Ostner, J., Carcy, S. & Muller, C. L. tascCODA: Bayesian Tree-Aggregated Analysis of Compositional Amplicon and Single-Cell Data. *Front Genet* **12**, 766405, doi:10.3389/fgene.2021.766405 (2021).
- 8 Maier, M. J. DirichletReg: Dirichlet Regression for Compositional Data in R. *Research Report Series* (2014).
- 9 Phipson, B. *et al.* propeller: testing for differences in cell type proportions in single cell data. *Bioinformatics* **38**, 4720-4726, doi:10.1093/bioinformatics/btac582 (2022).
- 10 Eraslan, G. *et al.* Single-nucleus cross-tissue molecular reference maps toward understanding disease gene function. *Science* **376**, eabl4290, doi:10.1126/science.abl4290 (2022).
- 11 Viktor Petukhov, A. I., Rasmus Rydbirk, Shenglin Mei, Lars Christoffersen, Konstantin Khodosevich, Peter V. Kharchenko. Case-control analysis of single-cell RNA-seq studies. *bioRxiv* (2022).
- 12 Yang, A. C. *et al.* Dysregulation of brain and choroid plexus cell types in severe COVID-19. *Nature* **595**, 565-571, doi:10.1038/s41586-021-03710-0 (2021).
- 13 Grubman, A. *et al.* A single-cell atlas of entorhinal cortex from individuals with Alzheimer's disease reveals cell-type-specific gene expression regulation. *Nat Neurosci* **22**, 2087-2097, doi:10.1038/s41593-019-0539-4 (2019).
- 14 Velmeshov, D. *et al.* Single-cell genomics identifies cell type-specific molecular changes in autism. *Science* **364**, 685-689, doi:10.1126/science.aav8130 (2019).
- 15 Ayhan, F. *et al.* Resolving cellular and molecular diversity along the hippocampal anterior-to-posterior axis in humans. *Neuron* **109**, 2091-2105 e2096, doi:10.1016/j.neuron.2021.05.003 (2021).
- 16 Bennett, D. A. *et al.* Religious Orders Study and Rush Memory and Aging Project. *J Alzheimers Dis* **64**, S161-S189, doi:10.3233/JAD-179939 (2018).
- 17 Ximerakis, M. *et al.* Single-cell transcriptomic profiling of the aging mouse brain. *Nat Neurosci* **22**, 1696-1708, doi:10.1038/s41593-019-0491-3 (2019).